

# Phylogeny of Paleozoic limbed vertebrates reassessed through revision and expansion of the largest published relevant data matrix

David Marjanović[1] and Michel Laurin[2]

[1] Science Programme "Evolution and Geoprocesses", Museum für Naturkunde—Leibniz Institute for Evolutionary and Biodiversity Research, Berlin, Germany
[2] Centre de Recherches sur la Paléobiologie et les Paléoenvironnements (CR2P), Centre national de la Recherche scientifique (CNRS)/Muséum national d'Histoire naturelle (MNHN)/Sorbonne Université, Paris, France

Corresponding author
David Marjanović,
david.marjanovic@gmx.at

## ABSTRACT

The largest published phylogenetic analysis of early limbed vertebrates (Ruta M, Coates MI. 2007. *Journal of Systematic Palaeontology* 5:69–122) recovered, for example, Seymouriamorpha, Diadectomorpha and (in some trees) Caudata as paraphyletic and found the "temnospondyl hypothesis" on the origin of Lissamphibia (TH) to be more parsimonious than the "lepospondyl hypothesis" (LH)—though only, as we show, by one step. We report 4,200 misscored cells, over half of them due to typographic and similar accidental errors. Further, some characters were duplicated; some had only one described state; for one, most taxa were scored after presumed relatives. Even potentially continuous characters were unordered, the effects of ontogeny were not sufficiently taken into account, and data published after 2001 were mostly excluded. After these issues are improved— we document and justify all changes to the matrix—but no characters are added, we find (Analysis R1) much longer trees with, for example, monophyletic Caudata, Diadectomorpha and (in some trees) Seymouriamorpha; *Ichthyostega* either crownward or rootward of *Acanthostega*; and Anthracosauria either crownward or rootward of Temnospondyli. The LH is nine steps shorter than the TH (R2; constrained) and 12 steps shorter than the "polyphyly hypothesis" (PH—R3; constrained). *Brachydectes* (Lysorophia) is not found next to Lissamphibia; instead, a large clade that includes the adelogyrinids, urocordylid "nectrideans" and aïstopods occupies that position. As expected from the taxon/character ratio, most bootstrap values are low. Adding 56 terminal taxa to the original 102 increases the resolution (and decreases most bootstrap values). The added taxa range in completeness from complete articulated skeletons to an incomplete lower jaw. Even though the lissamphibian-like temnospondyls *Gerobatrachus*, *Micropholis* and *Tungussogyrinus* and the extremely peramorphic salamander *Chelotriton* are added, the difference between LH (R4; unconstrained) and TH (R5) rises to 10 steps, that between LH and PH (R6) to 15; the TH also requires several more regains of lost bones than the LH. *Casineria*, in which we tentatively identify a postbranchial lamina, emerges rather far from amniote origins in a gephyrostegid-chroniosuchian grade. Bayesian inference (Analysis EB, settings as in R4) mostly agrees with R4. High posterior probabilities are found for

Lissamphibia (1.00) and the LH (0.92); however, many branches remain weakly supported, and most are short, as expected from the small character sample. We discuss phylogeny, approaches to coding, methods of phylogenetics (Bayesian inference vs. equally weighted vs. reweighted parsimony), some character complexes (e.g. preaxial/postaxial polarity in limb development), and prospects for further improvement of this matrix. Even in its revised state, the matrix cannot provide a robust assessment of the phylogeny of early limbed vertebrates. Sufficient improvement will be laborious—but not difficult.

## INTRODUCTION

This ancient inhabitant of the coal swamps of Nova Scotia, was, in short, as we often find to be the case with the earliest forms of life, the possessor of powers and structures not usually, in the modern world, combined in a single species. It was certainly not a fish, yet its bony scales, and the form of its vertebræ, and of its teeth, might, in the absence of other evidence, cause it to be mistaken for one. We call it a batrachian, yet its dentition, the sculpturing of the bones of its skull, which were certainly no more external plates than the similar bones of a crocodile, its ribs, and the structure of its limbs, remind us of the higher reptiles; and we do not know that it ever possessed gills, or passed through a larval or fish-like condition. Still, in a great many important characters, its structures are undoubtedly batrachian. It stands, in short, in the same position with the *Lepidodendra* and *Sigillariæ* under whose shade it crept, which though placed by palæo-botanists in alliance with certain modern groups of plants, manifestly differed from these in many of their characters, and occupied a different position in nature. In the coal period, the distinctions of physical and vital conditions were not well defined—dry land and water, terrestrial and aquatic plants and animals, and lower and higher forms of animal and vegetable life, are consequently not easily separated from each other.

– *Dawson (1863*: 23–24*) about *Dendrerpeton acadianum*

Homoplasy Is even More Common than I, or perhaps Anyone, Has ever Imagined
– section headline in *Wake (2009*: 343*)

What is required is a more complete discussion of the character coding of previous data matrices, and a thorough reanalysis based on those matrices. This would enable a well-founded discussion of lissamphibian origins in light of a supermatrix based on all the current and pertinent data.

– *Sigurdsen & Green (2011*: 459*)

Giant morphological data matrices are increasingly common in cladistic analyses of vertebrate phylogeny, reporting numbers of characters never seen or expected before.

However, the concern for size is usually not followed by an equivalent, if any, concern for character construction/selection criteria. Therefore, the question of whether quantity parallels quality for such influential works remains open.

– Simões et al. (2017: abstract)

Not too surprisingly, as it is yet a youthful paradigm shift, modern phylogenetic systematics is still evolving to improve on the lack of precision, rigour and objectivity it inherited from the pre-cladistic period. Furthermore, transforming a descriptive science (morphological description) bounded by language as a means of outlining empirical observations into hopefully objectively delimited characters and character states is a difficult task; every effort to do so is to be commended, while at the same time rigorously scrutinized and improved upon.

– Simões et al. (2017: 215)

Phylogenetic trees are hypotheses that attempt to explain a data matrix. Much work has gone, to great success, into the methods for generating and testing phylogenetic hypotheses from a given data matrix; and a matrix of molecular data can, apart from the problem of alignment, be largely taken for granted as a set of observed facts. But molecular data are not always available. In many cases phylogeneticists have to rely on morphological data—and a matrix of morphological data is a matrix of hypotheses. The characters, their states, and the relationships between the states are hypotheses that rely on hypotheses about homology, about independent evolution from other characters, about ontogeny and even preservation (especially in the case of fossils); the terminal taxa (operational taxonomic units—OTUs) are hypotheses that rely on hypotheses of monophyly, ontogeny and again preservation; and even given all these, each cell in a data matrix is still a hypothesis that relies on hypotheses of homology, ontogeny and preservation—some are close enough to observed facts, others less so. In addition, morphological data matrices can only be compiled by hand—there is no equivalent to sequencer machines or alignment programs. This makes human error inevitable. Consequently, morphological data matrices must not be taken for granted as sets of objective data; the hypotheses of which they consist must be identified and carefully tested.

The analysis by *Ruta & Coates (2007*—hereinafter RC07*)* has played a large role in shaping current ideas on the phylogeny and early evolution (Late Devonian to Cisuralian, with a few younger taxa) of the limbed vertebrates, including the origins of amniotes and lissamphibians. Being based on the largest matrix so far applied to this problem, its results have been widely treated as a consensus and even used as the basis for further work in evolutionary biology (*Bernardi et al., 2016*; *MacIver et al., 2017*). However, several conflicting phylogenetic results have persisted in other analyses based on different matrices (*Vallin & Laurin, 2004*; *Marjanović & Laurin, 2008*, *2009*; *Sigurdsen & Green, 2011*; *Pardo et al., 2017*; *Pardo, Small & Huttenlocker, 2017*: fig. 2, S6, S7). Although the large differences in character sampling between any two of these analyses may be the greatest contribution to their discrepancies (*Pardo et al., 2017*; compare also *Cau, 2018a*), it is also possible that some of the differences between these trees may be a function of taxon

sampling, analytical parameters like ordering, choice of optimization criterion (parsimony or Bayesian inference), correlation between characters (due to different treatments of ontogeny and heterochrony or other sources of large-scale convergence, or to outright duplication of characters in the same matrix), or accidental misscores (*Vallin & Laurin, 2004*; *Wiens, Bonett & Chippindale, 2005*; *Tykoski, 2005*; *Pawley, 2006*; *Marjanović & Laurin, 2008*, *2009*, *2013*; *Sigurdsen & Green, 2011*; *Langer et al., 2017*; *Spindler et al., 2018*: online resource 3). We have aimed to test this complex of hypotheses rigorously by reevaluating the matrix of RC07 in detail, and by adding taxa for a separate set of analyses.

## Aims

Exhaustive treatment of characters and taxa is the most appropriate way to disentangle contrasting phylogenetic signals in large matrices.

– RC07 (abstract)

*Ruta, Coates & Quicke (2003)* and RC07 presented two successive versions of a new matrix, discussed taxa and characters, analyzed their matrices with various methods and constraints, and used the resulting trees as a starting point for a review of the phylogeny of limbed vertebrates in general and the origin of lissamphibians in particular. Similarly, we have investigated the following questions.

### *Accuracy of analysis procedure*

• Did RC07 find all of the most parsimonious trees (MPTs) that fit their matrix?

This may seem trivial, but *Matsumoto et al. (2013)* reported that the software PAUP* 4.0b10 found three times as many MPTs as TNT 1.0 did when used on their matrix; conversely, *Schoch (2013*: 682*)* reported that PAUP 3.1 did not find any trees as short as those recovered by TNT; *Baron, Norman & Barrett (2017)* found only 93 of 16,632 MPTs, having neglected to run a second round of tree bisection and reconnection on their TNT trees (*Watanabe, 2017a*, *2017b*; *Langer et al., 2017*: supplementary information: 26); likewise using TNT, *Cau (2018a)* found only 3,072 of 10,872 MPTs (*Mortimer, 2018*; *Cau, 2018b*); and *Skutschas & Gubin (2012)* found that the "parsimony ratchet", a procedure for reducing calculation time (*Ruta, Coates & Quicke, 2003*), did not find any trees less than 35 steps longer than the MPTs. RC07 used the parsimony ratchet. We therefore repeated their analysis without using the ratchet (Analysis O1—see Table 1 and Fig. 1).

• What is the difference in tree length between the MPTs of RC07, which find Lissamphibia nested among the temnospondyls (Fig. 1), and the shortest trees compatible with their matrix that are constrained to place Lissamphibia among the "lepospondyls"?

RC07 reported that this difference is both nine steps (p. 85) and 15 steps (p. 86). The first resulted from an unpublished constraint compatible with the tree of *Laurin (1998a)*, which is not the only possibility for where and how Lissamphibia could be placed among the "lepospondyls". RC07 did not publish or describe the constraint they used for

**Table 1 Overview of analysis settings and results concerning lissamphibian origins.** See Table 3 for results concerning other questions. Steps were counted in PAUP* (*Swofford, 2003*), partial uncertainty was distinguished from polymorphism. "Parsimony" refers to equally weighted maximum parsimony, "bootstrap" to equally weighted maximum parsimony bootstrap, "Bayesian" to Bayesian inference.

| Matrix | Taxon sample | Analysis | Method | Constraint | Steps | Result | Figures |
|--------|-------------|----------|--------|-----------|-------|--------|---------|
| RC07 | RC07 (102 OTUs) | O1 | Parsimony | None | 1,621 | TH | 2 |
| | | O2 | | Against TH[1] | 1,622 | LH | 9 |
| | | O3 | | For PH | 1,633 | PH | – |
| | Revised | R1 | | None | 2,182 | LH | 10, 11 |
| | | R2 | | Against LH[2] | 2,191 | TH[3] | 12 |
| | | R3 | | For PH | 2,194 | PH | 13 |
| | | B1 | Bootstrap | None | 2,210 | LH | 18 |
| | Expanded (158 OTUs) | R4 | Parsimony | None | 3,011 | LH | 14 |
| | | R5 | | Against LH[2] | 3,021 | TH | 15, 16 |
| | | R6 | | For PH | 3,026 | PH | 17 |
| | | B2 | Bootstrap | None | 3,089 | LH | 19 |
| | | EB | Bayesian | None | (3,075)[4] | LH | 20, 21 |

Notes:
[1] This constraint is aimed at enforcing the LH, but it is also compatible with the (never proposed) "inverse polyphyly hypothesis". Where the salientians would be lepospondyls and the gymnophionomorphs (represented by *Eocaecilia*) would be temnospondyls.
[2] This constraint is aimed at enforcing the TH, but it is also compatible with the PH.
[3] A highly unusual version of the TH, see Results, Discussion and Table 3.
[4] This is the parsimony treelength calculated for this topology by PAUP*. The more appropriate Bayesian treelength is not comparable to the other lengths presented here.

the second. We therefore created a new constraint, which we describe explicitly below, and used it in a second analysis of their matrix (Analysis O2). A third (Analysis O3) is constrained against lissamphibian monophyly and is useful for certain comparisons.

### Accuracy of the matrix of RC07

Analogously to the problem of alignment in molecular analyses, morphological analyses begin with the construction of a dataset, where characters need to be defined in ways that prevent them from being redundant. (When the state of a character is predictable from the state of another character, the characters are redundant for the purposes of phylogenetic analysis; to use redundant characters amounts to counting the same character at least twice, doubling or multiplying its influence on the results.) Observations need to be interpreted in terms of these characters, and then these interpretations need to be inserted in the data matrix by hand. In our own practice, we have every once in a while caught ourselves making typographic errors, being momentarily confused about which state is 0 and which is 1 (because of faulty memory as well as conflicting conventions on how to assign such numbers—0 can for instance mean "absent" or "presumably plesiomorphic"), inserting the right value in the wrong column or line, and committing similar blunders; additionally, we have on occasion misinterpreted the descriptive literature and its illustrations (line drawings, but even photographs, can give misleading three-dimensional impressions), overlooked poorly known publications, had language barriers or conflicting terminologies prevent us from being sure if a published sentence said one thing or the opposite, or simply relied on the then current state of research that

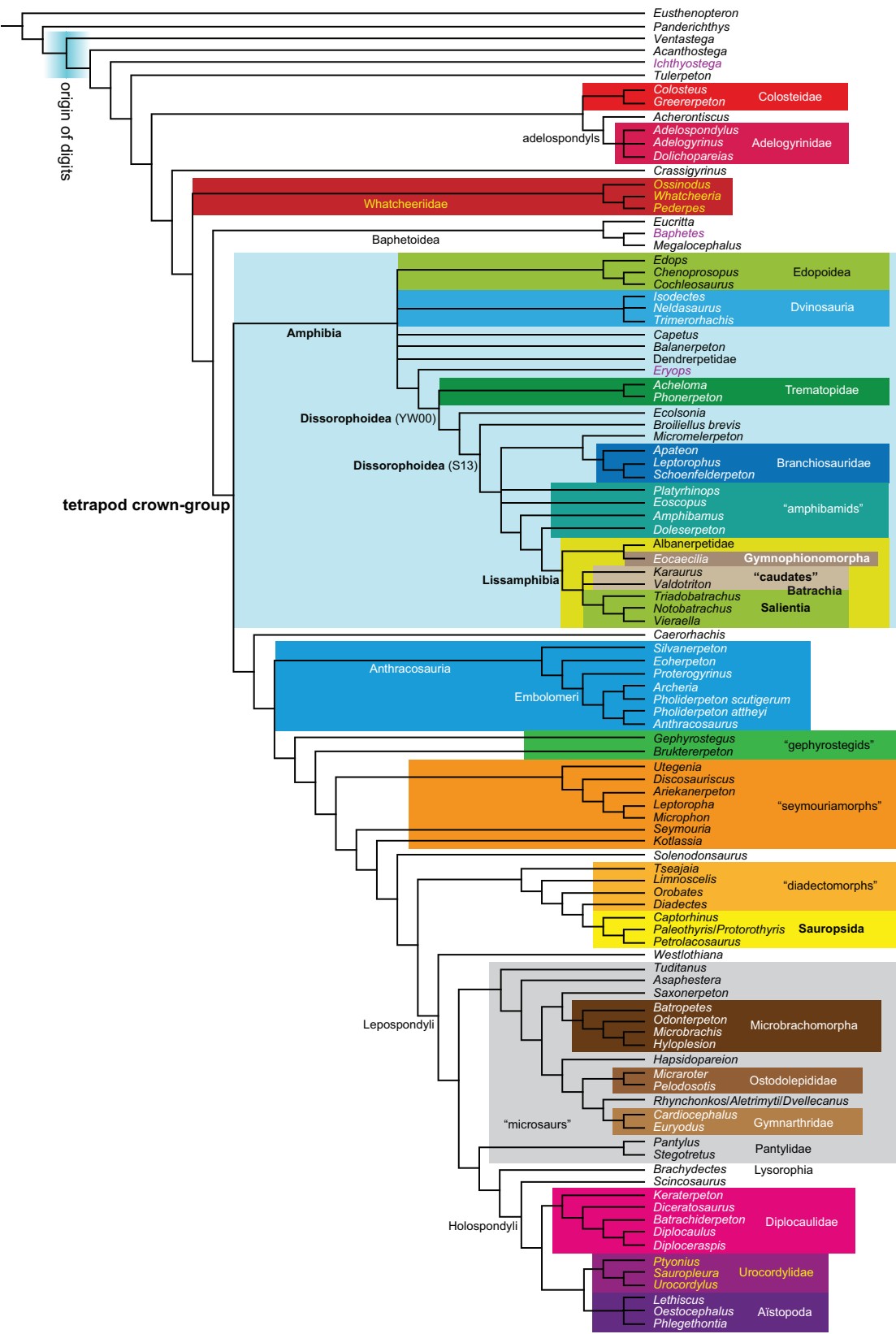

**Figure 1  Strict consensus of the MPTs (length: 1,621 steps including polymorphisms) found by RC07 and by our unconstrained reanalysis of their unchanged data matrix (Analysis O1; see Table 1).** The topology is identical to RC07: fig. 5, 6. Names of extant taxa are in boldface in this and

all following tree figures. In this figure and all following ones, Albanerpetidae and Dendrerpetidae are each a single OTU (called "Albanerpetontidae" and "*Dendrerpeton*" by RC07). The name Edopoidea (and, in some other figures, Eutemnospondyli) is placed under the assumption that *Mastodonsaurus* (not included in any of our analyses) would be closest to *Eryops* as found by Schoch (2013); "Dissorophoidea (S13)" is placed according to Schoch's (2013) definition, "Dissorophoidea (YW00)" is placed according to the definition by Yates & Warren (2000), and "Dissorophoidea (content)", only shown where different from Dissorophoidea (YW00), is the smallest clade that has the traditional contents of that taxon (trematopids, *Broiliellus*, amphibamids/branchiosaurids, *Micromelerpeton*). The origin of digits cannot be narrowed down to a single internode in this and several other figures. For easier orientation, *Ichthyostega*, *Baphetes* and *Eryops* are written in purple. The color scheme of the background boxes is consistent across all figures. Occasional abbreviations: Gymnophio., Gymnophionomorpha; Liss., Lissamphibia; micr. or microbr., microbrachomorphs; *Ph.*, *Pholiderpeton*. All "microsaurs" are underlain in the same shade of gray, but some (in some figures) are not labeled as such due to lack of space. 

was later overturned when the next publication came out. It stands to reason that these things also happen to other people.

Following the reevaluations by Marjanović & Laurin (2008, 2009: supplementary information) of the matrices by McGowan (2002) and Anderson et al. (2008a), those by Sigurdsen & Green (2011) of the matrices by Vallin & Laurin (2004), RC07 and Anderson et al. (2008a) and the one by Langer et al. (2017) of the matrix by Baron, Norman & Barrett (2017), we scrutinized the matrix of RC07 in the light of the following questions:

- Are there redundant characters or accidental misscores in the matrix of RC07?
- If there are any, and if we revise them, does that change the resulting trees?

These questions account for the bulk of the work we present here. All changes to the matrix, most of which are of these kinds, are documented and justified in App. S1, the commented character list, which comprises more than half of the total text of the present publication. The revised matrix is presented in human-readable form as App. S2, where the changes are highlighted in color, and in NEXUS format as Data S3.

We should stress that we did not make changes to the matrix in order to test whether they are sufficient for obtaining different MPTs from the ones found by RC07 (as a reviewer put it: what it takes to "break" their matrix). Neither did we restrict our changes to information Ruta & Coates could have known in 2007 (or 2006, when they submitted their manuscript); the context in which the matrix was made is not a subject of our study. Rather, we have tried to identify *all* redundant characters and *all* misscores (regardless of whatever their sources may be), deal with *all* of these potential problems to the best of our current abilities, perform new phylogenetic analyses on the revised matrix (Analyses R1–R3; see Table 1), and report how the resulting MPTs differ from the ones found by RC07 in lengths, topologies, indices and bootstrap support (Analysis B1). The question of how many additional steps are required to obtain different hypotheses of lissamphibian origins is tested by constrained analyses (R2, R3) as in RC07; these numbers are far lower than the total of our changes to the matrix.

In all likelihood, accidental misscores should be a good approximation to random noise. Such noise is expected to produce many weak false signals which cancel each other out instead of accumulating into a challenge to the true signal. However, when the true signal is weak to begin with (perhaps due to a character sample which is small enough to cause accidental sampling bias) and one or a few strong false signals are already present (due to large-scale evolutionary convergence or redundant characters), random noise

added to the true and false signals may change the balance from slightly in favor of the true signal to slightly in favor of a false signal—or indeed from one false signal to another, so that efforts to reduce the strength of the first false signal will not make the true signal stand out.

Our methods for identifying and attempting to deal with redundant characters—in some cases a hard problem on which we expect future advances—are explained below (Materials and Methods: Treatment of characters). This includes ontogeny-related characters: taxa known only from immature or paedomorphic individuals will predictably have "immature" states of many characters, with dramatic consequences for the resulting trees such as clustering of these taxa into spurious clades (*Wiens, Bonett & Chippindale, 2005*). Our approach to this difficult problem, modified from the recommendation of *Wiens, Bonett & Chippindale (2005*: 96*)*, was independently proposed by *Tykoski (2005)*, *Pawley (2006*: 206*)* and *Marjanović & Laurin (2008)*; it was explained in more detail by *Marjanović & Laurin (2013)* and is presented again below (Materials and Methods: Treatment of characters: "Ontogeny discombobulates phylogeny"). The changes made to the matrix for this reason are likewise documented and justified in App. S1; they are also marked in blue in App. S2 and counted in Data S4.

As mentioned, *Sigurdsen & Green (2011)* performed their own reevaluation of the matrix of RC07. That work, however, had a much more limited scope than ours (see *Marjanović & Laurin, 2013*, for discussion). We have incorporated most, though not all, of the changes to individual cells suggested in it (as discussed under the respective characters in App. S1). Unlike *Sigurdsen & Green (2011)*, we have not deleted characters of unclear value.

### Phylogeny of early limbed vertebrates

By total number of scores, the matrix of RC07 is the largest published one that concerns the phylogeny of early (roughly Paleozoic) limbed vertebrates other than amniotes. If we have come satisfactorily close to solving the problems presented above, our modified matrix should therefore be better suited to investigating the following questions, among others, than any other matrix published so far, even though it cannot treat all of them in sufficient depth. Compare Fig. 1:

- Are lissamphibians temnospondyls, "lepospondyls" or (diphyletically) both?
  - How strong is the support for each of these hypotheses?
- Are the albanerpetids lissamphibians? What are their closest relatives?
- Do the traditional diadectomorphs form a clade?
- Do the traditional "microsaurs" form a clade (including or excluding any lissamphibians)?
- Do the traditional "lepospondyls", or some of them, form a clade (including or excluding any lissamphibians)?
- Do the traditional seymouriamorphs form a clade?

- Are the traditional seymouriamorphs or any traditional "lepospondyls", especially "microsaurs", closer to Amniota?
- What can be said about temnospondyl phylogeny?
- Is Anthracosauria or Temnospondyli closer to Amniota?
- What are the phylogenetic positions of *Solenodonsaurus*, *Gephyrostegus*, *Bruktererpeton*, *Caerorhachis*, *Silvanerpeton* and *Tulerpeton* (all have been connected to anthracosaur origins at one point or another)?
- Are Adelogyrinidae and *Acherontiscus* "lepospondyls" or close to the colosteids?
- Is Colosteidae or Whatcheeriidae closer to the tetrapod crown-group?
- Is *Ichthyostega* or *Acanthostega* closer to the tetrapod crown-group?
- What happens to the above questions when taxa are added (Analyses R4–R6 and B2; see Table 1 and the "Phylogenetic background" section below)?
- The addition of taxa allows us to test further questions such as:
  - What is the phylogenetic position of Chroniosuchia and several other taxa?
  - Are the traditional diadectomorphs amniotes, or are they the closest relatives of Amniota?
  - Is *Casineria* close to amniote origins?

These questions are presented in more detail in the "Phylogenetic background" section below and reviewed in the Discussion (section "Phylogenetic relationships").

The addition of taxa required doubling the number of analyses of the revised matrix. To avoid another duplication, we did not add any characters; adding characters will be part of future work. However, we discuss a few characters—both inside and outside the present matrix—that have recently been connected to lissamphibian origins (Discussion: Characters: subsections other than the first and part of the second).

### The effects of different methods of analysis

- Does a Bayesian analysis of our revised matrix support a different tree than parsimony?

All of the analyses mentioned above used the non-parametric method somewhat misleadingly called "parsimony" or "maximum parsimony". For comparison, we also applied Bayesian inference to our revised matrix. The behavior of Bayesian inference under the conditions of this matrix are not well understood (Discussion: Bayesian inference and parsimony in comparison), and Bayesian analyses are time-consuming; we therefore ran only one analysis (under the same conditions as R4 and B2: enlarged taxon sample, no constraints) which we consider exploratory (Analysis EB; see Table 1). According to recent simulations, Bayesian inference has advantages over parsimony (*Wright & Hillis, 2014*; *O'Reilly et al., 2016*, *2018*; *Puttick et al., 2017*; but see *Simmons, 2012a*, *2012b*; *Brown et al., 2017*; *Goloboff, Torres & Arias, 2018*); in particular, it is much less sensitive to long-branch attraction, which may be a concern with some of the "weirder" taxa in our sample like adelospondyls, aïstopods or indeed lissamphibians.

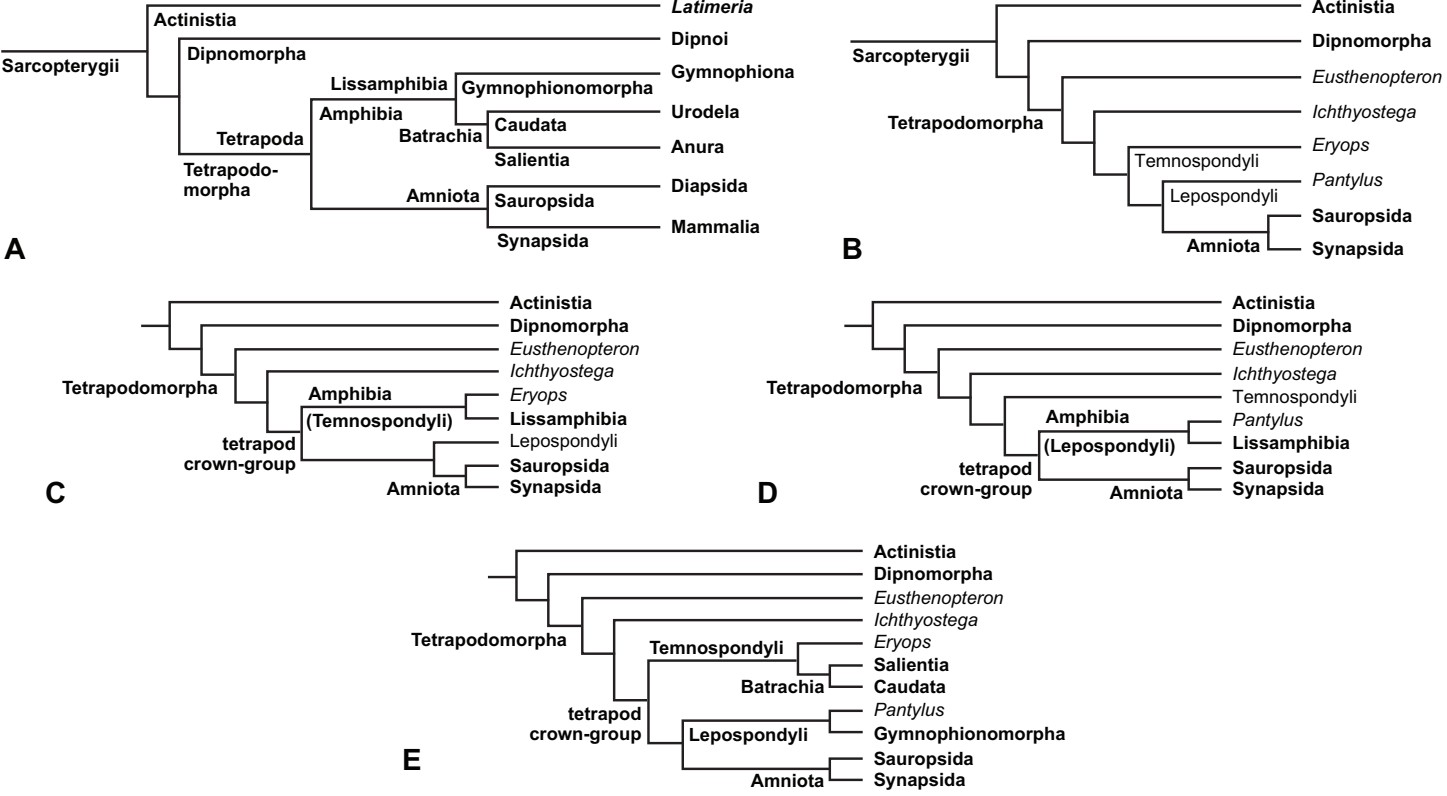

**Figure 2 Hypotheses on the origin(s) of the extant amphibians, and their compatibility with molecular and morphological data.** The names Amphibia and Lissamphibia do not apply in (E). (A) Consensus of all recent molecular studies (*Pyron, 2014*: supplementary file amph_shl.tre; *Irisarri et al., 2017*). (B) Part of the consensus of all recent morphological studies (e.g. our results, RC07 and references in both); modern amphibians not shown. (C) Extant amphibians added to (B) according to the "temnospondyl hypothesis"; note compatibility with (A). (D) "Lepospondyl hypothesis" mapped to (B); note compatibility with (A). (E) "Polyphyly hypothesis" mapped to (B); note lack of compatibility with (A). In earlier versions of the "polyphyly hypothesis", Caudata often lies next to Gymnophionomorpha instead of Salientia; in that case, the name Batrachia does not apply (or becomes a synonym of the tetrapod crown-group).

## Phylogenetic background

The early phylogeny of the limbed vertebrates contains a number of open questions on which there is either no consensus, or the existing consensus is weakly supported. Most famously, the origin of the modern amphibians (Lissamphibia and its possible member or sister-group Albanerpetidae—see below on that name) remains a vexing problem (Fig. 2). From the late 19th century to now, the modern amphibians have been considered temnospondyls by some (Fig. 2C—"temnospondyl hypothesis", abbreviated as TH below; most recently found by: RC07; *Sigurdsen & Green, 2011*; *Pardo, Small & Huttenlocker, 2017*: fig. S6; *Pardo et al., 2017*), lepospondyls by others (Fig. 2D—"lepospondyl hypothesis", abbreviated as LH below; *Vallin & Laurin, 2004*; *Pawley, 2006*: app. 16; *Marjanović & Laurin, 2008, 2009, 2013*), and polyphyletic by yet others (Fig. 2E—"polyphyly hypothesis", abbreviated as PH below; *Carroll, 2007*; *Huttenlocker et al., 2013*), with Salientia being nested among the temnospondyls, Gymnophionomorpha among the lepospondyls, and Caudata either in the lepospondyls (all early works, for example *Carroll & Holmes, 1980*) or in the temnospondyls (works published in the 21st century).

This particular question has far-reaching implications for the interpretation of most of our taxon sample. For example, the seymouriamorphs—considered close to amniote origins for most of the 20th century, though known to have gilled juveniles and possibly neotenes—indeed lie on the amniote stem under the TH (Fig. 1); from phylogenetic bracketing, it follows that the unfossilized parts of their anatomy and behavior were more amniote-like than found in lissamphibians and lay within the range of other crown-group tetrapods in the absence of fossil evidence to the contrary. Under the PH, the seymouriamorphs lie on the common stem of amniotes and caecilians to the exclusion of batrachians (salientians + caudates); the last common ancestor of the extant amphibians was thus the last common ancestor of all crown-group tetrapods, and those of its features retained by both batrachians and caecilians should be expected to have persisted in the seymouriamorphs unless there is evidence to the contrary. Under the LH, in contrast, the seymouriamorphs are (most likely, see below) stem-tetrapods, bracketed by extant tetrapods on only one side, and may have been less similar to extant tetrapods than the TH or the PH predict. Likewise, the diverse temnospondyls are stem-amphibians under the TH (Figs. 1 and 2C), bracketed by lissamphibians and amniotes among extant taxa; under the PH (Fig. 2E), they belong to the batrachian stem, bracketed by extant amphibians on both sides, so that a great many features found among extant amphibians should be expected to have been shared by temnospondyls (validating a large amount of existing literature and artwork); under the LH, however (Fig. 2D), they are not crown-group tetrapods at all, but lie fairly far rootward on their stem and should be expected to be unlike anything alive today in an unknown but large number of aspects. (The new version of the TH by *Pardo, Small & Huttenlocker, 2017*: fig. 2, S7 but not S6, makes much the same predictions as the PH in this regard because it places most temnospondyls in the amphibian crown group.)

Among other more or less open questions are the phylogenies of Temnospondyli (*Pawley, 2006*; *Ruta, 2009*; *Dilkes, 2015a*; *Pardo, Small & Huttenlocker, 2017*), "Lepospondyli" (RC07; *Huttenlocker et al., 2013*; *Marjanović & Laurin, 2013*; *Pardo, Small & Huttenlocker, 2017*: fig. S6) and Devonian limbed vertebrates (*Ahlberg & Clack, 1998*; *Clack et al., 2012a*; *Pardo et al., 2017*), as well as the relative positions of Anthracosauria and Temnospondyli (*Laurin & Reisz, 1999*; RC07; *Pardo et al., 2017*) and the position of Diadectomorpha inside or next to Amniota (*Berman, Sumida & Lombard, 1992*; *Berman, 2013*), not to mention the positions of confusing (*Andrews & Carroll, 1991*; *Smithson et al., 1994*; *Clack, 2001*; *Ruta, Milner & Coates, 2002*; *Vallin & Laurin, 2004*; RC07) or fragmentary Carboniferous taxa (*Smithson, 1980*; *Paton, Smithson & Clack, 1999*; *Bolt & Lombard, 2006*; *Clack et al., 2012b*, *2016*; *Sookias, Böhmer & Clack, 2014*).

Molecular data are of limited use for tackling these questions: of all tetrapodomorphs (tetrapods and everything closer to them than to lungfish), only frogs, salamanders, caecilians and amniotes still have living members (Fig. 2). Thus, molecular data cannot test, say, whether Amniota is closer to Anthracosauria or to Temnospondyli, or if the many disparate taxa classified as "Lepospondyli" constitute a clade, a grade, or a wastebasket. When it comes to the origins of the extant amphibians, molecular data can

distinguish the PH (Fig. 2E) from the other hypotheses, because the PH predicts that the extant amphibians are paraphyletic with respect to Amniota, while the other two hypotheses of course predict monophyly under recent conceptions of the affinities of temnospondyls, lepospondyls and amniotes (Fig. 2B; *Laurin, 2002*; see also *Marjanović & Laurin, 2007, 2013*); but molecular data cannot distinguish the two monophyly hypotheses, because too many relevant taxa have been extinct for too long. Finding the extant amphibians monophyletic with respect to Amniota (Fig. 2A), analyses of molecular data support both monophyly hypotheses equally (Figs. 2A, 2C and 2D); only paleontological data can distinguish them. Existing analyses of paleontological data, however, disagree greatly on this question (see above; Figs. 2C–2E) as well as on others. To some extent, no doubt, this is due to the many differences in their taxon and character samples. However, problems in datasets of the kinds presented above (Aims: Accuracy of the matrix of RC07) constitute another possible reason. When such misscores and miscodings are removed from matrices—without, as far as possible, changing the taxon or character sample—do the results change?

The largest published morphological data matrix that has been applied to the problems of the phylogeny of limbed vertebrates in general and the origin of the modern amphibians in particular is that by RC07; it supported the TH and is often cited for this result. Here, we reevaluate this matrix in order to test, and explain within the limitations of the dataset, to what degree this result—and others that together constitute the consensus tree of RC07 (fig. 5, 6; our Fig. 1)—continues to follow from their dataset after a thorough effort to improve the accuracy of the scoring and reduce character redundancy has been carried out to the best of our current knowledge.

Naturally, this effort will not suffice to solve the question of lissamphibian origins or any other of the many controversies in the phylogeny of early limbed vertebrates. A quick look at matrices such as those of *Ruta (2009)*, *Sigurdsen & Green (2011)*, *Maddin, Jenkins & Anderson (2012)*, *Sookias, Böhmer & Clack (2014)*, *Dilkes (2015a)*, *Clack et al. (2016)*, *Pardo, Small & Huttenlocker (2017)* or *Pardo et al. (2017)*, or at reinvestigations of anatomy such as those of *Witzmann (2007, 2011, 2013)*, *Bolt & Lombard (2010)*, *Mondéjar-Fernández, Clément & Sanchez (2014)*, *Dilkes (2015a)* or *Pardo et al. (2017)* and references therein will demonstrate that many characters which are known to carry phylogenetic signal for the present taxon sample remain absent from this matrix; this even includes some of the very few (41) characters used by *McGowan (2002)*. Adding them (as well as yet more taxa) will be part of future work, and may well lead to trees with a different topology. However, we think the present work forms a necessary step toward solving any of those problems. Further progress may come from larger matrices—if and only if the increase in the number of cells is not accompanied by a proportional decrease in the care that goes into scoring them (*Simões et al., 2017*).

Originally we did not intend to add any taxa to the matrix, just as we have not added any characters. However, soon after the work of RC07 was published, the intriguing amphibamid temnospondyl *Gerobatrachus* was described and was argued to add strong support to the PH (*Anderson et al., 2008a*). Phylogenetic analyses that included *Gerobatrachus* in different versions of the same matrix have supported the PH (*Anderson*

*et al., 2008a*), the LH (*Marjanović & Laurin, 2009*), or more recently the TH (*Maddin & Anderson, 2012*; *Maddin, Jenkins & Anderson, 2012*); the latter work even found *Gerobatrachus* to be nested within Lissamphibia (partially replicated by *Pardo, Small & Huttenlocker, 2017*: fig. S6B, S7B; trivially replicated by *Pardo, Small & Huttenlocker, 2017*: fig. 2, S7A, where most temnospondyls are crown-group amphibians; not replicated by *Pardo, Small & Huttenlocker, 2017*: fig. S6A or by *Pardo et al., 2017*, where *Gerobatrachus* and Lissamphibia are sister-groups as in *Maddin & Anderson, 2012*). Clearly, *Gerobatrachus* is too important to be left out. Following the examples of *Marjanović & Laurin (2008)* and *Langer et al. (2017)*, we have therefore performed a separate series of analyses (R4–R6, B2) for which we added *Gerobatrachus* to the matrix; at that opportunity, for the same series of additional analyses, we added a further 55 OTUs as detailed and justified in Materials and Methods: OTUs added for a separate set of analyses, bringing the total from 102 to 158.

In the Discussion section, we explore the relationships of the sampled taxa and the distributions of certain character states in the light of our findings and other recent publications. By presenting the current areas of uncertainty (some expected, some unexpected), we hope to highlight opportunities for future research.

## NOMENCLATURE

A few remarks are necessary to explain our use of certain terms. The list of abbreviations used in the text is located between the Conclusions and the Acknowledgments. Abbreviations not listed there that consist of at least three letters, at least one space and a number designate characters, following the practice of RC07; see App. S1 and below (Materials and Methods: Treatment of characters: Character interdependence; redundant characters).

### Taxonomic nomenclature

Without mentioning the fact that they were doing so, *Averianov & Sues (2012*: 466*)* corrected the spelling of Albanerpetontidae *Fox & Naylor, 1982*, to Albanerpetidae. We follow this in analogy to several corrections by *Martín, Alonzo-Zarazaga & Sanchiz (2012)* as discussed by *Marjanović & Laurin (2014*: 543*)*, who did not know of *Averianov & Sues (2012)* and therefore incorrectly claimed that "no other spelling [than Albanerpetontidae] has ever been used", as well as in analogy to several further corrections by *Schoch & Milner (2014)*. Assuming that this correction is a "justified emendation," the name Albanerpetidae must continue to be attributed to *Fox & Naylor, 1982* (*International Commission on Zoological Nomenclature, 1999*: articles 19.2, 29).

Likewise, the spelling Hapsidopareiontidae must be corrected to Hapsidopareiidae Daly, 1973: there is no basis for -ont- in Homeric Greek παρήϊον ("cheek") (*Perseus Digital Library, accessed 5 November 2017*). This name has been used so rarely that the question of common usage does not arise.

The name *Diploradus* Clack & Smithson in *Clack et al. (2016)*, was explained as follows (*Clack et al., 2016*: 3*)*: "Genus from *diplo* (Greek) 'double' and *radus* (Greek) 'row' referring to the double coronoid tooth row." We have not been able to find a word

similar to "radus" in the Greek or for that matter Latin dictionaries in the Perseus Digital Library ("Dictionary Entry Lookup" and "English-to-[Language] lookup" in the "General Search Tools" http://www.perseus.tufts.edu/hopper/search accessed 5 November 2017). The closest in form and meaning appears to be Latin *radius*, originally meaning "staff, rod". (The one language we have found where *rad* means "row" is Slovak.) However, this does not affect the validity of the name.

## Formal and informal phylogenetic nomenclature

Because of the length of this paper and because the International Code of Phylogenetic Nomenclature ("PhyloCode") has not yet taken effect, we refrain from proposing new clade names or definitions. Having, however, noticed that the names Adelogyrinidae and Adelospondyli currently refer to indistinguishable taxa, we follow *Ruta, Coates & Quicke (2003*, especially p. 284: "*Acherontiscus* is an adelospondyl"*)*, RC07: 81 (but not fig. 5) and *Coates, Ruta & Friedman (2008*: fig. 2: "Adelospondyli"*)* in informally referring to a clade composed of *Acherontiscus* and Adelogyrinidae as "adelospondyls" for brevity, always excluding the historically included lysorophians.

As in a previous paper (*Marjanović & Laurin, 2013*), where we failed to make this explicit, we use "modern amphibians" for Lissamphibia and its possible member or sister-group Albanerpetidae.

Several names for temnospondyl clades, including Temnospondyli itself, have been given very different definitions by *Yates & Warren (2000)* and *Schoch (2013)*. We have generally applied the definitions by *Yates & Warren (2000)*. The important exceptions are Dissorophoidea and Stereospondylomorpha, which we use in the text for clades containing all those OTUs that have traditionally been regarded as members of clades with these names and exclude most or all OTUs that have traditionally not been regarded as members. For Dissorophoidea, we indicate both definitions and our usage in the tree figures (our usage usually, but not always, coincides with the definition by *Yates & Warren, 2000*).

## Anatomical nomenclature

Consistently or nearly so, RC07 (as also *Pardo, Szostakiwskyj & Anderson, 2015*) exchanged the terms "skull roof" and "skull table". The former refers, in most other works, to the dermatocranium except its ventral side (the palate); the latter refers to the dorsal side of the skull roof, often demarcated from the lateral sides by distinct edges.

RC07 (and *Ruta, Coates & Quicke, 2003*) also almost consistently used "mesial" when they were aiming at "medial". This appears to be a common British practice; internationally, however, it is "medial" that means "toward the sagittal plane at a right angle to it", while "mesial" belongs to toothrow nomenclature and means "toward the symphysis along the curvature of the jaw", the opposite of the poorly chosen term "distal." Many instances of "mesial" apply to the lower jaw and actually mean "lingual", "proximal at a 90° angle to the curvature of the jaw", the opposite of "labial"; only at the symphysis is mesial medial.

We have tried to rectify these issues in the annotated character list (App. S1). However, we use the pairs "anterior–posterior" and "cranial/rostral—caudal" interchangeably (despite a preference for the latter) because there is, in our taxon sample, no danger of confusion.

Our ambiguous usage of "orbit" (*Marjanović & Laurin, 2008*, *2013*) confused *Pardo & Anderson (2016)*, who claimed that we had first thought that *Brachydectes* had very large eyes which filled its orbitotemporal fenestrae (ascribed to our 2008 paper) and that we had later amended this view (ascribed to 2013a); in reality, we explicitly stated (2008: 194–195) that we considered the "orbits (or 'orbitotemporal fenestrae' as they are sometimes called in salientians and caudates)" to have "presumably" contained jaw muscles in *Brachydectes*, not just the eyeballs, and we implied no changes in the later paper (*Marjanović & Laurin, 2013*: 239, 241). In the present work we consistently speak of "orbitotemporal fenestrae" to mean skull openings that appear to have contained the eyes as well as jaw muscles, regardless of the inferred homologies of these openings.

## MATERIALS AND METHODS

The data matrices were edited in successive versions of Mesquite up to 3.31 (*Maddison & Maddison, 2017*); this program was also used to display and visually compare trees and to optimize characters on them.

### Treatment of characters

#### *Character interdependence; redundant characters*

Characters in a data matrix for phylogenetic analysis are interdependent when a state of a character (other than "unknown") is predictable—without prior knowledge of phylogeny—from the state of another character. Because phylogenetic analysis operates on the assumption that all characters are independent of each other, the presence of interdependent characters in a matrix amounts to counting the same apomorphy at least twice, which can distort the resulting tree topology and will almost inevitably distort at least some of its support values. While this fact seems to be universally acknowledged in principle, we find (*Marjanović & Laurin, 2008*, *2009*, *2013*; and below) that it is underappreciated in practice.

Different kinds of character interdependence require different amounts of effort to detect. *O'Keefe & Wagner (2001*: 657; and references therein*)* distinguished "logical correlations among characters" from "[b]iological correlations"; *Pardo (2014*: 52–60*)* distinguished four kinds of interdependence. We call Pardo's first three kinds, which include logical interdependence, "redundancy" and biological interdependence "correlation" hereinafter.

It can be very difficult to determine whether characters are correlated; studies of development genetics are sometimes, perhaps often, required (*Kangas et al., 2004*; *Harjunmaa et al., 2014*). We expect, therefore, that all of our best efforts will be unable to completely eliminate character interdependence from any morphological matrix. However, many cases of redundant characters are much more obvious; and although RC07 noticed and removed several cases from the preceding version (*Ruta, Coates &*

*Quicke, 2003*), we present a considerable number of additional cases in App. S1. We have merged each pair (or multiple) of redundant characters that we identified into a single character.

To make our mergers more transparent, we have created abbreviations for merged characters from those of all their constituents: for example, PREMAX 1-2-3 (ch. 1) is built from the three characters PREMAX 1, PREMAX 2 and PREMAX 3 of RC07, and MAX5/PAL5 (ch. 22) consists of MAX 5 and PAL 5 of RC07. The extreme cases are VOM 5-10/PTE 10-12-18/INT VAC 1 (ch. 105), assembled from the six characters VOM 5, VOM 10, PTE 10, PTE 12, PTE 18 and INT VAC 1 of RC07, and EXOCC 2-3-4-5/ BASOCC 1-5 (ch. 134).

### Ordering of multistate characters

RC07 (p. 78), like *Ruta, Coates & Quicke (2003*: 271*)*, followed the widespread practice (*Sigurdsen & Green, 2011*; *Schoch, 2013*; stated but unexplained preference of *Simões et al., 2017*: 211; *Pardo et al., 2017*; *Pardo, Small & Huttenlocker, 2017*) of treating all multistate characters as unordered. Following another widespread practice, they did not spell out any reasons for this decision. Presumably they adhered to the common assumption (already identified and discussed by *Slowinski, 1993*) that ordering a character is to make an assumption, while "leaving" it unordered means not to make an assumption—which is incorrect (*Slowinski, 1993*). In particular, potentially continuous (clinal) characters should be ordered, because the basic assumption behind ordering—that it is easier to change from any state to a similar state than to a less similar one—has already been used to partition the observed spread of data into discrete states in the first place; it would be incoherent to reject this assumption in one place but not the other (*Slowinski, 1993*; *Wiens, 2001*). This also holds for certain meristic characters (*Wiens, 2001*). As advocated by *Slowinski (1993)*, *Wiens (2001)* and *Baron, Norman & Barrett (2017*: supplementary information: 4–9*)*, we have ordered many, but not all, multistate characters; see App. S1 for discussion of each case. Two characters (ch. 32, 134: PAR 2/POSFRO 3/INTEMP 1/SUTEMP 1, EXOCC 2-3-4-5/BASOCC 1-5) have part of their state range ordered (*Slowinski, 1993*: fig. 1a, d); this is accomplished by creating stepmatrices (App. 1: App.-Tables 2, 4). We have marked these decisions in the name of each multistate character by adding "(ordered)", "(unordered)" or "(stepmatrix)" to its end.

The consequences of ordering potentially continuous characters are largely unpredictable. Empirically, ordering such characters can reveal additional signal and thus increase the resolution of the consensus tree (*Slowinski, 1993*; *Fröbisch & Schoch, 2009a*; *Grand et al., 2013*; *Simões et al., 2017*: fig. 2b, 3b, 4; *Baron, Norman & Barrett, 2017*: fig. 1, extended data fig. 4); on the other hand, and even at the same time, it can also reveal previously hidden character conflict and thus decrease the resolution, showing that the original resolution was not supported by the data (*Slowinski, 1993*; *Marjanović & Laurin, 2008*; *Baron, Norman & Barrett, 2017*: fig. 1, extended data fig. 4). Both of these results are congruent with the finding of the simulation studies by *Grand et al. (2013)* and *Rineau et al. (2015)* that ordering clinal characters decreases the rate of artefactual resolution and increases the power to detect real clades.

## Continuous characters

Due to the limits of available computation power, we have not been able to analyze continuous characters "as such" (as recommended by *Simões et al., 2017*; *Brocklehurst, Romano & Fröbisch, 2016*; and references therein), but have broken them up into a small number of discrete states. Such characters were ordered if multistate (see above). Where RC07 had defined reproducible state limits that did not render the characters parsimony-uninformative (see below), we have not changed them; in the cases of PREMAX 7 and SKU TAB 1 (our ch. 3 and 95), which we have wholly recoded (see below: "Deleted, recoded and split characters"), we divided the observed range into several equal parts at round numbers because there are no gaps in the distribution. This is an obvious opportunity for future improvement. See *Brocklehurst, Romano & Fröbisch (2016)* for further discussion.

## Inapplicable characters and mergers

> Perhaps the eventual solution will be to write new algorithms for computer programs that will allow the characters to be coded independently but that will consider interactions between characters and count steps in some characters only on those portions of the tree on which they are applicable.
>
> – *Maddison (1993*: 580)

> Character-taxon matrices and their accompanying character lists should be viewed as formatted data, and not just a table of observations. That is, they should be constructed with an understanding of how that information will be interpreted by the algorithm that is receiving them. For many multistate characters, authors should consider how character state information is (or is not) distributed to other transformation series. The problem with "0" being used as a catch-all for anything that simply "isn't 1" should be borne in mind when using binary characters (see Discussion below).
>
> – *Brazeau (2011*: 494)

> *Maddison (1993)* concluded that addressing this problem would require modification of phylogenetic software; 25 years later, there are still few signs of progress on this problem.
>
> – *Brazeau, Guillerme & Smith (2017*: 5)

It is a common situation that a character is inapplicable to part of a taxon sample: when, say, the ectopterygoid bone in the palate is absent, it does not have a shape, and it is not toothed or toothless. *Strong & Lipscomb (1999)* reviewed the methods for dealing with such cases:

- Reductive coding: inapplicable scores are treated as missing data, using the symbols "?" or "-". (The latter designates a gap in a molecular sequence, which is treated either as missing data—the default setting in available software—or as a fifth nucleotide/21st amino acid—which is absence coding, see below.)
- Composite coding: characters that are inapplicable to any OTU are merged with the characters they depend on, producing multistate characters. For instance,

"ectopterygoid toothed (0); toothless (1)" might be merged with "ectopterygoid present (0); absent (1)" as the unordered multistate character "ectopterygoid toothed (0); toothless (1); absent (2)".

- Non-additive binary coding: the presence or absence of each character state is treated as a binary character of its own that can be scored for all OTUs. For example, "toothed ectopterygoid: present (0); absent (1)" and "toothless ectopterygoid: present (0); absent (1)" can be scored for all OTUs—taxa that lack an ectopterygoid have state 1 of both of these characters.

- Absence coding: inapplicability is coded as an additional state of each concerned character.

As *Strong & Lipscomb (1999)* pointed out, absence coding creates redundant characters by counting each condition that makes characters inapplicable several times, once for each character that it makes inapplicable. Absence coding must therefore be avoided.

Non-additive binary coding runs the same risk. In the example given above, an OTU cannot have state 0 of both characters: having state 0 of one unfailingly predicts state 1 of the other. The characters are thus redundant. It also treats non-homologous conditions as the same state, which runs the risk of creating spurious apomorphies which may increase artefactual resolution (*Brazeau, 2011*, and references therein); *Strong & Lipscomb (1999*: 368–369*)* went so far as to state that "non-additive binary coding denies homology and the hierarchical relationships between states. The result are cladograms and character interpretations that are absurd and inaccurate representations of our observations." *Brazeau (2011*: 495*)* further performed a reductio ad absurdum by asking what would happen if molecular data were coded as, for example, "adenine at site 121: absent (0); present (1)". where state 0 would treat guanine, cytosine, thymine, and the absence of site 121 (a deletion) as identical and primarily homologous for no defensible reason.

Composite coding eliminates the redundancy that is sometimes created by non-additive binary coding. Furthermore, it has the advantage over reductive coding that it prevents contradictory optimization of ancestors. With reductive coding, an ancestor may be optimized as having a toothed ectopterygoid and at the same time as having no ectopterygoid at all (*Brazeau, Guillerme & Smith, 2017*, and references therein); composite coding makes this impossible and may thus make arguably nonsensical trees less parsimonious. However, composite coding is not always feasible to its full extent.
In our example, if the character that represents the presence of the ectopterygoid is merged with the character that describes its dentition, what is to become of size and shape? Merging all of the affected characters would often yield a huge multistate character that would, in many cases, require a complex stepmatrix and be difficult to interpret in evolutionary terms (as already noted by *Maddison, 1993*), not to mention the adverse effects of stepmatrices on calculation times (D. Marjanović, pers. obs. 2008–2017).

RC07 used reductive, composite and non-additive binary coding in different cases. The numerous occurrences of non-additive binary coding may be a side effect of taking characters from diagnoses and synapomorphy lists (*Bardin, Rouget & Cecca, 2014*), where they are usually presented in the form of a single (presumedly derived) state, while

the other states are not mentioned. We have greatly reduced the amount of non-additive binary coding. There are a few cases we have left for the time being; these are cases where the effort needed to rectify the situation would probably be out of proportion to the gain in phylogenetic signal. Examples are PREMAX 4 (ch. 2 in App. S1), "Premaxilla with flat, expanded anteromedial dorsal surface and elongated along its lateral margin but not along its medial margin, when observed in dorsal aspect: absent (0); present (1)", PASPHE 11 (ch. 143), "Basipterygoid processes of the basisphenoid shaped like anterolaterally directed stalks, subtriangular to rectangular in ventral view and projecting anterior to the insertion of the cultriform process: absent (0); present (1)", TIB 6 (ch. 235), "Outline of tibia medial margin shaped like a distinct, subsemicircular embayment contributing to interepipodial space and the diameter of which is less than one-third of bone length: absent (0); present (1)", and TRU VER 30 (ch. 275), "Transverse processes stout and abbreviated, the length of which is less than 30% of neural arch height: absent (0); present (1)". We would need to survey the range of morphologies and in some cases their considerable ontogenetic transformations—see below—in detail in order to determine how many characters, let alone how many states, should be distinguished within state 0 of each of these. All these characters should, however, be reinvestigated in the future.

We have slightly redefined some characters to avoid predictable scores. An example is PAR 1, our ch. 31. In RC07 it was called "Parietal/tabular suture: absent (0); present (1)" (their ch. 38). When the parietal and the tabular are present but the supratemporal is absent, the parietal and the tabular inevitably touch, so the score of 1 is wholly predictable from the scores of the characters that code the presence/absence of the supratemporal (our ch. 32; ch. 63 of RC07) and the tabular (our ch. 53; ch. 67 of RC07); yet, RC07 scored 1 in almost all these cases. Conversely, of the nine OTUs which lack not only the supratemporal but also the tabular, eight were scored as unknown, but one (Albanerpetidae) was scored 0: lacking a tabular, it cannot have a parietal-tabular suture. If this is not a typographic or similar error, perhaps state 0 was in this case understood as every condition that is not state 1 (non-additive binary coding). We have made state 0 explicit in redefining the character as: "Supratemporal/postparietal suture (0); parietal/tabular suture (1)." This is reductive coding: if any of these four bones is absent, the character is inapplicable (which we spell out in App. S1 for this and all comparable characters). The scores we changed to unknown because our redefinition makes them unambiguously inapplicable—Albanerpetidae, all "microsaurs" except *Odonterpeton* (which lacks tabulars and was already scored as unknown), *Brachydectes*, *Scincosaurus*, all diplocaulids—are marked in green in App. S2 (and counted as such in Data S4), as are all scores that involve the redefinition, or possible redefinition, of a state of any character (see below: Modifications to individual cells).

Others we have simply interpreted as reductive instead of as non-additive binary. An example is POSORB 5, our ch. 64 (ch. 81 of RC07). Its name remains unchanged: "Postorbital/tabular suture: absent (0); present (1)." All OTUs scored in RC07 as having state 0 indeed lack a contact between the postorbital and the tabular; almost all of

them, however, have a supratemporal, and whenever the supratemporal is present, it lies between the other two bones and prevents them from reaching each other. No cases are known in our taxon sample, and no cases are known to us outside our taxon sample, where the postorbital or the tabular would reach around the supratemporal and separate it from the parietal or the squamosal. We consider state 0 predictable from the presence of the supratemporal (our ch. 32; ch. 63 of RC07), have changed it to unknown in all OTUs that have a supratemporal, mentioned this in App. S1, marked these changes in green in App. S2 and counted them accordingly in Data S4.

On a few occasions we have changed massively non-binary to composite coding. RC07 used, for example, the following characters (p. 110, italics omitted):

333. DIG 1. Digits: absent (0); present (1).

334. DIG 2. Manus with no more than four digits: absent (0); present (1).

335. DIG 3. Manus with no more than five digits: absent (0); present (1).

336. DIG 4. Manus with no more than three digits: absent (0); present (1).

Despite having "no more than" in their names, DIG 2–DIG 4 were scored in RC07 as if they meant "exactly four fingers per hand: no (0); yes (1)", "exactly five" and so on: each OTU scored as 1 for any of these three characters was scored as 0 for both of the others. In other words, state 1 of any of these three unfailingly predicted state 0 of the other two—and also state 1 of DIG 1. We have therefore merged DIG 2–DIG 4 and most of DIG 1 as follows (App. S1):

276. DIG 1-2-3-4: "Independent radials" (0); polydactyly (1); pentadactyly (2); tetradactyl forelimb (3); tridactyl forelimb (4) (ordered).

The term "independent radials" refers to Johanson et al. (2007). The complete absence of extremities, previously part of DIG 1(0), is now part of the limb-reduction/body-elongation character:

219. HUM 18/DIG 1: Forelimb absent (0); humerus present, length smaller (1) or greater (2) than combined length of two and a half mid-trunk vertebrae (ordered).

In a few other cases we have changed fully reductive to partially composite coding. This includes the following characters of RC07:

140. ECT 1. Separately ossified ectopterygoid: present (0); absent (1).

141. ECT 2. Ectopterygoid with (0) or without (1) fangs [ . . . ].

142. ECT 3. Ectopterygoid without (0) or with (1) small teeth (denticles) [ . . . ].

143. ECT 4. Ectopterygoid longer than/as long as (0) or shorter than (1) palatine.

144. ECT 5. Ectopterygoid with (0) or without (1) row of teeth (3+) [ . . . ].

145. ECT 6. Ectopterygoid/maxilla contact: present (0); absent (1).

146. ECT 7. Ectopterygoid narrowly wedged between palatine and pterygoid: no (0); yes (1).

The presence (ECT 1) and size of the ectopterygoid (ECT 4) are now a single character, to which we have added an additional state distinction (ultimately from *McGowan, 2002*):

115. ECT 1–4: Ectopterygoid at least as long as palatine (0); at least about a third as long as but shorter than palatine (1); at most about a third as long as palatine (2); absent (3) (ordered).

The interpretation of ECT 1 and ECT 4 as parts of a potentially continuous character fits the observation that taxa with very small ectopterygoids tend to be the closest relatives of those with no ectopterygoids and nested among taxa with middle-sized ones: absence, in this particular case, seems to be in effect a length of zero (as forelimb absence is a humerus length of zero in HUM 18/DIG 1). ECT 2, ECT 3 and ECT 5, however, are unaffected; they continue to be scored as unknown for OTUs that lack ectopterygoids. Restudy of the literature and a specimen has shown that states ECT 6(1) and ECT 7(1) can only be scored for at most one OTU each (*App. S1*); this makes the characters ECT 6 and ECT 7 parsimony-uninformative, so we have deleted them (see below).

Too recently for us to use, *Brazeau, Guillerme & Smith (2017)* published a new approach to dealing with reductively coded inapplicability in phylogenetics software. We are looking forward to further developments of its implementation. However, we strongly disagree that so-called "neomorphic characters" should be scored as having the presumedly plesiomorphic state when they are inapplicable. This requires identifying the plesiomorphic state in advance, which increases the danger that the phylogenetic analysis will conform to one's preconceptions just as much as an all-zero outgroup would. It is also much less easy than *Brazeau, Guillerme & Smith (2017*: 23*)* implied when they stated that in their analysis "every inapplicable token in each neomorphic character was replaced with the token corresponding to the presumed non-derived condition (typically 'absent')"—for example, our matrix contains many characters for the presence or absence of bones that are, in our taxon sample, plesiomorphically present and are apomorphically lost several times, while different taxon samples (e.g. vertebrates generally, or actinopterygians) would support the opposite polarization or none at all. Further, this method creates redundancy just like absence coding does: in the example by *Brazeau, Guillerme & Smith (2017*: table 1*)*, absence of the tail unfailingly predicts absence of eyespots on the tail.

### *"Ontogeny discombobulates phylogeny" (Wiens, Bonett & Chippindale, 2005)*

Heterochrony can result in misleading scores if morphologically immature (juvenile or paedomorphic) individuals are scored at face value, which can result in large-scale character correlation that can strongly distort phylogenetic trees (*Wiens, Bonett & Chippindale, 2005*). We have tried to deal with this problem as described by *Marjanović & Laurin (2008)* and in more detail by *Marjanović & Laurin (2013)*, modified from the recommendation of *Wiens, Bonett & Chippindale (2005*: 96*)* and independently proposed by *Tykoski (2005*: 276*)*: observed states are treated as unknown if they are restricted to immature stages in non-paedomorphic close relatives of the OTU in question. Thus, OTUs

known from growth series including skeletally mature individuals are therefore scored as only having the most mature state of ontogeny-affected characters instead of as polymorphic; OTUs known only from apparently morphologically immature individuals are scored as having the most mature observed state or any more mature one, as partial uncertainty (or complete uncertainty if the only observed state is the least mature one the character has to offer), instead of just the most mature observed state. *Pawley (2006*: 206*)* appears to have used the same approach, explaining for binary characters: "If it was possible that a derived [ . . . ] characteristic may be absent in a particular specimen due to the morphogenetic immaturity of the specimen, then the character state was coded "?", to avoid confusing morphogenetic immaturity with the plesiomorphic state."

The ontogeny of most of our OTUs and their closest relatives is insufficiently well known; despite this, and in spite of the additional complications discussed by *Marjanović & Laurin (2013)*, we think that this approach offers the greatest chance to escape character correlation caused by heterochrony. Each of the rather few cases (79 in the taxon sample of RC07, 18 in the taxa we added) is discussed in App. S1 and marked in blue in App. S2; Data S4 additionally lists our scores of deleted characters that would be marked in blue (three in the taxon sample of RC07, one in an OTU we added).

### Deleted, recoded and split characters

Our matrix has only 277 characters, a strong decrease from the 339 of RC07. For the most part, this is due to our mergers of redundant characters and does not entail a loss of information (see above; detailed in App. S1). The characters IFN 1, ILI 10, ISC 1, DOR FIN 1 and BAS SCU 1, however, were parsimony-uninformative in the matrix of RC07 and remain so after our corrections (all of them documented and justified in App. S1); we have deleted them rather than carrying these currently irrelevant characters along and inflating the character count (see *Marjanović & Laurin, 2013*, for discussion). The characters PREFRO 6, PREFRO 9, LAC 6, NOS 1, VOM 11, PAL 6, ECT 6, ECT 7, BASOCC 6 and DEN 1 were parsimony-informative as scored by RC07, but this is no longer the case after our corrections (likewise documented and justified in App. S1), so we have deleted them as well. (For LAC 6 and VOM 11, one of the two states turns out not to occur in the original taxon sample at all, and only once or never in the expanded taxon sample.) Conversely, the character ANG 3 was parsimony-uninformative in RC07, but is parsimony-informative in our matrix even for the original taxon sample (see App. S1); we have kept it (though merged it with ANG 2, now ch. 161).

Characters INT FEN 1 and TEETH 3 were composites of two independent characters each; we have split INT FEN 1 into the redefined INT FEN 1 (ch. 84) and the new MED ROS 1 (ch. 85), and TEETH 3 into the redefined TEETH 3 (ch. 183) and the new TEETH 10 (ch. 190). Similarly, we have reversed the merger of PIN FOR 1 and PIN FOR 2 of *Ruta, Coates & Quicke (2003)*, our ch. 91 and 92, into PIN FOR 2 of RC07. We have also partitioned several characters into more states than before.

Character PREMAX 7, a ratio of two measurements, was parsimony-uninformative as defined (one of the two original states is limited to an OTU that was scored as unknown

by RC07), but not as described or scored; we have measured all OTUs, defined new state limits (as discussed in App. S1: ch. 3), and rescored the entire taxon sample. SKU TAB 1, another ratio, was defined and described in contradictory ways, neither of which matched the scores; we have treated it the same way (see App. S1: ch. 95). The measurements, sources, calculations and state changes are presented and compared to the original and revised scores in Data S5; the ratios, sources and state changes are also shown in App.-Tables 1 (for PREMAX 7) and 3 (for SKU TAB 1), which are placed in App. S1 under the characters they refer to.

PTE 15 is not reproducible; we have not figured out, either from the description or from the scores, what exactly the difference between the two states was meant to be (see App. S1 for details). The most likely meaning would be a duplicate of PTE 14 (ch. 123); we have deleted PTE 15.

Similarly, TAB 4 was described in a confusing way (see App. S1). Most likely, it is either a duplicate of PAR 7 (ch. 36) or is scored in a way that strongly contradicts its definition. (It concerns the position of the suture between the tabular and the squamosal—most OTUs do not have such a suture, but were scored as known nonetheless.) We have deleted it as well.

### Blockwise scoring of taxa

In RC07, the binary character PREMAX 7 (our ch. 3, mentioned in the preceding subsection), was (except for a few cases of missing data) scored 1 in all amniotes, diadectomorphs, "microsaurs", seymouriamorphs, anthracosaurs, *Silvanerpeton*, *Gephyrostegus*, *Diplocaulus*, *Diploceraspis* and *Eocaecilia*, and state 0 in all other OTUs. We have measured every OTU anew (Data S5; App.-Table 1). As detailed in App. S1, we have not been able to find an interpretation of the description of this character by RC07, or the different description by *Ruta, Coates & Quicke (2003)*, that would generate such a neat pattern or anything close. No matter how we interpret the character, most of the original scores have no particular relationship with our measurements (App.-Table 1; Data S5). Evidently, most OTUs were scored after presumed close relatives instead of on their own terms.

We have not found evidence of blockwise scoring in any other character. In this matrix, fortunately, blockwise scoring does not seem to be a recurrent problem.

### Modifications to individual cells

All modifications (including those made by *Germain (2008a*: chapter V*)*, except where we found them unjustified) are documented and justified in App. S1, with citations of the literature and/or specimens we used to rescore each cell. For the sake of brevity, the original scores (RC07) are usually not mentioned there, only our modified ones are; scores we did not change are not mentioned in App. S1 except where they could be controversial.

All modifications to individual cells are further presented in App. S2 (deleted characters excepted) and Data S4, separated by type of change: Green font in App. S2 (second sheet of Data S4) marks new scores when a state is new, redefined or possibly redefined (in cases where we are not sure of the meaning intended by RC07) and is also used to mark

all but the most trivial newly recognized cases of inapplicability (usually changes away from non-additive binary coding) as well as, in App. S2 but not Data S4, the scores changed to unknown following the removal of all postcranial material from *Rhynchonkos* (see below under Treatment of OTUs: *Rhynchonkos*). Blue font (third sheet of Data S4) indicates scores changed in order to account for ontogeny (see above under "Ontogeny discombobulates phylogeny"); red font (first sheet of Data S4) shows scores changed at face value (unaffected by any redefinition or ontogenetic considerations). The distinction between green and red should not be mistaken for different degrees of certainty; those are instead discussed in App. S1. Only green is shown when red and green apply, only blue is shown when red or green and blue apply; blue font is also used for the added OTUs where appropriate (their other scores are all black).

POSPAR 9, our ch. 43, is an example of a character to which we have made changes of all three types. (See App. S1 for details, discussions and references.) RC07 (p. 96: ch. 54) called it "Postparietals without (0) or with (1) broad, concave posterior emargination." As the presence of such an emargination does not depend on the presence of postparietals or their participation in the dorsal surface of the skull, we have redefined the character (see App. S1) as: "Edge between the dorsal and the caudal surfaces of the skull without (0) or with (1) broad, concave posterior emargination in the central bones." This has allowed us to score *Triadobatrachus* (which lacks identifiable postparietals and has our state 0), *Captorhinus* (whose postparietals lie entirely on the occipital surface, that is, ventral of the emargination that gives it our state 1) and *Batropetes* (which lacks identifiable postparietals and has our state 1); the resulting changes (from original scores as unknown) are marked in green.

Even though RC07 scored state 1 exclusively for diplocaulids other than *Keraterpeton*, it is in fact present both under their definition and under ours in a large number of other OTUs. The resulting 20 changes are marked in red font; this includes three from state 0 to polymorphism, and four (to state 1) that we comment as "arguably borderline", "weakly", "borderline" or "marginally". Also marked in red is the change to unknown in *Tseajaia*, where, as we discuss in App. S1, the position of the suture between postparietal and tabular is unclear so that we cannot determine if the central bones (the postparietals in this case) have a concave or a straight caudal edge, that is, if they are wide enough to visibly participate in the curvature. Likewise red is the change from unknown to 0 in *Ventastega*, which was required under both definitions by material published only in 2008.

Finally, *Microphon* changed from state 0 (scored by RC07) to state 1 in ontogeny, so we have scored state 1 and marked this in blue. Only state 0 (scored by RC07) has been observed in its fellow seymouriamorph *Leptoropha*, but all known individuals are ontogenetically comparable to those of *Microphon* that show state 0; as we do not know when the skeletal ontogeny of *Leptoropha* would have ended, or if *Leptoropha* reached morphological maturity at all instead of remaining paedomorphic in the adult stage, we have scored it as unknown (i.e., uncertainty between the two available states) and marked this in blue as well.

While we have not systematically compared every cell in the matrix to the literature or specimens, our attention was not limited to particular taxa or characters; we ended up making changes to the scoring of all OTUs, and of all characters except 96 (FONT 1), 150 (PSYM 4), 187 (TEETH 7) and the deleted parsimony-uninformative ISC 1, DOR FIN 1 and BAS SCU 1. In the end we have probably compared almost all cells in the matrix to the literature and/or to specimens. Shying away from the inordinate amount of time this work ended up taking, we initially concentrated our efforts on characters and OTUs relevant to lissamphibian origins, as determined, mostly, by optimizations on trees in Mesquite (*Maddison & Maddison, 2017*). Soon, though, we branched out to characters with subjectively suspicious state distributions (it goes without saying that many turned out to be entirely or partially correct), characters we did not immediately understand, characters we recoded while merging them with others and taxa that had been redescribed (recently or sometimes not so recently). Furthermore, we investigated characters that supported conflicting or unusual hypotheses of relationships apart from lissamphibian origins, and the taxa implicated in these; examples are Seymouriamorpha and Diadectomorpha, both surprisingly found to be paraphyletic by RC07, and Adelogyrinidae and *Acherontiscus*, found for the first time to clade with colosteids rather than "lepospondyls" by RC07. Finally, we checked the scores of OTUs that temporarily did strange things in preliminary analyses of ours, like *Tulerpeton*, *Edops*, *Trimerorhachis* or *Ossinodus* (unsurprisingly, some of this strangeness has remained). Whenever we investigated a character, we usually verified its scores for all OTUs, and whenever we investigated an OTU, we verified its scores for all characters except as restricted in the two subsections below.

As expected, we found cells where the correct score may be a matter of interpretation rather than being straightforwardly testable. (These are discussed in App. S1, or at least marked by such terms as "borderline", "weakly", "arguably", "probably", "most likely" or "almost certainly"; see above for examples.) The homology of a few bones and processes is a matter of interpretation, and sometimes we had to deal with such factors as taphonomic distortion; we argue for our interpretations in App. S1, but we recognize that some of our arguments may not be significantly stronger than potential counterarguments.

However, we must stress that a very large proportion of our changes to the matrix—most of the 2404 (see Data S4) that are marked in red in App. S2 and the 67 changes to deleted characters that would be marked red (Data S4), as well as many of the green or blue ones where those colors also apply—concern cases where scores in RC07 contradict their definitions and descriptions of the characters, according to all sources we had access to (literature and/or personal observation of specimens), without RC07 having provided an explanation (e.g. a new interpretation of homology or personal observation of specimens). We expect that many readers will share our surprise at this fact, and invite them to double-check our claims in App. S1.

### *Literature*

For the following OTUs—and quite possibly others—we have compared most or all cells to the literature (see below for specimens). Only the most important sources are cited here, a few more scores were changed based on sources cited in App. S1:

- *Panderichthys* (*Vorobyeva & Schultze, 1991*; *Boisvert, 2005*, *2009*; *Brazeau & Ahlberg, 2006*; *Boisvert, Mark-Kurik & Ahlberg, 2008*; *Ahlberg, 2011*—mostly occiput, extremities and girdles)
- *Ventastega* (*Ahlberg, Lukševičs & Lebedev, 1994*; *Lukševičs, Ahlberg & Clack, 2003*; *Ahlberg et al., 2008*)
- *Acanthostega* (*Coates, 1996*; *Porro, Rayfield & Clack, 2015*)
- *Ichthyostega* (skull roof: *Clack & Milner, 2015*; lower jaw: *Ahlberg & Clack, 1998*; *Clack et al., 2012a*; humerus: *Jarvik, 1996*; *Callier, Clack & Ahlberg, 2009*; *Ahlberg, 2011*; *Pierce, Clack & Hutchinson, 2012*; vertebrae: *Pierce et al., 2013*)
- *Tulerpeton* (*Lebedev & Coates, 1995*)
- *Colosteus* (*Langston, 1953*; *Hook, 1983*; *Bolt & Lombard, 2010*)
- *Greererpeton* (*Smithson, 1982*; *Godfrey, 1989*; *Bolt & Lombard, 2001*, *2010*)
- *Crassigyrinus* (*Panchen, 1985*; *Panchen & Smithson, 1990*; *Clack, 1998*)
- *Whatcheeria* (*Lombard & Bolt, 1995*, *2006*)
- *Baphetes* (*Beaumont, 1977*; *Milner & Lindsay, 1998*; *Milner, Milner & Walsh, 2009*)
- *Megalocephalus* (*Beaumont, 1977*)
- *Eucritta* (*Clack, 2001*)
- *Edops* (*Romer & Witter, 1942*)
- *Chenoprosopus* (*Hook, 1993*; *Reisz, Berman & Henrici, 2005*—previously missing data, including all postcranial material, and data mentioned as having been hitherto misinterpreted)
- *Cochleosaurus* (*Sequeira, 2004*, *2009*)
- *Isodectes* (*Sequeira, 1998*)
- *Neldasaurus* (*Chase, 1965*)
- *Trimerorhachis* (*Pawley, 2007*; *Milner & Schoch, 2013*)
- *Balanerpeton* (*Milner & Sequeira, 1994*)
- Dendrerpetidae (*Carroll, 1967*; *Milner, 1980*, *1996*; *Godfrey, Fiorillo & Carroll, 1987*; *Holmes, Carroll & Reisz, 1998*; *Robinson, Ahlberg & Koentges, 2005*—especially to make sure no polymorphisms were overlooked; see below and *Schoch & Milner, 2014*)
- *Eryops* (*Sawin, 1941*; *Pawley & Warren, 2006*)
- *Acheloma* (*Olson, 1941*; *Maddin, Reisz & Anderson, 2010*; *Polley & Reisz, 2011*—mostly filled in previously missing data)
- *Phonerpeton* (*Dilkes, 1990*)
- *Broiliellus* (*Carroll, 1964*; *Schoch, 2012*)
- *Amphibamus* (*Daly, 1994*)
- *Doleserpeton* (*Sigurdsen, 2008*; *Sigurdsen & Bolt, 2009*, *2010*; *Sigurdsen & Green, 2011*: app. 2)
- *Eoscopus* (*Daly, 1994*)

- *Platyrhinops* (*Clack & Milner, 2010*; remaining missing data filled in from *Hook & Baird, 1984*, and from *Werneburg, 2012a*, where not likely ontogeny-dependent)
- *Micromelerpeton* (*Boy, 1972*, *1995*; *Schoch, 2009*)
- *Apateon, Leptorophus, Schoenfelderpeton* (*Boy, 1972*, *1986*, *1987*; *Werneburg, 1991*, *2007a*; *Schoch & Fröbisch, 2006*; *Schoch & Milner, 2008*; *Fröbisch & Schoch, 2009b*)
- Albanerpetidae (*Estes & Hoffstetter, 1976*; *Fox & Naylor, 1982*; *Gardner, 2001*; *McGowan, 2002*; *Gardner, Evans & Sigogneau-Russell, 2003*; *Venczel & Gardner, 2005*; *Maddin et al., 2013a*; note that we follow this latter reference in interpreting the "cultriform process of the parasphenoid" (*McGowan, 2002*: fig. 13) as a hyobranchial element, contra *Marjanović & Laurin, 2008*)
- *Eocaecilia* (*Jenkins, Walsh & Carroll, 2007*)
- *Triadobatrachus* (*Rage & Roček, 1989*; *Roček & Rage, 2000*; *Sigurdsen, Green & Bishop, 2012*; *Ascarrunz et al., 2016*)
- *Valdotriton* (*Evans & Milner, 1996*)
- *Karaurus* (*Ivachnenko, 1978*)
- *Caerorhachis* (*Ruta, Milner & Coates, 2002*)
- *Bruktererpeton* (*Boy & Bandel, 1973*)
- *Gephyrostegus* (*Carroll, 1970*; *Godfrey & Reisz, 1991*; *Klembara et al., 2014*)
- *Eoherpeton* (*Panchen, 1975*; *Smithson, 1985*)
- *Proterogyrinus* (*Holmes, 1980*, *1984*)
- *Archeria* (*Romer, 1957*; *Holmes, 1989*)
- *Pholiderpeton attheyi* (*Panchen, 1972*)
- *Anthracosaurus* (*Panchen, 1977*; *Clack, 1987a*)
- *Pholiderpeton scutigerum* (*Clack, 1987b*)
- *Solenodonsaurus* (*Laurin & Reisz, 1999*; *Danto, Witzmann & Müller, 2012*—mostly filled in previously missing data)
- *Kotlassia* (*Bystrow, 1944*; *Bulanov, 2003*)
- *Discosauriscus* (*Klembara, 1997*, *2009*; *Klembara & Bartík, 2000*)
- *Seymouria* (*White, 1939*; *Laurin, 1996a*, *2000*; *Klembara et al., 2005*, *2006*, *2007*)
- *Diadectes* (*Case, 1910*, *1911*; *Case & Williston, 1912*; *Olson, 1947*; *Moss, 1972*; *Berman, Sumida & Lombard, 1992*; *Berman, Sumida & Martens, 1998*; *Berman et al., 2004*)
- *Limnoscelis* (*Fracasso, 1983*; *Berman & Sumida, 1990*; *Reisz, 2007*; *Berman, Reisz & Scott, 2010*; *Kennedy, 2010*)
- *Westlothiana* (*Smithson et al., 1994*)
- *Batropetes* (*Glienke, 2013*, *2015*)
- *Stegotretus* (*Berman, Eberth & Brinkman, 1988*)
- *Rhynchonkos* (CG78; *Szostakiwskyj, Pardo & Anderson, 2015*)
- *Microbrachis* (CG78; *Vallin & Laurin, 2004*; *Olori, 2015*)

- *Hyloplesion* (CG78; *Olori, 2015*)
- *Brachydectes* (*Wellstead, 1991*; *Pardo & Anderson, 2016*)
- *Acherontiscus* (*Carroll, 1969a*)
- *Adelospondylus, Adelogyrinus, Dolichopareias* (*Andrews & Carroll, 1991*)
- *Scincosaurus* (*Milner & Ruta, 2009*)
- *Diplocaulus* (postcranium and lower jaw: *Williston, 1909*; *Douthitt, 1917*)
- *Diploceraspis* (mostly postcranium and lower jaw: *Beerbower, 1963*)
- *Lethiscus* (*Wellstead, 1982*; *Anderson, Carroll & Rowe, 2003*; *Pardo et al., 2017*)
- *Oestocephalus* (*Carroll, 1998a*; *Anderson, Carroll & Rowe, 2003*; *Anderson, 2003a*)
- *Phlegethontia* (*Anderson, 2002*, *2007a*)
- *Notobatrachus* (*Estes & Reig, 1973*; *Báez & Basso, 1996*; *Báez & Nicoli, 2004*, *2008*)
- *Vieraella* (*Estes & Reig, 1973*; *Báez & Basso, 1996*)
- *Ossinodus* (*Warren & Turner, 2004*; *Warren, 2007*; *Bishop, 2014*)
- *Pederpes* (*Clack & Finney, 2005*)
- *Orobates* (*Berman et al., 2004*; *Nyakatura et al., 2015*: digital model)
- *Ariekanerpeton* (*Laurin, 1996b*; *Klembara & Ruta, 2005a*, *2005b*)
- *Leptoropha* (*Bulanov, 2003*)
- *Microphon* (*Bulanov, 2003*, *2014*)
- *Capetus* (*Sequeira & Milner, 1993*)
- *Tseajaia* (*Moss, 1972*; *Berman, Sumida & Lombard, 1992*)
- *Utegenia* (*Laurin, 1996c*; *Klembara & Ruta, 2004a*, *2004b*)

Parts of the above list surprise us. For instance, *Ruta, Coates & Quicke (2003)* cited *Ahlberg, Lukševičs & Lebedev (1994)* as their source for scoring *Ventastega*, but only used that publication to code not much more than half of the bones that are described and illustrated in that work; large parts of the skull, for example almost the entire palate, were scored as entirely unknown by *Ruta, Coates & Quicke (2003)* and RC07 for reasons that we could not determine. This goes so far that in some instances some of the bones belonging to a skull fragment were scored while others sutured to them in the same fragment were not.

RC07 cited the paper by *Sequeira (2004)*, but only for its phylogenetic analysis, not for its redescription of the skull of *Cochleosaurus* based on "several large, presumably adult, skulls" (*Sequeira, 2004*: abstract)—all previous descriptions "were based almost entirely on small, subadult [ . . . ] specimens" (ibid.). Instead, RC07 kept the outdated scores of *Cochleosaurus* by *Ruta, Coates & Quicke (2003)*. In total, to the best of our knowledge, RC07 made only two changes to individual cells of the preceding version, the two mentioned on their p. 93 for PREMAX 1 (ch. 1).

*Ruta, Coates & Quicke (2003)* cited *Boy & Bandel (1973)* as their source for the scoring of *Bruktererpeton*. That publication is written in German, which may explain some of the differences between it and the scoring by *Ruta, Coates & Quicke (2003)* and RC07.

(Fortunately, German is D. M.'s native language.) Some of the figures, however, contradict the matrix by RC07 as well.

Despite the overlap in authors, many discrepancies exist between the matrix by RC07 and the detailed redescriptions of *Caerorhachis* (*Ruta, Milner & Coates, 2002*—not cited by RC07, but cited as "2001", the year of publication originally intended by the journal, by *Ruta, Coates & Quicke, 2003*), *Silvanerpeton* (*Ruta & Clack, 2006*—not cited, not even as "in press"), *Ariekanerpeton* (*Klembara & Ruta, 2005a*, *2005b*—not cited, although personal observations are cited) and *Utegenia* (*Klembara & Ruta, 2004a*, *2004b*—cited).

*Ruta, Coates & Quicke (2003)* scored *Kotlassia* after the description by *Bystrow (1944)*; RC07 did not change this. Bystrow explicitly synonymized *Kotlassia* and *Karpinskiosaurus* and did not document for all parts of his description on which specimen(s) they were based. *Bulanov (2003)* disagreed with Bystrow and separated the two taxa again, considering them rather distant relatives, but he merely mentioned the existence of postcrania of *Kotlassia* (the entire monograph describes only the cranial anatomy of a wide range of seymouriamorphs). Similarly, *Klembara (2011)* redescribed only the skull of *Karpinskiosaurus*. A useful description of the postcrania of either taxon does not exist as far as we know. We have accepted the scores by RC07 at face value for the time being and scored all postcranial characters of *Karpinskiosaurus*, a taxon we have added to the matrix (see below), as unknown, except for those few that *Bystrow (1944)* explicitly commented on.

### Specimens

"This has not been a literature-based exercise", wrote *Ruta, Coates & Quicke (2003*: 292*)* in their conclusions; and while they and RC07 did have to rely on the literature to score some taxa (if only because the taxon sample is simply too large for anyone to visit all relevant specimens within realistic time and budget constraints), both of their publications contain lists of specimens they had studied firsthand. Strangely, however, the results of almost none of these observations are mentioned in the papers; there is no way of extracting information on which characters are described or illustrated incorrectly in the literature or which ones are visible in the fossils but not presented in a publication (so that scoring from the literature would produce spurious missing data). We must insist that a data matrix is not an appropriate place to publish new observations: in a matrix there is no way to tell if a surprising score is a new observation or just a typographic error, a momentary confusion of contradictory conventions about which state is called 0 and which is called 1, or one of a number of other very common problems (see above and *Marjanović & Laurin, 2013*, for discussion). For each of the scores we have changed based on our observations of specimens, we have therefore documented the change in App. S1 along with the inventory numbers of specimens that show the states we have scored.

For the following taxa in the original sample we have compared most or all cells to specimens (fossils or casts) which are, where necessary, mentioned under the respective characters in App. S1:

- *Ichthyostega* (palate only)
- *Edops* (as *Romer & Witter (1942)* noted, the sutures on the only known adult skull roof—that of MCZ 1378—are for the most part extremely difficult to trace, so we have accepted their interpretations wherever D. M. was unable to confirm them; D. M. did not detect any contradictions)
- *Isodectes* (postcranium only)
- *Ecolsonia* (previously missing data only)
- *Triadobatrachus*
- *Karaurus* (everything except the palate, which is not reproduced by the cast we had access to)

*Diceratosaurus* is about to undergo redescription by Angela Milner, D. M., Florian Witzmann and likely Jason Anderson's lab; we have not preempted that work, but only changed the scores of characters of particular interest based on D. M.'s observations of specimens, mostly filling in missing data—and often confirming the redescription by *Jaekel (1903*; not cited by RC07 or *Ruta, Coates & Quicke, 2003)*.

The complete list of specimens we have used to change scores in the matrix forms Table 2. We cannot provide a list of all specimens we have seen; specimens that agree with the scores of RC07 are therefore only mentioned in App. S1 when a score could be controversial, or when other specimens of the same OTU show another state and we have consequently scored polymorphism. However, whenever D. M. visited a collection, he saw all or almost all specimens that are broadly relevant to this study. One exception to this is the NMS, where the collection is kept in a building on the other side of the city and specimens that are not on exhibit are brought to the museum building only on request; the only NMS specimen D. M. has seen is *Casineria* (which we added to the matrix as described below). The other exception is the Anderson lab at the University of Calgary, which currently houses many important specimens of which D. M. has only taken the time to study a referred specimen of *Asaphestera* in any detail.

### The albanerpetid neck

Though this concerns very few characters, our coding of the albanerpetids (a single OTU as in RC07) assumes our reinterpretation of their unique atlas-axis complex. As in mammals, this complex accommodated dorsoventral and lateral movements of the head at separate joints. Traditionally (*Estes & Hoffstetter, 1976*; *Fox & Naylor, 1982*), this complex is considered to consist of the atlas (a complete vertebra consisting of a centrum and fused, fully formed, full-size neural arch), the axis (a centrum that lacks any trace of a neural arch), and the third vertebra (again a complete vertebra consisting of a centrum and fused, fully formed, full-size neural arch). The "axis" is commonly sutured to the "third vertebra". Dorsoventral movements of the head occurred between the skull and the atlas, lateral ones between the atlas and the "axis". By comparison to amniotes and "microsaurs", we think it is more parsimonious to interpret the "axis" as only the intercentrum of the axis, even though intercentra are otherwise unknown in albanerpetids, while the "third vertebra" would be the pleurocentrum and neural arch of the axis.

**Table 2 List of specimens used by D. M. to change scores in the matrix (as cited in App. S1) or to score added taxa.** The OTUs are in the same order as in the matrix, which is unchanged from the non-alphabetical version of the matrix file of RC07 for the shared taxa (*Acanthostega* through *Tseajaia*). Specimens that do not contradict any scores of RC07 (including "unknown") or the literature on the added taxa (*Nigerpeton* through *Casineria*) are not listed except in cases of polymorphism; a few such specimens are mentioned in the text.

| OTU | Specimens |
|---|---|
| *Acanthostega* | TMM 41766-1 (cast of MGUH VP 6033[*], formerly "A. 33") |
| *Ichthyostega* | AMNH 23100; MCZ 3361; TMM 41224-2 (all of them casts of MGUH VP 6055, formerly "A. 55") |
| *Greererpeton* | TMM 41574-1 |
| *Edops* | MCZ 1378, 1769, 1781, 1782, 6489, 6493, 7126, 7128, 7136, 7143, 7158, 7162, 7197, 7258, 7259, 7264, 7274; USNM 23309 |
| *Chenoprosopus* | CM 34909; USNM 437646 |
| *Isodectes* | CM 81430, 81512; MCZ 6044 (cast of USNM 4481); unnumbered MCZ cast of AMNH 6935 before etching; USNM 4471, 4474, 4555 |
| *Trimerorhachis* | AMNH 4565[*], 4572; TMM 40031-80, 40031-81, 40998-39 |
| *Eryops* | AMNH 4180, 4183, 4186, 4189[*], 4673, 23529; MCZ 1126, 1129, 1536, 2588, 2638, 2682, 2766; TMM 31225-33, 31226-12, 31227-11, 31227-14, 40349-20; USNM uncatalogued "Texas '84 #40", "Texas '86 #77" |
| *Phonerpeton* | AMNH 7150; MCZ 1414, 1419[*], 1485, 1548, 1771, 2313, 2474; USNM 437796 |
| *Ecolsonia* | CM 38017, 38024 |
| *Broiliellus brevis* | MCZ 3272 |
| *Doleserpeton* | AMNH 24969, 29466, 29470; BEG 40882-25 |
| *Platyrhinops* | AMNH 2002 |
| *Eocaecilia* | MNA V8066[*] (Museum of Northern Arizona; formerly MCZ 9010) |
| *Karaurus* | unnumbered MNHN cast[1] of PIN 2585/2[**] |
| *Triadobatrachus* | MNHN.F.MAE126[**] (natural mold and cast)[1] |
| *Archeria* | MCZ 2049, other MCZ specimens[2] |
| *Gephyrostegus* | MB.Am.641; TMM 41733-1 (cast) |
| *Seymouria* | BEG 30966-176 |
| *Diadectes* | AMNH 4352, 4839; BEG 31222-56 |
| *Paleothyris* | TMM 45955-2 (cast of MCZ 3482) |
| *Batropetes* | *B. palatinus*: MB.Am.1232 (including casts) |
| *Tuditanus* | CM 29592 |
| *Stegotretus* | CM 34901 (all others were on loan) |
| *Asaphestera* | NMC 10041 (National Museum of Canada; currently on loan to J. Anderson, University of Calgary) |
| *Microbrachis* | MB.Am.840 |
| *Hyloplesion* | NHMW 1983/82/54, other NHMW specimens |
| *Odonterpeton* | USNM 4465+4467[**3] |
| *Scincosaurus* | MB.Am.29 |
| *Diceratosaurus* | AMNH 6933[*]; CM 25468, 26231, 29593, 29876, 34617, 34656, 34668, 34696, 34670, 67157, 67169, 72608, 81504, 81507, 81508; MB.Am.776, MB.Am.778 |
| *Ptyonius* | MCZ 3721 (cast of "AMNH 6871 (85466)") |
| *Sauropleura* | CM 25312 |
| *Lethiscus* | MCZ 2185[**] |
| *Phlegethontia* | USNM 17097 |
| *Tseajaia* | CM 38033 |
| *Nigerpeton* | MNN MOR 69[*] (including unpublished intercentrum and unprepared skull fragments), 70, 82, 83, 108 |
| *Saharastega* | MNN MOR 73[**] (including two unpublished mandible pieces) |

(Continued)
| Table 2 (continued). | |
|---|---|
| **OTU** | **Specimens** |
| *Iberospondylus* | PU-ANF 2, 14*, 15[4] |
| Caseasauria | *Eothyris*: MCZ 1161**; *Oedaleops*: unnumbered MCZ cast of UCMP 35758* (University of California Museum of Paleontology) |
| *Sparodus* | NHMW 1899/0003/0006 (Figs. 3 and 4) |
| *Carrolla* | TMM 40031-54** |
| *Sclerocephalus* | MB.Am.1346 |
| *Chelotriton* | MB.Am.45 (natural mold and cast) |
| *Trihecaton* | CM 47681*, 47682 (probably part of the same individual) |
| St. Louis tetrapod | MB.Am.1441** (natural mold and cast) |
| *Casineria* | NMS G 1993.54.1** (part and counterpart; Figs. 5–7) |

**Notes:**

[*] Type specimen of the type species.

[**] Type and only known specimen of the only known species.

[1] Observed together with M. L.

[2] All were used to score the same characters.

[3] The apparently unpublished specimens CM 81525 and CM 81526 are also labeled *Odonterpeton*, and are kept together with a note by Baird (dated 1991) which says that further small skeletons attributed to *Brachydectes* may belong to *Odonterpeton* as well. More likely, CM 81525—a limbless skeleton without a trace of limbs or girdles, whose skull is only preserved as an indistinct impression—is instead a juvenile aïstopod with ribs that are not yet k-shaped, and the disarticulated CM 81526 is a juvenile *Brachydectes* after all (D. Marjanović, per. obs. 2016; J. Pardo, personal communication after seeing photos taken by D. M.); a more detailed study is in preparation.

[4] All three were observed together with Rodrigo Soler-Gijón; additionally, they had been studied earlier by M. L. (*Laurin & Soler-Gijón, 2001, 2006*).

Comparison to temnospondyls is more difficult; in both *Doleserpeton* (*Sigurdsen & Bolt, 2010*) and *Gerobatrachus* (*Anderson et al., 2008a*), the only dissorophoids with pleurocentrum-dominant vertebrae, the axis is very incompletely preserved, but the relative sizes of pleuro- and intercentra elsewhere in the column are at least not incompatible with our interpretation.

### The skull roof of Brachydectes

I have not hesitated to criticise former interpretations of structures and sutures, and I need not add that my own are likewise open to criticism.

– *Romer (1930*: 80)

*Wellstead (1991)* reconstructed several unusual features in the skull roof of *Brachydectes*. We previously tried (*Marjanović & Laurin, 2008, 2013*) to resolve these problems by reinterpreting the homologies of the bones that *Wellstead (1991)* had identified. For example, the supposed postparietals do not meet in Wellstead's reconstruction, but are separated by a contact between the suproccipital and the parietals; we suggested that they were tabulars instead, while the supposed tabulars would be postorbitals (thought absent by Wellstead), and postparietals would be absent. As a byproduct, this would restore normal spatial relationships to the posttemporal foramen identified by *Wellstead (1991)*.

Based on new material and µCT scans, *Pardo & Anderson (2016)* have shown that *Wellstead (1991)* had overlooked a few sutures and misjudged certain 3D relationships; our reinterpretation of Wellstead's reconstruction is therefore moot. In particular, Wellstead correctly identified the postparietals, which do, however, meet in the midline in most or all specimens for a variable part of their length; there is no posttemporal

foramen at all; and postorbitals are, like in lissamphibians, absent. We have updated the scores of *Brachydectes* following *Pardo & Anderson (2016)*.

## Treatment of OTUs

OTUs are hypotheses; the delimitations of some require comments.

### Paleothyris/Protorothyris

One OTU of RC07 is (explicitly) a composite of *Paleothyris acadiana* and *Protorothyris archeri*. *Protorothyris* is, according to the only phylogenetic analysis that has included both so far (*Müller & Reisz, 2006*), more closely related to *Petrolacosaurus*—another OTU in this matrix—than to *Paleothyris*; *Paleothyris* could even be more closely related to *Captorhinus*—also included in this matrix—than to either *Protorothyris* or *Petrolacosaurus*.

Still, these two species are morphologically similar, and most of their differences could not be scored in our matrix, so we have not made a special effort to weed out scores based on *Protorothyris*. Our changes to this OTU, however, are exclusively based on the description of *Paleothyris* by *Carroll (1969b)* and D. M.'s observations of *Paleothyris* specimens, especially TMM 45955-2 (a cast of MCZ 3482, which is on display and was not accessible at the time), cursorily confirmed against MCZ 3473, MCZ 3475, MCZ 3477, MCZ 3481 (the holotype), MCZ 3483, MCZ 3486, MCZ 3487, MCZ 3488, MCZ 3490, MCZ 3492 and the apparently uncatalogued unlabeled specimen B-17 (also kept at the MCZ). Some of these changes are from known to unknown, so they likely concern characters whose state is known in the more completely preserved *Protorothyris* but not in *Paleothyris*.

### Dendrerpetidae

*Milner (1996)* revised the specimens from Joggins (Nova Scotia, Canada) that were at that time included in *Dendrerpeton acadianum* (following *Carroll, 1967*), but had originally been described as several genera with many species. "The majority of specimens are still attributable to *Dendrerpeton acadianum* but three other forms are represented by one or two specimens each. These are *Dendrerpeton confusum* sp. nov., based on a single large skull, *Dendrerpeton helogenes* (Steen) comb. nov., based on two specimens, and an indeterminate cochleosaurid, based on a set of cranial fragments" (*Milner, 1996*: abstract). One of the specimens newly referred to *D. helogenes* was the well preserved skull that had been described by *Godfrey, Fiorillo & Carroll (1987)* as *D. acadianum*. "Subsequently HOLMES et al. [ = Holmes, Carroll & Reisz] (1998) described an exquisitely preserved skeleton of a temnospondyl from Joggins and referred it to *Dendrerpeton acadianum*. By doing so, they lumped all the Joggins material together again. In this work, it is argued that the Joggins material includes two distinct genera, *Dendrerpeton* sensu MILNER 1996 and *Dendrysekos* (*Dendrerpeton helogenes* of MILNER 1996), which differ in a number of characters. The specimens of GODFREY et al. [ = Godfrey, Fiorillo & Carroll] (1987) and HOLMES et al. (1998) are referable to *Dendrysekos*. This effectively means that *Dendrysekos* rather than *Dendrerpeton* has formed the outgroup in many recent cladistic analyses of temnospondyls" (*Schoch & Milner, 2014*: 25).

*Ruta, Coates & Quicke (2003)* did not cite *Milner (1996)*. Their "*Dendrerpeton acadianum*" OTU, kept unchanged by RC07 (who did not cite *Milner (1996)* either), is in part based on their personal observations of specimens, of which at least some—notably the lectotype—really do belong to *D. acadianum* according to *Milner (1996)* and *Schoch & Milner (2014)*. However, *Ruta, Coates & Quicke (2003)* also used and cited the descriptions by *Godfrey, Fiorillo & Carroll (1987)* and *Holmes, Carroll & Reisz (1998)*. Assuming that the personal observations were in fact used to score the taxon and not merely to cursorily confirm the literature, this OTU is thus a chimera of *Dendrerpeton acadianum* and *Dendrysekos helogenes*.

*Schoch & Milner (2014)* continued to maintain that *Dendrerpeton* and *Dendrysekos* are sister-groups; for this reason, we have not split the OTU for the time being, but merely renamed it Dendrerpetidae, the name recommended by *Schoch & Milner (2014)*, and compared it to the literature to make sure no polymorphisms had been overlooked (see above and *Watanabe, 2016*). In future work, however, it may be a good idea to treat all dendrerpetid species separately (including *Dendrerpeton rugosum* from Jarrow in Ireland: *Milner, 1980*; *Schoch & Milner, 2014*): *Ruta (2009)* added *Dendrerpeton confusum* to a temnospondyl matrix derived from the matrix of RC07 and found that it was not the sister-group of the "*D. acadianum*" OTU. We have not done this yet because the present matrix most likely does not contain enough characters to place them accurately with respect to each other.

(Whether the "*D. acadianum*" OTU of *Ruta (2009)* included any other dendrerpetids is unclear to us. The matrix of *Ruta (2009)* is a slight modification of that of *Ruta & Bolt (2006)*. The latter's app. 1 lists all OTUs, except *D. confusum*, and the literature sources used to score them. For "*D. acadianum*", these are *Carroll, 1967*; *Milner, 1980*, *1996*; *Godfrey, Fiorillo & Carroll, 1987*; and *Holmes, Carroll & Reisz, 1998*. In addition, this OTU is marked with an asterisk for having "been examined directly by one or both authors (using either casts or original specimens)"; which specimens those are is not mentioned, but presumably they are the same as those consulted by *Ruta, Coates & Quicke (2003)*. However, given the absence of *D. confusum* from this appendix, we wonder if the appendix was written before Ruta & Bolt decided to separate the two *Dendrerpeton* OTUs; if so, it is possible that the "*D. acadianum*" OTU really is just *D. acadianum* and not a chimera.)

*Balanerpeton* has sometimes (*Pawley, 2006*; *Clack et al., 2012b*; *Dilkes, 2015a*), but not always (*Milner & Sequeira, 1994*; *Ruta, 2009*), been found as the sister-group of "*Dendrerpeton*"; we therefore follow *Schoch & Milner (2014)* in not including it in the Dendrerpetidae OTU.

### Rhynchonkos

*Szostakiwskyj, Pardo & Anderson (2015)* have shown that previous conceptions of *Rhynchonkos* (e.g. CG78; considered to contain a single species) were chimeric. They restricted *Rhynchonkos* to its holotype, a skull with lower jaw. The other skulls were referred to the new taxa *Aletrimyti* and *Dvellecanus*; a further lower jaw turned out to belong to none of the three. *Szostakiwskyj, Pardo & Anderson (2015)* did not mention the

postcranial material that had been referred to *Rhynchonkos* (e.g. CG78). A poorly preserved but largely articulated presacral skeleton (CG78: fig. 67) belongs to the same specimen as the referred skull of *Aletrimyti* (*Szostakiwskyj, Pardo & Anderson, 2015*: 5). We therefore wondered whether to restrict our *Rhynchonkos* OTU to *Rhynchonkos* or instead to *Aletrimyti* (the type material of which is also slightly better preserved than that of *Rhynchonkos*). J. Pardo (pers. comm. 2015) has, however, pointed out that the postcranial material needs to be revisited. We have therefore restricted the *Rhynchonkos* OTU to the type material of *Rhynchonkos* and scored all postcranial characters as unknown (changes marked in green in App. S2 but not counted in Data S4).

We considered adding *Aletrimyti* and *Dvellecanus* as separate OTUs to our analyses with added taxa. Most likely, however, this would require also adding several at least superficially similar "microsaurs" (*Nannaroter, Tambaroter, Huskerpeton, Proxilodon*) and several characters; we prefer to leave this to future work.

RC07 had scored a few characters that remain unknown for all of the specimens referred to *Rhynchonkos* by CG78. An example is CAU FIN 1 (our ch. 277; see App. S1), for which they scored state 1, the absence of lepidotrichia in the tail—to the best of our knowledge, only the most proximal tail vertebrae are represented in the previously attributed material (CG78: 103–104, 109–110, fig. 67–72). This is, incidentally, a common issue that affects several other OTUs and several other tail characters (App. S1).

### Taxa added as parts of existing OTUs

We have interpreted almost all OTUs as genera (or larger taxa in the cases of Albanerpetidae and Dendrerpetidae) rather than species. This has allowed us to fill in some missing data and to approach more plesiomorphic morphotypes. We do not think the polymorphisms this has occasionally introduced are a problem; quite the opposite—they are a better representation of the scope of the matrix. With the exceptions of *Megalocephalus* (*Milner, Milner & Walsh, 2009*), *Broiliellus* (*Schoch, 2012*; *Holmes, Berman & Anderson, 2013*; *Dilkes, 2015b*), *Oestocephalus* (*Anderson, 2003a*) and of course *Pholiderpeton*, the monophyly of the few genera in this matrix that are not monospecific is fairly obvious, at least with respect to the other OTUs in this matrix, and has not been disputed in the literature. *Pholiderpeton* was already coded as two separate OTUs (*Pholiderpeton scutigerum, Pholiderpeton attheyi*) by RC07 and indeed *Ruta, Coates & Quicke (2003)*. *Megalocephalus* is, if at all, only paraphyletic with respect to *Kyrinion* (*Milner, Milner & Walsh, 2009*) which is not included in this matrix (we have refrained from adding it—see below—because it would, given the present character sample, provide almost no new information on baphetid morphodiversity and thus baphetid relationships); also, we have not used any information from ?*M. lineolatus*, but only from the type species *M. pachycephalus*. We have not used information from any species referred to *Broiliellus* other than the one used by RC07, *B. brevis*—which is, incidentally, not the type species of *Broiliellus* and might therefore end up outside that taxon. In the figures, App. S2 and Data S4, we explicitly call this OTU "*Broiliellus brevis*". We have also ignored the incompletely preserved ?*Oestocephalus nanus* (*Boyd, 1982*), which is likely closer to *Phlegethontia* than to *Oestocephalus* (*Anderson, 2003a*), and the

well preserved and stunningly well prepared but inadequately described ?*O. guettleri* (*Krätschmer, 2006*) and have—no doubt like RC07—restricted ourselves to the type species, *O. amphiuminus*. (Naturally, the intriguing large skull referred to *O.* by *Anderson, Pardo & Holmes (2018)* was unknown to us; we are looking forward to its complete description.) Because information about the specimen called *Scincosaurus spinosus* was unavailable to us, we only used the type species, *S. crassus* (used by RC07). Further, while the monophyly of *Micromelerpeton* has not been doubted, we have ended up using only *M. credneri* (the type species, used by RC07) because the other species do not add any information, being known from fewer individuals which are all skeletally immature.

*Milner & Schoch (2013)* found that all recognized species of *Trimerorhachis* except *T. sandovalensis* form a clade which may or may not be the sister-group of *T. sandovalensis* to the exclusion of *Neldasaurus*. Because *Neldasaurus* is an OTU in this matrix, we have not used information from ?*T. sandovalensis* to score the *Trimerorhachis* OTU. We considered adding ?*T. sandovalensis* as an OTU to our analyses with added taxa, but its scores would hardly differ from the *Trimerorhachis* OTU at all. Because *Milner & Schoch (2013)* presented a phylogeny of the four species in the undoubted *Trimerorhachis* clade, we have avoided scoring *Trimerorhachis* as polymorphic and instead tried to identify the plesiomorphic condition (as we have done with Albanerpetidae and *Batropetes*); this concerns very few characters, however.

## OTUs added for a separate set of analyses

Several theoretical considerations suggest that taxon exemplars should be as diverse as possible [ . . . ] [five references]. Importantly, a recent study based on simulations of true phylogenies (*Salisbury & Kim, 2001*) indicates that dense and random taxon sampling increases the probability of retrieving correctly the plesiomorphic condition of characters as well as the ancestral state near the tree root. Furthermore, *Salisbury & Kim's (2001)* simulations show that in the analysis of small clades, estimates of ancestral states are strongly affected by cladogram topology and by the number of descendent [sic] branches in progressively more distal internal nodes.

– *Ruta, Coates & Quicke (2003*: 255*)*

Also, taxon removal because of incomplete preservation and missing character scores may be undesirable, because such taxa may have a positive effect on cladogram resolution [ . . . ] [five references].

– *Ruta, Coates & Quicke (2003*: 257*)*

We recommend the inclusion of as many fossils as possible in any phylogenetic analysis.

– *Conrad & Norell (2015)*

Numerous phylogenetic analyses demonstrate that the systematic position of *Paranthodon* is highly labile and subject to change depending on which exemplifier for the clade Stegosauria is used. The results indicate that the use of a basal exemplifier may not result in the correct phylogenetic position of a taxon being recovered if the taxon displays character states more derived than those of the basal exemplifier, and we

recommend the use, minimally, of one basal and one derived exemplifier per clade.
– *Raven & Maidment (2018*: abstract*)*

Analogously to that of *Marjanović & Laurin (2008*, *2009)*, the most important of the aims of the present work is to find out which hypothesis on the origin of the modern amphibians the matrix by RC07 supports after revision if we keep, as far as possible, its taxon and character sample intact. However, the Early Permian (Cisuralian) *Gerobatrachus* was described too late (*Anderson et al., 2008a*) to be included in the matrix of RC07, yet it is highly relevant because the phylogenetic analysis included in its description supported the PH. Being a temnospondyl with many similarities to lissamphibians, *Gerobatrachus* may also be considered to support the TH over the LH; indeed, the analysis by *Maddin, Jenkins & Anderson (2012)* found it to be the oldest known lissamphibian while supporting the TH. However, this is based on a further development of the matrix by *Anderson et al. (2008a)* that did not take the reevaluations by *Marjanović & Laurin (2009)* and *Sigurdsen & Green (2011)* into account; and *Marjanović & Laurin (2009)* found the LH despite including *Gerobatrachus*. Following the example of *Marjanović & Laurin (2008)*, we have therefore added *Gerobatrachus* to the present matrix in order to test if it changes the results.

At this opportunity we also added the following taxa:

### Further temnospondyls other than stereospondylomorphs

*Micropholis* is an amphibamid, which means that the TH and the PH consider it close to the ancestry of some or all extant amphibians. It is the only dissorophoid known to have two widely spaced occipital condyles (like the modern amphibians), and it shares the Early Triassic age of the oldest known lissamphibians (the stem-salientians *Triadobatrachus* and *Czatkobatrachus*), unlike the Cisuralian *Doleserpeton* and *Gerobatrachus* or the Pennsylvanian *Amphibamus*. Growth series spanning most of the skeleton are known (*Schoch & Rubidge, 2005*). It is particularly surprising that RC07 did not add *Micropholis* to their matrix; they speculated that it might "turn out to occupy a more derived position than *Doleserpeton* on the amphibian stem group" (RC07: 86) because of its stratigraphic position.

*Nigerpeton* was described (*Steyer et al., 2006*; *Sidor, 2013*) as a cochleosaurid edopoid temnospondyl. The edopoids are thought to be close to the base of Temnospondyli—indeed, some analyses of temnospondyl phylogeny have used one or more edopoids as the outgroup or part thereof (*Schoch & Witzmann, 2009a*, *2009b*)—so we expected an influence on the position of Temnospondyli and on the interrelationships of its largest constituent groups. The skull and lower jaw have been described, though their surface is mostly badly preserved. D. M. has seen all known specimens—including postcranial material—and compared all scores to them; in App. S1 we report a number of previously unpublished observations.

*Saharastega*, of which likewise the entire skull is known, though the preservation of the surface is again bad (*Damiani et al., 2006*), has had an unstable phylogenetic position close to the root of Temnospondyli. Based on a few character states that are rather odd for a temnospondyl, it has even been suggested to be a seymouriamorph,

though not in the peer-reviewed literature (*Yates, 2007*). It almost certainly is a temnospondyl, but, as for *Nigerpeton*, we expected an influence on the relationships and the large-scale phylogeny of the temnospondyls. In addition, this is of course an opportunity to clarify the position of *Saharastega* itself. D. M. has seen the only known specimen and compared all scores to it; in App. S1 we report a number of previously unpublished observations.

*Iberospondylus* is a rather early temnospondyl known from most of the skull and various postcranial remains, preserving an unusual combination of plesiomorphies and apomorphies such as the dissorophoid-like dorsal process on the quadrate. The eight phylogenetic analyses that have included it so far (*Laurin & Soler-Gijón, 2001*, *2006*; *Pawley, 2006*: fig. 44; *Schoch & Witzmann, 2009a*, *2009b*; *Dilkes, 2015a*; *Pardo, Small & Huttenlocker, 2017*: fig. 2, S7) have given seven different results, and two of them were based on very small matrices. Both of us have studied all three specimens.

*Tungussogyrinus* has occasionally been considered a caudate, which would be highly interesting considering its Early Triassic age. *Werneburg (2009)* redescribed it as the sister-group to all other branchiosaurids, but noted similarities to lissamphibians, especially one apomorphy shared with Salientia that is coded in this matrix. RC07 (p. 86) mentioned *Tungussogyrinus* together with *Micropholis*.

*Acanthostomatops* (*Witzmann & Schoch, 2006a*) is a Cisuralian zatracheid temnospondyl thought to lie close to Dissorophoidea and/or *Eryops*. As pointed out but not tested by *Witzmann & Schoch (2006a*: 365*)*, it may thus influence temnospondyl phylogeny as well as the position of Lissamphibia.

*Palatinerpeton* is a temnospondyl known from an incompletely preserved, immature skeleton (*Boy, 1996*). Endowed with an unusual combination of characters and suggested to be a stereospondylomorph (*Schoch & Milner, 2000*) or in a trichotomy with Stereospondylomorpha and an Eryopidae-Zatracheidae-*Parioxys* clade (*Boy, 1996*; see "Taxa that were not added", below, for *Parioxys*), it has the potential to influence temnospondyl phylogeny.

*Erpetosaurus*, redescribed by *Milner & Sequeira (2011)* on the basis of many flattened specimens that range from skull fragments to incomplete articulated skeletons, is a dvinosaurian temnospondyl, thought to be most closely related to *Isodectes* in the taxon sample of RC07. We have added it in order to confuse things—in other words, to test the robustness of this matrix against homoplasy: *Erpetosaurus* shows a confusing mix of unexpected plesiomorphies and unexpected similarities to colosteids (among others).

*Mordex* was neglected for a long time because it is mostly known from larvae that were confused with *Branchiosaurus*, *Platyrhinops* and other dissorophoids from the same site. Recently, however, it has been understood as the oldest known trematopid (*Milner & Sequeira, 2003*; *Milner, 2007*; *Werneburg, 2012a*); *Milner (2007)* called it "most similar to *Ecolsonia*" (tentatively confirmed by *Schoch, 2012*, and *Schoch & Milner, 2014*), which implies that adding *Mordex* to the present analysis will influence the phylogenetic position of *Ecolsonia*. Although it is not very well known—the largest specimen is an incomplete skull roof, the others are all much smaller and incompletely ossified (*Werneburg, 2012a*), it differs from all other taxa in this matrix.

*Branchiosaurus* used to be the name given to almost all skeletally immature temnospondyls from the Czech Pennsylvanian, as well as occasionally the Pennsylvanian and Cisuralian of other places. Building on and greatly expanding the work of *Milner & Sequeira (2003)*, *Werneburg (2012a)* tried to sort out this confusion and provided a preliminary redescription of the type species, *B. salamandroides* from the Czech Republic. This redescription confirms that *Branchiosaurus* differs in several features from all other branchiosaurids (see *Schoch & Milner, 2008*); some of these features are coded in the present matrix. *Branchiosaurus* is further necessary to test the phylogenetic position of *Tungussogyrinus* (see above). As *Werneburg (2012a)* noted, *B. fayoli* from Commentry and "*B.* aff. *fayoli*" from Montceau-les-Mines (both France) need to be redescribed before it can be assessed whether they form a clade with *B. salamandroides*; we have therefore not used any information from them.

### Stereospondylomorphs

Temnospondyl phylogeny is a vexing question; most mathematically possible permutations of Edopoidea (which may not be a clade; *Pawley, 2006*: chapter 6), *Capetus*, Dendrerpetidae + *Balanerpeton* (together mono- or paraphyletic), Dvinosauria, Eryopidae, Stereospondylomorpha and Dissorophoidea have been supported by recent phylogenetic analyses, and a few taxa have been found close to Eryopidae or as early stereospondylomorphs. This was, in part, discussed by *Ruta, Coates & Quicke (2003)* (as well as by *Pawley, 2006*: chapter 5; *Ruta, 2009*; *Schoch, 2013*; *Dilkes, 2015a*; and references in all five). It is strange, then, that the matrices of *Ruta, Coates & Quicke (2003)* and RC07 do not contain a single possible stereospondylomorph despite the presence of representatives of all other clades listed above. We have therefore added:

*Sclerocephalus*, variously considered an eryopid relative and/or a stereospondylomorph, is known from complete articulated skeletons and was redescribed by *Meckert (1993*; shoulder girdle and forelimb only*)* and *Schoch & Witzmann (2009a)*; D. M. compared some scores to the specimen MB.Am.1346 (a skull roof in dorsal view, a left lower jaw in medial and a right lower jaw in lateral view). Because *Sclerocephalus* is a weakly supported clade of four species (*Schoch & Witzmann, 2009a*)—ignoring the poorly known ?*S. stambergi*, whose phylogenetic position is even less clear (*Klembara & Steyer, 2012*), we have only considered the type species, *S. haeuseri*, which also happens to be the best-known one.

*Cheliderpeton vranyi*, the type species of *Cheliderpeton*, was redescribed by *Werneburg & Steyer (2002)*. Like *Sclerocephalus*, it has variously been considered an eryopid relative or a stereospondylomorph. The present matrix may be able to decide this question. Further, Werneburg considered *Cheliderpeton* an intasuchid, Steyer preferred to consider it an archegosaurid (*Werneburg & Steyer, 2002*); this question is testable in this matrix insofar as *Cheliderpeton* may form an exclusive clade with *Archegosaurus* (see below) or not. Growth series of the skull roof and a few other bones are known. We have not been able to use information from the putative second species, *Cheliderpeton lellbachae*, which "[i]n most respects [ . . . ] resembles *Sclerocephalus haeuseri*" and needs to be (re)described (*Schoch & Witzmann, 2009b*: 122).

*Glanochthon* was variously referred to *Archegosaurus*, *Actinodon*, *Sclerocephalus* or *Cheliderpeton* till *Werneburg & Steyer (2002)* recognized it as distinct from all four and *Schoch & Witzmann (2009b)* redescribed it. (For example, it appears as "*Cheliderpeton latirostre*" in the analysis by *Ruta, 2009*.) Found to be closer to *Archegosaurus* and Stereospondyli than *Cheliderpeton* and *Sclerocephalus* by *Schoch & Witzmann (2009b)*, the two species (*G. latirostris*, *G. angusta*; *Schoch & Witzmann, 2009b*) are known from growth series of many skulls (including lower jaws: *Boy, 1993*) and most of the postcranium except, apparently, the tail.

*Archegosaurus* is known from complete articulated skeletons and was redescribed by *Witzmann (2006*: head skeleton*)* and *Witzmann & Schoch (2006b*: postcranium*)*. Together with the other potential archegosaurids, all of which are much less completely known, it is considered a close relative of Stereospondyli. We have not used information from *Memonomenos*, which was often referred to *Archegosaurus* in earlier times but belongs elsewhere in the tree (*Schoch & Milner, 2000*: fig. 52; *Schoch & Witzmann, 2009b*).

*Platyoposaurus*, known from rich cranial and postcranial material (*Efremov, 1932*; *Konzhukova, 1955*; *Gubin, 1991*), is superficially very similar to *Archegosaurus* and was found as its sister-group by *Pardo, Small & Huttenlocker (2017*: fig. 2, S7*)*, but may actually be closer to Stereospondyli (*Schoch & Witzmann, 2009b*; *Pereira Pacheco et al., 2017*).

*Konzhukovia* has been allied to *Archegosaurus* and to Stereospondyli in various sources. *Gubin (1991)* illustrated (drawings 6, 15) and described the well preserved type skull of the type species, *K. vetusta*; we have not used other information except for two comments on the same species by *Pereira Pacheco et al. (2017*: app. 2*)*. First, further information on *K. vetusta* is not available (*Gubin (1991*: fig. 2 of plate II*)* only showed a small photo of an additional skull fragment of *K. vetusta* and merely mentioned the existence of a lower-jaw fragment and postcranial fragments); second, the referral of *K. tarda* and a fortiori *K. sangabrielensis* to *Konzhukovia* is doubtful (*Pereira Pacheco et al., 2017*, found *K. tarda* outside a clade consisting of *K. vetusta* and *Tryphosuchus paucidens*, and *K. sangabrielensis* outside a clade containing all three); third, *K. sangabrielensis* is only known from a partial skull that hardly seems to differ from *K. vetusta* in characters included in this matrix, while *Gubin (1991*: fig. 1 of plate II*)* only showed small photos of a skull of *K. tarda* and merely mentioned the existence of another skull. The diagnoses of the three species do not differ in characters that the present matrix contains, and the illustrations apparently do not either (except for the lateral-line grooves of *K. sangabrielensis*, which *K. vetusta* actually shares: *Pereira Pacheco et al., 2017*: app. 2).

*Lydekkerina* is within a few internodes of the origin of Stereospondyli according to all recent analyses. It is known from complete skeletons that are unusually well ossified for a stereospondyl. The postcranium was recently described by *Pawley & Warren (2005)* and *Hewison (2008)*, and the skull and lower jaw by *Jeannot, Damiani & Rubidge (2006)* and *Hewison (2007)*. *Shishkin, Rubidge & Kitching (1996)* showed that various proportions of the skull roof of *Lydekkerina* are paedomorphic, a hypothesis further supported for other characters by *Hewison (2007)*; this probably does not affect any

characters in this matrix. Miniaturization by progenesis may explain the well ossified endochondral bones.

*Australerpeton* was recently identified as a rhinesuchid stereospondyl (*Eltink et al., 2016*) after being thought to lie close to *Archegosaurus* by *Schoch & Milner (2000)*. Almost the entire skeleton is known (*Barberena, 1998*; *Dias & Schultz, 2003*; *Eltink & Langer, 2014*; *Eltink et al., 2016*).

### Chroniosuchians

Chroniosuchia is an enigmatic clade that has long been known from fragmentary remains. Only recently has more complete material been published, without, however, clarifying the phylogenetic position of the group. These animals have occasionally (*Laurin, 2000*; *Klembara, Clack & Čerňanský, 2010*) been considered embolomeres (thus anthracosaurs) mainly because of their embolomerous centra, but their confusing mosaic of character states is compatible with a number of other phylogenetic positions as well; indeed, *Klembara et al. (2014)* recovered them as the sister-group to a gephyrostegid-seymouriamorph clade, *Witzmann & Schoch (2017)* found them in three positions (next to Anthracosauria, *Silvanerpeton* or a clade containing *Gephyrostegus*, seymouriamorphs, amniotes and "microsaurs" among others), and *Schoch, Voigt & Buchwitz (2010)* and *Buchwitz et al. (2012)* even found them to be lepospondyls. Moreover, the two chroniosuchians we added could influence the positions of the embolomeres, *Gephyrostegus*, *Bruktererpeton*, *Silvanerpeton*, perhaps *Caerorhachis*, and possibly *Solenodonsaurus* in our tree.

*Chroniosaurus* is by far the most thoroughly described representative (*Clack & Klembara, 2009*; *Klembara, Clack & Čerňanský, 2010*) of the clade, with most of the skeleton being preserved. The specimens described and figured by *Clack & Klembara (2009)* did not preserve much of the lower jaws; *Clack & Klembara (2009*: 21*)* therefore stated that they had scored the reconstruction by *Ivachnenko & Tverdochlebova (1980)* in their phylogenetic analysis. This reconstruction (*Ivachnenko & Tverdochlebova, 1980*: drawing 16) would allow unambiguous scoring of our characters 147 (PSYM 1) and 155 (SPL 2), which *Clack & Klembara (2009)* scored as unknown, and 156 (SPL 3–4), which they scored as having our state 1 or 2. However, we have kept only the scores of *Clack & Klembara (2009)* because all of the drawings in *Ivachnenko & Tverdochlebova (1980)* are reconstructions which often do not indicate which parts are actually known; the characters in question concern those parts of the lower jaw that are most likely to be incompletely preserved or crisscrossed by fractures (as chroniosuchian dermal bones usually are). We should note that the reconstruction of the skull proper in all three views (dorsal, ventral, lateral: drawings 1a, 6, в) differs appreciably from those of *Clack & Klembara (2009)* and *Klembara, Clack & Čerňanský (2010)* in the shape and proportions of the skull, the course of some sutures, and the size, shape and relative positions of all openings, in two cases even their presence. The text (*Ivachnenko & Tverdochlebova, 1980*: 22) only briefly describes the chroniosuchian lower jaw in general, without mentioning which information is based on *Chroniosaurus* or *Chroniosuchus*, and does not refer to the characters in question.

Specimen drawings or photos of the lower jaw of any chroniosuchian have never been published to our knowledge, except for part of the labial side of that of *Chroniosaurus* in *Clack & Klembara (2009*: fig. 2*)* which does not provide further information.

*Bystrowiella*, originally described from rather limited postcranial material (*Witzmann, Schoch & Maisch, 2008*), has recently become another well-understood chroniosuchian following the discovery of cranial as well as further postcranial material (*Witzmann & Schoch, 2017*). As a bystrowianid, it shortens the potentially long branch of the chroniosuchid *Chroniosaurus*.

### "Microsaurs"

*Utaherpeton* was a surprising omission by RC07, given the facts that it is among the oldest known "microsaurs" and has been considered a basal "microsaur" (*Carroll, Bybee & Tidwell, 1991*; *Carroll & Chorn, 1995*; *Anderson, 2001*; *Anderson et al., 2008a*) or the sister-group of *Microbrachis* and thus a basal member of the "microsaur"-lysorophian-lissamphibian clade (*Vallin & Laurin, 2004*); in more recent analyses it fell out close to a "nectridean"-aïstopod-lysorophian-lissamphibian clade (*Marjanović & Laurin, 2009*: supplementary information 2) or as a "lepospondyl" outside the major groups (*Pardo, Small & Huttenlocker, 2017*: fig. S6, and references therein). Similarities to the "nectrideans" were already noted in the original description (*Carroll, Bybee & Tidwell, 1991*). A variety of interesting effects on lepospondyl intra- and perhaps even interrelationships could be expected from its addition to the present matrix based on the two descriptions (*Carroll, Bybee & Tidwell, 1991*; *Carroll & Chorn, 1995*). We have tentatively accepted the referral by *Carroll & Chorn (1995)* of the specimen they described to *Utaherpeton*. *Carroll & Chorn (1995)* did not offer any evidence for this referral other than the fact that their specimen is a "microsaur" from the same locality as *Utaherpeton*, and did not discuss or even mention the question. However, the unusual proportions of the hindlimb mentioned in the diagnosis by *Carroll, Bybee & Tidwell (1991)* do appear to fit, the difference in the shape of the interclavicle may well be ontogenetic, and while the figures do not make clear if either specimen has precisely 26 presacral vertebrae as described, they do not rule out that both at least have the same or almost the same count.

The "Goreville microsaur", too, is among the oldest known "microsaurs". Although it was deliberately not named by *Lombard & Bolt (1999)*, it differs from all other OTUs in this matrix; its somewhat unusual combination of plesio- and apomorphic character states (noted in its description) could change the topology of the tree. Eight badly preserved specimens, amounting to most of the skeleton, are known.

*Sparodus* is another "microsaur" that is not obviously deeply nested in one of the universally recognized clades, showing affinities to both Pantylidae and Gymnarthridae but having fewer apomorphies than the undisputed members of both (and being older than most of them). Personal observation by D. Marjanović of NHMW 1899/0003/0006 (Figs. 3 and 4), the referred specimen described by *Carroll (1988)*, shows that *Carroll's (1988*: fig. 1*)* drawing and reconstruction of the skull are probably optimistic, unless the

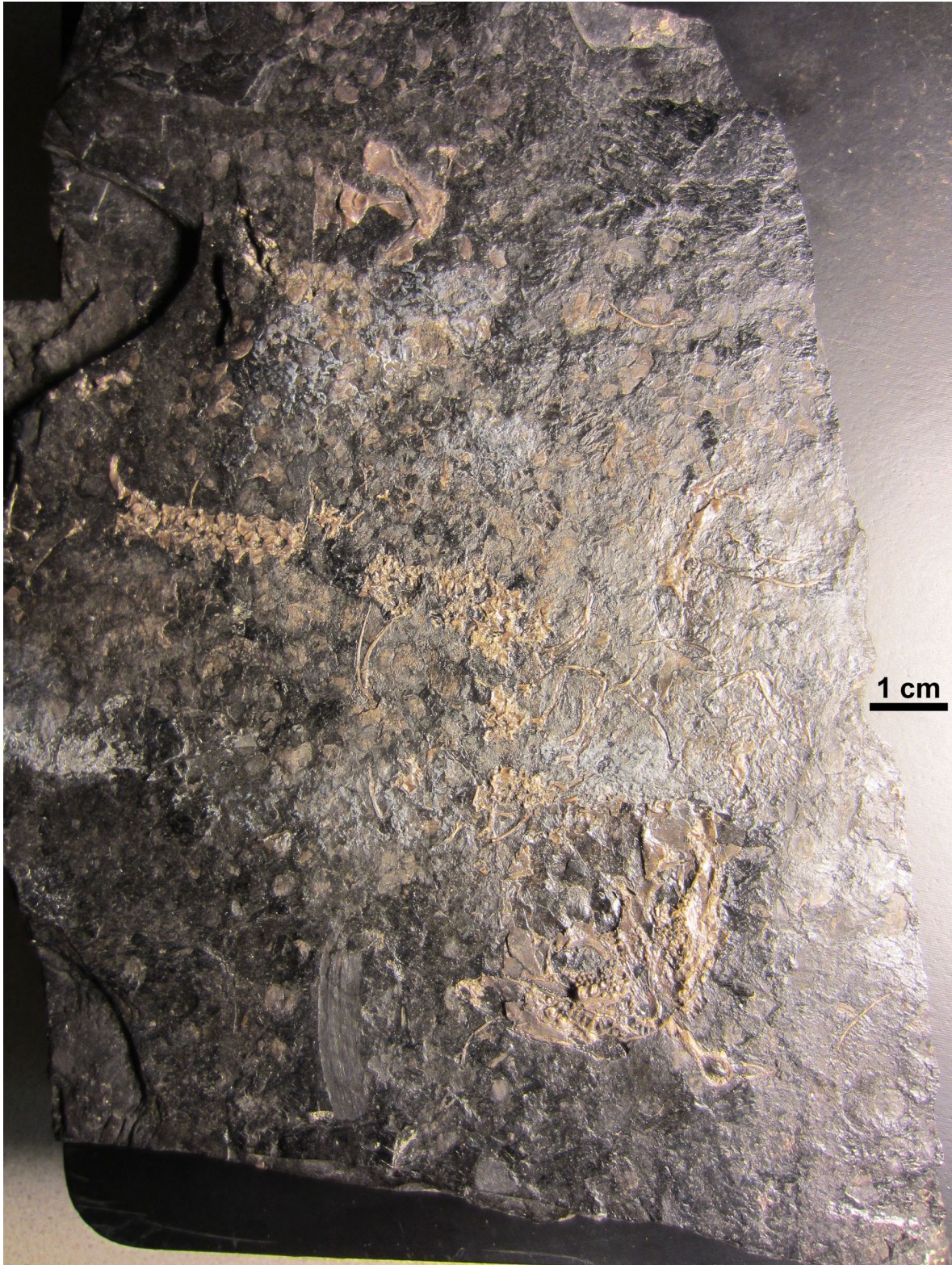

**Figure 3 NHMW 1899-0003-0006 (formerly 1899-III-6), a specimen referred to *Sparodus validus*.** For an interpretative drawing, see *Carroll (1988*: pl. XII*)*, which, however, makes the vertebrae appear much flatter than they are and omits the unusually well preserved scales. Some of the scales are only visible as striations in the matrix (see Fig. 4), which is usual in other "microsaurs" and in micromelerpetid temnospondyls; but most retain thick bone. Photo taken by D. M.

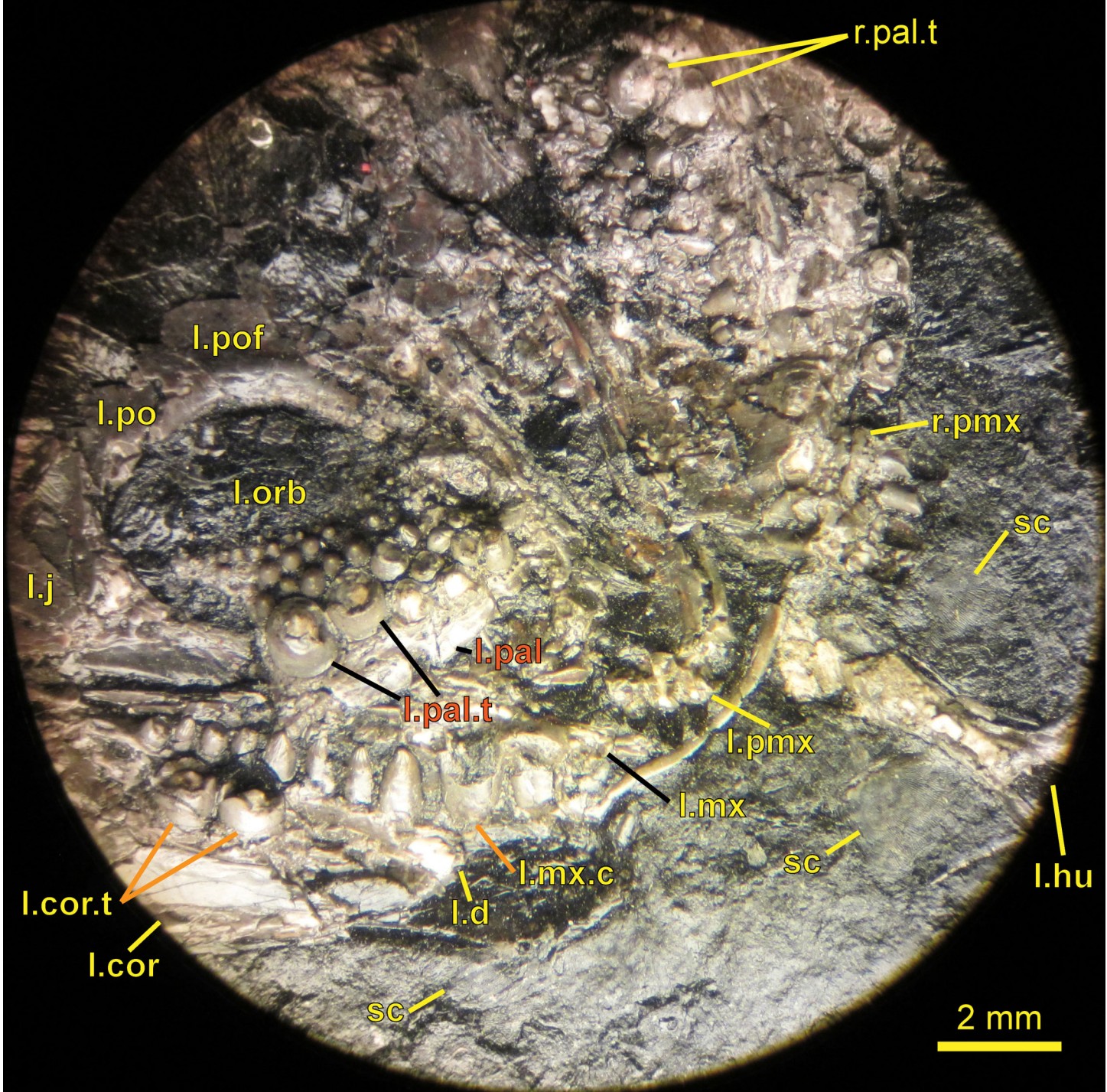

**Figure 4 Ventral view of the palate and lingual view of the left lower jaw of the *Sparodus* specimen shown in Fig. 3.** *Carroll (1988*: fig. 1A*)* provided an interpretative drawing of a latex cast. The photograph was taken by holding a digital camera to an ocular of a binocular microscope. Abbreviations: l, left; r, right; c, largest caniniform tooth (on the maxilla); t, tusk (one of two per bone); cor, unidentified coronoid; d, dentary; hu, humerus; j, jugal; mx, maxilla; orb, orbit; pal, palatine; pmx, premaxilla; po, postorbital; pof, postfrontal; sc, scales preserved as striations. Photo taken by D. M.

sutures are better visible on the latex cast Carroll used (which we have not seen) than on the fossil itself. However, this concerns very few characters.

*Carrolla* is a brachystelechid "microsaur" known only from a skull with lower jaw. Few differences between it and *Batropetes* (the brachystelechid that was already included in the matrix) are known, and few of those concern characters in the present matrix, but the redescription by *Maddin, Olori & Anderson (2011)* based on high-resolution computed tomography has made a large amount of information available. This is especially important because it allows us to better test the results by *Vallin & Laurin (2004)*, who found Brachystelechidae to be the sister-group of Lysorophia + Lissamphibia, as well as the results by *Maddin, Jenkins & Anderson (2012)*, who found the brachystelechids to be nested in a clade (Recumbirostra) that contains all the other "microsaurs" with a snout tip that protrudes beyond the premaxillary toothrow (state PREMAX 8(1) in the present matrix). D. M. has seen the only known specimen and was consequently able to score some characters that concern externally visible anatomy and were insufficiently addressed by *Maddin, Olori & Anderson (2011)*; these scores are described in App. S1.

*Crinodon* is a "microsaur" that is rich in plesiomorphies. Known from a well-preserved dermatocranium and partial lower jaws (CG78)—parts of the body that the present craniocentric matrix is well equipped to deal with—it may help shed light on "microsaurian" intra- and interrelationships. CG78 classified it as a tuditanid, but the two "tuditanids" in the present matrix (*Tuditanus* and *Asaphestera*) were not found as sister-groups by RC07, leaving us wondering whether the diagnostic characters of "Tuditanidae" might all be symplesiomorphies or homoplasies.

*Trihecaton* is a large "microsaur" known from an articulated skeleton that lacks the skull (apart from the maxilla, whose dorsal edge is not prepared), part of the lower jaw, most of the distal limbs and much of the tail—although further preparation would probably reveal additional bones. Presumably because most of the skull is unknown, *Trihecaton* has never featured in a phylogenetic hypothesis since CG78 gave it its own family in Tuditanomorpha; yet, it differs from all other OTUs in this matrix, so we see no reason to continue its exclusion from phylogenetic analyses. D. M. scored *Trihecaton* directly from the two known specimens (which most likely belong to the same individual); it is possible that additional preparation has been done since the description by CG78.

*Quasicaecilia*, known from an isolated skull without lower jaws (except an articular fragment), was redescribed by *Pardo, Szostakiwskyj & Anderson (2015)*. It is a very small brachystelechid that may contribute to resolving the relationships of brachystelechids, other burrowing "microsaurs", lysorophians and lissamphibians; additionally or alternatively, it could highlight the effects of miniaturized taxa on phylogenetic analyses.

*Llistrofus* is very similar to *Hapsidopareion* in the original taxon sample, but while *Bolt & Rieppel (2009)* found very few differences in their redescription of the skull, they also found very few synapomorphies. The postcranium is scored after CG78.

### Synapsids

The diadectomorphs (in the original and the expanded sample: *Diadectes*, *Orobates*, *Tseajaia*, *Limnoscelis*) are usually thought to lie on the amniote stem. It has, however, been proposed (*Berman, Sumida & Lombard, 1992*; *Berman, 2013*) that they are actual amniotes—in particular, the closest known relatives of Synapsida, thus part of Theropsida, the sister-group of Sauropsida (*Goodrich, 1916*). Surprisingly, all unquestioned amniotes in the matrix by RC07—*Petrolacosaurus*, *Paleothyris*, *Captorhinus*—are sauropsids; the lack of unambiguous theropsids means that the original taxon sample is unable to test the mentioned hypothesis, even though its large amount of non-amniote OTUs would have made it very well suited for such a purpose. The two synapsid OTUs we have added help to solve this problem. Furthermore, they might further influence diadectomorph monophyly—which was not found by *Ruta, Coates & Quicke (2003)*, *Pawley (2006)*, RC07, or even *Germain (2008a)*.

We have included *Eothyris*, *Oedaleops* and *Eocasea* as a single OTU called Caseasauria. *Eothyris* is known from an isolated, almost complete skull and lower jaw (*Reisz, Godfrey & Scott, 2009*); *Oedaleops* is known from a less complete skull and lower jaw, additional skull fragments, vertebrae, ribs, and girdle and limb elements (*Reisz, Godfrey & Scott, 2009*; *Sumida, Pelletier & Berman, 2014*; we follow the latter in not including the ilium tentatively referred by *Langston, 1965*); *Eocasea* is known from a skeleton lacking the shoulder girdle, the forelimbs, the distal part of the tail and most of the skull (*Reisz & Fröbisch, 2014*). The overlapping parts of these three taxa score almost identically in our matrix; where they disagree, we have simply scored polymorphism (and noted this in App. S1), because the relationships of these taxa to each other and to the herbivorous members of Caseidae, which are not considered here, remain unclear (*Reisz, Godfrey & Scott, 2009*; *Sumida, Pelletier & Berman, 2014*; *Reisz & Fröbisch, 2014*; *Brocklehurst, Romano & Fröbisch, 2016*; though see *Brocklehurst et al., 2016*). Some data of *Eothyris* not presented in the cited publications were filled in by D. M. by observation of the holotype; they agree with the conditions in *Oedaleops* (*Reisz, Godfrey & Scott, 2009*) and *Eocasea* (*Reisz & Fröbisch, 2014*). We did not consider *Callibrachion* or *Datheosaurus*, which are poorly preserved and incompletely ossified (*Spindler, Falconnet & Fröbisch, 2016*), or the recently (re)described *Vaughnictis* (*Brocklehurst et al., 2016*); they should be taken into account in future analyses, however, because all the analyses by *Brocklehurst et al. (2016*: fig. 10, 11*)* and *Spindler et al. (2018*: fig. 30*)* fully resolved the phylogenetic positions of all taxa mentioned in this paragraph.

Supplementing the Caseasauria OTU is *Archaeovenator*, the oldest known varanopid and sister-group to all other varanopids, known from a largely complete, mostly articulated skeleton described by *Reisz & Dilkes (2003)* and recently found nearly as close to the origin of Synapsida as Caseasauria (*Reisz & Fröbisch, 2014*: app. S5; *Brocklehurst et al., 2016*; *Spindler et al., 2018*).

### Lissamphibians

*Liaobatrachus* is an Early Cretaceous salientian known from plentiful, very well preserved articulated skeletons (*Dong et al., 2013*). Although it was found to form a trichotomy

with the two component clades of the crown-group of frogs, which means it is not very close to the root of Salientia, it may still influence the position of Salientia in particular and Lissamphibia in general, because few of the more rootward salientians are well preserved. *Dong et al. (2013)* lumped what had been described as five monospecific genera (*Liaobatrachus*, *Dalianbatrachus*, *Callobatrachus*, *Mesophryne*, *Yizhoubatrachus*) as three species of *Liaobatrachus* and added a fourth. *Chen et al. (2016)* and *Gao & Chen (2017)* preferred to consider *Liaobatrachus* and *Dalianbatrachus* nomina dubia and included the other three nominal genera as separate OTUs in their phylogenetic analyses; rather than finding them as a clade, they found them as a grade on the bombinanuran stem just within the anuran crown-group. However, there are no other crown-group frogs in our matrix; if the mentioned analyses are correct, our *Liaobatrachus* OTU merely bears the wrong name and contains too much missing data. Further, *Gao & Chen (2017)* found little if any support in this part of the tree, while *Chen et al. (2016)* did not perform any robustness analyses; both analyses contain less than twice as many characters as taxa and were focused on more highly nested crown-group frogs; the arrangement of *Mesophryne*, *Yizhoubatrachus* and *Callobatrachus* as an uninterrupted grade differs starkly from the positions found in earlier literature (summarized in *Marjanović & Laurin, 2014*: fig. 3), which were widely dispersed through the salientian stem and the base of the crown; and both analyses peculiarly lack *L. zhaoi Dong et al., 2013*, which is preserved in three dimensions and therefore is the source of most of the scores of our *Liaobatrachus* OTU.

*Beiyanerpeton* is a neotenic salamandroid of apparently Late Jurassic age (*Gao & Shubin, 2012*), in any case older than its Early Cretaceous relative *Valdotriton* which was already included in the matrix. It is the only known lissamphibian, extant or extinct, to possess grooves for the lateral-line organ on the skull. Among the lissamphibians in the matrix, it is the only one known to possess opisthotics that are not fused to the prootics or exoccipitals (or otherwise absent). These factors make it important enough to add it to this matrix (which contains both of these characters), even though its paedomorphosis makes some characters difficult to interpret for purposes of phylogenetics (see *Gao & Shubin, 2012*: supplementary information).

The even older (Late or Middle Jurassic) *Pangerpeton* was found by *Jia & Gao (2016a)* to be a stem-hynobiid salamander, similarly close to the root of Urodela as *Beiyanerpeton* and *Valdotriton*. Several features of the only known skeleton (which lacks most of the tail, but is accompanied by a body outline) suggest that it had undergone metamorphosis, although it remained aquatic (*Wang & Evans, 2006*). Indeed, in having the jaw joints and the occiput in the same vertical transverse plane, it is more peramorphic than *Beiyanerpeton*, *Valdotriton* and even *Karaurus*. The skeleton is preserved in ventral view, so that most of the skull roof remains unknown. Still, *Pangerpeton* does not score redundantly with any other OTU in this matrix.

Even more peramorphic than *Pangerpeton* is *Chelotriton*. Scoring of this late Oligocene pleurodeline salamandrid (newt) is based mostly on MB.Am.45 (*Marjanović & Witzmann, 2015*). This specimen exhibits peramorphic traits that are otherwise rare or even entirely unknown in salamanders (in a few cases even in lissamphibians as a

whole), such as very long ribs, some of which are ventrally curved, and jaw joints that lie far caudal of the occiput. In 1981, it was misidentified as a "Dissorophid temnospondyl" "?Amphibamus" on a label. MB.Am.45 is indeed strikingly amphibamid-like in appearance, with some similarity also to certain diplocaulid "nectrideans." We have added *Chelotriton* to the matrix to see the effects, if any, of its unexpected peramorphic features on caudate and lissamphibian intra- and interrelationships. For this purpose, we took MB.Am.45 (scored mostly from the silicone cast MB.Am.45.3) at face value wherever possible rather than scoring polymorphism; missing data were filled in from the less extremely peramorphic *Chelotriton* individuals described and figured by *Roček & Wuttke (2010)* and *Schoch, Poschmann & Kupfer (2015)*.

### Seymouriamorph

*Karpinskiosaurus* may help clarify seymouriamorph phylogeny and relationships. The skulls of the type species, *K. secundus*, including an adult, non-paedomorphic one, were redescribed by *Klembara (2011)*; according to the same work, the type specimen of *K. ultimus* requires revision, so this species is not considered here. Much of the axial skeleton was illustrated and commented on by *Bystrow (1944)*.

### Undoubted colosteids

*Deltaherpeton* is one of the oldest known colosteids, represented by a skull roof and lower jaw (*Bolt & Lombard, 2001*, *2010*). It shows a few plesiomorphies as well as several unique features that are absent or not preserved in *Colosteus* and *Greererpeton*. *Bolt & Lombard (2010)* drew attention to this fact, but then undertook only a comparison instead of a phylogenetic analysis to justify their conclusion that "[t]he morphology of *Deltaherpeton* and the revised data presented for colosteids do not clarify the relationship of colosteids to other early tetrapods" (*Bolt & Lombard, 2010*: abstract). Indeed, there are characters that indicate a very rootward position of Colosteidae (as found by RC07), but others support a position more crownward than those of Whatcheeriidae and *Crassigyrinus* (as found by *Ruta, 2009*, if we assume that the temnospondyls would be close to or within the crown), and a few have led to the traditional concept of colosteids as temnospondyls (as the sister-group to all other members of that clade).

    *Pholidogaster* has been known for over 150 years; it was redescribed by *Romer (1964)* and, after additional preparation of the skull and shoulder girdle, *Panchen (1975)*. Its identity as a colosteid has not been questioned since the latter publication; and yet, to the best of our knowledge, it has never been included in a phylogenetic analysis. Although it is generally less well known than *Colosteus*, *Greererpeton* and *Deltaherpeton*, it scores differently from all three in our matrix; unlike in any known specimen of the other three colosteids (*Bolt & Lombard, 2010*), the septomaxilla and parts of its surroundings are preserved. Its age, close to that of *Deltaherpeton*, makes it a particularly interesting potential source of information on the phylogeny and affinities of Colosteidae.

### Anthracosaurs

*Holmes & Carroll (2010)* described an unusually small but adult (or nearly so) incomplete skeleton of an anthracosaur (embolomere) from Joggins (Late Carboniferous). Unfortunately, NSM 994 GF 1.1 cannot be assigned to a named genus, because it does not overlap with diagnostic parts of the only known anthracosaur from the same site, *Calligenethlon*; however, because it is considerably more complete than any specimen which can be assigned to *Calligenethlon* (all of which are slightly smaller and possibly less mature), we have added it to this matrix.

*Palaeoherpeton* is less well known than the originally included anthracosaurs, but the material was very thoroughly described by *Panchen (1964*, with a correction 1972*)* and scores differently from any other OTU in our matrix.

*Neopteroplax*, an anthracosaur described by *Romer (1963)* from a largely complete skull with lower jaw, has been mostly ignored ever since. In spite of having a skull shape quite similar to that of *Pholiderpeton attheyi*, however, *Neopteroplax* shows a number of features that are unusual for embolomeres; it makes an interesting addition to the question of anthracosaur phylogeny and affinities. We consider the referral of isolated jaw fragments and centra (not clear if pleuro- or intercentra) from a younger, very distant site to *Neopteroplax* (*Romer, 1963*: 451) highly doubtful; for the centra in particular it requires a rather long chain of inference from a single potentially diagnostic feature (the distance between the teeth). The jaw fragments would not add any information to our scoring of *Neopteroplax*; we have ignored the centra to avoid creating a chimera.

### Aïstopods

The skull of *Coloraderpeton* was redescribed by *Anderson (2003a)* and reinterpreted from μCT data by *Pardo et al. (2017)*. Like *Lethiscus* (*Pardo et al., 2017*), it preserves unexpected plesiomorphies; among those not known in *Lethiscus* are an at least mostly enclosed mandibular lateral-line canal and a bone that is most parsimoniously interpreted as a preopercular. Unfortunately, information on the postcranium is very scarce in publications (*Carroll, 1998b*; *Anderson, 2003a*; *Anderson, Carroll & Rowe, 2003*), and D. M. only briefly saw the specimens (at the CM) years before we decided to add *Coloraderpeton* to the matrix.

*Pseudophlegethontia* (*Anderson, 2003b*) may fill the morphological gap between *Phlegethontia* and the other aïstopods; we have reinterpreted a few characters (as detailed in App. S1) in the light of *Lethiscus* and *Coloraderpeton* (*Pardo et al., 2017*).

### Devonian enigmas

*Ruta, Coates & Quicke (2003*: 262*)* listed *Metaxygnathus* among Devonian taxa that are "known mainly from lower jaw rami and/or incomplete postcranial remains" and "are omitted" from their matrix, presumably because they are so incompletely known. However, the isolated lower jaw ramus called *Metaxygnathus* differs from all other lower jaws (isolated or not) in this matrix. Indeed, one character, ANG 3, seems to have been deliberately included to potentially hold *Acanthostega* and *Metaxygnathus* together in

"future, expanded versions of our matrix" (RC07: 103). Our source was the redescription by *Ahlberg & Clack (1998)*.

*Ymeria* is an "*Ichthyostega*-grade" animal known from premaxillae, a palate in dorsal view and lower jaws (plus shoulder girdle remains that are too fragmentary to be scored in this matrix). It was found one node rootward of *Ichthyostega* in those analyses by *Clack et al. (2012a)* and *Sookias, Böhmer & Clack (2014)* that had enough resolution to tell.

*Densignathus*, much like *Metaxygnathus*, is known only from lower-jaw material (*Daeschler, 2000*; *Ahlberg, Friedman & Blom, 2005*) and was listed among the excluded taxa by *Ruta, Coates & Quicke (2003*: 262*)*, but differs from all other lower jaws in the matrix.

*Elginerpeton* has been described, from successively greater amounts of isolated material and jaw fragments, as a Devonian animal close to the origin of limbs (*Ahlberg, 1995*, *1998*; *Ahlberg & Clack, 1998*; *Ahlberg, Friedman & Blom, 2005*). It could serve to establish character polarities for the other Devonian OTUs. Following *Ahlberg, Friedman & Blom (2005)* and *Ahlberg (2011)*, we have included the postorbital bone and the postcranial material described by *Ahlberg (1998)*, except for the supposed humerus.

### Mississippian enigmas

*Sigournea* is known only from an isolated lower jaw (*Bolt & Lombard, 2006*). Although similarities to the baphetoids, especially *Spathicephalus* (see below), and to the equally mysterious jaw material called *Doragnathus* (see below) have been noted (*Bolt & Lombard, 2006*; *Milner, Milner & Walsh, 2009*), these conjectures have only been tested twice in a phylogenetic analysis. *Sookias, Böhmer & Clack (2014)* found it in the whatcheeriid region of the tree (more rootward than *Baphetes*, the only included baphetoid). That analysis did not include *Doragnathus* and had a generally insufficient taxon and, likely, character sample to address this question. *Clack et al. (2016)*, who included *Doragnathus*, found *Sigournea* as a colosteid (unweighted parsimony) or in several places in the grade between the more rootward whatcheeriids and the more crownward colosteids (Bayesian inference, as well as parsimony with various degrees of reweighting). Our matrix contains enough lower-jaw and tooth characters to show that *Sigournea* is distinct from any other taxon for which more than a few such characters can be scored.

*Doragnathus* is another mysterious animal known only from lower jaws and parts of upper jaws. It has only once before (*Clack et al., 2016*) been included in a phylogenetic analysis, even though it was described quite some time ago (*Smithson, 1980*), and even though it differs from *Sigournea*, *Spathicephalus* (see below) and all other potential relatives in characters that are included in the present matrix. As recommended by *Smithson & Clack (2013)*, we have not scored the isolated postcranial material that was found at the same site and may or may not belong to the same taxon (described and illustrated by *Smithson & Clack, 2013*).

*Spathicephalus* is a baphetoid known from skull and lower-jaw material with several strange autapomorphies (*Baird, 1962*; *Beaumont & Smithson, 1998*; *Smithson et al., 2017*). Still, in a few characters it may be more plesiomorphic than at least *Baphetes* and

*Megalocephalus*, which may help test baphetoid relationships (see *Milner, Milner & Walsh, 2009*); it may also influence the positions of *Sigournea* and *Doragnathus*.

MB.Am.1441, the "St. Louis tetrapod", is a natural mold of a skull with lower jaws in ventral view that was described by *Clack et al. (2012b)*. The description noted several similarities to colosteids, but also to temnospondyls, and ultimately did not commit to either hypothesis of relationships; the phylogenetic analysis in that paper, which used a very small taxon sample and a rather repetitive character sample, found it to be the sister-group of a novel (*Ptyonius*, (*Adelogyrinus*, *Greererpeton*)) clade, which together with the "St. Louis tetrapod" formed the sister-group to the only two included temnospondyls—an unusual arrangement as well. The matrix presented here may be better suited to resolving the relationships of MB.Am.1441, which D. M. was able to study firsthand (both the natural mold, MB.Am.1441.1, and the silicone cast, MB.Am.1441.2). Despite the small number of characters that can be scored, the "St. Louis tetrapod" differs from all other OTUs in this matrix.

*Perittodus*, *Diploradus* and *Aytonerpeton* are three of the many new taxa that were briefly presented by *Clack et al. (2016)*. Their middle Tournaisian age lies in Romer's Gap, which is a temporal as well as morphological gap between the Devonian and the post-Tournaisian limbed vertebrates. *Perittodus* is mostly known from a very plesiomorphic, at most "*Ichthyostega*-grade" lower jaw; *Diploradus* has a lower jaw reminiscent of *Sigournea* and *Doragnathus*, but also preserves fragments of the rest of the skeleton; *Aytonerpeton* is known from scattered postcrania as well as a snout which resembles the "St. Louis tetrapod" and preserves, as we discuss in App. S1 (ch. 6, 7), the youngest anterior tectal identified in any limbed vertebrate.

*Casineria* is generally thought to be close to the origin of amniotes or at least seymouriamorphs (*Paton, Smithson & Clack, 1999*; *Clack et al., 2012b*, *2016*; *Witzmann & Schoch, 2017*: fig. 16). In stark contrast, *Pawley (2006*: 207*)* reported that it scored identically to *Caerorhachis* in her matrix, apart from the (quite different) distribution of missing data, and considered them "indistinguishable based on the available evidence" (*Pawley, 2006*: 195, 239), pointing out further that the supposedly amniote-like features of *Casineria* have a wider distribution. In her phylogenetic analysis, it came out as a temnospondyl (in a trichotomy with *Caerorhachis* and a clade formed by all other temnospondyls). D. M. has seen the only known specimen, a headless and largely tailless, mostly articulated skeleton. We present new photos (Figs. 5–7), interpretations and comparisons to the description and redescription of *Caerorhachis* (*Holmes & Carroll, 1977*; *Ruta, Milner & Coates, 2002*) in the Discussion (The interrelationships of Anthracosauria, *Silvanerpeton*, *Caerorhachis*, Gephyrostegidae, *Casineria* and Temnospondyli); notably, we identify what seems to be a postbranchial lamina on the cleithrum (Fig. 7).

### Pennsylvanian enigma

The "Parrsboro jaw" is an incomplete impression of a lower jaw of Pennsylvanian age, consistently called NSM 987GF65 in the original description (*Godfrey & Holmes, 1989*) but NSM 987GH65 in the partial redescription (*Sookias, Böhmer & Clack, 2014*). It has a unique combination of characters and differs from all other OTUs in this matrix, which may therefore have the potential to resolve the relationships of this mysterious

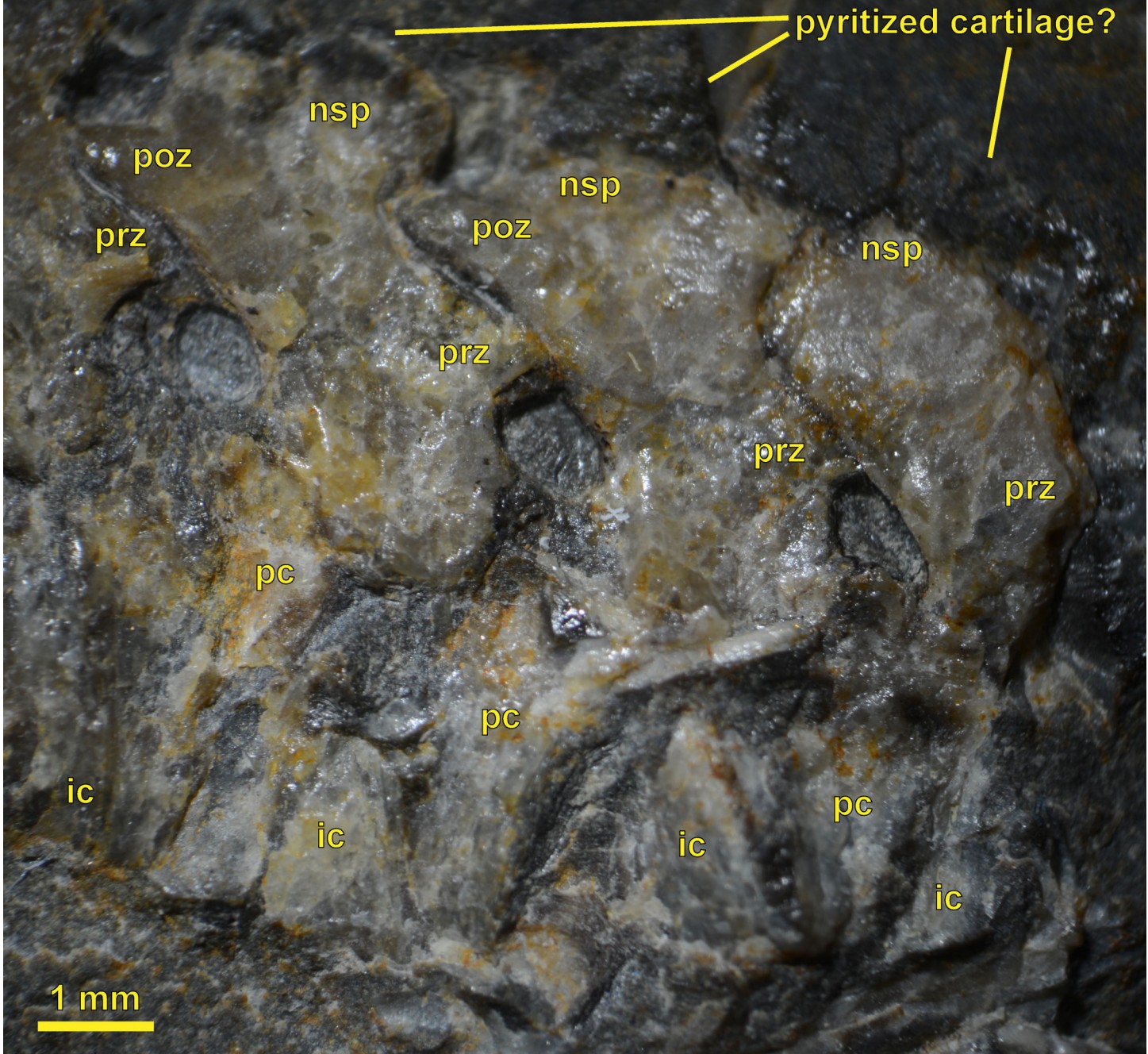

**Figure 5 Caudal trunk vertebrae of NMS G 1993.54.1cp, the holotype of *Casineria kiddi* (counterpart plate), in partly right lateral, partly interior view.** The neural arches end dorsally in an ossification front marked by black material (an iron sulfide?) just dorsal to the level of the postzygapophyses, indicating that the individual was immature; the pleuro- and intercentra are reminiscent of those of the anthracosaur *Proterogyrinus* and the temnospondyl *Neldasaurus* (*Chase, 1965*), although their precise shapes are difficult to determine because the vertebrae are split lengthwise and probably recrystallized. Abbreviations: ic, intercentrum; nsp, neural spine; pc, pleurocentrum; poz, postzygapophysis; prz, prezygapophysis. The scale bar is approximate. Photo taken by D. M.

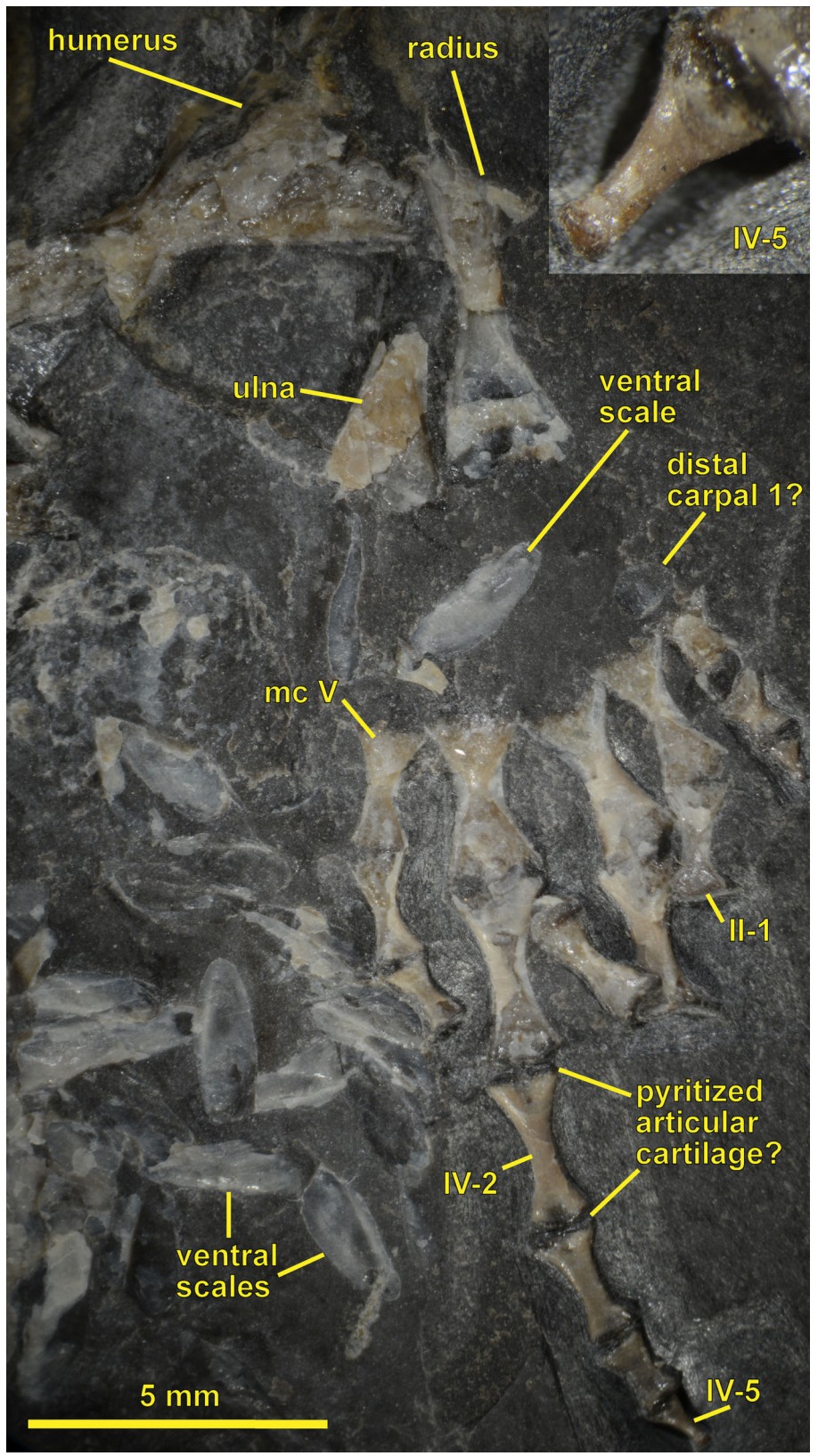

**Figure 6 Left forelimb of NMS G 1993.54.1, the holotype of *Casineria kiddi*, in plantar view.** The carpus is unossified except for what may be an incipiently mineralized distal carpal 1. The inset shows

phalanx IV-5 at a higher magnification (photographed from a different angle; length approximately 1 mm); note that the tip, although curved plantarly, is blunt, unlike in amniotes and contrary to *Paton, Smithson & Clack (1999)*. Abbreviation: mc, metacarpal. The scale bar is approximate. Photos taken by D. M.                                           

fragment that has warranted redescription but defied the phylogenetic analysis of *Sookias, Böhmer & Clack (2014*: fig. 5A*)*.

### Taxa that were not added

We have not added taxa that likely could not be handled by the character sample of the present matrix (e.g., too deeply nested taxa, such as any further baphetoids, cochleosaurids, eryopids, dvinosaurs, dissorophids, stereospondylomorphs, "microsaurs", amniotes or lissamphibians beyond those we did add, or taxa too far outside the clade of limbed vertebrates). Still, a few taxa would have been intuitive candidates for addition, but we have left them out for the following reasons:

Because of the state of preparation of the specimens, the original description of the enigmatic and interesting "anthracosaur" *Eldeceeon* (*Smithson, 1994*) is very preliminary and contains little information; following the discovery of additional specimens, *Eldeceeon* is being redescribed according to *Ruta & Clack (2006)* and *Clack & Milner (2015)*.

Although *Madygenerpeton* is only the second chroniosuchian for which a thorough description exists, we have not included it because it seems to lie at the tip of a long branch. According to the oldest phylogenetic analysis of Chroniosuchia (*Schoch, Voigt & Buchwitz, 2010*), it is highly nested within that clade as the sister-group of *Chroniosaurus*, so that its seemingly plesiomorphic aquatic adaptations are probably reversals and would likely, perhaps together with the many other autapomorphies that *Madygenerpeton* has, distort the tree in general and the position of *Chroniosaurus* in particular. Indeed, *Madygenerpeton* lacks traces of the lateral-line organ (*Schoch, Voigt & Buchwitz, 2010*; D. Marjanović, pers. obs. 2014) which would be expected to be present in a primarily aquatic vertebrate; the dorsal rims of the eye sockets are raised high above the skull table, implying a crocodile-like predator that looked for terrestrial prey rather than staying underwater, an interpretation further corroborated by the nostrils which are apparently raised above the roof of the snout. In temnospondyls, for example, the nostrils are always well below eye level, and, with the borderline exception of *Glaukerpeton* (*Werneburg & Berman, 2012*), the eye sockets are never noticeably raised (D. Marjanović, pers. obs., and see below). The second analysis that included *Madygenerpeton* (*Buchwitz et al., 2012*) instead found it as the sister-group to all other chroniosuchians, but—much like its predecessor—had a very small sample of outgroups. Conversely, the present matrix undersamples characters relevant to chroniosuchian phylogeny: the osteoderms and the antorbital fenestra, for example, are not coded. Indeed, a test run failed to find *Madygenerpeton* as a chroniosuchian. Only osteoderms and the mentioned skull roof (with distorted parts of the palate) are known of *Madygenerpeton*.

The temnospondyl *Parioxys ferricolus*, variously connected to *Eryops* and/or Dissorophoidea and more specifically found as the sister-group of *Iberospondylus* by *Pawley (2006*: chapter 5*)*, is most likely a chimeric taxon (J. Pardo, pers. comm. 2012).

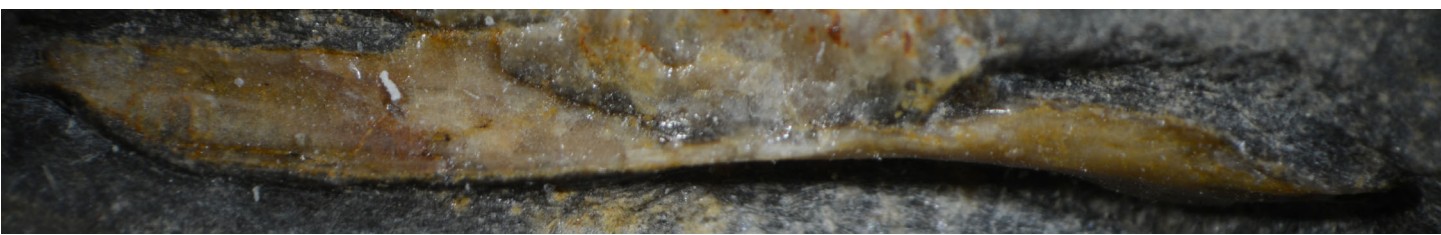

**Figure 7 Right cleithrum of NMS G 1993.54.1, the holotype of *Casineria kiddi*, in caudal view, showing postbranchial lamina (compare fig. 70-2.4 of *Pawley, 2006*).** Ventral end on the left. Total length about 11.5 mm. Photo taken by D. M.

The lectotype (AMNH 4309) is covered by a crust that would be difficult to prepare away and completely obscures the sutures as well as most other details (D. Marjanović, pers. obs. 2013; *Schoch & Milner, 2014*: 77), and referred specimens (AMNH 2445 and almost all of the MCZ material) are being restudied by M. Ruta (D. Marjanović, pers. obs. of loan slips 2013). "Preparation of USNM material has revealed many similarities to *Cacops*" (a dissorophid like *Broiliellus*; *Schoch & Milner, 2014*: 77); it remains to be seen if the same will hold true of the lectotype. AMNH 7118, the type and only known specimen of *P. bolli*, consists only of encrusted (D. Marjanović, pers. obs. 2013) assorted postcrania, and whether it (or any other material currently assigned to *Parioxys*) can be referred to *Parioxys* awaits investigation.

*Onchiodon* is well known, but so similar to the even better known *Eryops* (*Werneburg, 2007b*) that it would most likely not add any new information to this matrix. Pretty much the same holds for the less well known *Glaukerpeton* (*Werneburg & Berman, 2012*). The rest of Eryopidae—and also the diversity within *Eryops*—is very poorly understood and currently undergoing revision (*Werneburg, 2007b*, *2012b*; *Werneburg & Berman, 2012*; *Schoch & Milner, 2014*; *Rasmussen, Huttenlocker & Irmis, 2016*).

Comparison to character lists in *Fröbisch & Reisz (2012)*, *Schoch (2012)*, *Maddin et al. (2013b)* and *Holmes, Berman & Anderson (2013)* shows that many characters relevant to dissorophoid phylogeny are lacking from this matrix. Adding these characters and a better and more even sample of dissorophoid temnospondyls—micromelerpetids (in our matrix represented only by *Micromelerpeton*), dissorophids (*Broiliellus brevis*), trematopids (original sample: *Ecolsonia*, *Acheloma*, *Phonerpeton*; added: *Mordex*) and amphibamids (*Amphibamus*, *Eoscopus*, *Doleserpeton*, *Platyrhinops*, three "branchiosaurids"; added: *Gerobatrachus*, *Micropholis*, two more "branchiosaurids")—will be part of future work. As *Schoch (2012)*, *Werneburg (2012a)*, *Maddin et al. (2013b)* and *Holmes, Berman & Anderson (2013)* pointed out, some of these taxa need to be (or are being) redescribed, and some material probably even needs additional preparation, which is in many cases as difficult to do as for *Parioxys* (see above). In particular, as mentioned above, we have not used information from species currently referred to *Broiliellus* other than *B. brevis*.

*Kirktonecta* (*Clack, 2011a*), "the oldest known microsaur", is so poorly preserved (and split through the bone like *Eldeceeon*) that it would not differ from several other OTUs in this matrix if we added it.

The characters in which *Czatkobatrachus* differs from *Triadobatrachus* (*Evans & Borsuk-Białynicka, 2009*) are not represented in this matrix. It would be possible to

score *Czatkobatrachus* for a few characters that are unknown in *Triadobatrachus*, but in all of these, *Czatkobatrachus* has the state expected (under all hypotheses) for the ancestral lissamphibian and the ancestral salientian, so it is not relevant here.

"Several other taxa [of Mesozoic salamanders] (e.g., *Laccotriton*, *Sinerpeton*, and *Jeholotriton*) are excluded from this analysis because they are anatomically uncertain and are currently under taxonomic revision" (*Gao & Shubin, 2012*: supplementary information: 3). This revision is ongoing (*Jia & Gao, 2016b*).

Adding *Chinlestegophis* (*Pardo, Small & Huttenlocker, 2017*) to our matrix would currently be pointless in the absence of stereospondyls (other than *Lydekkerina* and *Australerpeton* in the expanded taxon sample) and characters pertinent to stereospondyl phylogeny.

## Phylogenetic analyses

### Maximum-parsimony analyses

These analyses were conducted in PAUP* 4.0b10 (*Swofford, 2003*) on a desktop computer with an Intel® Core™2 Duo processor (2.67 GHz) and 3.87 GB of usable RAM. Test runs in PAUP* 4.0a158 (*Swofford, 2017*) were sometimes slower and sometimes found fewer MPTs. Alpha versions of PAUP* 4.0 up to a158 were, however, used to generate initial tree figures for further processing from the results of all phylogenetic and robustness analyses.

We did not use any additional weighting procedures (such as reweighting, see Discussion: Reweighting and equal weighting in comparison); each character contains at least one state transition that costs one step, and there are no state transitions that cost less.

Six analyses (Table 1) of our modified matrix were conducted with (Analyses R4–R6) and without the added taxa (Analyses R1–R3), and with and without backbone constraints. The first constraint (Fig. 8A), used in Analyses R2 and R5, forced the dissorophoid temnospondyl *Doleserpeton* to be closer to the three salientians (*Triadobatrachus*, *Notobatrachus*, *Vieraella*; their monophyly with respect to *Doleserpeton* was specified, but not their relationships to each other) than the lysorophian lepospondyl *Brachydectes*, making the LH impossible but allowing both the TH and the PH. The second constraint (Fig. 8B), used in Analyses R3 and R6, only allowed the PH by additionally forcing the gymnophionomorph *Eocaecilia* to be closer to *Brachydectes* than to *Doleserpeton*. Both constraints contain *Eusthenopteron* in the outgroup position. In order to find all optimal islands and all optimal trees within each island, each heuristic search used 10,000 addition-sequence replicates (with random addition sequences), each of which was restricted to 50 million rearrangements by tree bisection and reconnection (25 million turned out to be too few to find all MPTs in some cases). This limit was hit in almost all replicates of the analyses with added taxa and close to two thirds of the replicates in the analyses without added taxa. Analysis R1 took 25:58:25 of calculation time, Analysis R6 lasted for 47:25:19.

We also reanalyzed (Analysis O1; Table 1) the original, unmodified matrix of RC07 without any constraints to test whether their procedure for reducing calculation time (the "parsimony ratchet"; see *Ruta, Coates & Quicke, 2003*) had overlooked any MPTs. This was deemed important because, when *Skutschas & Gubin (2012)* applied
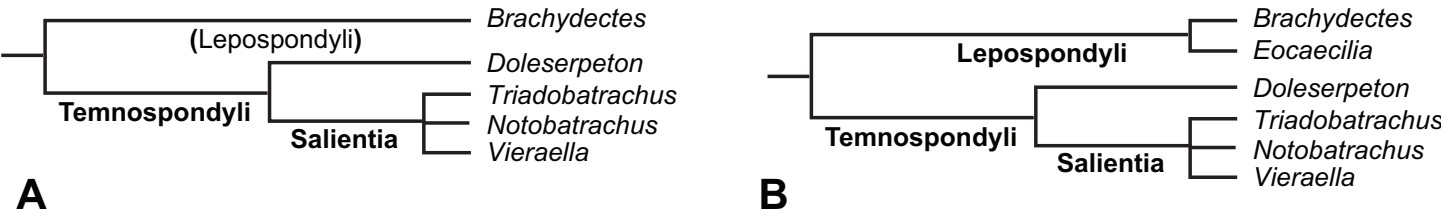

**Figure 8 Topological constraints used in Analyses O2, O3, R2, R3, R5 and R6 (see Table 1).** These are backbone constraints: the OTUs not mentioned in the constraints were free to be positioned anywhere within the constraint tree (including at its root), and the polytomies were allowed to be resolved in all mathematically possible ways. (A) Constraint against the LH, allowing both TH and PH; used in Analyses R2 and R5 (both of which found the TH). Whether Lepospondyli is extinct depends on the unconstrained positions of the caudates. With the positions of *Brachydectes* and *Doleserpeton* exchanged, this constraint instead enforces the LH and was used in Analysis O2. (B) Constraint for the PH, used in Analyses O3, R3 and R6.

the same procedure to their own matrix which was small enough for a branch-and-bound analysis, it found a single tree that was no less than 35 steps longer than the MPTs found by branch-and-bound—a method which is guaranteed to find all MPTs. (To our surprise, *Skutschas & Gubin (2012)* reported the length difference without commenting on it and without discarding the crassly suboptimal tree or offering a reason why it should be retained in consideration.) Calculation time with 5,000 addition-sequence replicates and up to a hundred million rearrangements per replicate was 21:48:07.

On p. 85, RC07 reported that constraining their analysis to find a topology congruent with that of *Laurin (1998a)*—a version of the LH—required "the addition of a mere nine steps" and yielded 60 suboptimal trees. On the next page, however, they wrote: "Using our own data, the 720 suboptimal trees placing lissamphibians with lepospondyls are 15 steps longer than the MPTs from the original analysis." (RC07 did not publish or describe the constraint they used to obtain this result.) To resolve this contradiction, we reanalyzed (Analysis O2; Table 1) the original, unmodified matrix of RC07 under a backbone constraint that required the salientians to be closer to *Brachydectes* than to *Doleserpeton*. *Eusthenopteron* was again included in the constraint in the outgroup position. The salientians are *Triadobatrachus*, *Notobatrachus* and *Vieraella*; their monophyly with respect to *Doleserpeton* was specified, but not their relationships to each other (Fig. 8A with the positions of *Brachydectes* and *Doleserpeton* exchanged). Calculation time was 23:48:00.

For completeness and to be able optimize characters on a tree supporting the PH, we reanalyzed (Analysis O3; Table 1) the same matrix under a backbone constraint that required *Eocaecilia* to be closer to *Brachydectes* than to *Doleserpeton* and the salientians (Fig. 8B). Calculation time was 23:22:13.

### Robustness analyses

José Grau (then Museum für Naturkunde, Berlin) kindly performed two parsimony bootstrap analyses on our modified matrix (Table 1), one each for the original (Analysis B1) and for the expanded taxon sample (Analysis B2). Both were carried out on an Intel® Xeon® CPU E5-4667 v4 (2.20 GHz, 512 GB of available RAM) using PAUP* 4.0a158 for Unix/Linux (*Swofford, 2017*), and used 1,000 bootstrap replicates consisting of 500 addition-sequence replicates which were limited to 50 million rearrangements each.

To keep this within a reasonable timeframe, J. Grau divided each analysis into 100 runs of 10 bootstrap replicates, which were run on a total of 50 processors by means of GNU parallel v20130522, followed by concatenation and calculating the majority-rule consensus to obtain the final bootstrap tree. To maximize the amount of visible information, we included not only clades with bootstrap values of 0.5 and above in the bootstrap trees, but also clades with lower bootstrap values that do not contradict any of those with higher ones (PAUP* setting: contree all/majrule=yes le50=yes; equivalent to bootstrap keepall=yes for a non-parallel analysis).

The total running time for each bootstrap run was between 27 and 38 h; the consensus took around 40 min to compute. Each bootstrap run produced between 30,111 and 114,768 trees; the consensus was calculated from a total of 6,870,699 trees.

In order to test whether the differences between MPTs from different analyses with the same taxon sample and the same reversibility coding are statistically significant, we ran the tests implemented in PAUP*: a Kishino–Hasegawa test, a Templeton test and a winning-sites test. The *p*-value we report for each is the probability (Kishino–Hasegawa) or approximate probability (Templeton, winning sites) of finding a more extreme test statistic; the null hypothesis is that the differences between the two tested trees are indistinguishable from random. The MPTs used for these tests are part of the data file (Data S3) and were chosen to differ as little as possible from each other topologically.

Eduardo Ascarrunz (pers. comm. 2016) and the reviewer Michael Buchwitz kindly notified us of problems with our first (2015) applications of these tests. *Goldman, Anderson & Rodrigo (2000)* and *Planet (2006)* have shown that all three tests, as implemented in PAUP* and elsewhere, are inappropriate to use for our questions and indeed for the vas majority of the questions they have been applied to in the phylogenetics literature. To the best of our knowledge, better tests are not available for practical purposes. Thus, we cannot reject the abovementioned null hypothesis in any case. However, the error caused by using these tests is one-sided: the *p*-values found by better tests are always greater than the *p*-values found by the inappropriately used one-tailed tests—greater by an amount that cannot be estimated in advance (*Goldman, Anderson & Rodrigo, 2000*). This allows us to look at the problem from the other side and distinguish the cases where the null hypothesis can *certainly* not be rejected from those where it remains rejectable: when the inappropriate tests do not reject the null hypothesis, we can be certain that a better test would not reject it either, but when the inappropriate tests do reject the null hypothesis, the null hypothesis may or may not be rejected by a better test.

PAUP* performs two-tailed versions of all three tests; we report *p*-values from the one-tailed versions, derived by halving the *p*-values put out by PAUP* (*Goldman, Anderson & Rodrigo, 2000*; *Planet, 2006*).

### Measure of similarity

Valentin Rineau (CR2P, Paris) kindly calculated ITRIs for selected pairwise comparisons of trees in the software LisBeth 01/2013 (see *Zaragüeta-Bagils et al., 2012*). The ITRI (*Grand et al., 2013*) is a measure of the similarity of a tree in question to a reference tree (not the other way around, which could give a different value). If the tree in question

contains all of the three-item statements (3is) that constitute the reference tree, meaning that it is identical to or a superset of the reference tree, its ITRI compared to that reference tree is 100%. However, the ITRI is not simply the fraction of shared 3is. Combinations of 3is imply further 3is in the same tree; this limitation of the degrees of freedom is used to weight the ratios of shared 3is before calculating the ITRI (*Grand et al., 2013*).

Previously (*Grand et al., 2013*; *Rineau et al., 2015*), the ITRI was used for cases where the true tree was known (e.g. simulations). This is of course not the case here. In each of our comparisons we have used the "older" tree (higher up in Table 1) as the reference tree: O trees as the references for R trees, O1 as the reference for O2, O1 and O2 as the references for O3 (in two separate comparisons) and so on; O1 is the reference tree in each comparison that contains it, EB never is. Thus, we quantify how similar "newer" trees are to "older" ones.

If the reference tree is larger than the tree in question, the latter's ITRI can never reach 100% because it cannot contain all 3is found in the reference tree. We have always used reference trees equal in size to or smaller than the trees in question (see Table 1), so such cases do not arise here.

From each analysis we used only one tree. Analysis EB (see the following section) of course yielded a single tree; from the parsimony analyses we used the same MPTs as for the statistical tests described in the preceding section. Because those trees (included in Data S3) were chosen to be as similar as possible, this ensures that the ITRIs we find are upper bounds on a range. Lower bounds could only be found by an exhaustive search. We did not compare strict consensus trees because they are poorly resolved (which would exaggerate the similarities) in some of the very ways the ITRI is designed to avoid. The majority-rule consensus is not a good representation of the result of a parsimony analysis (see *Marjanović & Laurin, 2018*); the Adams consensus must not be mistaken for an actual tree.

### Exploratory Bayesian analysis

As a preliminary test of whether the method of analysis has an impact on our results, J. Grau kindly ran an unconstrained Bayesian-inference analysis on our revised matrix including the added taxa (Analysis EB; Table 1). He used MrBayes 3.2.6 (*Huelsenbeck et al., 2015*) in parallel on 16 AMD Opteron™ 6172 processors (2.1 GHz), using 16 GB of RAM. Instead of the usual two, we chose to use four simultaneous runs; all other settings remained at the factory defaults, notably the specification of the first quarter of the generations as burn-in. Calculation time was 20 h and 6 min.

MrBayes cannot handle stepmatrices. We therefore had to split the two characters with stepmatrices into two to three characters each, which were ordered or unordered. The results of this non-trivial task are documented in App. S1 under characters 32 and 134; the scores of the new characters are wholly predictable from those of the stepmatrix characters used for all other analyses, and are therefore not presented separately except in the matrix file (Data S6). The total number of characters for this analysis was 280.

The average standard deviation of the split frequencies decreased to 0.05 between 6 and 7 million generations into the analysis. Because it had not reached stationarity,

we extended the analysis in the hope of reaching the recommended value of 0.01; after some 20 million generations of fluctuation around 0.013, we ended the analysis after a total of 40 million generations (including 10 million of burn-in). Convergence was achieved according to two other metrics: the plot of the logarithm of probability against generation number (excluding the burn-in) did not show a trend, and the potential scale reduction factor of the tree length was 1.000 as recommended.

## RESULTS

A very brief overview and comparison of our results is presented as Table 3. The results of statistical tests for lack of distinguishability of topologies are reported below and shown in Table 4; the similarities of topologies are reported below as ITRI and shown in Table 5.

### Reanalyses of the matrix of *Ruta & Coates (2007)*

Our unconstrained reanalysis of the original, unmodified matrix (Analysis O1) found 324 MPTs, as RC07 did; their strict consensus (Fig. 1) is identical to that reported by RC07 (fig. 5, 6). Their length is 1,621 steps when we distinguish polymorphism from partial uncertainty (both occur in the matrix; PAUP* setting: pset mstaxa=variable), but 1,584 steps—as reported by RC07—when we treat both as partial uncertainty (pset mstaxa=uncertain). The indices we find are slightly different from those reported by RC07, and differ again depending on the treatment of polymorphism: when polymorphism is treated as uncertainty, the ci excluding the parsimony-uninformative characters is 0.2281 rather than 0.22, the ri is 0.6768 rather than 0.67 and the rc is 0.1564 rather than 0.15; when polymorphism and uncertainty are distinguished, the ri is unaffected, but the ci excluding the parsimony-uninformative characters rises to 0.2458 and the rc becomes 0.1683.

Our analysis constrained for the LH (Analysis O2) found 60 MPTs. Their strict consensus (Fig. 9) is quite unlike what RC07 reported on p. 86, and also unlike that of *Laurin (1998a)*, part or all of which RC07 used as a constraint on p. 85: we find Lissamphibia as the sister-group of Albanerpetidae, both of which together form the sister-group to Holospondyli ("nectrideans" including aïstopods). All of these together are found as the sister-group of *Brachydectes*; in other words, lepospondyl topology is identical to that in the shortest trees, except for the addition of Lissamphibia + Albanerpetidae next to Holospondyli, and except for the loss of resolution in Urocordylidae. The resolution of Temnospondyli is greatly improved. The length of these suboptimal trees is 1,622 steps (1,585 when polymorphism is treated as uncertainty); ci excluding the parsimony-uninformative characters = 0.2280 (polymorphism treated as uncertainty) or 0.2457 (polymorphism and uncertainty distinguished), ri = 0.6766, rc = 0.1562 or 0.1681.

In short, the matrix of RC07 supports the temnospondyl hypothesis over the lepospondyl hypothesis by one single step, rather than either nine or fifteen steps as originally claimed (RC07: 85, 86).

To test this result, we manually drew a most parsimonious rooted tree and a rooted tree that fulfills the constraint in Mesquite (*Maddison & Maddison, 2017*). Mesquite reported

**Table 3 Summary and comparison of analysis results.** See Table 1 for the settings of these analyses and for the numbers of the figures that represent the trees. The "modern amphibians" column gives the closest one or two relatives of the clade of modern amphibians (TH, LH) or of the two clades of modern amphibians (PH); the Batrachia column indicates whether Caudata and Salientia form a clade to the exclusion of *Eocaecilia*; the "closer to Amniota" column shows whether the temnospondyls (T) are closer to Amniota than *Caerorhachis*, Anthracosauria and *Silvanerpeton* (CAS) are or more distant than all of them (except in B1, where *Caerorhachis* is a temnospondyl); the *Kotlassia* column states whether *Kotlassia* is crownward of, rootward of or a member of Seymouriamorpha; the *Limnoscelis* column states whether *Limnoscelis* is crownward of, rootward of or a member of Diadectomorpha; the last column states whether *Ichthyostega* is rootward or crownward of *Acanthostega* and *Ventastega*. Note that whenever Amniota is closer to Temnospondyli than to CAS, *Caerorhachis* is found as a temnospondyl. Brachystelechidae contains *Batropetes*, *\*Carrolla* and *\*Quasicaecilia*. OTUs unique to the expanded taxon sample are marked with an asterisk. #, symbol of the analysis (as used in the text and in Tables 1 and 4–12); AS, Anthracosauria and *Silvanerpeton* (*Caerorhachis* is found as a temnospondyl); CAS, *Caerorhachis*, Anthracosauria and *Silvanerpeton*; Diad, Diadectomorpha; Seym, Seymouriamorpha; T, Temnospondyli; (U, A), a clade formed exclusively by Urocordylidae and Aïstopoda. Numbers in curly braces are bootstrap percentages, shown in boldface if 50 or higher, except for Analysis EB, where they are posterior probabilities (in %), shown in boldface if 75 or higher.

| # | Modern amphibians | Batrachia | Albanerpetidae closest to: | Closer to Amniota: | Kotlassia | Limnoscelis | Adelospondyls closest to: | Ichthyostega |
|---|---|---|---|---|---|---|---|---|
| O1 | TH: *Doleserpeton* | Yes | *Eocaecilia* | CAS | Crownward | Rootward | Colosteidae | Crownward |
| O2 | LH: Holospondyli | Yes | Lissamphibia | CAS | Crownward | Rootward | Colosteidae | Crownward |
| O3 | PH: *Doleserpeton; Brachydectes,* Holospondyli or both | Yes | *Eocaecilia* | CAS | Crownward | Rootward | Colosteidae | Crownward |
| R1 | LH: (Adelospondyls (U, A)) | Yes | Batrachia | (1) CAS; (2) T | (1) Rootward; (2) Seym. | Diad. | (U, A) | (1) Crownward; (2) rootward |
| R2 | TH: *Phlegethontia*[1] | Yes | Lissamphibia | T | Rootward | Diad. | [1] | (1) Crownward; (2) rootward |
| R3 | PH: *Doleserpeton;* (adelospondyls (U, A))[2] | No[2] | Caudata | (1) CAS; (2) T | (1) Rootward; (2) Seym. | Diad. | (U, A) | (1) Crownward; (2) rootward |
| B1 | LH {84}[3]: (*Brachydectes* + *Batropetes*) {20}[4] | Yes {77} | Batrachia {53} | AS {12} | Seym. {28} | Diad. {59} | Holospondyli + crownward {35; 25}[5] | Crownward {47} |
| R4 | LH: (*Brachydectes* + Brachystelechidae) | Yes | Batrachia | CAS | Seym. | Crownward | (U, A) | Crownward |
| R5 | TH: *Doleserpeton* or *\*Gerobatrachus* | No | *Eocaecilia* | CAS | Seym. | Crownward | (U, A) | Crownward |
| R6 | PH: *Doleserpeton;* Aïstopoda[6] | Yes[6] | *Valdotriton* | CAS | Seym. | Crownward | (U, A) | Crownward |
| B2 | LH {21[3], 25[7]}: (*Brachydectes* + Brachystelechidae) {15}[4] | Yes {54} | Lissamphibia {67} | T {6} | Seym. {30} | Diad. {24} | all other amphibians {20; 10}[5] | Crownward of *Ventastega* {11} and *Acanthostega* {10} |
| EB | LH {92}: (*\*Carrolla* + *\*Quasicaecilia*) {88}[4] | Yes {81} | Batrachia {55} | T {48} | Seym. {99} | Diad. {96} | ("U," A) {48; 69; 86}[8] | Crownward of *Ventastega* and *Acanthostega* {both 76} |

**Notes:**
1. An entirely novel arrangement where Lissamphibia and the strongest "lepospondyls" are nested among the dvinosaurian temnospondyls, closest to *Trimerorhachis*; see text and Fig. 12.
2. Salientia (as constrained) is nested within Temnospondyli (next to *Doleserpeton*), while *Eocaecilia* (as constrained), Caudata and Albanerpetidae are nested within Lepospondyli.
3. Dissorophoidea.
4. 20 (B1) and 15 (B2) are the bootstrap percentages of the smallest clade that contains *Batropetes*, *Brachydectes* and Lissamphibia, which has a PP of **92**; (*Batropetes* + *Brachydectes*) respectively (Brachystelechidae + *Brachydectes*) have 37 (B1) or 23 (B2), while (Lissamphibia (*\*Carrolla*, *\*Quasicaecilia*)) has a PP of **88**.
5. The highest bootstrap percentage that keeps the adelospondyls away from Colosteidae is 35 (B1) or 20 (B2), the highest that keeps them away from (U, A) is 25 (B1) or 10 (B2).
6. Salientia (as constrained) and Caudata including Albanerpetidae are nested within Temnospondyli (next to *Doleserpeton*), while *Eocaecilia* (as constrained) is nested within Lepospondyli.
7. Amphibamidae (not including *\*Micropholis*).
8. The urocordylids come out as paraphyletic (PP = 65). The PP of the adelospondyls being the sister-group of (*Ptyonius* (*Urocordylus* (*Sauropleura*, Aïstopoda))) is 48, the PP of that clade + *\*Utaherpeton* is 69, the highest PP that keeps them away from Colosteidae is **86**.

**Table 4 Summary of parsimony support for hypotheses on lissamphibian origins.** See Table 1 for analysis settings. The second column subtracts the MPT length found by the second analysis in each line from that found by the first. The rightmost column summarizes the *p*-values from the one-tailed Kishino–Hasegawa, Templeton and winning-sites tests (listed in the text; our use of the tests is discussed in Materials and Methods: Robustness analyses). Bootstrap values of 0.5 or higher are given in boldface, as are *p*-values that do not certainly fail to reject the null hypothesis at the (arbitrary) 0.05 level; the null hypothesis is that the topology differences cannot be distinguished from random variations. Note that a cutoff value of 0.1 instead of 0.05 would have accepted the comparisons of O3 to O1 and O2, but otherwise given the same results, except that the Templeton and the winning-sites test distinguish R4 and R6 at a minimum of *p* = 0.0565 and *p* = 0.0966. Conversely, a cutoff value of 0.01 would have rejected all.

| Analyses | Difference in number of steps | Highest bootstrap support for difference in topology | Difference in topology indistinguishable from random at *p* = 0.05? |
|---|---|---|---|
| **O1 (TH), O2 (LH)** | 1 | < 0.5 (RC07: app. 4) | Certainly (*p* > 0.44) |
| **O1 (TH), O3 (PH)** | 12 | **0.67** (modern amphibians—RC07: app. 4) | Certainly (0.06 > *p* > 0.054) |
| **O2 (LH), O3 (PH)** | 11 | **0.67** (modern amphibians—RC07: app. 4) | One test: certainly, **two others: not certainly** (0.065 > *p* > **0.046**) |
| **R1 (LH), R2 (TH)** | 9 | 0.47 (*Isodectes* + *Neldasaurus*) | Certainly (0.225 > *p* > 0.133) |
| **R1 (LH), R3 (PH)** | 12 | **0.84** (Dissorophoidea) | **Not certainly (0.035 > *p* > 0.017)** |
| **R2 (TH), R3 (PH)** | 3 | **0.84** (Dissorophoidea) | Certainly (*p* > 0.4) |
| **R4 (LH), R5 (TH)** | 10 | 0.40 (*Doleserpeton* + *\*Gerobatrachus*) | Certainly (0.227 > *p* > 0.183) |
| **R4 (LH), R6 (PH)** | 15 | **0.67** (modern amphibians) | Certainly (0.133 > *p* > 0.056) |
| **R5 (TH), R6 (PH)** | 5 | **0.67** (modern amphibians) | Certainly (*p* > 0.31) |

**Table 5 Maximum similarities of trees from our analyses expressed as ITRIs (in %).** The trees used here are the ones used for the statistical tests for lack of distinguishability (text and Table 4), meaning they are chosen for being as similar as possible and are included in Data S3. The ITRI is asymmetric; the columns show the reference trees, the lines represent the trees compared to each reference tree. Comparisons represented by empty cells were deemed uninteresting and not made. See Tables 1 and 3 or the text for more information about the analyses.

| ↓ compared to → | O1 (TH) | O2 (LH) | O3 (PH) | R1 (LH) | R2 (TH) | R3 (PH) | R4 (LH) | R5 (TH) |
|---|---|---|---|---|---|---|---|---|
| **O2 (LH)** | 86.7 | | | | | | | |
| **O3 (PH)** | 95.5 | 92.8 | | | | | | |
| **R1 (LH)** | 72.0 | 92.8 | | | | | | |
| **R2 (TH)** | 76.7 | 82.0 | | 86.5 | | | | |
| **R3 (PH)** | | | 78.9 | 95.7 | 85.3 | | | |
| **R4 (LH)** | 75.6 | 88.2 | 92.7 | | | | | |
| **R5 (TH)** | 90.2 | 80.1 | | | 79.8 | | 87.8 | |
| **R6 (PH)** | | | 88.0 | | | 88.6 | 87.5 | 97.5 |
| **EB (LH)** | 71.2 | 81.3 | | 96.7 | | | 84.2 | |

the same lengths as PAUP* (distinguishing polymorphism from partial uncertainty, as Mesquite always does).

This result is perhaps not as surprising as it may seem. Although they only published the values for seven nodes, RC07 (app. 4—p. 122) did conduct a bootstrap and a decay analysis. While Lissamphibia (bootstrap percentage = 67, Bremer index = 8) and most of the nodes within it are well supported, the sister-group relationship of Lissamphibia and *Doleserpeton* has a Bremer index of 1 and "no boots[t]rap support compatible with a 50% majority-rule consensus", and the sister-group relationship of both of these to *Amphibamus* has a Bremer index of 2 and "no bootstrap support in a 50% majority-rule consensus". Values for more rootward nodes were not reported, but are not likely to be any higher given the limited resolution of this part of the tree (Fig. 1).

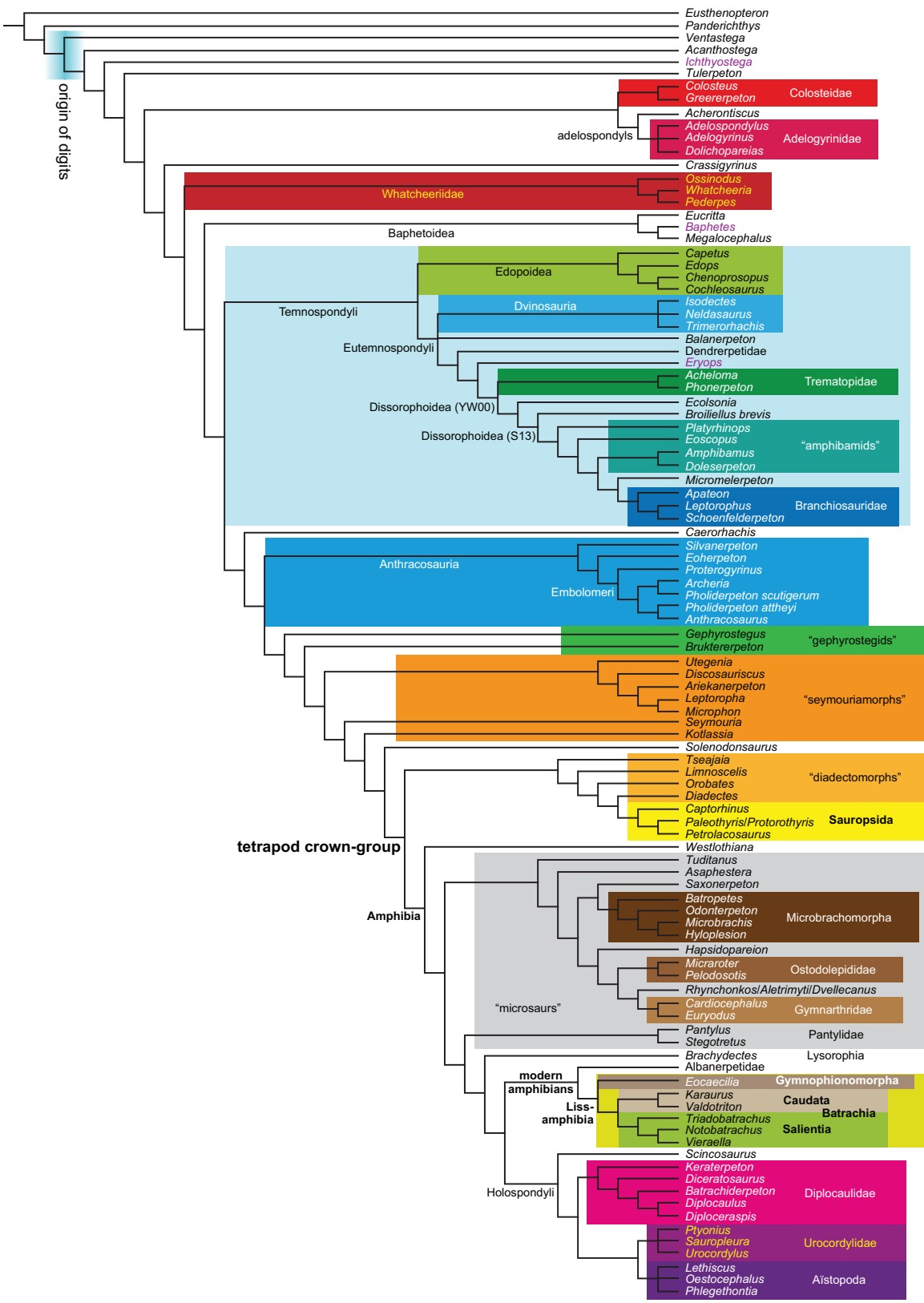

**Figure 9** **Strict consensus of the MPTs (length: 1,622 steps including polymorphisms) found by our reanalysis of the unchanged matrix of RC07 constrained for the LH (Analysis O2).** The constraint forced the lysorophian "lepospondyl" *Brachydectes* to be closer to the three

salientians (*Triadobatrachus*, *Notobatrachus*, *Vieraella*) than the dissorophoid temnospondyl *Doleserpeton* (Fig. 8A with the positions of *Brachy-dectes* and *Doleserpeton* exchanged).

For completeness and for the purposes of the next section, we also ran an analysis constrained for the PH (Analysis O3). It yielded 4,627 MPTs of 1,633 steps (1,596 when polymorphism is treated as uncertainty), 12 steps more than the MPTs from O1; ci excluding the parsimony-uninformative characters = 0.2264 (polymorphism treated as uncertainty) or 0.2468 (polymorphism and uncertainty distinguished), ri = 0.6737, rc = 0.1545 or 0.1662. Although less well resolved among seymouriamorphs, diadectomorphs and "lepospondyls", the strict consensus is very similar to those of O1 and O2; Salientia and the two caudates form a trichotomy nested next to *Doleserpeton* as in O1, Albanerpetidae and *Eocaecilia* form a clade nested in a trichotomy with Holospondyli and *Brachydectes* (all three resolutions occur in different MPTs). Indeed, some MPTs are identical to those from O2 except for the position of Batrachia. In some MPTs, the "lepospondyl" clade (including Albanerpetidae and *Eocaecilia* but not the adelospondyls) and *Solenodonsaurus* lie outside the "seymouriamorph"-"diadectomorph"-amniote clade.

Unsurprisingly, trees from O1 and O2 are statistically indistinguishable: $p = 0.4500$ (Kishino–Hasegawa and Templeton tests), $p = 0.5000$ (winning-sites test). Trees from O2 and O3, however, may be distinct: $p = 0.0468$ (Kishino–Hasegawa), 0.0467 (Templeton), 0.0637 (winning sites). Interestingly, despite being one step farther apart than O2 and O3, trees from O1 and O3 are not distinguishable at the 0.05 level: $p = 0.0545$ (Kishino–Hasegawa), 0.0547 (Templeton), 0.0599 (winning sites). The tree from O2 used for these tests (included in Data S3) has an ITRI of 86.7% compared to the tree from O1; the tree from O3 has an ITRI of 95.5% compared to O1 and an ITRI of only 92.8% compared to O2, fitting the tests for lack of distinguishability.

Data S7 contains the unmodified matrix, the constraints, our other analysis settings, a MPT (O1, 1,621 steps), a tree that fulfills the constraint for the LH (O2, 1,622 steps) and two that fulfill the constraint for the PH (O3, 1,633 steps), the first chosen to be as similar as possible to those from O1 and O2 and used in the statistical tests and in the section following below, the other very different in topology (together, the differences between these two trees account for almost all the polytomies in the strict consensus). Executing this file in PAUP* repeats all three analyses and performs the statistical tests on the trees that are already stored in the file.

## Amount, distribution and impact of revised scores

Excepting the deleted postcranium of *Rhynchonkos* (Materials and Methods: Treatment of OTUs: *Rhynchonkos*), App. S2 contains 4,125 colored scores (2,404 red, 1,642 green, 79 blue) for the original taxon sample. (See Materials and Methods: Modifications to individual cells or below for the meanings of the colors; the scores were counted in Data S4). Additionally, 67 changes to deleted characters are counted as red in Data S4, five as green and three as blue; the total number of individual changed scores in the matrix is thus 4,200.

Table 6 shows the OTUs with the highest numbers of red changes (in App. S2, that is, not counting deleted characters). These are due to contradictions between the scores of RC07 and their character or state definitions, without any redefinitions by us. At the

**Table 6 OTUs with the highest numbers (up to a rank of 20) of score changes that are marked red in App. S2, as counted in Data S4.** In these cases the score in RC07 disagrees with the definition or description of the respective character in RC07, without further interpretation of the character. References are provided in the text (Materials and Methods: Modifications to individual cells) and in App. S1; M. L. coauthored the 2016 redescription of *Triadobatrachus* (*Ascarrunz et al., 2016*).

| Rank | OTU | Number of red scores | Relevance to lissamphibian origins if any, main sources of changes |
|------|-----|---------------------|-------------------------------------------------------------------|
| 1 | *Ventastega* | 104 | New material described 2008; description 1994 had only partially been used. |
| 2 | *Lethiscus* | 96 | Redescribed 2017; redescription 2003 had not been used either. |
| 3 | *Cochleosaurus* | 68 | Postcranium described 2009; skull redescription 2004 had not been used either. |
| 4 | *Batropetes* | 67 | Close to **Lissamphibia** in LH. Redescribed 2013, 2015. |
| 5 | *Kotlassia* | 58 | Redescription 2003 had not been used. |
| 6 | *Trimerorhachis* | 57 | Redescribed 2007, 2013; pers. obs. |
| 7 | *Ossinodus* | 54 | New material described 2007, 2014. |
| 8 | *Doleserpeton* | 52 | Close to **Lissamphibia** in TH. Redescribed 2008, 2010. |
| 9 | Albanerpetidae | 50 | Next to or in **Lissamphibia**. New material described 2013; descriptions 2002, 2003 and 2005 had not been used. |
| 10 | *Isodectes* | 49 | Personal observation of nearly complete undescribed postcrania. |
| 11 | *Orobates* | 45 | Digital model 2015 showing previously unpublished data. |
| 12 | *Diplocaulus* | 43 | Descriptions 1909, 1917 had not been used. |
| 13 | *Platyrhinops* | 42 | Close to **Lissamphibia** in TH. Redescribed 2010, 2012. |
| 13 | *Cardiocephalus* | 42 | Effect of scoring both species instead of just one. |
| 15 | *Edops* | 41 | Personal observation especially of neglected postcrania briefly described in 1942. |
| 16 | *Eocaecilia* | 40 | **Lissamphibian.** Redescribed 2007; pers. obs. |
| 17 | *Acheloma* | 38 | Second species described 2010, 2011. |
| 17 | *Triadobatrachus* | 38 | **Lissamphibian.** Redescribed 2012, 2016 (with digital model); pers. obs. |
| 19 | *Baphetes* | 35 | Redescribed 2009; new material published 1998 had not been fully taken into account either. |
| 20 | *Limnoscelis* | 34 | Redescribed 2007, 2010; redescription 1983 (thesis) had only partially been used. |

top lies *Ventastega*, due to the description of new material by *Ahlberg et al. (2008)*, but also due to the fact that *Ruta, Coates & Quicke (2003)* only scored about half as much as they could have based on the original description by *Ahlberg, Lukševičs & Lebedev (1994)* as mentioned above (Materials and Methods: Modifications to individual cells: Literature). The close second is *Lethiscus*, whose skull was redescribed from μCT data by *Pardo et al. (2017)*, though some of the changes we had earlier made based on *Wellstead (1982*; also describing the postcranium*)* and *Anderson, Carroll & Rowe (2003)* were confirmed by the new paper. Data published after 2001—not always after 2006—account for most of the rest of the table as well, although we should highlight the fact that most of the 43 red changes we made to *Diplocaulus* come from the detailed descriptions of the lower jaw and postcranium by *Williston (1909)* and *Douthitt (1917)*, neither of which was cited by RC07 or *Ruta, Coates & Quicke (2003)*.

Table 7 does the same for characters. The highest rank by a large margin belongs to ch. 257 (TRU VER 4), "Haemal spines not fused (0) or fused (1) to caudal centra." RC07 considered state 1 to be "almost exclusively observed in nectrideans", but this is not defensible unless the character is extensively reinterpreted. It is possible that RC07 only meant "pleurocentra" by "centra", and identified the single-piece centra of "nectrideans" as pleurocentra (but did not score them accordingly, except in *Scincosaurus*;

see ch. 259, TRU VER 7, in App. S1). Indeed, it has sometimes been assumed that hemal arches are homologous to intercentra or parts thereof (most recently by *Olori, 2015*: 57). However, that is not the case, nor do hemal arches always fuse to intercentra when intercentra are present (*Pawley & Warren, 2005*; *Schoch, 2006*: fig. 6H; *Witzmann & Schoch, 2006a*; *Holmes & Carroll, 2010*; apparently *Clack, 2011b*). Therefore, we have taken the definition literally and scored state 1 whenever the hemal spines are fused to intercentra or to single-piece centra, keeping it for all "nectrideans" but expanding it to most of the rest of the sampled OTUs with known tails—and scoring it as unknown in 13 OTUs that had been given state 0 despite lacking sufficiently preserved tails, as well as in the one OTU (*Triadobatrachus*) that had a tail without any hemal arches and to which this character, here originally scored 0 as well, therefore cannot apply. At the third rank in Table 7, the problem of the homology of the "dorsal process" of the ilium has a somewhat similar impact on ch. 225 (ILI 3); the other character at the third rank, the mandibular lateral-line canal (ch. 101: SC 2), was misscored for almost all "microsaurs", apparently because *Ruta, Coates & Quicke (2003)* had misinterpreted statements by CG78 about pits and grooves clearly referring, from context, to ornament or nerves and blood vessels. All characters mentioned in this paragraph are discussed at some length in App. S1.

For the characters in the rest of Table 7, we cannot find such a single main cause or any common pattern for the incorrect scores. Literature from 2006 or later is the main source for changes in only one, literature from 2003 or later only in a total of three of these characters.

Green changes are at least in part due to redefinitions or possible redefinitions of states (or entire characters). As we made such changes to characters rather than to taxa, many of the OTUs with the highest numbers of such scores (Table 8) are among those we rescored wholesale. Indeed, many are familiar from Table 6—less than half of the 20 OTUs in the latter (*Edops*, *Cochleosaurus*, *Isodectes*, *Doleserpeton*, *Eocaecilia*, *Cardiocephalus*, *Diplocaulus* and *Ossinodus*) are missing from the former. Also well represented in Table 8, however, are well known OTUs to which many characters are inapplicable; this is because we have changed many scores from known to unknown in attempts to move away from non-additive binary coding or avoid redundancy. Thus, five of the seven modern amphibian OTUs and all three aïstopods have made it into the 20 highest ranks, and at the very top lies *Brachydectes*, which combines unusual anatomy, lost skull bones and a recent redescription (*Pardo & Anderson, 2016*).

Naturally, the two characters with the most green changes (Table 9) are the two we split off from others (Materials and Methods: Treatment of characters: Deleted, recoded and split characters), one of them having fresh scores for all 102 OTUs. We redefined and completely rescored the next six as well; similar but less drastic changes, such as reinterpretation as reductive instead of non-additive binary, account for the rest of the table.

Blue changes represent our attempts to take ontogeny into account. These affect mostly "branchiosaurs" and "discosauriscids" (Table 10), and characters that depend on the extent of ossification (Table 11). Further notable are the two characters describing the lateral-line canals at ranks 3 and 5 in Table 11, which are affected by the disappearance of grooves from the skull (ch. 100, SC 1) and the lower jaw (ch. 101, SC 2) in the ontogeny

**Table 7 Characters with the highest numbers of score changes (up to a rank of 20) that are marked red in App. S2, as counted in Data S4.** In these cases the score in RC07 disagrees with the definition or description of the respective character in RC07, but we did not redefine any states, and ontogeny had no impact on our score changes. Columns O and R show the relative support for the hypotheses on lissamphibian origins in the original (O; in case of mergers we present the net total support from all component characters) and our revised matrix (R) found by optimizing each character on the first three trees in Data S7 (from Analyses O1–O3) and Data S3 (from Analyses R1–R3). LP 1: the character has one fewer step on the trees supporting the LH and the PH in Data S7 (column O) or Data S3 (column R) than on the respective tree supporting the TH; T 4, P 1: the character has one fewer step on the tree supporting the PH than on the one supporting the LH, and four fewer on the one supporting the TH than on the one supporting the LH; and so on; –: no difference between the three trees (equal to LTP 0).

| Rank | Character | Number of red scores | O | R | Main sources of changes |
|---|---|---|---|---|---|
| 1 | **257**: TRU VER 4 | 50 | – | – | Description says "centra"; had been scored only for pleurocentra, possibly under untenable assumption that hemal arches are homologous to intercentra or parts thereof. Discussed in App. S1. |
| 2 | **215**: HUM 13 | 37 | – | – | 20th-century literature. |
| 3 | **101**: SC 2 | 34 | – | – | Literature of all ages; misinterpretation by *Ruta, Coates & Quicke (2003)* of wording in CG78, discussed in App. S1. |
| 3 | **225**: ILI 3 | 34 | – | – | Homology across the taxon sample left unclear by RC07, discussed in App. S1. |
| 5 | **40**: POSPAR 3-6 | 32 | LP 1 | – | Literature mostly from 2003 and later. |
| 6 | **276**: DIG 1-2-3-4 | 30 | – | – | Mostly 20th-century literature; pers. obs. |
| 7 | **100**: SC 1 | 29 | – | – | Literature of all ages. |
| 7 | **102**: VOM 1-13 | 29 | T 2 | – | Literature mostly from 2003 and later. |
| 7 | **119**: PTE 3-9 | 29 | LT 1 | LP 1 | Literature of all ages and pers. obs. |
| 10 | **134**: EXOCC 2-3-4-5/ BASOCC 1-5 | 26 | T 2 | LT 1 | Literature of all ages and pers. obs. |
| 11 | **146**: JAW ART 1/SQU 2/DEN 8 | 25 | L 1 | T 4, P 1 | 20th-century literature. |
| 12 | **17**: PREFRO 10 | 24 | T 1 | – | 20th-century literature. |
| 12 | **66**: POSORB 7 | 24 | – | T 1 | Literature of all ages. |
| 12 | **219**: HUM 18/DIG 1 | 24 | T 1 | LP 3 | 20th-century literature. |
| 15 | **34**: PAR 5 | 23 | – | LP 1 | Literature of all ages. |
| 16 | **27**: FRO 2 | 22 | L 2, P 1 | LP 1 | Literature of all ages. |
| 16 | **43**: POSPAR 9 | 22 | – | – | Literature of all ages. |
| 16 | **130**: CHO 1 | 22 | – | – | Literature of all ages. |
| 16 | **272**: TRU VER 27 | 22 | – | – | Artificial missing data filled in from literature of all ages and pers. obs. |
| 20 | **105**: VOM 5-10/PTE 10-12-18/INT VAC 1 | 21 | T 2 | T 2 | Literature mostly from 2006 and earlier. |

of "branchiosaurs" and "discosauriscids"; also at rank 5 lies ch. 146 (JAW ART 1/SQU 2/DEN 8), which describes the gradual caudal shift of the jaw joint as the suspensorium grows with positive allometry in the ontogeny of many vertebrates.

## Analyses of our revised matrix

Note that the tree lengths we report throughout are those given by PAUP* (distinguishing polymorphism from partial uncertainty). Mesquite consistently reports one fewer step for trees from Analyses R1–R3 (original taxon sample; see Table 1) and one or two fewer steps for trees from Analyses R4–R6 (expanded taxon sample); apparently, the programs handle the stepmatrices differently. (As mentioned, PAUP* and Mesquite report the same lengths for the MPTs of Analyses O1–O3, where stepmatrices do not occur.)

**Table 8 OTUs with the highest numbers of score changes (up to a rank of 20) that are marked green in App. S2, as counted in Data S4.** These changes involve, or potentially involve, redefinitions of character states. "Complete enough to be affected by many redefinitions" applies to all OTUs in this table and is only mentioned in the absence of other major causes. M. L. is one of the coauthors of the 2016 redescription of *Triadobatrachus* (*Ascarrunz et al., 2016*).

| Rank | OTU | Number of green scores | Relevance to lissamphibian origins if any, main causes of changes |
|---|---|---|---|
| 1 | *Brachydectes* | 33 | Close to **Lissamphibia** in LH. Very many scores inapplicable. Redescribed 2016. |
| 2 | *Batropetes* | 29 | Close to **Lissamphibia** in LH. Many scores inapplicable. Redescribed 2013, 2015. |
| 3 | *Lethiscus* | 27 | Unusual anatomy, some scores inapplicable. Redescribed 2017; redescription 2003 had not been used either. |
| 4 | *Trimerorhachis* | 23 | Some unusual anatomy. Redescribed 2007, 2013. |
| 5 | *Baphetes* | 22 | Unusual anatomy, a few scores inapplicable. Redescribed 2009; new material published 1998 had not been fully taken into account either. |
| 5 | *Valdotriton* | 22 | **Lissamphibian.** Very many scores inapplicable. |
| 5 | *Gephyrostegus* | 22 | Skull redescribed 2014; redescription of axial skeleton 1991 had not been used either. |
| 8 | *Eryops* | 21 | Appendicular skeleton redescribed 2006. |
| 8 | *Archeria* | 21 | Complete enough to be affected by many redefinitions. |
| 10 | *Megalocephalus* | 20 | Unusual anatomy, a few scores inapplicable. |
| 10 | *Cochleosaurus* | 20 | Postcranium described 2009; skull redescription 2004 had not been used either. |
| 10 | *Acheloma* | 20 | Redescribed 2010, 2011. |
| 10 | Albanerpetidae | 20 | Next to or in **Lissamphibia.** Very many scores inapplicable. New material described 2013; descriptions 2002, 2003 and 2005 had not been used. |
| 10 | *Triadobatrachus* | 20 | **Lissamphibian.** Very many scores inapplicable. Redescribed 2012, 2016 (with digital model); pers. obs. |
| 10 | *Eoherpeton* | 20 | Complete enough to be affected by many redefinitions. |
| 10 | *Kotlassia* | 20 | Redescription 2003 had not been used. |
| 10 | *Limnoscelis* | 20 | Redescribed 2007, 2010; redescription 1983 (thesis) had only partially been used. |
| 10 | *Phlegethontia* | 20 | Unusual anatomy, very many scores inapplicable. |
| 10 | *Orobates* | 20 | Digital model 2015. |
| 20 | *Ventastega* | 19 | New material described 2008; description 1994 had only partially been used. |
| 20 | *Acanthostega* | 19 | Skull redescribed 2015. |
| 20 | *Ichthyostega* | 19 | Partial redescriptions 2009, 2011, 2012 twice, 2013, 2015, pers. obs. |
| 20 | Dendrerpetidae | 19 | Effect of making sure all species are scored; occiput redescribed 2005. |
| 20 | *Platyrhinops* | 19 | Close to **Lissamphibia** in TH. Redescribed 2010, 2012. |
| 20 | *Karaurus* | 19 | **Lissamphibian.** Very many scores inapplicable; pers. obs. |
| 20 | *Oestocephalus* | 19 | Unusual anatomy, some scores inapplicable. |
| 20 | *Notobatrachus* | 19 | **Lissamphibian.** Very many scores inapplicable; redescribed 2004, 2008. |

Comparison of the results to literature other than RC07 is part of the Discussion (section "Phylogenetic relationships").

### Analysis R1

The unconstrained analysis of our presumably improved matrix without added taxa yielded 1,120 MPTs with a length of 2,182 steps (a drastic increase over the 1,621 steps required by the unmodified matrix of RC07), a ci of 0.2181 (likewise revealing increased character conflict), a hi of 0.8327, a ri of 0.62118 and a rc of 0.1355.

Marjanović and Laurin (2019), *PeerJ*, DOI 10.7717/peerj.5565

**Table 9  Characters with the highest numbers of score changes (up to a rank of 20) that are marked green in App. S2, as counted in Data S4.** These characters have redefined or possibly redefined states. Columns O and R as in Table 7.

| Rank | Character | Number of green scores | O | R | Main cause of changes |
|------|-----------|------------------------|-----|-----|------------------------|
| 1 | **85**: MED ROS 1 | 102 | | – | Split off from INT FEN 1 (ch. 84, O: L1, R: –). |
| 2 | **190**: TEETH 10 | 99 | | – | Split off from TEETH 3 (ch. 183, O: T1, R: TP 2). |
| 3 | **95**: SKU TAB 1 | 98 | – | LP 2 | States redefined, completely rescored; see text. |
| 4 | **25**: MAX 8 | 97 | T 1 | – | States redefined to make scores reproducible, completely rescored. |
| 5 | **3**: PREMAX 7 | 94 | – | – | States redefined, completely rescored; see text. |
| 6 | **1**: PREMAX 1-2-3 | 87 | LP 1 | LP 1 | States redefined to account for entire taxon sample, completely rescored. |
| 7 | **145**: PASPHE 14 | 86 | T 1 | P 1 | States redefined to account for entire taxon sample, completely rescored. |
| 8 | **10**: NAS 5 | 68 | TP 1 | T 1 | Redefined to make scores reproducible and avoid massive redundancy, completely rescored. |
| 9 | **64**: POSORB 5 | 62 | – | – | Interpreted as reductive to avoid massive redundancy, see text. |
| 10 | **122**: PTE 13 | 48 | P 1 | – | Interpreted as reductive to avoid redundancy. |
| 11 | **124**: PTE 16 | 44 | – | – | Interpreted as reductive to avoid redundancy. |
| 12 | **83**: NOS 4 | 39 | P 1 | – | Redefined to increase applicability and avoid correlation with snout length. |
| 12 | **131**: CHO 2 | 39 | TP 1 | LP 1 | States defined to make scores reproducible. |
| 14 | **56**: TAB 6 | 37 | – | – | Redefined to make more useful. |
| 14 | **69**: SQU 3 | 37 | LP 1 | P 1 | States defined to make scores reproducible and avoid non-additive binary coding. |
| 14 | **231**: FEM 1-2-6 | 37 | L 1 | LP 1 | States redefined to be primarily homologous across taxon sample and to avoid redundancy. |
| 17 | **253**: CER VER 4 | 31 | LP 1 | L 1 | Interpreted as reductive to avoid redundancy. |
| 18 | **123**: PTE 14 | 29 | LT 1 | – | States defined to make scores reproducible. |
| 19 | **265**: TRU VER 13-14 | 26 | L 1 | – | Single-piece centra no longer assumed to be necessarily pleurocentra. |
| 20 | **31**: PAR 1 | 22 | T 1 | LP 1 | Redefined to avoid redundancy. |

Figures 10 and 11 presents the similarities and differences between the MPTs, based on inspection of the strict consensus and the first, the 25th, the 75th, the 125th, the last and every 50th MPT (1, 25, 50, 75, 100, 125, 150, 200 ... 1,000, 1,050, 1,100, 1,120) in Mesquite. They show that, in all MPTs, *Panderichthys* is the most rootward member of the ingroup as in RC07. More crownwards, however, the positions of *Ichthyostega*, *Ventastega* and *Acanthostega* are not fully resolved: *Ventastega* can be the sister-group of *Acanthostega* or one node more rootward; *Ichthyostega* can be more crownward than both, more rootward than both, or (Fig. 11) between the two when they are not sister-groups. Interestingly, a *Ventastega–Acanthostega* clade is only recovered when *Ichthyostega* lies crownward of it.

The next more crownward branch is the Mississippian Whatcheeriidae (resolved as in RC07), followed crownward by the Devonian *Tulerpeton*. Next more crownward is *Crassigyrinus*, followed by Colosteidae (which was instead rootward of Whatcheeriidae in RC07).

Crownward of these, the MPTs share a consistent backbone, which is a Hennigian comb of—from rootward to crownward—Baphetidae, Anthracosauria, *Silvanerpeton*, *Gephyrostegus*,

**Table 10 Complete list of OTUs with score changes that are marked blue in App. S2, as counted in Data S4.** These changes involve interpretations of ontogeny or heterochrony.

| Rank | OTU | Number of blue scores | Relevance to lissamphibian origins if any, main causes of changes |
|---|---|---|---|
| 1 | *Schoenfelderpeton* | 8 | Close to **Lissamphibia** in TH. Skeletally immature, possibly neotenic. |
| 1 | *Micromelerpeton* | 8 | Somewhat close to **Lissamphibia** in TH. Known mostly from immature individuals. |
| 3 | *Apateon* | 7 | Close to **Lissamphibia** in TH. Known mostly from immature individuals, one species known to be neotenic (*Fröbisch & Schoch, 2009b*). |
| 3 | *Utegenia* | 7 | Skeletally immature, possibly neotenic. |
| 5 | *Discosauriscus* | 6 | Known mostly from immature individuals. |
| 6 | *Ariekanerpeton* | 5 | Skeletally immature, possibly neotenic. |
| 6 | *Microphon* | 5 | Skeletally immature, possibly neotenic. |
| 8 | *Amphibamus* | 4 | Close to **Lissamphibia** in TH. Known mostly from immature individuals. |
| 8 | *Leptorophus* | 4 | Close to **Lissamphibia** in TH. Skeletally immature, possibly neotenic. |
| 8 | *Leptoropha* | 4 | Skeletally immature, possibly neotenic. |
| 11 | *Eucritta* | 3 | Skeletally immature, known from incomplete growth series. |
| 11 | *Trimerorhachis* | 3 | Slow-growing; postcranium redescribed 2007 with reports of more mature individuals than known before. |
| 13 | *Doleserpeton* | 2 | Close to **Lissamphibia** in TH. New growth stage of ch. 238 described in 2010; ch. 252 known in subadult but not adult specimens. |
| 13 | *Pederpes* | 2 | Known humerus (ch. 205) and ulna (ch. 224) incompletely ossified. |
| 15 | *Crassigyrinus* | 1 | Known ulna incompletely ossified (ch. 224). |
| 15 | *Chenoprosopus* | 1 | Ch. 69 changes state in ontogeny. |
| 15 | *Broiliellus* | 1 | Known humerus incompletely ossified (ch. 205). |
| 15 | *Archeria* | 1 | Ch. 205 changes state in ontogeny. |
| 15 | *Kotlassia* | 1 | State of ch. 105 probably not mature in known specimen. |
| 15 | *Batropetes* | 1 | Close to **Lissamphibia** in LH. Ch. 263 changes state in ontogeny. |
| 15 | *Hapsidopareion* | 1 | Somewhat close to **Lissamphibia** in LH. State of ch. 95 probably not mature in known specimens. |
| 15 | *Microbrachis* | 1 | Ch. 263 changes state in ontogeny. |
| 15 | *Hyloplesion* | 1 | Ch. 122 changes state in ontogeny. |
| 15 | *Odonterpeton* | 1 | Known ulna incompletely ossified (ch. 224). |
| 15 | *Silvanerpeton* | 1 | Known ulna incompletely ossified (ch. 224). |

*Solenodonsaurus*, Seymouriamorpha, and then the crown-group consisting of Amphibia, Diadectomorpha and Sauropsida (Figs. 10 and 11). Parts of this arrangement contradict the findings of RC07 (Fig. 1); the position of *Solenodonsaurus* is altogether novel.

In different MPTs, Temnospondyli is either rootward of Anthracosauria (one node crownward of Baphetidae) as in RC07, or crownward of Anthracosauria (one node rootward of *Solenodonsaurus*). These two positions are part of two classes of topologies that have limited overlap other than the mentioned backbone and the almost complete resolution of the tetrapod crown-group.

When Temnospondyli is rootward of Anthracosauria (Fig. 11), *Eucritta* is one node rootward of Baphetidae; *Edops* is the sister-group of all other temnospondyls, which form a Hennigian comb of Cochleosauridae, *Eryops*, *Capetus*, Dvinosauria,

**Table 11 Complete list of characters with score changes that are marked blue in App. S2, as counted in Data S4.** These changes involve interpretations of ontogeny or heterochrony. Columns O and R as in Table 7.

| Rank | Character | Number of blue scores | O | R | Causes of changes |
|---|---|---|---|---|---|
| 1 | **205**: HUM 2 | 9 | – | LP 1 | Distal end of humerus generally ossifies late, if ever. |
| 1 | **224**: ULNA 1 | 9 | L 1 | – | Proximal end of ulna generally ossifies late. |
| 3 | **100**: SC 1 | 6 | – | – | Lateral-line canals often disappeared in ontogeny. |
| 3 | **105**: VOM 5-10/PTE 10-12-18/INT VAC 1 | 6 | T 2 | T 2 | Interpterygoid vacuities closed in seymouriamorph and possibly baphetoid ontogeny. |
| 5 | **101**: SC 2 | 5 | – | – | Lateral-line canals often disappeared in ontogeny. |
| 5 | **146**: JAW ART 1/SQU 2/DEN 8 | 5 | L 1 | T 4, P 1 | Jaw joint very often moves caudally in ontogeny. |
| 5 | **263**: TRU VER 11 | 5 | – | – | Neural arches fuse to centra during ontogeny in amniotes, at least some "microsaurs" and probably seymouriamorphs. |
| 8 | **22**: MAX 5/PAL 5 | 2 | T 1 | LT 1 | *Apateon* changed to state 2 in ontogeny, *Schoenfelderpeton* only known from morphologically immature individuals showing the immature state 1. |
| 8 | **27**: FRO 2 | 2 | L 2, P 1 | LP 1 | *Apateon* changed to state 2 (already scored in RC07) in ontogeny; *Leptorophus* and *Schoenfelderpeton* only known from less mature individuals. |
| 8 | **43**: POSPAR 9 | 2 | – | – | *Microphon* changed from state 0 to 1 in ontogeny; *Leptoropha* only known from individuals that have state 0 and are ontogenetically comparable to those of *Microphon* with state 0. |
| 8 | **69**: SQU 3 | 2 | LP 1 | P 1 | Temporal embayment often became narrower notch in ontogeny, affecting *Chenoprosopus* and *Micromelerpeton*. |
| 8 | **189**: TEETH 9 | 2 | – | T 1 | Number of maxillary teeth increased in ontogeny in *Apateon* and *Leptorophus*. |
| 8 | **261**: TRU VER 9 | 2 | LP 1 | LP 1 | Trunk pleurocentra fused middorsally in seymouriamorph ontogeny, including *Utegenia* and probably *Ariekanerpeton*. |
| 14 | **14**: PREFRO 3 | 1 | P 1 | – | *Micromelerpeton* changed from 1 to 0 in ontogeny. |
| 14 | **23**: MAX 6 | 1 | – | – | *Schoenfelderpeton* scored as unknown following closure of maxillary arcade in ontogeny of *Apateon*. |
| 14 | **37**: PAR 8 | 1 | T 1 | LP 2 | Suture became more interdigitated in ontogeny of *Apateon*. |
| 14 | **54**: TAB 2-3-9 | 1 | – | – | Tabular "horns" became longer and more pointed in ontogeny of *Apateon*. |
| 14 | **66**: POSORB 7 | 1 | – | T 1 | Postorbital became wider in ontogeny of *Apateon*. |
| 14 | **72**: JUG 2-6 | 1 | LP 1 | – | *Schoenfelderpeton* skeletally too immature to be scored. |
| 14 | **74**: JUG 4 | 1 | – | T 1 | *Eucritta* has juvenile eye size. |
| 14 | **87**: ORB 2 | 1 | P 1 | P 2, T 1 | *Eucritta* has juvenile eye size. |
| 14 | **88**: ORB 3/LAC 5 | 1 | – | – | *Eucritta* has juvenile eye size. |
| 14 | **95**: SKU TAB 1 | 1 | – | LP 2 | Postorbital part of *Hapsidopareion* skull became longer in ontogeny. |
| 14 | **103**: VOM 3 | 1 | – | LP 1 | *Amphibamus* grew vomerine tusks in ontogeny. |
| 14 | **110**: PAL 1 | 1 | – | LP 1 | *Amphibamus* grew palatine tusks in ontogeny. |
| 14 | **113**: PAL 7 | 1 | P 2, L 1 | P 1 | Shape of palatine, unknown in the most mature *Amphibamus* specimens, may depend on tusks. |
| 14 | **122**: PTE 13 | 1 | P 1 | – | Interpterygoid vacuities of *Hyloplesion* become wider in ontogeny. |
| 14 | **137**: PASPHE 2-12 | 1 | – | LT 1 | *Amphibamus* changed from state 2 to 1 in ontogeny. |

(Continued)

Marjanović and Laurin (2019), *PeerJ*, DOI 10.7717/peerj.5565

| Rank | Character | Number of blue scores | O | R | Causes of changes |
|---|---|---|---|---|---|
| 14 | **196**: INTCLA 3 | 1 | – | T 1 | *Schoenfelderpeton* skeletally too immature to be scored. |
| 14 | **204**: HUM 1 | 1 | – | – | *Trimerorhachis* acquired state 1 rather early in ontogeny. |
| 14 | **215**: HUM 13 | 1 | – | – | Entepicondyle kept ossifying in large *Trimerorhachis*. |
| 14 | **238**: FIB 3 | 1 | – | – | *Doleserpeton* grew a ridge in ontogeny. |
| 14 | **250**: RIB 7 | 1 | T 1 | T 1 | Ribs of *Micromelerpeton* grew longer. |
| 14 | **252**: CER VER 3 | 1 | LT 1 | L 1 | Ontogeny of *Doleserpeton* unclear. |
| 14 | **262**: TRU VER 10 | 1 | – | – | *Utegenia* scored as unknown following swelling of neural arches in ontogeny of *Discosauriscus*. |

Dendrerpetidae and Dissorophoidea, where *Balanerpeton* can be the sister-group of either of the last two. *Caerorhachis* is one node rootward of the anthracosaurs (as in RC07); within the latter, *Pholiderpeton scutigerum* is the sister-group of either (*Proterogyrinus* + *Archeria*) or (*Pholiderpeton attheyi* + *Anthracosaurus*), both unlike in RC07. *Bruktererpeton* is either the sister-group of *Gephyrostegus* or one node more rootward (never crownward as in RC07), both lying rootward of *Solenodonsaurus* and crownward of *Silvanerpeton*; the traditional seymouriamorphs show a remarkable diversity of mono- and paraphyletic arrangements, though never the one found by RC07.

When Temnospondyli is crownward of Anthracosauria (Fig. 10), *Eucritta* is a non-baphetid baphetoid (as in RC07). *Caerorhachis* is one node rootward of the anthracosaurs in some MPTs (Fig. 10: main tree and upper inset), in which case Dissorophoidea forms the sister-group to all other temnospondyls (among which Dvinosauria is highly nested), or to all except Dvinosauria, or to all except (*Isodectes* + *Neldasaurus*), in which latter case *Trimerorhachis* is the sister-group of a highly nested (*Balanerpeton* + Dendrerpetidae) clade. In the other MPTs (Fig. 10: lower inset), *Caerorhachis* is the sister-group of all other temnospondyls, among which Dvinosauria is sister to the remainder, followed by Dissorophoidea. *Edops* is far from the temnospondyl root in both cases. Anthracosauria is resolved as in RC07; *Bruktererpeton* and *Gephyrostegus* are sister-groups and lie one node rootward of Temnospondyli; the traditional seymouriamorphs form a monophyletic Hennigian comb, except that *Kotlassia* is either one node rootward of Seymouriamorpha or a member of it (the sister-group to the rest)—a subset of the resolutions in the other class of topologies.

In both cases (Figs. 10 and 11), the dissorophoid *Ecolsonia* can emerge as a trematopid or (as in RC07) one node closer to all other dissorophoids.

All MPTs support the LH. Lissamphibia is highly nested among the "lepospondyls"; within Temnospondyli, Amphibamidae is monophyletic and highly nested.

The tetrapod crown-group consists of Amphibia (the lissamphibian total group, here including all "lepospondyls") and the sister-groups Diadectomorpha and Sauropsida. Sauropsida is resolved as in RC07; Diadectomorpha is resolved as (*Limnoscelis* (*Tseajaia* (*Orobates*, *Diadectes*))).

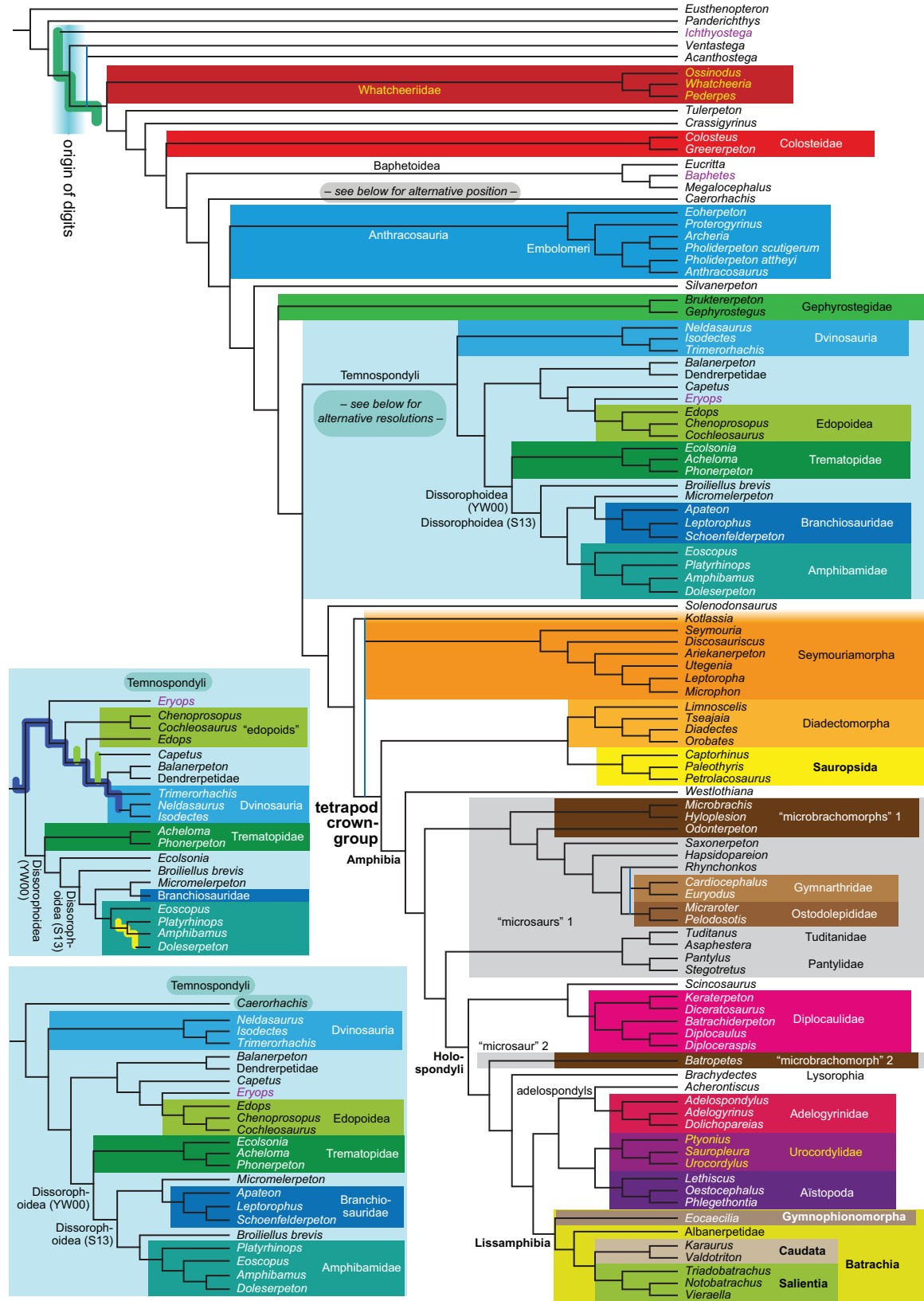

**Figure 10** Representation of some of the MPTs (length: 2,182 steps) from Analysis R1, performed on our revised matrix—specifically the ones where Anthracosauria is rootward of Temnospondyli. The remaining MPTs from R1, where Anthracosauria is crownward of Temnospondyli, are

shown in the next figure. The taxon sample was unchanged from RC07; no constraint was enforced. This tree, like those in the following figures, aims to represent all of the information contained in the MPTs, more than any consensus tree can. The uninterrupted part of the tree (black lines), polytomies excepted, is present in all MPTs represented in this figure as a backbone on which the branches connected only by colored underlays have varying positions; all nodes underlain in color are absent from the strict consensus. The teal underlay in this figure connects the two positions of *Ichthyostega* that occur in different MPTs: one node more rootward, or one node more crownward, than *Ventastega* and *Acanthostega*; the green underlay connects the two positions of *Capetus* in the upper inset, the blue one the two of the (*Neldasaurus* + *Isodectes*) clade, the yellow one the two of *Doleserpeton*. In this and the following figures, branches with only two neighboring positions in different MPTs are indicated with blue lines: for example, the clade of all seymouriamorphs except *Kotlassia* is the sister-group of *Kotlassia* or of the tetrapod crown-group. Trichotomies that are resolved in all three possible ways in different MPTs (none in this figure) are shown in the usual way (without a blue line). Equally parsimonious alternatives that would be confusing if shown on the same tree are shown in the insets to the left, concerning in this case Temnospondyli and *Caerorhachis*, which may or may not (with equal parsimony) be a temnospondyl.

Unchanged from RC07, *Westlothiana* is the most rootward amphibian and is followed crownward by two "microsaur" clades, which comprise all "microsaurs" in the taxon sample except (unlike in RC07) *Batropetes*; the composition and especially the topologies of these clades are quite different from those found by RC07, however. The larger and more rootward clade has a basal split into (*Microbrachis* + *Hyloplesion*) on one side and (*Odonterpeton* (*Saxonerpeton* (*Hapsidopareion* (Gymnarthridae, Ostodolepididae)))) on the other, where *Rhynchonkos* is the sister-group of Gymnarthridae or (Gymnarthridae + Ostodolepididae); the smaller and more crownward one is composed of Pantylidae (as in RC07) and Tuditanidae (*Tuditanus* + *Asaphestera*).

Next more crownward lie the sister-groups *Scincosaurus* and Diplocaulidae. Diplocaulidae is resolved as in RC07, except that *Keraterpeton* and *Diceratosaurus* are sister-groups.

The mentioned brachystelechid "microsaur" *Batropetes* and the lysorophian *Brachydectes* lie successively closer to the modern amphibians. However, the sister-group of Lissamphibia is a clade that contains, one could say, all the strangest "lepospondyls": the adelospondyls (which, in RC07, were the sister-group of the colosteids) form the sister-group to Urocordylidae + Aïstopoda. Unlike in RC07, *Adelospondylus* is resolved as the sister-group to the other adelogyrinids and *Lethiscus* as the sister-group of the other aïstopods.

Lissamphibia is fully resolved as (*Eocaecilia* (Albanerpetidae (Caudata, Salientia))).

The tree from R1 included in Data S3 has an ITRI of 72.0% compared to the tree from O1 (supporting the TH) and an ITRI of 92.8% compared to the tree from O2 (LH).

### Analysis R2

Constraining the analysis of the original taxon sample against the LH yielded 64 MPTs with a length of 2,191 steps—nine more than in the unconstrained Analysis R1—and a ci of 0.2173, a hi of 0.8334, a ri of 0.6191, and a rc of 0.1345. Figure 12 presents the similarities and differences between the MPTs, based on inspection of all of them in Mesquite. All MPTs have a bundle of highly unusual features.

Specifically, when the salientians move closer to *Doleserpeton* than to *Brachydectes*, they drag not merely the other lissamphibians along, but also the aïstopods, urocordylids and adelospondyls; and this assemblage nests not within the amphibamids or the

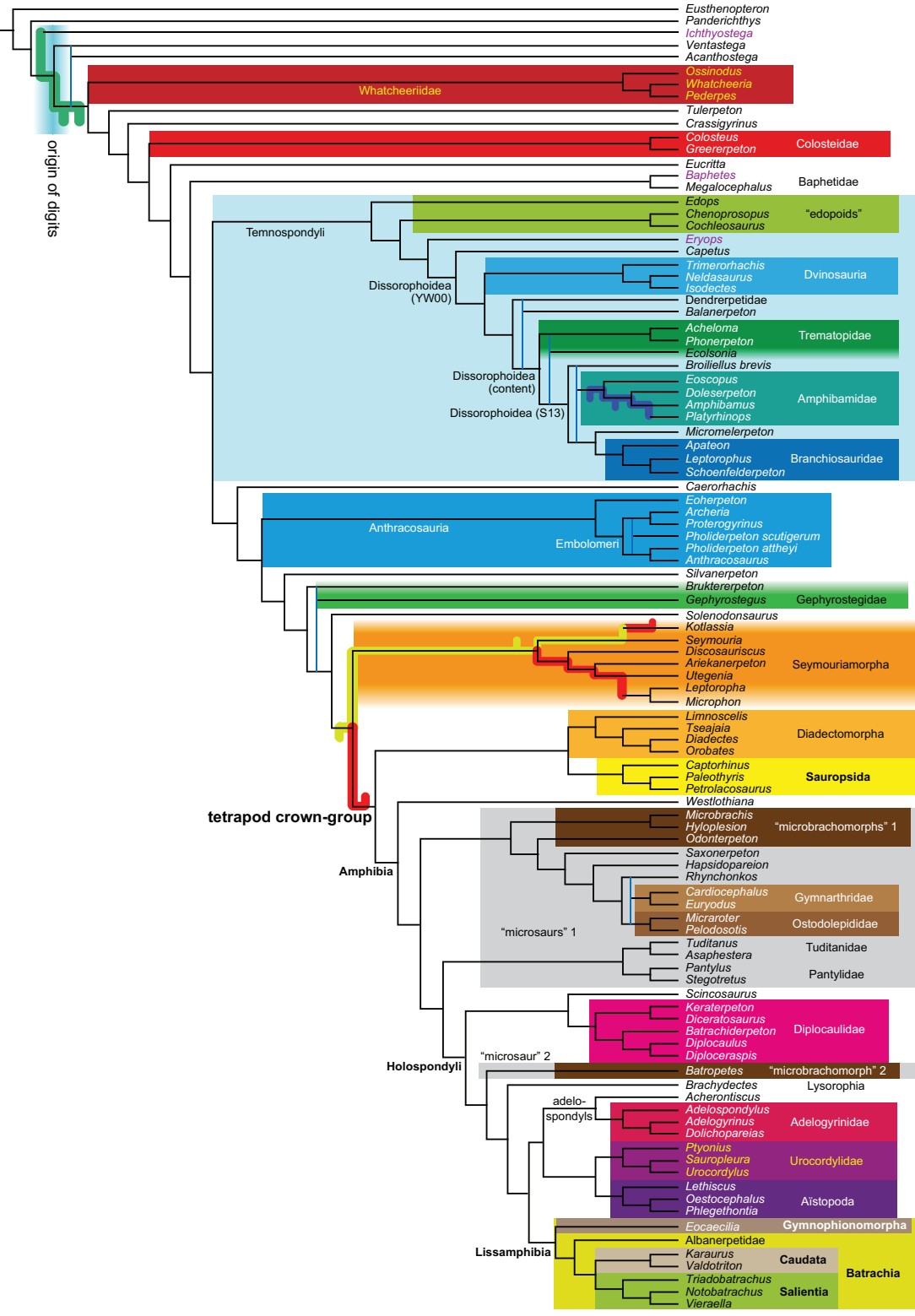

**Figure 11 Representation of those MPTs from Analysis R1 where Anthracosauria lies crownward of Temnospondyli.** See legend of Fig. 10 for more information. Note the additional position of *Ichthyostega* (immediately crownward of *Ventastega* and rootward of *Acanthostega*) and the fixed position of *Caerorhachis* (outside of Temnospondyli, though close to it). Purple underlay: *Platyrhinops* (three positions); yellow: *Kotlassia* (three positions); red: (*Leptoropha + Microphon*) (three positions).

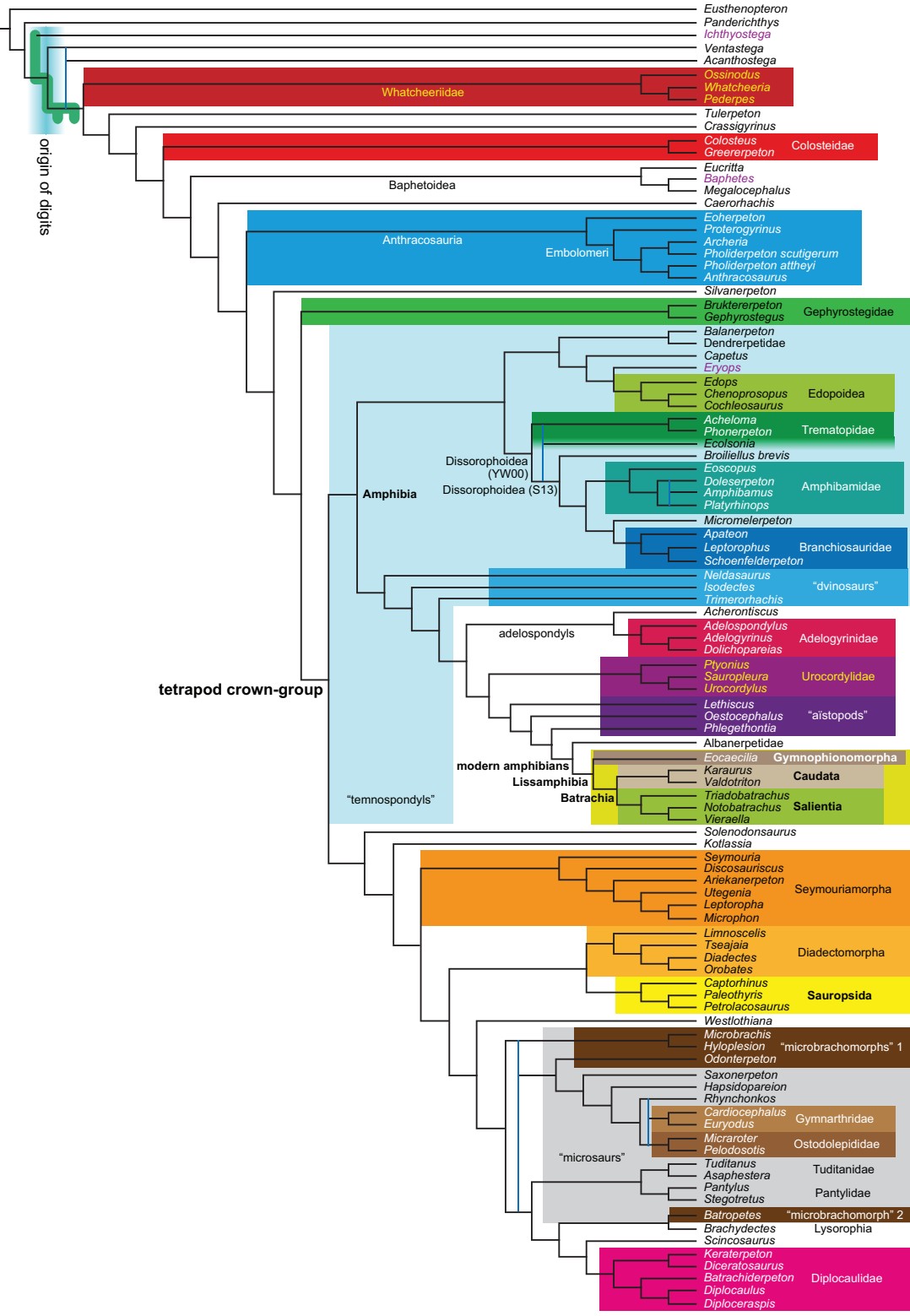

**Figure 12 Representation of all MPTs (length: 2,191 steps) from Analysis R2, performed on our revised matrix.** The taxon sample was unchanged from RC07. A constraint forced the three salientians (*Triadobatrachus*, *Notobatrachus*, *Vieraella*) to be closer to the dissorophoid temnospondyl *Doleserpeton* than to the lysorophian "lepospondyl" *Brachydectes*; this allowed both the TH and the PH. Teal underlay as in Figs. 10 and 11.

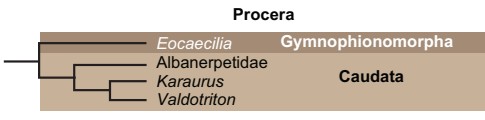

**Figure 13 Representation of all MPTs (length: 2,194 steps) from Analysis R3, performed on our revised matrix.** Only the parts different from the results of R1 (Figs. 10 and 11) are shown. Outside the shown clades, the entire range of resolutions of R1 also occurs in R3. Procera lies in the place where Lissamphibia is found in R1. The taxon sample was unchanged from RC07. A constraint forced the dissorophoid temnospondyl *Doleserpeton* to be closer to the three salientians (*Triadobatrachus*, *Notobatrachus*, *Vieraella*) than the lysorophian "lepospondyl" *Brachydectes*, and additionally forced the gymnophionomorph *Eocaecilia* to be closer to *Brachydectes* than to *Doleserpeton*; this allowed only the PH.

dissorophoids more generally, but within the dvinosaurs. The modern amphibians form the sister-group to the aïstopod *Phlegethontia*; all together are joined on the outside by *Oestocephalus*, then *Lethiscus*, then Urocordylidae (where *Ptyonius* is resolved as the sister-group to the rest), then the adelospondyls, and then the dvinosaurian temnospondyl *Trimerorhachis*, followed by *Isodectes* and then *Neldasaurus*. The clade that contains all non-dvinosaurian temnospondyls is still recovered as in Analysis R1, but is almost fully resolved as in a subset of the trees from Analysis R1, with highly nested edopoids. Temnospondyli lies crownward of Gephyrostegidae, *Silvanerpeton*, Anthracosauria and (most rootward) *Caerorhachis*. *Eucritta* is the sister-group of Baphetidae. In some trees, the (*Microbrachis* + *Hyloplesion*) clade falls out as the sister-group to all lepospondyls except *Westlothiana*.

Other than this, the topology is consistent with trees from Analysis R1 (Fig. 10). The clade formed by Diplocaulidae and *Scincosaurus* does not follow the other holospondyls into Temnospondyli, but stays behind as the sister-group of (*Batropetes* + *Brachydectes*). *Ichthyostega* and *Ecolsonia* have the same positions as in Analysis R1; *Kotlassia* is always rootward of Seymouriamorpha; Lissamphibia and Albanerpetidae are sister-groups in all MPTs.

Statistical tests find the difference between a tree from Analysis R2 and a tree from Analysis R1 insignificant at a $p$ level of 0.05: $p$ is 0.2248 under the Kishino–Hasegawa test, 0.1854 under the Templeton test and 0.1332 under the winning-sites test (Table 4). The same tree from R2 has an ITRI of 76.7% compared to a tree from O1, 82.0% compared to O2 and 86.5% compared to R1.

### Analysis R3

An analysis of the original taxon sample that was constrained to be compatible with the PH yielded 736 MPTs with a length of 2,194 steps—only three steps more than those from Analysis R2, 12 more than those from Analysis R1—and a ci of 0.2170, a hi of 0.8336, a ri of 0.6184 and a rc of 0.1342.

Judging from the first, the last and every 25th MPT, the different MPTs (Fig. 13) are very similar to those of R1. Salientia forms the sister-group to *Doleserpeton* and is nested within the amphibamid dissorophoid temnospondyls, not within the dvinosaurian

temnospondyls as it is in R2. All other modern amphibians have the same positions as in R1.

The difference between a tree from Analysis R3 and a tree from Analysis R1 is potentially significant at the level of $p = 0.05$ according to the statistical tests: $p = 0.0352$ (Kishino–Hasegawa), 0.0362 (Templeton), 0.0175 (winning sites). The difference between Analyses R3 and R2 is far from significance: $p = 0.4074$ (Kishino–Hasegawa), 0.4255 (Templeton), 0.5000 (winning sites). The same tree from R3 has an ITRI of 78.9% compared to O3, 95.7% compared to R1 and 85.3% compared to R2.

### Analysis R4

From here on, the OTUs we have added are marked with an asterisk; taxa absent even from the expanded taxon sample are marked with two asterisks.

An unconstrained analysis of the increased taxon sample found 401 MPTs with a length of 3,011 steps, a ci of 0.1860, a hi of 0.8788, a ri of 0.6186, and a rc of 0.1150.

Figure 14 presents the similarities and differences between the MPTs, based on the strict and Adams consensus trees and on the first, the last and every 10th tree.

Despite containing several OTUs known only from isolated lower jaws, the Devonian area of the tree is remarkably well resolved (Fig. 14): *Elginerpeton, *Metaxygnathus, (Acanthostega + Ventastega), *Ymeria, *Perittodus, Ichthyostega, Whatcheeriidae, *Densignathus, Tulerpeton and Crassigyrinus are successively more crownward. This pattern is partially obscured by Ossinodus, which has four positions: as a whatcheeriid (the sister-group to the rest as in R1), as the sister-group of *Densignathus, or one node more crownward or more rootward than the latter.

Crownward of Crassigyrinus follows a dichotomy between an unusually enlarged Temnospondyli and its sister-group, which contains the tetrapod crown-group (all MPTs support the LH).

Within Temnospondyli, the sister-group of all others is Eucritta, followed by a novel and fully resolved colosteid-baphetid clade. The *"St. Louis tetrapod" (MB.Am.1441; Clack et al., 2012b) is the sister-group to all other colosteids, among which *Pholidogaster is the sister-group to (*Aytonerpeton (*Deltaherpeton (Colosteus, Greererpeton))). Within Baphetidae, Baphetes, Megalocephalus, *Diploradus and *Sigournea are successively closer to *Doragnathus and *Spathicephalus.

The traditional temnospondyls form a clade which is resolved as an unusual Hennigian comb where Edops, Cochleosauridae, Eryopiformes, Capetus, Dvinosauria, (Balanerpeton + Dendrerpetidae) and *Iberospondylus are successively closer to Dissorophoidea. *Palatinerpeton has two positions as the sister-group of Dendrerpetidae or of (*Iberospondylus + Dissorophoidea). *Nigerpeton and *Saharastega form a novel clade with three novel positions: as the sister-group of Eryops, of Stereospondylomorpha or of all traditional temnospondyls except Edops and Cochleosauridae.

Dissorophoidea is fully resolved. Trematopidae, including *Mordex and Ecolsonia, is the sister-group of the remainder, in which *Micropholis clusters with the "branchiosaurs" of the original taxon sample, while (*Branchiosaurus + *Tungussogyrinus) is the

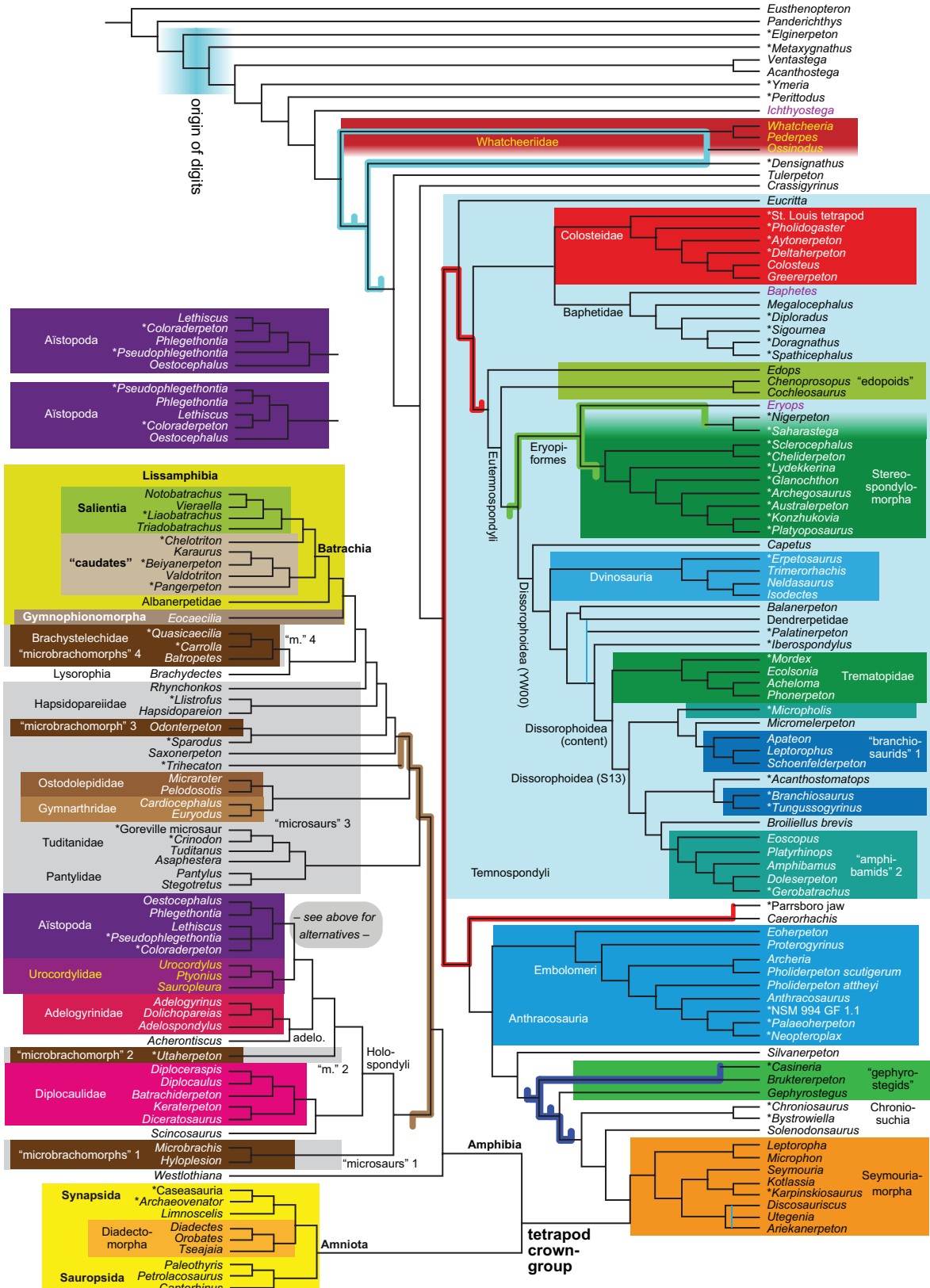

**Figure 14 Representation of all MPTs (length: 3,011 steps) from Analysis R4 (revised matrix, expanded taxon sample, no constraint).** The insets at the left show the two remaining parsimonious alternatives to the three aïstopod topologies shown in the main tree. In this and the following

figures, the names Stereospondylomorpha and Limnarchia are placed according to prevailing usage, not according to the definitions by *Yates & Warren (2000)* or *Schoch (2013)*. According to the definition by *Yates & Warren (2000)*, Stereospondylomorpha would in this analysis probably exclude *Sclerocephalus* and *Cheliderpeton*, possibly also *Glanochthon*; all three were included in the different topology of *Yates & Warren (2000*: fig. 1*)*. Limnarchia would be a synonym of Stereospondylomorpha as used here, but would exclude the originally included dvinosaurs. Stereospondylomorpha as redefined by *Schoch (2013)* would have the contents shown here, but this does not apply to some of the following figures. Similarly, Dvinosauria would under its original definition (*Yates & Warren, 2000*) be a synonym of Dissorophoidea as defined in the same work; as redefined by *Schoch (2013)*, it would have the contents shown here, but not quite in Fig. 16. The teal underlay connects the four positions of *Ossinodus* (a whatcheeriid sister to the rest, one node more crownward than Whatcheeriidae, sister to *Densignathus*, or one node more crownward than it). The two positions of the "Parrsboro jaw" (see text) are connected by the red underlay. The green underlay connects the two positions of (*Nigerpeton* + *Saharastega*), the dark blue one the four of *Casineria*, the brown one the two of *Trihecaton*.

sister-group of *Acanthostomatops*. The latter three together form the sister-group of (*Broiliellus* + Amphibamidae).

The *Parrsboro jaw (*Godfrey & Holmes, 1989*; *Sookias, Böhmer & Clack, 2014*) is found in two very distinct positions: as the sister-group to the clade of all traditional temnospondyls, or next to *Caerorhachis*.

*Caerorhachis* (and optionally the *Parrsboro jaw) is one node crownward of Temnospondyli. It is followed crownward by Anthracosauria, *Silvanerpeton*, *Bruktererpeton*, *Gephyrostegus*, Chroniosuchia (*Chroniosaurus*, *Bystrowiella*), *Solenodonsaurus* and Seymouriamorpha. Anthracosauria is fully resolved as in RC07, with the added taxa forming a clade (*NSM 994 GF 1.1 (*Palaeoherpeton*, *Neopteroplax*)) that nests with *Anthracosaurus*. *Casineria* has four positions: one node rootward of *Bruktererpeton*, one node crownward of *Gephyrostegus*, between the two, or as the sister-group of *Bruktererpeton*.

Just outside the tetrapod crown-group lies Seymouriamorpha, in which (*Leptoropha* + *Microphon*) and a (*Seymouria* (*Karpinskiosaurus*, *Kotlassia*)) clade lie successively closer to Discosauriscidae. Within the crown, the amniote + diadectomorph clade is fully resolved; curiously, however, the added OTUs *Caseasauria and *Archaeovenator emerge as the sister-group to *Limnoscelis*, followed by the clade of the other diadectomorphs and then by Sauropsida—in other words, the diadectomorphs are found as amniotes, specifically as synapsids that lack the synapsid condition (*Berman, 2013*), but they do not form a clade (unlike in *Berman, 2013*).

On the amphibian side of the crown, *Westlothiana* remains the sister-group to the remainder, which has a topology somewhat reminiscent of *Vallin & Laurin (2004)*. Instead of having a "microsaur" grade at the base as in Analyses R1–R3, this clade shows a basal dichotomy: (Holospondyli (*Microbrachis*, *Hyloplesion*)) lies on one side, while the other branch groups the remaining "microsaurs" and *Brachydectes* (which is the only lysorophian even in the expanded taxon sample) with the fully resolved Lissamphibia.

The basal dichotomy of Holospondyli is into a (*Scincosaurus* + Diplocaulidae) clade, which is fully resolved as in R1–R3, and a clade of (*Utaherpeton (adelospondyls (Urocordylidae, Aïstopoda))). Aïstopoda is poorly resolved.

On the other side of the amphibian branch, Pantylidae and Tuditanidae are sister-groups like in R1–R3; Tuditanidae is augmented by *Crinodon and the *Goreville microsaur. A (Gymnarthridae + Ostodolepididae) clade lies next to the remainder, in which *Saxonerpeton*, an unexpected (*Odonterpeton* + *Sparodus*) clade, an expected

(*Hapsidopareion* + *Llistrofus*) clade, *Rhynchonkos* and finally a clade composed of *Brachydectes* and Brachystelechidae (composed of *Batropetes* + (*Carrolla* + *Quasicaecilia*); named after a synonym of *Batropetes*) are successively closer to Lissamphibia, unlike in R1. Within Lissamphibia, Albanerpetidae emerges as the sister-group of Batrachia. The closest relative of Salientia is the caudate *Chelotriton*; together they are the sister-group to a clade formed by all other caudates, in which *Karaurus* is as highly nested as possible.

*Trihecaton* has two positions as the sister-group of (Holospondyli (*Microbrachis*, *Hyloplesion*)) or one node rootward of *Saxonerpeton*.

A tree from R4 has an ITRI of 75.6% compared to O1, 88.2% compared to O2 and 92.7% compared to R1.

### Analysis R5

An analysis of the increased taxon sample with a constraint against the LH found 6,778 MPTs which have 3,021 steps—10 more than the MPTs of Analysis R4—as well as a ci of 0.1854, a hi of 0.8792, a ri of 0.6170, and a rc of 0.1144.

As shown by comparison of the strict and Adams consensus trees and the first, the last and every 250th MPT (Figs. 15 and 16), all of which support the TH, the constraint has not had a strong effect on the tree; most of the differences to the results of R4 consist of loss of resolution among the temnospondyls. This is in stark contrast to earlier versions of this matrix (*Marjanović & Laurin, 2015, 2016*) or to the differences between R1 and R2.

The clade of *Nigerpeton* and *Saharastega* gains two new positions within Stereospondylomorpha (Fig. 15: lower inset; Fig. 16: middle inset); in one of them (Fig. 16), Dvinosauria joins it. These two clades can also form a clade or grade just crownward of Cochleosauridae, followed crownward by *Eryops* and then Stereospondylomorpha (Fig. 16: main tree, upper inset); or they can form a clade next to Stereospondylomorpha, which then, too, lies crownward of *Eryops* (Fig. 16: upper inset). The trematopids sometimes (Fig. 16) form a grade (with *Ecolsonia* at its crownward end as in RC07 and some MPTs of R1), *Tungussogyrinus*, *Branchiosaurus* and *Micropholis* gain positions within the non-trematopid non-lissamphibian dissorophoid grade, *Acanthostomatops* sometimes leaves it. Unlike in Analysis R2, which was performed under the same constraint, Lissamphibia is not nested among the dvinosaurs, and no "lepospondyls" are found among the temnospondyls; instead, either *Doleserpeton* or *Gerobatrachus*, but never both together, is the sister-group of Lissamphibia. Lissamphibia is resolved with a Procera topology, where an (Albanerpetidae + *Eocaecilia*) clade is nested within a caudate grade or next to a caudate clade, which always excludes *Chelotriton*, the sister-group to all other procerans.

*Trihecaton* has a single position as the sister-group of (Holospondyli (*Microbrachis*, *Hyloplesion*)). The other large lepospondyl clade is slightly rearranged from R4: the sister-group of the *Brachydectes*-brachystelechid clade is Hapsidopareiidae, followed by (*Odonterpeton* (*Goreville microsaur, *Sparodus*)), then (*Rhynchonkos* (Gymnarthridae, Ostodolepididae)), and then *Saxonerpeton* followed by (Tuditanidae + Pantylidae) as in R4.

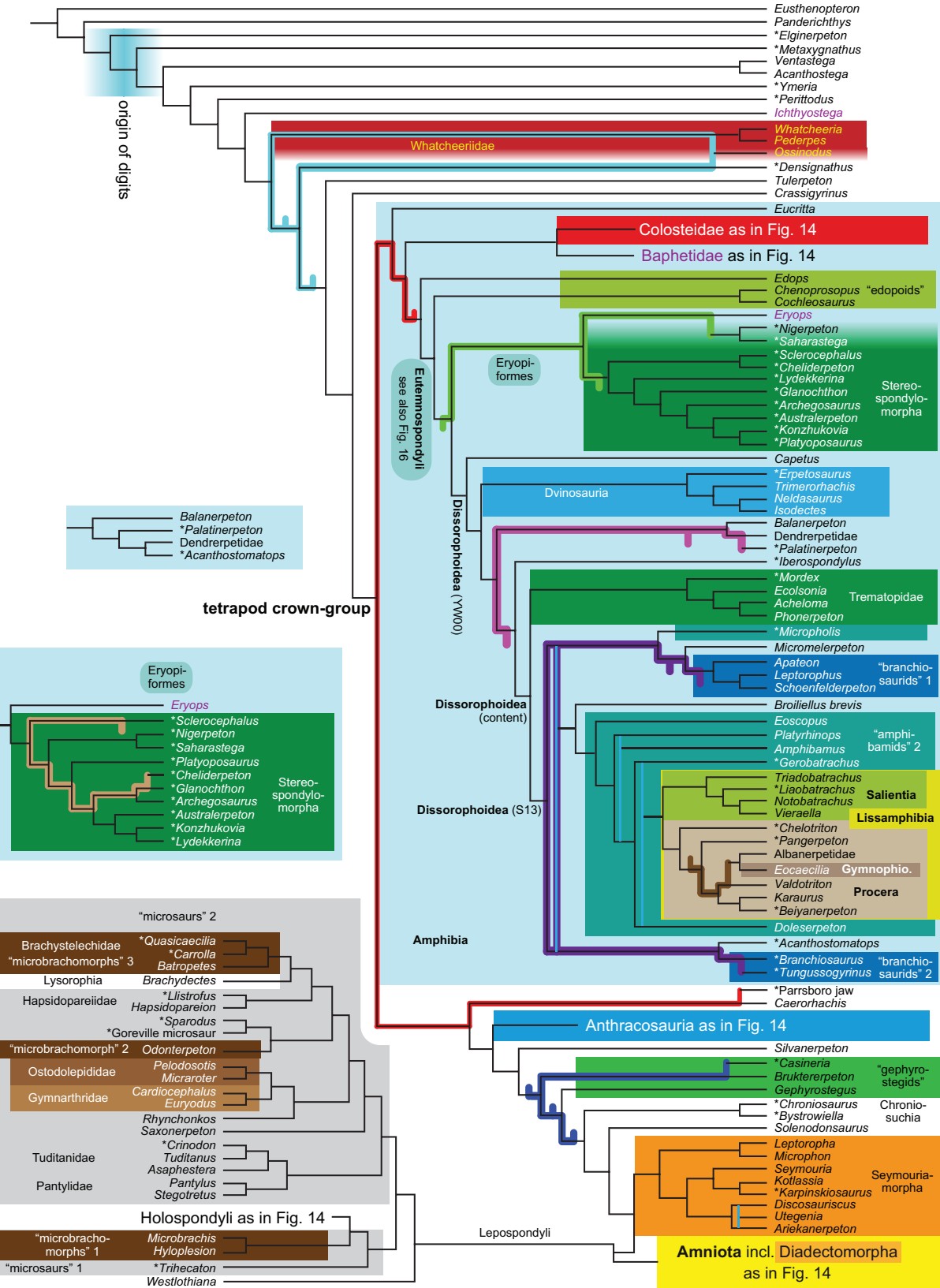

**Figure 15** Representation of some MPTs (length: 3,021 steps) resulting from Analysis R5 (revised matrix, expanded taxon sample, constraint against the LH as in Analysis R2). The insets at the left show two equally parsimonious alternative to parts of the temnospondyl topology shown in

the main tree. The remaining MPTs from R5 are represented in Fig. 16; the differences are again limited to eutemnospondyl phylogeny (Eutemnospondyli being the sister-group of *Edops*, as defined by *Schoch, 2013*). Teal, red, green and dark blue underlays as in Fig. 14, though note a fourth position for (*\*Nigerpeton* + *\*Saharastega*) in the lower inset. Further underlay colors: magenta: *\*Palatinerpeton* (three positions); violet: *\*Tungussogyrinus* (3); light brown: *\*Cheliderpeton* (2); dark brown: (*Eocaecilia* + Albanerpetidae) (2).

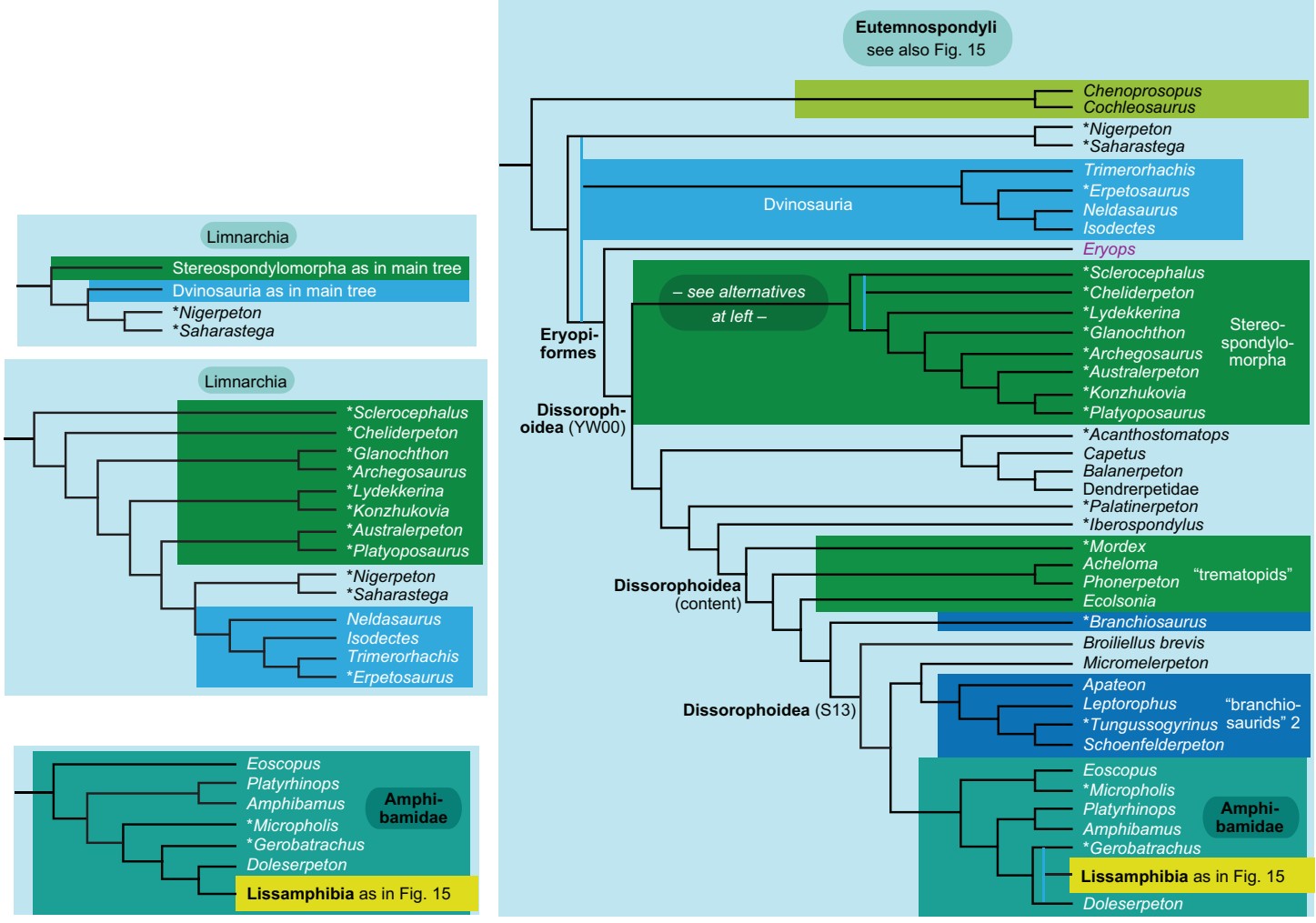

**Figure 16** Eutemnospondyl phylogeny from those MPTs from Analysis R5 that are not represented in Fig. 15; see the legend of Fig. 15 for more information.

The statistical tests find the difference between a tree from R5 and a tree from R4 insignificant: $p = 0.1831$ (Kishino–Hasegawa), 0.2278 (Templeton), 0.0975 (winning sites). The same tree from R5 has an ITRI of 90.2% when compared to a tree from O1, 80.1% compared to O2, 79.8% compared to R2 and 87.8% compared to R4.

### Analysis R6

With a constraint for the PH, an analysis of the increased taxon sample found 1,816 MPTs with 3,026 steps—five more than those from Analysis R5, 15 more than those from Analysis R4. The MPTs have a ci of 0.1851, a hi of 0.8794, a ri of 0.6162 and a rc of 0.1140.
Inspection of the first, the last and every 50th MPT (we later examined every 10th MPT up to the 640th for the position of *Cheliderpeton*) shows topologies mostly identical to those of R4, allowing us to greatly simplify Fig. 17. In addition to its positions in R4, the *Nigerpeton-*Saharastega* clade can lie inside a rearranged Stereospondylomorpha (as in some MPTs of R5). Batrachia is nested as the sister-group to *Doleserpeton*, followed by *Gerobatrachus*. Within Batrachia, *Chelotriton* is the sister-group of Salientia. Albanerpetidae is highly nested in the clade of all remaining caudates, as the sister-group of *Valdotriton*.

Lepospondyli is resolved much like in Analysis R1, with *Eocaecilia* as the sister-group of Aïstopoda; *Brachydectes* sometimes enters Brachystelechidae as the sister-group of *Batropetes*. Aïstopoda is better resolved than in R4 and R5 in that there is always a dichotomy between (*Oestocephalus* + *Phlegethontia*) and the remainder. *Trihecaton* lies next to Gymnarthridae or next to *Micraroter*.

Trees from R6 and R4 are indistinguishable at the level of $p = 0.05$: $p = 0.1328$ (Kishino–Hasegawa), 0.0565 (Templeton), 0.0966 (winning sites). Unsurprisingly, trees from R6 and R5 are hardly distinguishable: $p = 0.3515$ (Kishino–Hasegawa), 0.3174 (Templeton), 0.4538 (winning sites). The same tree from R6 has an ITRI of 88.0% compared to O3, 88.6% compared to R3, 87.5% compared to R4 and 97.5% compared to R5.

### Bootstrap analyses B1 and B2

Figures 18 and 19 present the bootstrap trees for the original and the expanded taxon samples, respectively (Analyses B1 and B2). They are fully resolved because clades with bootstrap values below 0.5 (50%) are included if they do not contradict the majority-rule consensus (see above). In Fig. 18 (Analysis B1), support is skewed toward peripheral nodes, while the "trunk" of the tree has bootstrap percentages well below 50. Still, together with many uncontroversial results, the position of Whatcheeriidae rootward of Colosteidae, *Crassigyrinus* and even *Tulerpeton* is supported (67%, 54% and 57%, respectively) as well as the membership of *Ossinodus* (52%), the position of *Eucritta* as a baphetoid (55%), Dissorophoidea sensu lato excluding Lissamphibia (84%), *Ecolsonia* as a trematopid (56%), Trematopidae as the sister-group to Dissorophoidea sensu stricto (55%), Amphibamidae excluding Branchiosauridae (60%), *Micromelerpeton* as the sister-group of Branchiosauridae (61%), *Bruktererpeton* and *Gephyrostegus* as sister-groups (53%), the monophyly of Diadectomorpha (59%) and Sauropsida (80%) as well as their sister-group relationship (68%), *Microbrachis* and *Hyloplesion* as sister-groups (67%), Lissamphibia including Albanerpetidae (65%), Batrachia + Albanerpetidae (53%), Batrachia excluding Albanerpetidae (77%), Caudata (74%) and the sister-group relationship of *Scincosaurus* and Diplocaulidae (56%). Further, *Acanthostega* as crownward of *Ventastega* (45%), *Ichthyostega* as crownward of both (47%), *Balanerpeton* + Dendrerpetidae (49%), *Isodectes* + *Neldasaurus* (47%), the adelospondyl clade (47%) and the sister-group relationship of Urocordylidae and Aïstopoda (43%) are each almost certainly better supported than any single alternative. In contrast, *Kotlassia* is found within Seymouriamorpha only in 28% of the replicates.

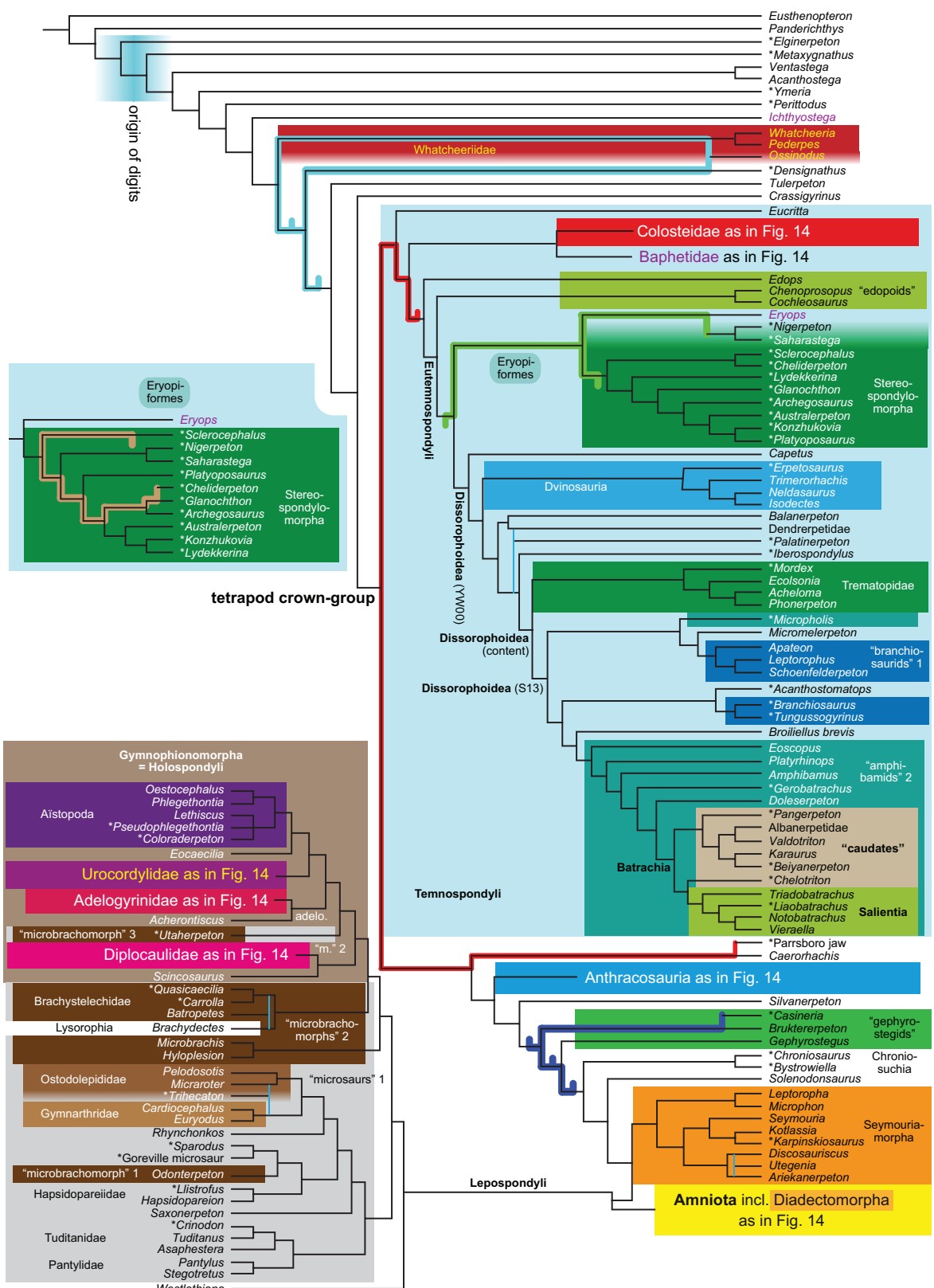

**Figure 17** Representation of all MPTs (length: 3,026 steps) from Analysis R6 (revised matrix, expanded taxon sample, constraint for the PH as in Analysis R3). Underlays as in Fig. 15.

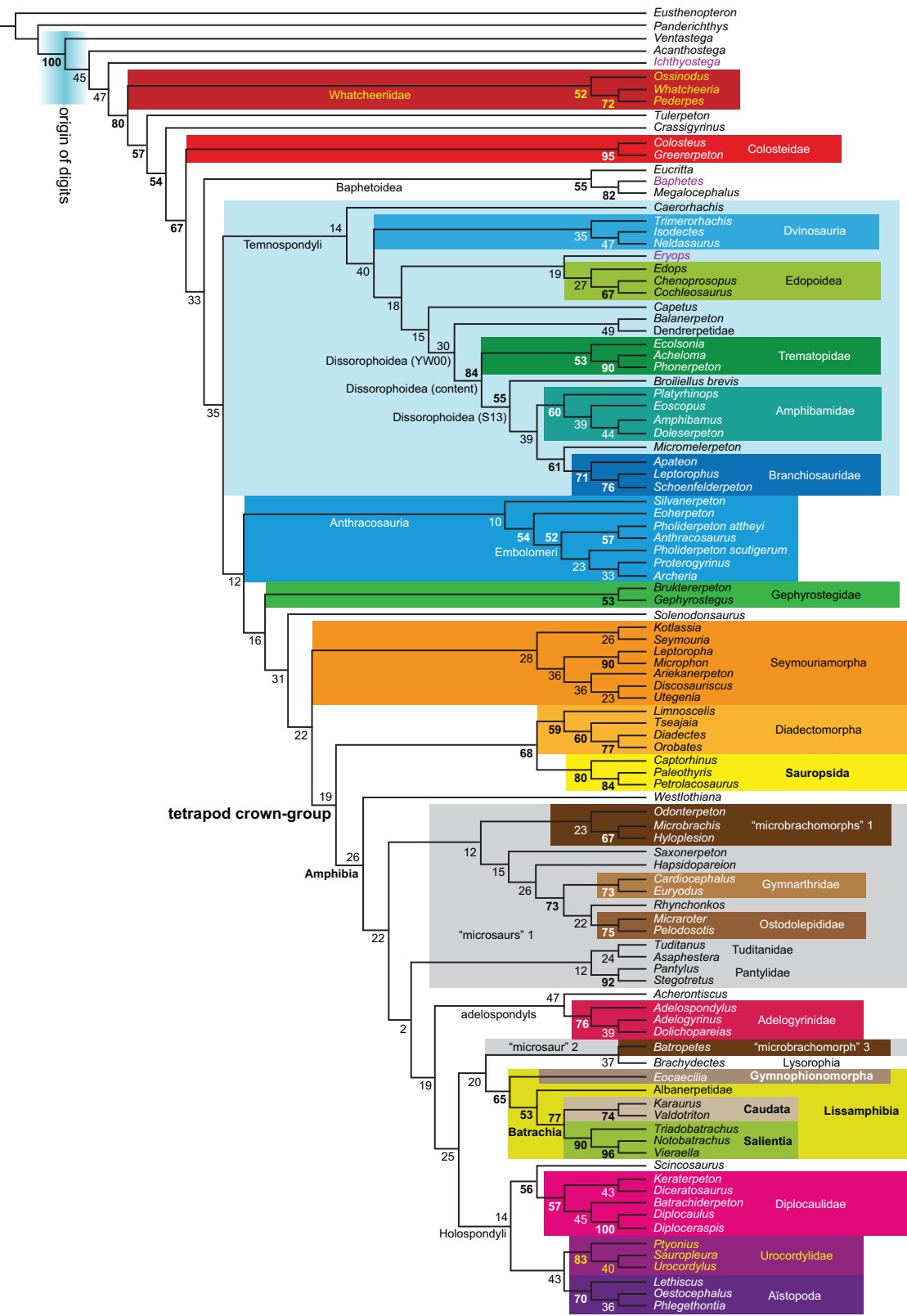

**Figure 18** **Bootstrap tree from Analysis B1 (revised matrix, original taxon sample, no constraint; compare Fig. 10).** Clades with a bootstrap percentage below 50 are included if they are compatible with those above 50; percentages of 50 and above are in boldface.



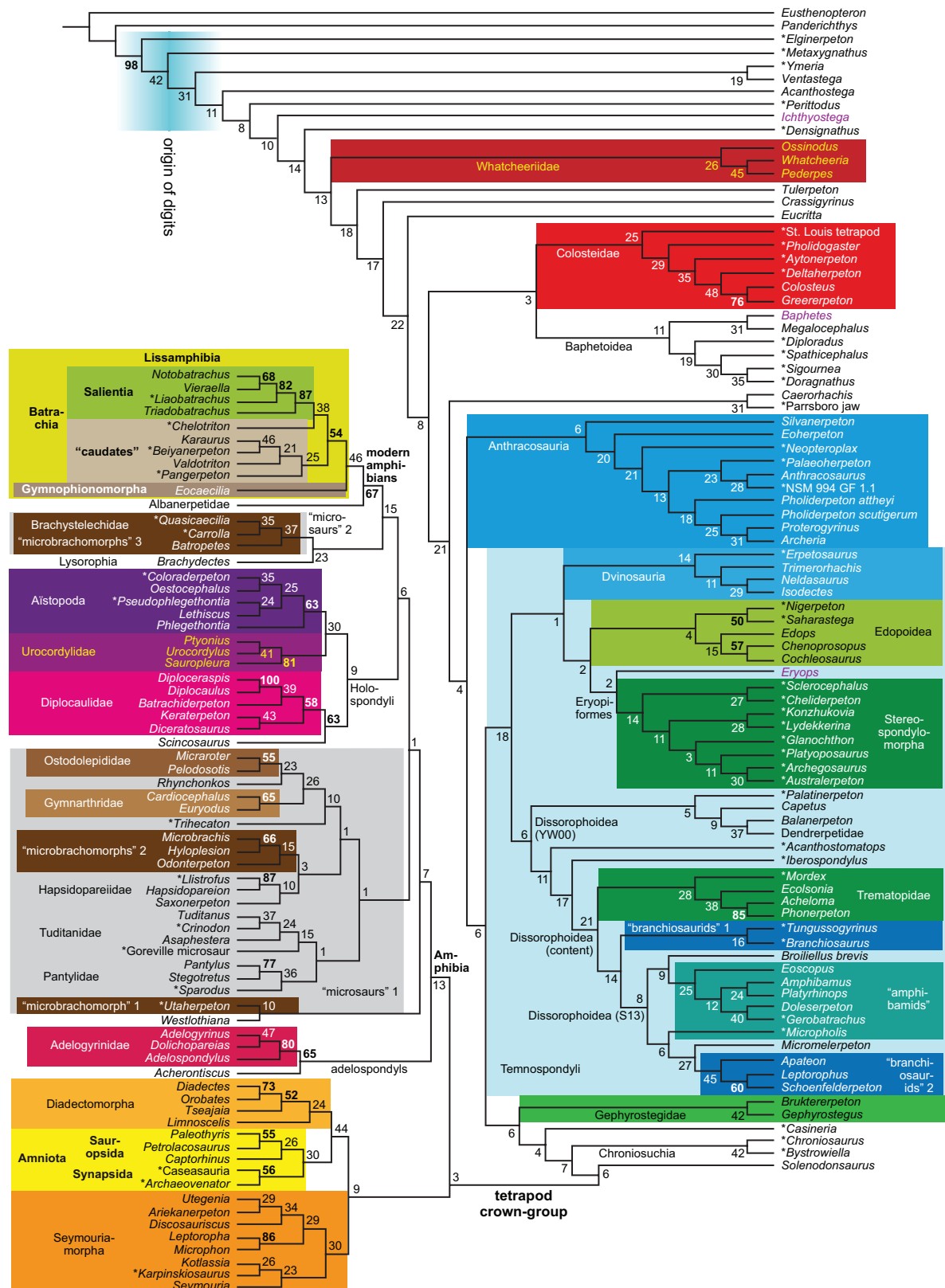

**Figure 19** Bootstrap tree from Analysis B2 (revised matrix, expanded taxon sample, no constraint; compare **Fig. 14**). See legend of **Fig. 18** for more information.

Adding taxa (Analysis B2) slightly increases the bootstrap support of four nodes, but—unsurprisingly—generally depresses the support of almost the entire tree (Fig. 19). Of its potentially controversial branches, only the sister-group relationship of *Nigerpeton and *Saharastega (50%), that of Microbrachis and Hyloplesion (66%), that of Scincosaurus and Diplocaulidae (63%), the clade of modern amphibians (Albanerpetidae + Lissamphibia: 67%) and Batrachia (54%) are found by 50% or more of the bootstrap replicates; Lissamphibia is found in 44% of the replicates, Karaurus + *Beiyanerpeton in 46%, Chroniosuchia and Gephyrostegidae in 42% each, and the adelospondyl clade (40%) and Doleserpeton + *Gerobatrachus in 40% each. The addition of two synapsids (Synapsida: 56%) has made Limnoscelis a diadectomorph (24%) outside of Amniota (30%) and reduced the support for the monophyly of Sauropsida to 26%, although support for Amniota + Diadectomorpha is considerably higher at 44%.

### Analysis EB

The results of our exploratory Bayesian analysis (Figs. 20 and 21) are remarkably similar to those of Analysis B2, which was conducted under the same settings. The most conspicuous difference may be how strong the support for the LH is: a node with a posterior probability (PP) of 92% (Fig. 20) would need to be broken to move the modern amphibians into Temnospondyli, not to mention one each with PPs of 88%, 76%, 74%, 73% and finally 64% (Dissorophoidea incl. *Iberospondylus).

Posterior probabilities of 75% or higher support *Elginerpeton crownward of Panderichthys (100%) and outside the rest of the ingroup (77%), Acanthostega crownward of Ventastega (75%), Ichthyostega crownward of Acanthostega (76%), Tulerpeton crownward of all whatcheeriids and *Densignathus (77%), Colosteidae including the *St. Louis tetrapod (97%) crownward of Whatcheeriidae and Crassigyrinus (86%) and rootward of Caerorhachis (again 86%), *Sigournea + *Doragnathus + *Spathicephalus + *Diploradus (87%) lying next to Megalocephalus (82%) in Baphetidae (again 82%), Anthracosauria (97%), Embolomeri (99%), Proterogyrinus outside a clade of all other embolomeres (94%), Balanerpeton + Dendrerpetidae (83%), Dvinosauria (77%), *Nigerpeton + *Saharastega (99%), *Konzhukovia + *Platyoposaurus (98%) next to *Australerpeton (81%), Ecolsonia as a trematopid (94%), Phonerpeton + Acheloma (100%), Kotlassia inside Seymouriamorpha (99%), Discosauriscidae excluding Kotlassia, Seymouria and *Karpinskiosaurus (91%), Leptoropha + Microphon (100%), *Caseasauria + *Archaeovenator (75%), Diadectomorpha (96%), a clade composed of Limnoscelis and the two diadectids (76%), Tuditanidae (89%) containing the *Goreville microsaur + *Crinodon (88%), Gymnarthridae (96%), Ostodolepididae (87%), Hapsidopareion + *Llistrofus (100%), Microbrachis + Hyloplesion (97%), *Sparodus as a pantylid (88%), Pantylus + Stegotretus (93%), a "Holospondyli" clade including all "nectrideans", "aïstopods", adelospondyls, *Utaherpeton, Brachydectes, "brachystelechids" and modern amphibians (76%), the adelospondyl clade (100%), Adelogyrinidae (95%), Urocordylidae incl. Aïstopoda (96%), Aïstopoda (100%), Scincosaurus + Diplocaulidae (86%), Diplocaulidae (100%), Diplocaulus + Diploceraspis (100%) as the sister-group of Batrachiderpeton (94%), a clade of Brachydectes, the "brachystelechids" and the modern

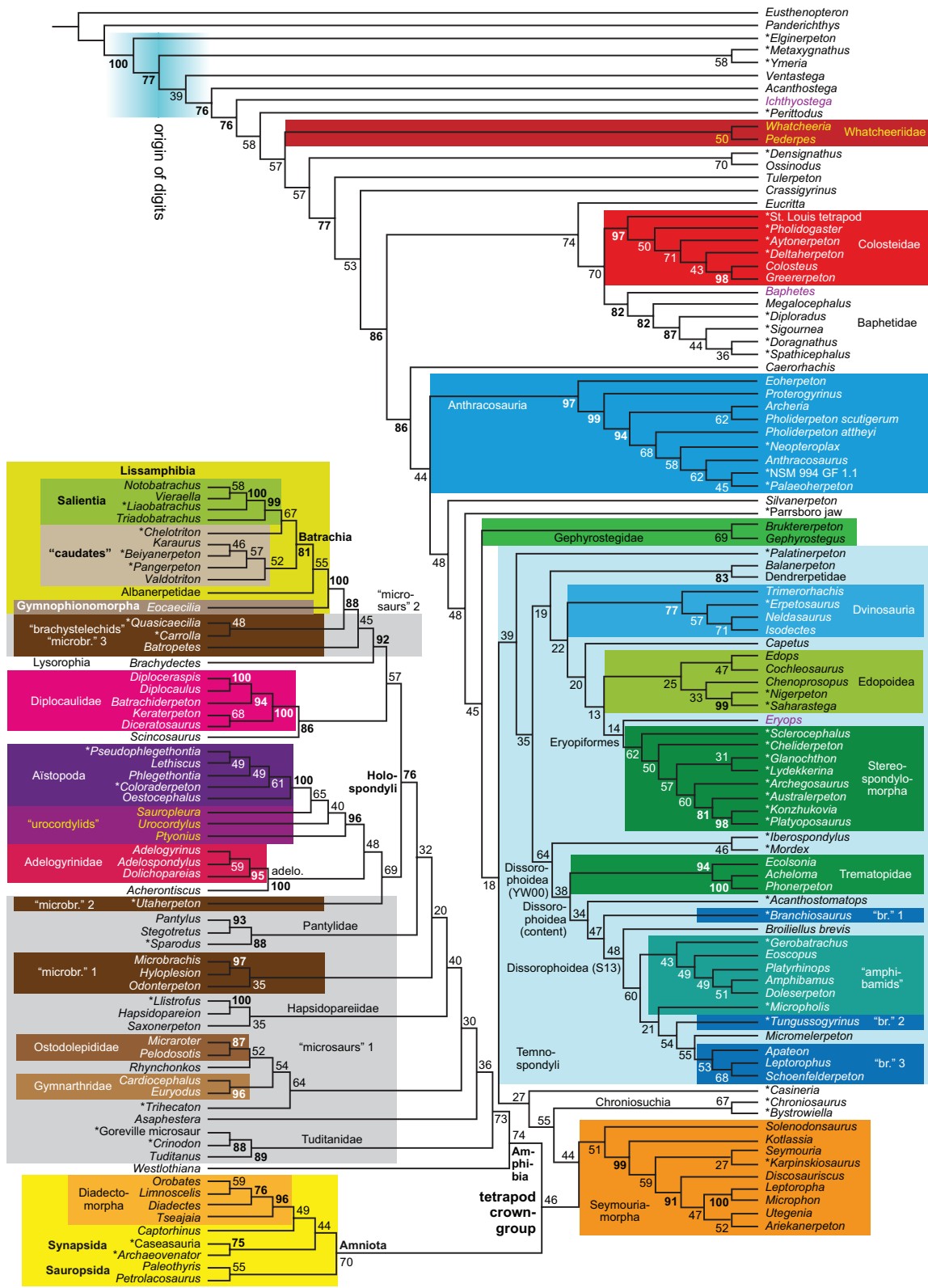

**Figure 20 Topology and posterior probabilities from Analysis EB (revised matrix, expanded taxon sample, no constraint; compare Figs. 14 and 19).** The numbers are posterior probabilities (in %), in boldface if 75 or higher. For branch lengths see Fig. 21. Abbreviations: br., branchiosaurids; microbr., microbrachomorphs.

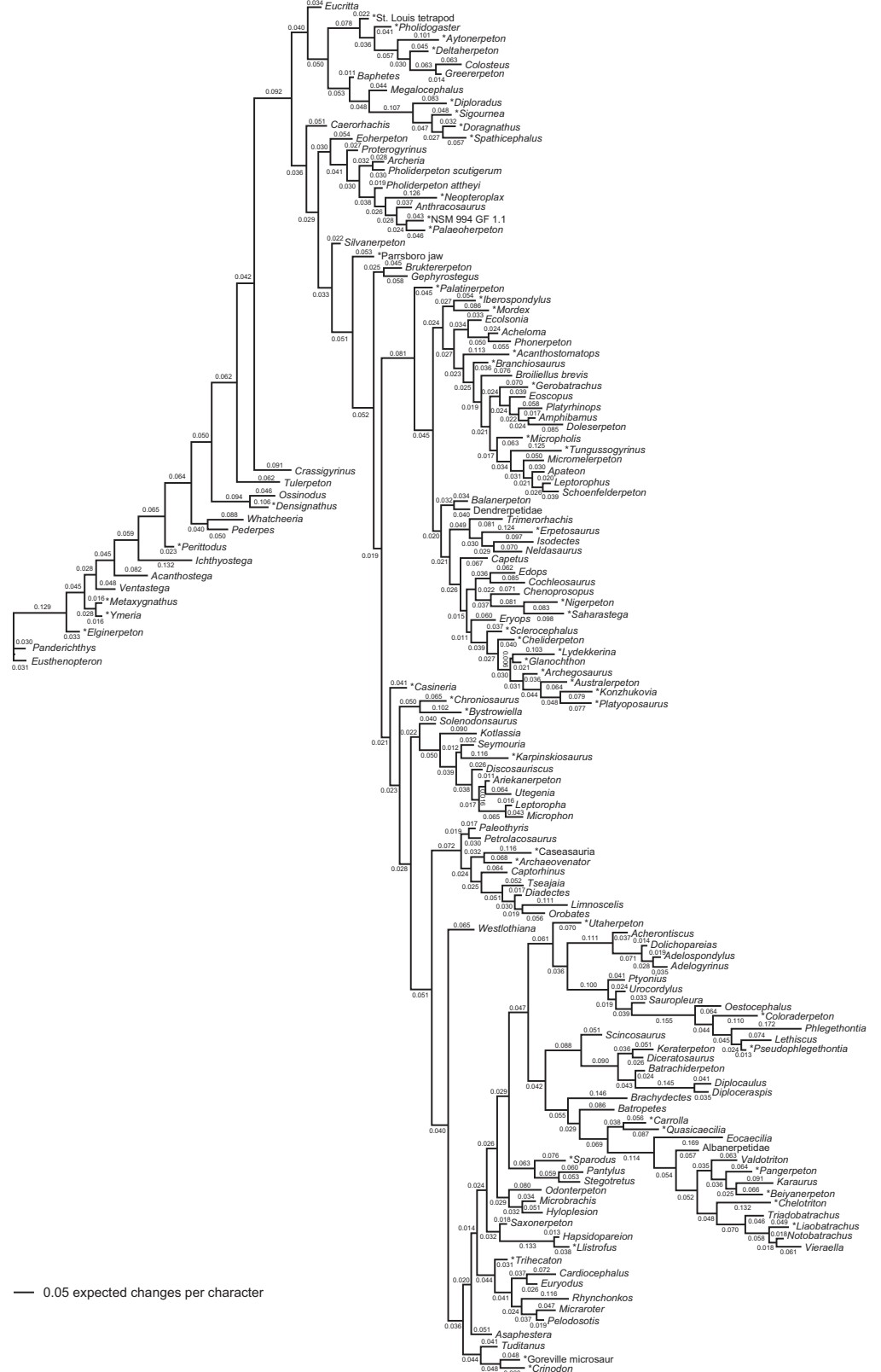

**Figure 21  Branch lengths from Analysis EB.** For nomenclature and branch support see Fig. 20.

amphibians (92%), a novel clade formed by the modern amphibians, *Quasicaecilia* and *Carrolla* to the exclusion of *Batropetes* and *Brachydectes* (88%), Lissamphibia (100%), Batrachia (81%), Salientia excluding *Chelotriton* (99%) and also excluding *Triadobatrachus* (100%). Like the *Eucritta*-Colosteidae-Baphetidae clade, Amphibia including *Westlothiana* has 74% posterior probability, the clade of all remaining amphibians has 73%, the colosteid clade of *Aytonerpeton*, *Deltaherpeton*, *Colosteus* and *Greererpeton* has 71%, as does *Isodectes* + *Neldasaurus*; Amniota incl. Diadectomorpha has 70%, as do Colosteidae + Baphetidae and *Ossinodus* + *Densignathus*. Gephyrostegidae reaches 69%, Chroniosuchia 67%.

Posterior probabilities below 50% are most common in Temnospondyli and its surroundings as well as in the "microsaur backbone". The strange finding of urocordylid paraphyly is weakly supported (65%). The *Ossinodus* + *Densignathus* clade is poorly supported as crownward of Whatcheeriidae (57%), *Perittodus* as rootward of it (again 57%), and *Perittodus* as crownward of *Ichthyostega* (58%); Chroniosuchia has weak support (55%) for its position crownward of Temnospondyli and *Casineria*. *Solenodonsaurus* is kept together with Seymouriamorpha by a posterior probability of 51%.

The longest branch is the terminal branch of *Phlegethontia* (0.172 expected changes per character; Fig. 21), followed by those of *Eocaecilia* (0.169), *Brachydectes* (0.146), *Ichthyostega* and *Chelotriton* (0.132 each), *Neopteroplax* (0.126), *Tungussogyrinus* (0.125) and *Erpetosaurus* (0.124). Internal branches with at least 0.1 expected changes per character are limited to Aïstopoda (0.155), *Diplocaulus* + *Diploceraspis* (0.145), *Hapsidopareion* + *Llistrofus* (0.133), the ingroup except *Panderichthys* (0.129), Lissamphibia (0.114), the adelospondyls (0.111), the clade of non-traditional baphetids (0.107) and Urocordylidae incl. Aïstopoda (0.100).

The tree has an ITRI of 71.2% compared to O1, 81.3% compared to O2, 96.7% compared to R1 and 84.2% compared to R4.

## DISCUSSION

Reevaluating the matrix of RC07 has revealed additional character conflict and polymorphism (on the latter's impact see *Watanabe, 2016*) and greatly increased the length of the MPTs. Evidently, the MPTs found by RC07 painted an oversimplified picture of the evolution of the limbed vertebrates.

The addition of taxa in Analyses R3–R6, B2 and EB has had unexpected effects that may have improved the reliability of the tree. As previously demonstrated (*Mortimer, 2006*; *Butler & Upchurch, 2007*; *Raven & Maidment, 2018*), every OTU in a data matrix can influence the position of every other OTU in the resulting trees.

The ITRIs (Table 5) show that our revisions of the scores had a noticeable but moderate impact on tree topology. Comparisons among trees from the original matrix (O1–O3) always reveal ITRIs above 85%, and similarly, nine of the 11 comparisons performed among trees obtained from the revised matrix (R1–R6, EB) yield ITRIs above 85%; in contrast, of the 12 comparisons of trees from the revised matrix to trees from the original matrix, only four show ITRIs above 85%, and two have the lowest values of the entire Table (71.2% and 72%). The method of analysis apparently played a lesser role:

the two comparisons of the Bayesian tree (EB) to parsimony trees based on the same matrix (with or without added taxa) yield ITRIs between 84% and 97%. The topological constraints also apparently played only a moderate role in determining tree similarity. This is shown by the fact that comparisons between trees supporting the same hypothesis (TH, LH or PH) have similarly large and widely overlapping ranges: 76.7–90.2% (TH), 81.3–96.7% (LH) and 78.9–88.6% (PH). Further, some trees supporting different hypotheses have high ITRIs (e.g., R6, supporting the PH, has an ITRI of 97.5% with respect to R5, which supports the TH), while other such comparisons yield very low ITRIs (e.g., 72% for R1 [LH] compared to O1 [TH]). Strikingly, among the trees that support the LH, EB is much more similar to R1 (96.7%), which contains fewer taxa, than to R4 (84.2%), while R4 has an intermediate ITRI of 92.7% with respect to R1. Thus, the ITRIs suggest that the observed similarities between trees result from a complex interplay between all the factors mentioned above, and possibly others not considered here.

Unstable areas of the tree and other phenomena highlight promising areas for future research. These include redescription of *Westlothiana* (currently being undertaken following the discovery of additional specimens: M. Ruta, pers. comm. 2015; *Clack & Milner, 2015*), **Eldeceeon* (currently being undertaken, see Materials and Methods: Treatment of OTUs: Taxa that were not added), the "microsaurs" *Asaphestera*, *Tuditanus*, *Odonterpeton* and *Trihecaton* (see below), **Sauravus* (the presumed sister-group of *Scincosaurus*), *Casineria* (especially in order to determine whether it is distinguishable from *Caerorhachis*), *Utaherpeton* (see below), **Macrerpeton* (see below), and quite possibly others.

## Bias in the matrices?

Given the fact that the present work bears on one of the most controversial questions that remain in vertebrate phylogeny, the origin of the modern amphibians, it is not surprising that several colleagues have wondered during the long genesis of this work whether various kinds of bias are present in the original matrix or in our revision of it. To some extent, we can test this: if our changes have mostly gone in one direction (say, increasing support for the LH or decreasing support for the TH), then either the matrix of RC07 was in some way biased against that direction, or our changes have been biased in that direction, or both. While we could perhaps not objectively distinguish between these three possibilities, we can address whether our revisions show such a preferential direction in the first place.

Our answer is firmly negative (see next section, "Bias in the scores?"). Indeed, we consider one of the most important results of our present work, together with the work of *Sigurdsen & Green (2011)*, *Langer et al. (2017)* and *Spindler et al. (2018*: online resource 3*)* as well as our previous work (*Marjanović & Laurin, 2008*, *2009*), to be the observation that accidental, unsystematic misscores are very common in published matrices—common enough to change the most parsimonious topologies.

Naturally, in many cases, RC07 simply could not have avoided incorrect scores because the correct ones were only published after 2006. We have not differentiated these in App. S2 or Data S4, because our goal was to reevaluate the matrix and the trees that result

from it, not the context in which it was made. Needless to say, however, this information could be extracted from the dates of the sources we cite in App. S1; some of it is summarized in Tables 6–8 and 10.

### Bias in the scores?

Tables 7, 9 and 11 list the characters at the top 20 ranks of changes marked red in App. S2, the characters at the top 20 ranks of changes marked green and all characters with changes marked blue, respectively (as explained in Results: Amount, distribution and impact of revised scores). In columns O and R of each table, we present by how many steps each of these characters supports the LH, the TH and/or the PH over its alternatives. Strikingly often, for example for the top four characters in Table 7, that number is zero both before (O) and after all our revisions put together (R) even if those revisions have caused dozens of changes. In all three tables, the only numbers larger than two in columns O and R are found in characters HUM 18/DIG 1 (ch. 219), which favored the TH over both of its alternatives by one step in RC07 (HUM 18 by one step, DIG 1 by none), but now disfavors it, again compared to both alternatives, by three steps after 24 red changes (Table 7), and JAW ART 1/SQU 2/DEN 8 (ch. 146), which favored the LH over both alternatives by one step in RC07, but now favors the PH by one step over the LH and the TH by four steps over the LH after 25 red and five blue changes (Tables 7 and 11), not to mention a green one (App. S2, Data S4). It is evident from Tables 7, 9 and 11 that support for all three hypotheses was present in the matrix of RC07, and that our revision has both added and removed support for all three, all in similar amounts. Table 9 shows in particular that many of our redefinitions of characters, including the most drastic ones, have had negligible effects, or none, on support for the hypotheses about the origins of the modern amphibians; Table 11 shows the same for our approach to ontogeny and heterochrony, and Table 7 for the many, many scattered changes that do not depend on interpretations of characters.

To quantify this impression, we performed binomial tests on the data from Tables 7, 9 and 11, both separately and on all these data simultaneously (Table 12), in QuickCalcs (*Motulsky, 2018*). As reported in Table 12, none of the separate tests are significant ($p > 0.10$ in all cases), and neither is the test on the pooled data ($p = 0.6835$). The method of our test consists in counting how many parsimony steps were gained by the LH and the TH by our rescoring. For instance, if the original score of a character in RC07 favored the TH over the LH by one step and if, after our rescoring, it favored the LH over the TH by one step, we scored a two-step difference in favor of the LH. We did this over all the characters listed in Tables 7, 9 and 11, and tested, in each case, if the observed distribution could be explained by changes that randomly favored either of the hypotheses ($H_0$). In these tests, we ignored characters whose rescoring did not alter the relative support of LH and TH. We did not test if our scores altered the relative support of the PH because there is no reason to expect bias in favor of the PH either in RC07 or in our revision (Tables 1 and 4). We performed all tests as two-tailed because there are a priori reasons to think that we could have favored either hypothesis: the LH because this is the hypothesis that we have supported in the past, and the TH because we tried to avoid biasing the results (e.g. by bold

**Table 12 Tests of the hypotheses that our revisions of scores have favored the LH or the TH.**
These tests rest on a binomial distribution of the probability of finding patterns at least as asymmetrical as the ones observed in Tables 7, 9 and 11 separately and together (last test). $H_0$ is that the changes are random, $H_1$ is that they are biased in favor of the LH or the TH; all four tests fail to reject $H_0$ ($0.10 < p < 0.69$; two-tailed because $H_1$ consists of two diametrically opposed hypotheses corresponding with the two tails of the distribution). We have not tested bias in favor of the PH because there is no reason to expect it either in RC07 or in our revision (Tables 1 and 4). "Changes" are the changes between columns O and R of the respective Tables. See the text for more information.

**Table 7**

| Rank | Character | Changes favoring LH | TH |
|------|-----------|---------------------|-----|
| 5 | 40 | | 1 |
| 7 | 102 | 2 | |
| 7 | 119 | 1 | |
| 10 | 134 | 2 | |
| 11 | 146 | | 5 |
| 12 | 17 | 1 | |
| 12 | 66 | | 1 |
| 12 | 219 | 4 | |
| 15 | 34 | 1 | |
| 16 | 27 | | 1 |
| **Total** | | **11** | **8** |
| ***p* = 0.6476** | | | |

**Table 9**

| Rank | Character | Changes favoring LH | TH |
|------|-----------|---------------------|-----|
| 3 | 95 | 2 | |
| 4 | 25 | 1 | |
| 7 | 145 | 1 | |
| 12 | 131 | 2 | |
| 14 | 69 | | 1 |
| 19 | 265 | | 1 |
| 20 | 31 | 2 | |
| **Total** | | **8** | **2** |
| ***p* = 0.1094** | | | |

**Table 11**

| Rank | Character | Changes favoring LH | TH |
|------|-----------|---------------------|-----|
| 1 | 205 | 1 | |
| 1 | 224 | | 1 |
| 5 | 146 | | 5 |
| 8 | 22 | 1 | |
| 8 | 27 | | 1 |
| 8 | 69 | | 1 |
| 8 | 189 | | 1 |

(Continued)

| Table 12 (continued). | | | |
|---|---|---|---|
| **Table 11** (continued) **Rank** | **Character** | **Changes favoring** | |
| | | **LH** | **TH** |
| 14 | 37 | 3 | |
| 14 | 66 | | 1 |
| 14 | 72 | | 1 |
| 14 | 74 | | 1 |
| 14 | 87 | | 1 |
| 14 | 95 | 2 | |
| 14 | 103 | 1 | |
| 14 | 110 | 1 | |
| 14 | 113 | | 1 |
| 14 | 196 | | 1 |
| 14 | 252 | 1 | |
| **Total** | | **10** | **15** |
| **p = 0.4244** | | | |
| **Global test** | | | |
| | | **Changes favoring** | |
| | | **LH** | **TH** |
| Table 7 | | 11 | 8 |
| Table 9 | | 8 | 2 |
| Table 11 | | 10 | 15 |
| **Total** | | **29** | **25** |
| **p = 0.6835** | | | |

interpretations of morphological data) in favor of the LH, which may have resulted in a bias against the LH. Regardless, performing the tests as one-tailed yields $p > 0.05$ in all cases.

There is very little evidence, too, of disagreements between Ruta & Coates and us about how to interpret the homology of the morphological features coded here. Almost all of them belong to TRU VER 4, our ch. 257, which is discussed above (Results: Amount, distribution and impact of revised scores) and at greater length in App. S1; Table 7 shows that the 50 changes we have made to its scores (all red, no green or blue ones: App. S2, Data S4) have not had any impact on its irrelevance to the origins of the modern amphibians. Even the character for which the OTUs were evidently scored blockwise (PREMAX 7, our ch. 3; see Materials and Methods: Treatment of characters: Blockwise scoring of taxa and App. S1) has no net bearing on this question either before or after our extensive revision of this character (Table 9).

Further evidence of accidental, unsystematic misscores is constituted by the numerous discrepancies (App. S1) between the matrix of RC07 and the careful, detailed, splendidly illustrated works by the same authors (*Lebedev & Coates, 1995*; *Coates, 1996*; *Ruta, Milner & Coates, 2002*; *Klembara & Ruta, 2004a*, *2004b*, *2005a*, *2005b*; *Ruta & Clack, 2006*; *Milner & Ruta, 2009*) that we have used as sources for our revision.

Therefore, we are certain that the vast majority of, at least, the straightforwardly indefensible scores in the matrix of RC07 are typographic or similar errors as described in the Introduction (Aims: Accuracy of the matrix of RC07). Their large number simply underscores that morphological phylogenetics, as great as its rewards are, is extremely work-intensive—not so much in terms of difficulty as in terms of sheer amount of time required.

### Bias in character selection?

While the character sample is much smaller than it could be (discussed below: Characters: Persisting problems with the character sample), we see no evidence to suggest that it is biased for or against a hypothesis on lissamphibian origins. Characters supporting all three of these hypotheses are represented in similar numbers (Tables 7, 9 and 11), and we have not noticed any glaring omissions of relevant characters. It is not the case either that characters supporting one of these hypotheses are systematically overcounted by being duplicated or multiplied as redundant characters. Instead, even though we have found and redefined or merged several redundant characters all across the matrix, there is evidence for systematic avoidance of redundancy: the matrices of *Ruta, Coates & Quicke (2003)* and RC07 lack many characters that had been used in previous matrices for analysis of the phylogeny of limbed vertebrates and that would have been obvious choices to include except for the fact that they would be redundant with others. For example, the length ratio of the antorbital and postorbital parts of the skull has been popular (to pick some of the most recent examples: *Clack et al., 2012b*: ch. 328; *Clack et al., 2016*: ch. 164 and 165; *Pardo et al., 2017*: ch. 32; *Pardo, Small & Huttenlocker, 2017*: ch. 25; *Spindler et al., 2018*: ch. 3), but would have made a correlated mess of FRO 2, TAB 7, NOS 4, ORB 5, SKU TAB 1, VOM 1, VOM 13 (our ch. 27, 57, 83, 90, 95, 102, again 102) and possibly others. We think it was deliberately kept out of the matrix for this reason. Of the redundant characters that did make it into the matrix of *Ruta, Coates & Quicke (2003)*, several were eliminated by RC07; an example is PAL 3 (ch. 129 of *Ruta, Coates & Quicke, 2003*: "Palatine excluded from (0) or contributing to (1) interpterygoid vacuities"), which was almost identical to VOM 10 (ch. 124 of *Ruta, Coates & Quicke, 2003*: "Vomer in contact with anterior ramus of pterygoid (0) or not (1)"; ch. 129 of RC07: "Vomer contact with pterygoid palatal ramus: present (0); absent (1)").

## Methods of phylogenetic analysis

### Bayesian inference and parsimony in comparison

We consider our Bayesian analysis exploratory because the behavior and performance of Bayesian inference on datasets like ours have not been studied. Firstly, the amount and distribution of missing data may be a matter of concern. We are aware of three studies of their effects on Bayesian inference:

*Wright & Hillis (2014)* explicitly intended to study the performance of Bayesian inference with morphological data. They simulated matrices of exclusively binary characters and scored all characters that evolved at a given rate as unknown for selected taxa; this may be somewhat realistic for molecular data, where different genes (each with its own rate) may have been sequenced for different taxa in a supermatrix, but makes limited sense for morphological data, where missing data are clustered by body

parts, not by rate of evolution. *Wright & Hillis (2014)* found that Bayesian inference outperforms parsimony under the conditions they studied.

Using contrived, simulated and empirical DNA datasets, *Simmons (2012a*, *2012b)* studied what happens when different taxa are scored for different characters so that some taxa have no scored characters in common at all, as may (again) happen when different genes have been sequenced for different taxa in a supermatrix. Given a matrix with non-overlapping taxa, parsimony cannot find only a single MPT which has them as sister-groups; instead, it will return consensus trees with polytomies when ever such taxa are not kept far enough apart by character states they share with other taxa. Parametric methods (Bayesian inference and maximum likelihood) instead try to compensate and can therefore find a single optimal tree which contains such taxa as sister-groups. Sometimes this turns out to be correct; however, *Simmons (2012a*, *2012b)* found that this situation routinely caused parametric methods to find strong support for wholly spurious clades—even under ideal conditions (no homoplasy, a perfectly fitting model of evolution, a perfect alignment, no rate heterogeneity). Moreover, which spurious clades were found was very sensitive to small changes in the scores of a matrix, despite the high support values. For datasets with missing data that are distributed as in his studies, *Simmons (2012a*, *2012b)* recommended that nodes and support values found by parametric analyses in parts of the tree that are not resolved in the strict consensus of a parsimony analysis of the same dataset should only be accepted after special scrutiny of their causes.

In our matrix, missing data are clustered by body parts, so the findings of *Wright & Hillis (2014)* may not translate to our situation; all OTUs except *Casineria* and the isolated lower jaws we added are scored at least for part of the dermal skull, so *Simmons's (2012a*, *2012b)* almost opposite conclusion may not translate to our situation either.

Of the four most recent studies of the performance of Bayesian inference on matrices of morphological characters, two (*O'Reilly et al., 2016*; *Puttick et al., 2017*) did not mention the problem of missing data at all, and *Goloboff, Torres & Arias (2018)* omitted it from most of their comparisons of Bayesian inference to other methods. *O'Reilly et al. (2018*: 106*)* stated that "our experiments do not attempt to simulate non-contemporary taxa or address the problem of missing data, qualities of palaeontological data that are of a level of complexity that is beyond the current debate." For us, of course, this is *most* of the current debate; we are quite surprised that *O'Reilly et al. (2018)* chose to publish their statement in the journal *Palaeontology*, whose very name suggests non-contemporary taxa and missing data.

A second issue was raised by *Puttick et al. (2017)*, who attempted to investigate the impact of tree shape (full symmetry vs. maximum asymmetry) on the performance of different methods. They found that all methods performed badly on the most basal nodes of a Hennigian comb, and that Bayesian inference performed least badly. *Goloboff, Torres & Arias (2018*: 420*)* showed that *Puttick et al. (2017*; later also *O'Reilly et al., 2018)* had actually tested something completely different: their symmetric tree had unitary branch length, while their asymmetric tree was ultrametric, so that the least nested

terminal branches were many times longer than the internal branches (which had unitary length). Their experiment thus amounts to a test of susceptibility to long-branch attraction, not to tree shape. The trees we find are much closer to the asymmetric than to the symmetric end of the spectrum; the branch lengths, however, are much closer to unitary than to ultrametric, with short terminal branches making up the majority of the Devonian part and being scattered over the rest of the "trunk" (Fig. 21). Indeed, apart from the adelospondyls and the aïstopods, our Bayesian tree (Fig. 21) places most OTUs remarkably close to their relative stratigraphic positions.

Thirdly, it remains unknown whether Bayesian inference outperforms parsimony on datasets whose amount and distribution of homoplasy is like that of ours. For matrices with average evolutionary rates approaching three changes per character per tree, the performance of Bayesian inference and parsimony converges when the number of characters increases toward 1,000 (O'Reilly et al., 2016, 2018; Puttick et al., 2017; perhaps also Wright & Hillis, 2014: fig. 6). Even with its many multistate characters, our matrix does not contain a number of states equivalent (or close) to 1,000 binary characters. However, as listed above, the consistency indices are below 0.22 in all our parsimony analyses and below 0.19 for all those with the expanded taxon sample; this translates to an average of about five changes per character per tree, well outside the range studied by Wright & Hillis (2014: two to three changes). Similarly, O'Reilly et al. (2016, 2018) and Puttick et al. (2017) studied datasets with consistency indices between 0.26 and 1.0; most of this range occurs only in very small (or contrived) datasets. Individual characters in our matrix show a wide range of rates—some change states only once per tree, some change states 30–40 times on every tree, the extreme being 51 (Analyses R1, R3) to 73 changes (R4) for SKU TAB 1 (ch. 95, five states).

To keep their calculations in a feasible timeframe, O'Reilly et al. (2016, 2018) and Puttick et al. (2017) reduced all their parsimony results to the majority-rule consensus. This must have overestimated the precision of parsimony and may have underestimated its accuracy.

Goloboff, Torres & Arias (2018) noted that all preceding studies had assumed a common branch length parameter for all characters to simulate their datasets, so that if one character had an increased probability of changing along a given branch, all others had proportionally increased probabilities of changing along that same branch. As Goloboff, Torres & Arias (2018) pointed out, this is highly unrealistic for morphological data; indeed, it does not fit our matrix (where different characters are stable or labile in different parts of the same tree). O'Reilly et al. (2018) responded to several criticisms by Goloboff, Torres & Arias (2018), but left this one unaddressed.

Finally, ordered characters (let alone stepmatrices) were not used in any of the studies cited here, so the impact of ordering remains unknown. In Wright & Hillis (2014) and O'Reilly et al. (2016), all characters were binary; in Goloboff, Torres & Arias (2018: 411), interestingly, all characters had four states, which is far from realistic as well.

Parametric methods of phylogenetics are generally less sensitive to long-branch attraction than the non-parametric method called "maximum parsimony" is (as inadvertently confirmed by Puttick et al., 2017 and O'Reilly et al., 2018). Long-branch

attraction is not unknown in morphological data. Indeed, we suspect that long-branch attraction could be responsible for parts of our MPTs; in particular, our results concerning the placement of adelospondyls, aïstopods (*Pardo et al., 2017*) and urocordylids (*Pardo et al., 2018*) should be taken with caution, as discussed below. However, these very results are supported by Analysis EB (posterior probability of 76% for all these taxa being highly nested amphibians close to Lissamphibia, and 69% for their forming an exclusive clade with *Utaherpeton*; Fig. 20), and the internal branches in that region of the tree are long but not extremely long (Fig. 21). Although we are looking forward to further developments of parametric methods and further tests of their performance with paleontological datasets, the extent to which the trees we present here are wrong will more likely be discovered through improvements to our matrix than improvements to the methods of analysis. Indeed, before we updated the scores of the aïstopod *Lethiscus* after *Pardo et al. (2017)* and added two more (*Coloraderpeton* and *Pseudophlegethontia*), the abovementioned PP were 98% and 97%, respectively; this may have been a case of "garbage in, garbage out", a law to which no method is immune.

### Reweighting and equal weighting in comparison

Our review of their datasets indicates that [...] Goloboff *et al.* (2017) [...] their simulated datasets are not individually empirically realistic, with many matrices dominated by characters with very high consistency and an unrealistically small proportion of characters exhibiting high levels of homoplasy. The datasets simulated by Goloboff *et al.* (2017) have qualities that strongly bias in favor of parsimony phylogenetic inference, and implied-weights parsimony in particular, as the presence of large numbers of characters that are congruent with the tree allows implied weights to increase the power of these 'true' congruent characters. This effect will not be possible when increased levels of homoplasy are present [...].

– O'Reilly et al., 2018: 106–107, about Goloboff, Torres & Arias, 2018

Any method of phylogeny inference works when its assumptions are met. Therefore, the fact that the method variously called reweighting, implied weighting or a posteriori weighting is logically circular (see below) is not an argument against using it when its assumptions are met; when they are, it outperforms all others (*Goloboff, Torres & Arias, 2018*). We think they are not met in our dataset and have therefore not used this method.

Specifically, in the empirical matrices *Goloboff, Torres & Arias (2018)* cited and the matrices they simulated, there is an exponential or nearly exponential distribution of characters so that a plurality of characters has, on the MPTs found by unweighted parsimony, no homoplasy at all, that is, as few steps as theoretically possible for a parsimony-informative character with the given number of states, in other words a consistency index of 1; next most common are characters with one extra step ($c_i = 0.5$), then two and so on (*Goloboff, Torres & Arias, 2018*: especially fig. 1). In our matrix, homoplasy-free characters are much less common (Data S8). Given the original taxon sample (Analysis R1; Fig. 22A), the distribution zigzags around an exponential one, but the most common number of extra steps is one, followed by two and only then (by a

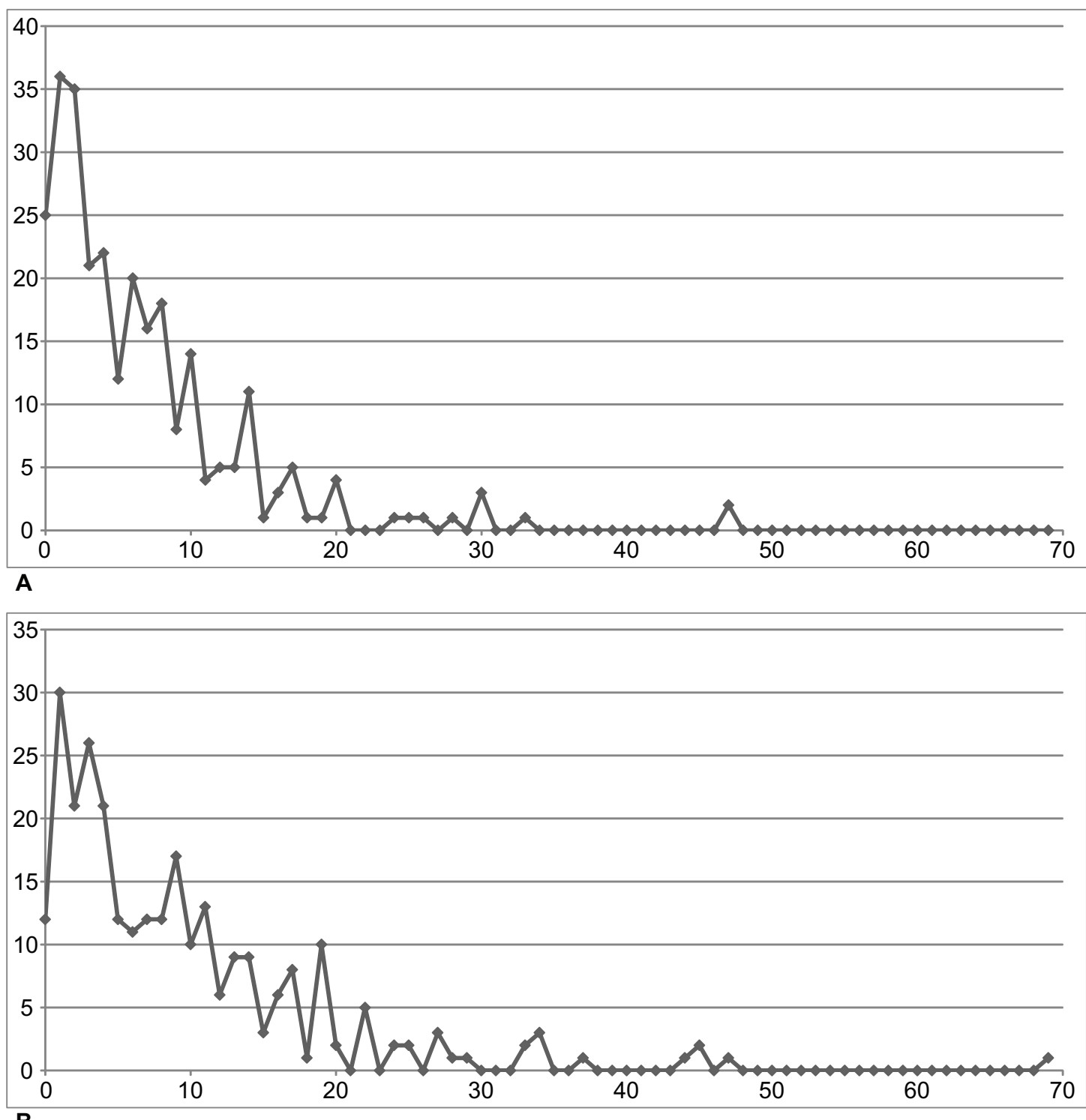

**Figure 22 Homoplasy distribution in our matrix.** The number of extra steps (*x*-axis) is the number of observed steps (in Mesquite, on the tree in Data S3 for each analysis) minus the minimum possible number of steps, which is the number of states minus one (for ordered and unordered characters as well as for both of our stepmatrix characters). The number of characters that have each number of extra steps is plotted on the *y*-axis. The line between the data points is meaningless, but makes it easier to compare the distributions to an exponential curve. Compare *Goloboff, Torres & Arias (2018*: fig. 1(a)*)*. (A) Original taxon sample (Analysis R1); the highest number of extra steps is 47, but we plot to 69 for comparison to (B). (B) Expanded taxon sample (Analysis R4); the highest number of extra steps is 69.

large margin) by zero; when taxa (and thus homoplasy) are added (Analysis R4; Fig. 22B), the most common number is still one, then three, then two and four (equally), then nine, then eleven, then zero, five, seven and eight (all equally), then the rest vaguely approaches an exponential distribution with a wide variance. In both taxon samples, several binary characters have the maximum possible number of steps given our matrix (i.e., of one state, no two occurrences are unambiguously optimized as homologous, or those that are are canceled out by reversals).

Even if it retrieves an accurate topology, we think that reweighting is prone to underestimating homoplasy and overestimating the support for this topology. Accurate topologies are not the only goal of our research.

(Reweighting attempts to downweight characters depending on how homoplastic they are. The circularity consists in the fact that the amount of homoplasy for each character is determined by optimizing the characters on the MPTs found by unweighted parsimony, which amounts to assuming that these trees are correct or nearly so. By downweighting characters that are incongruent with those trees, reweighting reduces character conflict and thus increases support and resolution, but not necessarily accuracy. As *Goloboff, Torres & Arias (2018)* emphasized, unweighted and reweighted parsimony are equally sensitive to long-branch attraction; furthermore, characters can be falsely congruent with each other for reasons other than long-branch attraction—redundancy and correlation in particular.)

## Phylogenetic relationships

In this section we mention the bootstrap percentages (BPO and BPE from Analyses B1 and B2, shown in Figs. 18 and 19, respectively) that support or contradict clades found by our analyses. Most of them are below 50, especially for the expanded taxon sample (Fig. 19); this highlights the suboptimal ratio between number of characters and number of taxa, as well as character conflict and wildcard taxa. As in Figs. 18 and 19 as well as Tables 3 and 4, bootstrap percentages of 50 and above are presented in boldface.

Likewise, we mention the PPs (in %: PP from Analysis EB, shown in Fig. 20) that support or contradict clades found by our analyses. As in Fig. 20 and Table 3, PPs of 75 and above are presented in boldface.

### *Devonian taxa, Whatcheeriidae and* Perittodus

At the base of the ingroup, RC07 found the "textbook" Hennigian comb (Figs. 1 and 23A): (*Panderichthys* (*Ventastega* (*Acanthostega* (*Ichthyostega* (*Tulerpeton*, post-Devonian clade))))). However, the position of *Ichthyostega* soon began to be questioned (*Ahlberg et al., 2008*: supplementary information; *Clack et al., 2012a*) as more information about it was discovered and published (*Callier, Clack & Ahlberg, 2009*; *Ahlberg, 2011*; *Clack et al., 2012a*; *Pierce, Clack & Hutchinson, 2012*; *Pierce et al., 2013*; *Clack & Milner, 2015*). In our analyses of the original taxon sample (Fig. 23B), *Ichthyostega* is equally parsimoniously crownward of *Ventastega* (BPO = 47; BPE = 11; PP = **76**) and *Acanthostega* (BPO = 47; BPE = 10; PP = **76**), which are—only in this case—sometimes found as sister-groups (contradicted by a BPO of 45, a BPE of 19 and a PP of **75**), or

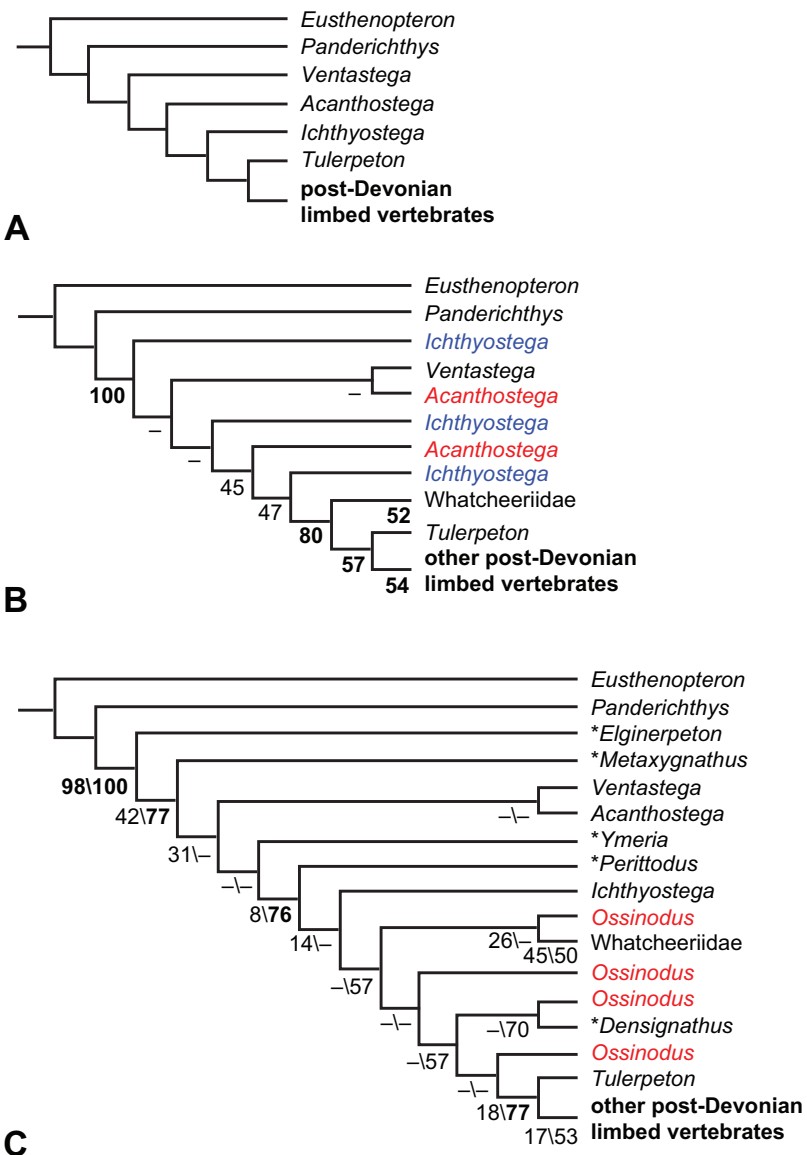

**Figure 23 Hypotheses on the relationships of Devonian limbed and possibly limbed vertebrates.** (Whatcheeriidae and *Perittodus, which have limbs, are only known from the Carboniferous with certainty.) Equally parsimonious positions of the same OTU are highlighted in color in this and the following figures. (A) RC07 and references therein. (B) Our results from the same taxon sample as RC07 (Analyses R1–R3). Numbers below internodes are BPO, in boldface if 50 or higher. (C) Our results from the expanded taxon sample (R4–R6). Numbers below internodes are BPE\PP, BPE in boldface if 50 or higher, PP in boldface if 75 or higher. Analysis B2 places *Ymeria next to *Ventastega* (BPE: 19), *Acanthostega* one node crownward of them (BPE: 11) and *Densignathus one node rootward of Whatcheeriidae (BPE: 13); EB places *Ymeria next to *Metaxygnathus (PP: 58), *Acanthostega* one node crownward of *Ventastega* (PP: **76**) and *Perittodus one node crownward of *Ichthyostega* (PP: 58).

rootward of *Acanthostega* but still crownward of *Ventastega*, or rootward of both.
We will not be surprised if the last of these three options will be upheld by analyses with larger character and outgroup samples, as indeed it has been (in the absence of *Ventastega*)

in the analysis of *Pardo et al. (2017)*. The picture of *Ichthyostega* as the Godzilla of mudskippers (*Pierce, Clack & Hutchinson, 2012*; see also *Coates & Clack, 1995*), an animal that acquired an amphibious lifestyle wholly independently from more crownward walking and limbless tetrapodomorphs that lived after Romer's Gap, may well be accurate.

However, in our expanded taxon sample (Fig. 23C), the position of *Ichthyostega* is stabilized crownward of *Ventastega*, *Acanthostega*, *Ymeria* and even the Tournaisian *Perittodus* (BPE = 10; contradicting a PP of 58, which holds *Perittodus* crownward).

The added Devonian taxa are all poorly known, as is *Perittodus*. Yet, all of them have fully resolved positions. This appears to be due to a combination of the relatively dense sample of lower-jaw characters and our similarly dense taxon sample.

Within this mostly Devonian grade, the Carboniferous whatcheeriids nest rootward of *Densignathus* (PP = 57; against a BPE of 13). This is particularly intriguing in the light of the "whatcheeriid-grade" skull bones that were found with *Densignathus*; we are much less certain than *Daeschler, Clack & Shubin (2009)* that they do not actually belong to *Densignathus*. *Ossinodus* is found as a whatcheeriid in the original taxon sample (BPO = **52**; BPE = 26), specifically as the sister-group to the other two as in RC07 (BPO = **72**; BPE = 45). This contrasts with the topology (*Pederpes* (*Ossinodus*, *Whatcheeria*)) found by *Pawley (2006)* before the recent descriptions of further *Ossinodus* material (*Warren, 2007*; *Bishop, 2014*). When we add taxa, *Ossinodus* gains three additional positions around *Densignathus* (PP for the two as sister-groups = 70; against a BPE of 13).

In all analyses, the Devonian *Tulerpeton* emerges crownward of Whatcheeriidae (BPO = **57**, BPE = 18; PP = **77**), neither rootward as in RC07 and *Pawley (2006)* nor as its sister-group as in *Ruta & Bolt (2006)* and the almost identical *Ruta (2009)* or in *Clack & Klembara (2009)*. Creating the former topology in Mesquite adds two steps each to Analyses R1 and R4; for the latter, three extra steps need to be added to Analysis R1 and four to R4 (three are required to make *Tulerpeton* a member of an *Ossinodus*-*Densignathus* clade one node crownward of Whatcheeriidae).

Taken at face value, our results support the idea (*Anderson et al., 2015*; *Clack et al., 2016*) that more than one clade of limbed vertebrates survived the Hangenberg event at the Devonian-Carboniferous boundary. We eagerly await the forthcoming redescription of **Elpistostege* based on a whole articulated CT-scanned skeleton (*Cloutier & Béchard, 2013*; *Cloutier et al., 2016*); the improved sample of Devonian stem-tetrapods and probably characters in the accompanying phylogenetic analysis will likely contribute to a better understanding of this part of the tree.

### More Mississippian mysteries

RC07 found Colosteidae, *Crassigyrinus* and Whatcheeriidae to be successively more crownward (Figs. 1 and 24A). Instead, like *Ruta (2009)*, we find them successively more rootward in all analyses (Figs. 23B, 23C, 24B and 24C; PP = **86**; BPO = **67** for Colosteidae + more crownward taxa including *Eucritta*; BPE = 22 for *Eucritta* + more crownward taxa including Colosteidae). This agrees with other recently published analyses; we attribute this to our corrections. For example, RC07 (and *Ruta, Coates & Quicke, 2003*) had scored *Whatcheeria* as lacking the preopercular bone in the skull (state PREOPE 1(1),

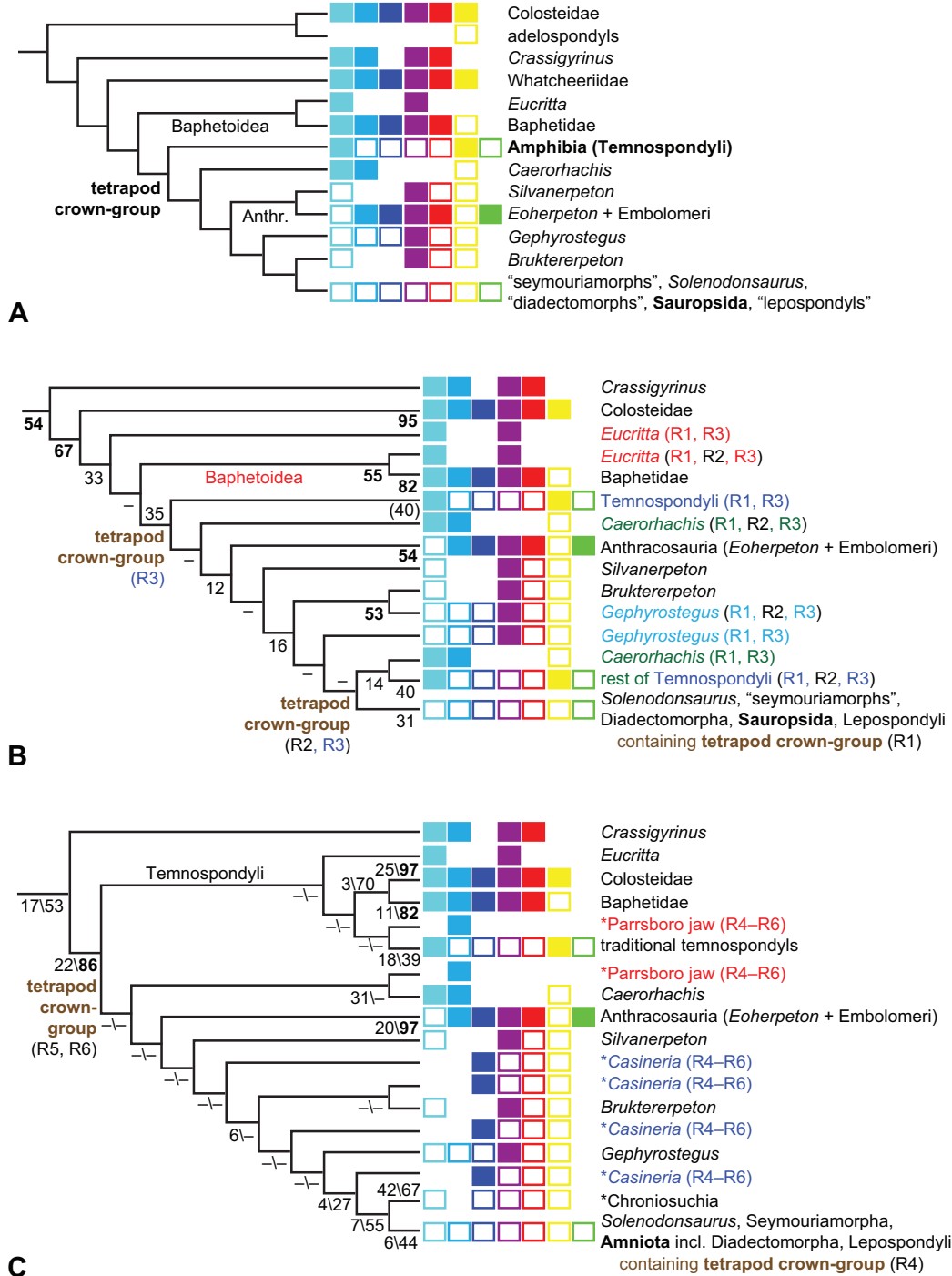

**Figure 24** **Hypotheses on the relationships of post-Devonian limbed vertebrates, and distribution of several character states.** (A) RC07; the entire clade shown here equals the "post-Devonian limbed vertebrates" of Fig. 23A. Anthr., Anthracosauria. (B) Our results from the original taxon sample (R1–R3). The entire clade shown here equals the "other post-Devonian limbed vertebrates" of Fig. 23B. Numbers are BPO, in boldface if 50 or higher. Analysis B1 finds *Caerorhachis* next to the other temnospondyls, but places Temnospondyli in the more rootward of the two indicated positions and also finds *Silvanerpeton* next to the other anthracosaurs (BPO: 10). (C) Our results from the expanded taxon sample (R4–R6). The entire clade shown here equals the "other post-Devonian limbed vertebrates" of Fig. 23C. In (B) and (C), the extant amphibians are shown indirectly as the three positions of the tetrapod crown-

group; although not rendered in boldface, "(rest of) Temnospondyli" is extant in R2, R3, R5 and R6, and "Lepospondyli" is extant in R1, R3, R4 and R6. Parentheses show which of these positions are found in which analyses. The numbers of these analyses are written in black if these analyses find a single position; for any position found as the single position by any analyses, the parentheses themselves are also black. Numbers are BPE\PP, BPE in boldface if 50 or higher, PP in boldface if 75 or higher. The rectangles indicate state 0 (filled) or 1 (empty) of characters 31 (PAR 1—cyan), 147 (PSYM 1—teal), 192 (CLE 2—blue), 212 (HUM 10—purple), 214 (HUM 12-15—red), 260 (TRU VER 8—orange) and 277 (CAU FIN 1—green); absence of a rectangle means missing data. Where known, Devonian limbed vertebrates have state 0 of each of these characters. Character 31: supratemporal/postparietal suture (0) or tabular/parietal suture (1); 147: presence (0) or absence (1) of parasymphysial; 192: presence (0) or absence (1) of postbranchial lamina on cleithrum; 212: humerus not (0) waisted (1); 214: humerus L-shaped (0) or not (1); 260: absence (0) or presence (1) of fusion between left and right pleurocentra in ventral midline; 277: presence (0) or absence (1) of tail fin skeleton (supraneural radials, lepidotrichia). See App. S1 for more precise definitions and discussion. Note that, of all adelospondyls, character 260 is only known in *Acherontiscus*; the adelogyrinids share state 1 if their single-piece centra are pleurocentra, but they have state 0 if their centra are intercentra, a question we cannot presently decide. Although all anthracosaurs in this matrix are unknown for character 277, we here show state 0 because of **CM 34638 (*Clack, 2011b*). Further derivations occur within some of the composite clades shown here: the highly nested temnospondyl *Micromelerpeton* is polymorphic for 31, and the diadectomorph *Tseajaia* has state 31(0); *Pederpes* (nested within Whatcheeriidae) has state 192(1); Urocordylidae and the diplocaulid *Keraterpeton* ("lepospondyls") have state 212(0); *Eusthenopteron*, *Panderichthys* and the embolomeres *Archeria* and *Pholiderpeton scutigerum* have state 212(1), while *Limnoscelis* and *Orobates* (Diadectomorpha) as well as the diplocaulids *Keraterpeton* and *Diceratosaurus* have state 212(0); and the highly nested temnospondyls *Doleserpeton* and \**Gerobatrachus* have state 260(1). State 2 of the ordered character 214 is not shown (*Eucritta*, which has state 0 or 1, is therefore shown as unknown); all the OTUs that have this state (an extra-long humerus) are nested well within clades for which 214(1) is plesiomorphic.

which is shared by the colosteids and *Crassigyrinus*), so that its presence (PREOPE 1 (0)) in *Pederpes* appeared as an isolated reversal. This was apparently due to a misreading of *Clack (1998)*, who explicitly confirmed that the preopercular is present in *Whatcheeria* as described by *Lombard & Bolt (1995)* and *Bolt & Lombard (2000)*; see ch. 81 (PREOPE 1) in App. S1 for discussion. The third sampled whatcheeriid-grade animal, *Ossinodus*, was scored as unknown, which was correct at the time; it is now known to have retained the preopercular as well (*Warren, 2007*), and we have scored it accordingly.

RC07 also found *Eucritta* as the sister-group of *Baphetes* + *Megalocephalus*, as did *Ruta (2009)*, *Milner, Milner & Walsh (2009)*, *Witzmann & Schoch (2017)* and the original full description by *Clack (2001)*. Given the original taxon sample, we find this as one of two options (BPO = **56**). Unexpectedly, the other option is one node rootward of Baphetidae; a position comparable to this is the only one found when we increase the taxon sample (BPE = 8), except in Analysis EB, which finds a (*Eucritta* (Baphetidae, Colosteidae)) clade with a PP of 74.

Like *Eucritta*, the extremely broad-headed \**Spathicephalus* has been considered a non-baphetid baphetoid (*Beaumont & Smithson, 1998*; *Milner, Milner & Walsh, 2009*; following the nomenclature of *Milner, Milner & Walsh, 2009*). \**Spathicephalus* is consistently a baphetoid in our analyses (BPE = 30; PP = **87**); however, it is highly nested within Baphetidae (PP = **87**; against a BPE of 31), closer to the long-snouted *Megalocephalus* than to the broad-headed *Baphetes* (PP = **82**). *Milner, Milner & Walsh (2009)* sampled

baphetids and baphetoid-related characters more densely than we did; however, it is possible that these advantages are outweighed by the disadvantages of their much smaller outgroup sample (only *Acanthostega* and *Crassigyrinus*), the fact that they did not order any characters (of the 24 characters, seven are continuous or meristic multistate characters), redundancy between ch. 3 and 4, and the scoring of juvenile morphology as adult (affecting at least ch. 19).

Further factors that may affect the position of *Spathicephalus* may be the enigmatic *Doragnathus* and *Sigournea*, the latter known from an isolated lower jaw, the former from fragments of lower and upper jaws. Both are found as baphetoids in all those of our analyses that include them (BPE = 30; PP = **87**), indeed as baphetids closer to *Megalocephalus* than *Baphetes* is (PP = **82**; against a BPE of 31), forming a clade with *Spathicephalus* (BPE = 30; PP = 44) and *Diploradus* (BPE = 19; PP = **87**). Similarities between all these taxa have long been noted, but only tested once in a phylogenetic analysis (even apart from the fact that *Diploradus* was very recently published): using a rather small character sample, *Clack et al. (2016)* found the three as a grade not very close to Baphetidae in some of their analyses, but widely scattered in others. Although our results remain to be tested with a larger sample of characters and of baphetoids, we think that *Doragnathus*, *Sigournea* and *Diploradus* will likely be upheld as baphetoids, possibly as baphetids and likely as a clade with *Spathicephalus* (BPE = 19; PP = **87**). It is noteworthy in this respect that *Smithson & Clack (2013)* noted similarities to baphetoids in the isolated postcranial material from the site where *Doragnathus* was found; perhaps it belongs to *Doragnathus* after all.

A very unexpected finding of our analyses with added taxa (Fig. 24C) is a clade composed of Colosteidae and Baphetidae (BPE = 3; PP = 70), which is furthermore a member of Temnospondyli (the sister-group to all others combined; contradicting a BPE of 20 and a PP of **86**). This ties into the problem of the mutual positions of Temnospondyli and Anthracosauria discussed below.

We are looking forward to the ongoing redescription of *Crassigyrinus* based on computed tomography. The abstract by *Porro, Clack & Rayfield (2015)* promises previously unknown character states, but the only new information it makes explicit is that "all three coronoids bear teeth"; we were unable to use this information, because our matrix contains separate characters for the presence or absence of tusks, denticles and toothrows.

### Colosteidae

Given the original taxon sample, Colosteidae is consistently found in a position one node rootward of Baphetoidea (or *Eucritta* followed by Baphetidae) and one node crownward of *Crassigyrinus* (Fig. 24B), as found, for example, by *Pawley (2006)* and *Ruta (2009)* and discussed immediately above. Adding taxa (Fig. 24C) pushes them and the baphetids further crownward, where both together form a surprising clade of temnospondyls (discussed below).

Among the added taxa are two uncontroversial colosteids. Analyses R4–R6 and EB, the first phylogenetic analyses to include them, unsurprisingly find *Pholidogaster*

(BPE = 30; PP = 50) and *Deltaherpeton* (BPE = 48; PP = 71) as successively closer to *Colosteus* + *Greererpeton* (a clade with a BPE of **76** and a PP of **98**).

Also included in the enlarged taxon sample are *Erpetosaurus* and the unnamed *St. Louis tetrapod (MB.Am.1441). The latter was described (*Clack et al., 2012b*) as having several similarities to colosteids, but also some to temnospondyls, while lacking several features otherwise found in colosteids; the former was most recently redescribed as a dvinosaurian temnospondyl with convergent similarities to colosteids, including features that the *St. Louis tetrapod lacks (*Milner & Sequeira, 2011*). In our analyses, the *St. Louis tetrapod consistently emerges as the sister-group to all other colosteids (BPE = 25; PP = **97**), while *Erpetosaurus* is a dvinosaur (BPE = 14, PP = **77**; BPE = 20 and PP = **86** for not lying next to Colosteidae, BPE = 25 and PP = **97** for not being nested within it).

One feature that may be relevant to the relationships of the *St. Louis tetrapod is not coded in this matrix: as pointed out by *Clack et al. (2012b)*, denticle-covered platelets of dermal bone that fill the interpterygoid vacuities are only known in that specimen and in temnospondyls. We speculate that such platelets appear more or less automatically when the interpterygoid vacuities are wide enough (the *St. Louis tetrapod has the proportionally widest ones of any colosteid) and denticles are present throughout the roof of the mouth (they are entirely absent in *Batropetes*, *Carrolla*, apparently *Quasicaecilia*, *Diplocaulus*, *Diploceraspis* and lissamphibians except *Eocaecilia*). This character deserves further investigation. It may be relevant that *Eocaecilia*, which fairly clearly lacks such platelets, has denticles on the pterygoids and the parasphenoid but not on the vomers or palatines (*Jenkins, Walsh & Carroll, 2007*).

We have also added *Aytonerpeton*, which *Clack et al. (2016)* described as sharing several similarities with the colosteids, but found far from Colosteidae in most of their analyses. One of these features—the premaxillary caniniform region, not coded in our matrix—represents an intermediate state between plesiomorphic homodonty and the premaxillary tusk which is found in traditional colosteids (and *Erpetosaurus*), where it is accommodated by a notch in the dentary. This notch is a character in our matrix, and scored as absent for *Aytonerpeton* (*Clack et al., 2016*); yet, *Aytonerpeton* nests surprisingly highly within Colosteidae, as the sister-group of (*Deltaherpeton* (*Colosteus*, *Greererpeton*)) to the exclusion of *Pholidogaster* (BPE = 35; PP = 71), which has the tusk and the notch, and the *St. Louis tetrapod, which has a W-shaped notch (see ch. 153, DEN 4, in App. S1 for discussion). To test this further, it will probably be necessary to fully describe *Aytonerpeton* and to reinvestigate the difficult specimens of *Pholidogaster*.

Whether the middle Tournaisian *Aytonerpeton*, which is older than the Viséan *St. Louis tetrapod, can claim the title of oldest known colosteid depends on two other specimens: the ***"Type 3 humerus" from Blue Beach in Nova Scotia (middle Tournaisian) and possibly ***"Ribbo" from the early middle Tournaisian site of Willie's Hole in (Old) Scotland (*Anderson et al., 2015*; *Clack et al., 2016*).

### The interrelationships of Anthracosauria, Silvanerpeton, Caerorhachis, Gephyrostegidae, Casineria and Temnospondyli

> Although [anthracosaurs have] long [been] associated with the amniote stem, this link appears to be increasingly tenuous.
>
> – *Coates, Ruta & Friedman, 2008*: 579

Ever since the early 20th century, Anthracosauria (which traditionally encompassed at least Embolomeri and the more recently discovered *Eoherpeton*) has generally been considered closer to Amniota than Temnospondyli is. The analysis by RC07 supported this "textbook" consensus (Fig. 24A). However, both the anthracosaurs and the temnospondyls share features with Amniota, Diadectomorpha, "Lepospondyli", Seymouriamorpha and/or *Solenodonsaurus* that the other taxon lacks (Fig. 24). Consequently, Temnospondyli has been found closer to Amniota than Anthracosauria is in a few analyses (*Ahlberg & Clack, 1998*; *Laurin & Reisz, 1999*: bootstrap tree and discussion; *Pawley, 2006*; *Klembara et al., 2014*, with poor support; some MPTs of *Pardo et al., 2017*).

Our results are best considered inconclusive on this point. Analyses R1 and R3 find Anthracosauria equally parsimoniously rootward (BPE = 6; PP = 48) or crownward of Temnospondyli (BPO = 12), while R2 fixes the rootward and R4–R6 the crownward position.

Part of the reason may be character conflict: not only Anthracosauria and Temnospondyli have conflicting combinations of character states, but so do other taxa found in the same area of the tree (Fig. 24).

*Silvanerpeton*, found as an anthracosaur (the sister-group to all others) by RC07 (BPO = 10; BPE = 6), lies one node more crownward (PP = 48) in all of our analyses except B1 and B2 (Figs. 24B and 24C). This is tenuous, however; earlier versions of our matrix (*Marjanović & Laurin, 2015*, *2016*) placed *Silvanerpeton* as an anthracosaur, one node more crownward or one node more rootward in different analyses or often in the same ones, indicating that its position depends on changes to a few scores as well as the taxon sample and constraints on other taxa.

*Caerorhachis* is variously considered to be close to the origin of temnospondyls and/or anthracosaurs (*Ruta, Milner & Coates, 2002*, and references therein; *Clack et al., 2016*). RC07 found *Caerorhachis* one node closer to Amniota than Temnospondyli and one node farther away than Anthracosauria (Figs. 1 and 24A). This is the only position found by Analyses R4–R6 (Fig. 24C). In Analyses R1 and R3, however, *Caerorhachis* can also be the sister-group to all other temnospondyls (Fig. 10: lower inset, 24B; BPO of Temnospondyli including *Caerorhachis*: 14), as found by *Pawley (2006)* and suggested earlier by *Godfrey, Fiorillo & Carroll (1987)*. Analyses R2, B2 and EB place *Caerorhachis* rootward of both Temnospondyli (BPE = 6; PP = 48) and Anthracosauria (BPE = 4; PP = 44).

RC07, and more recently *Witzmann & Schoch (2017)*, found *Gephyrostegus* one node rootward of *Bruktererpeton*. In Analyses R1–R3, they form either a clade (BPO = **53**; BPE = 42; PP = 69) as found by *Klembara et al. (2014)* in their redescription of the skull of *Gephyrostegus*, or (in some MPTs from R1 and R3) *Bruktererpeton* is one node

rootward of *Gephyrostegus* (Fig. 24B). Only the latter arrangement occurs in R4–R6 (Fig. 24C). As found by RC07 and *Witzmann & Schoch (2017)*, *Gephyrostegus* and *Bruktererpeton* are crownward of Anthracosauria in all analyses. A sister-group relationship of *Gephyrostegus* (or Gephyrostegidae) to Anthracosauria, as found by *Pawley (2006)*, does not occur (though it is only contradicted by BPO = 16; BPE = 6; PP = 48); it may be relevant that *Pawley's (2006)* taxon sample lacked *Silvanerpeton* and *Bruktererpeton*.

*Casineria* emerges in the gephyrostegid grade (Fig. 24C): variously one node crownward of *Gephyrostegus* (BPE = 4 and PP = 27 for lying crownward of Gephyrostegidae, BPE = 7 and PP = 55 for lying rootward of Chroniosuchia), one node rootward of *Bruktererpeton*, between the two, or as the sister-group of *Bruktererpeton*. This is novel, but perhaps not too surprising. Originally, *Casineria* was thought to lie very close to the origin of Amniota (*Paton, Smithson & Clack, 1999*; against a BPE of 9 and a PP of 55). *Clack et al. (2012b)* and *Witzmann & Schoch (2017*, using a very similar matrix*)* found it slightly more distant, as a "lepospondyl" (more precisely in a clade with *Westlothiana* and both of the sampled "microsaurs"; against a BPE of 13 and a PP of 55). *Clack et al. (2016)* increased the distance slightly further, finding it as a seymouriamorph (against a BPE of 9 and a PP of 55) or in the same grade (against a BPE of 7 and still the same PP of 55) in all analyses. In the meantime, *Pawley (2006*: 207*)* noted that *Casineria* scored identically to *Caerorhachis* in her matrix, apart from the (quite different) distribution of missing data; the matrix contains an unusually large amount of postcranial characters, well suited for the headless skeleton called *Casineria*. In our matrix, for which D. M. studied the only known *Casineria* specimen (part and counterpart: NMS G 1993.54.1p, NMS G 1993.54.1cp) for a full day, *Caerorhachis* and *Casineria* are likewise indistinguishable. *Pawley's (2006*: fig. 62*)* phylogenetic analysis featured a trichotomy of *Casineria*, *Caerorhachis* and all other temnospondyls, with strong bootstrap and Bremer support for Temnospondyli (fig. 63). Between *Caerorhachis* on one side and the seymouriamorph-"lepospondyl"-amniote clade on the other, the positions we find for *Casineria* lie in the middle.

*Paton, Smithson & Clack (1999)* emphasized that *Casineria* had gastrocentrous vertebrae and claws, highlighting especially the latter as an amniote-like feature. *Pawley (2006*: 239; see also 195*)* remarked: "As in *Caerorhachis bairdi*, none of the postcranial characteristics claimed to be 'reptiliomorph' in *Casineria kiddi* are truly apomorphic for the amniote lineage. All are present in temnospondyls [...], or potentially may be present in basal temnospondyls (including the five[-]digit manus), because they are plesiomorphic for early tetrapods." We confirm that the vertebrae of *Casineria* with their large intercentra are more reminiscent of *Caerorhachis* and animals in vaguely the same grade like the anthracosaur *Proterogyrinus* or, to a lesser degree, the temnospondyl *Neldasaurus* than of amniotes, seymouriamorphs or even chroniosuchians (Fig. 5); indeed, they differ from those of *Caerorhachis* at most in a slightly lesser degree of ossification. Although the terminal phalanges are distally pinched and curved plantarly, they are neither pointed as expected of a claw, nor rounded as incorrectly shown in the interpretative drawing by *Paton, Smithson & Clack (1999*: fig. 3b*)*. Instead, their ends are squared off

(Fig. 6). This shape is intriguingly similar to the outline drawings of the terminal phalanges of *Caerorhachis* by *Holmes & Carroll (1977*: fig. 13*)*, who unfortunately did not provide more detailed illustrations or any description. These two character complexes, thus, are not obstacles to placing *Casineria far from amniote origins. Moreover, if our very tentative identification of distal carpal 1 in Fig. 6 is correct, the first distal carpal began to ossify before the fourth in *Casineria, a plesiomorphic condition not found in amniotes or diadectomorphs (see below, Discussion: Characters: Preaxial polarity in limb development).

Going beyond her matrix, *Pawley (2006*: 195, 239*)* called *Caerorhachis* and *Casineria "indistinguishable based on the available evidence". Now that D. M. has seen the *Casineria specimen (Figs. 5–7), we can only find four differences to the thorough redescription of *Caerorhachis* by *Ruta, Milner & Coates (2002)*: *Caerorhachis* is larger and has more elongate (but otherwise identical) ventral scales, a slightly more prominent dorsal process on the ilium and taller ossified parts of the neural spines. All four can effortlessly be ascribed to ontogeny. Indeed, the neural spines of *Casineria are capped by a black material (Fig. 5; likely an iron sulfide) that is also found on the incompletely ossified ends of long bones (Fig. 6) and may be related to cartilage decay. (For scale ontogeny, see *Witzmann (2007*, *2011).*) Lack of difference is not a synapomorphy; our analyses do not currently find *Caerorhachis* and *Casineria as sister-groups—if only, perhaps, because the skull of *Casineria is unknown and postcranial characters are underrepresented in our matrix, and the two specimens come from close but not identical ages and localities. Still, the possibility should be seriously considered that *Casineria, once redescribed, could turn out to be either a synonym or the closest known relative of *Caerorhachis*. Rearranging a tree from Analysis R4 to form a clade of *Caerorhachis*, *Casineria and optionally the *Parrsboro jaw (see below) takes two extra steps in Mesquite, moving this clade next to the traditional temnospondyls (*Pawley, 2006*) takes one more; both actions contradict nodes with a BPE of 6 and a PP of 48.

Character conflict in this part of the tree concerns, among others, features that are probably relevant to the origin of terrestriality (Fig. 24). At least one anthracosaur (**CM 34638: *Clack, 2011b*; see also *Holmes & Carroll, 2010*) retained a dermal and endochondral tail fin skeleton (state CAU FIN 1(0) in this matrix, see ch. 277 in App. S1); unknown in *Silvanerpeton* (*Ruta & Clack, 2006*), *Bruktererpeton* and *Gephyrostegus*, it is absent in all temnospondyls that are well enough known to tell. The anthracosaur *Archeria* is known (*Pawley, 2006*: chapter 6) to retain the postbranchial lamina on the cleithrum (CLE 2(0), ch. 192); due to preservation and publication bias (the lamina is practically only visible in cranial or caudal view), this character is unknown in many OTUs in this matrix, but the lamina is absent in *Gephyrostegus* and all sufficiently well known unquestioned temnospondyls (*Pawley, 2006*: chapter 6). Because D. Marjanović (pers. obs. 2014) has identified what seems to be a postbranchial lamina quite similar to that of *Archeria* (depicted by *Pawley, 2006*: fig. 70-2.4) on the cleithrum of *Casineria, presented here in Fig. 7, we have scored *Casineria as possessing the lamina; this will require further investigation. If correct, it is a further argument for removing *Casineria from amniote origins. Anthracosauria, *Silvanerpeton* and Gephyrostegidae retain flat humeri (HUM 10(0), ch. 212), while *Casineria (D. Marjanović, pers. obs.) shares a

waisted (HUM 10(1)), twisted humerus shape with all skeletally mature temnospondyls (unknown in *Caerorhachis*); the anthracosaurs *Eoherpeton*, *Proterogyrinus* and *NSM 994 GF 1.1 additionally retain an L-shaped humerus (HUM 12-15(0), ch. 214) where the entepicondyle is about as long (measured along its proximal margin) as the part of the humerus proximal to it, which is not seen in *Archeria* or *Pholiderpeton scutigerum* (unknown in *Pholiderpeton attheyi*, *Anthracosaurus*, *Palaeoherpeton* and *Neopteroplax*) or in the other taxa listed in this paragraph. Finally, pleurocentra that fuse in the ventral midline (TRU VER 8(1), ch. 260) are found in all the taxa listed in this paragraph except the unquestioned temnospondyls. (Among the latter, TRU VER 8(1) does occur in *Doleserpeton* and *Gerobatrachus*, as well as in some **brachyopid stereospondyls and some **tupilakosaurid dvinosaurs, but these are too highly nested to be relevant here (*Warren, 1999*; *Werneburg et al., 2007*; *Warren, Rozefelds & Bull, 2011*).) However, while state TRU VER 8(1) presumably increases the resistance of the vertebral column to compression, this may be dorsoventral compression from bearing weight just as well as mediolateral compression from, for example, anguilliform swimming. Various amphibious and terrestrial temnospondyls, on the other hand, elaborated the neural arches and (in Dissorophidae) neomorphic osteoderms into a weight-bearing apparatus that evidently did not need extensively ossified centra; furthermore, the vertebral columns of *Gephyrostegus* and *Bruktererpeton* in particular look rather supple in reconstructions despite showing state TRU VER 8(1).

Outside our matrix, character conflict in this part of the tree is further increased by the sacral vertebra described by *Holmes, Godfrey & Baird (1995*: 919, fig. 6) which shows a unique combination of similarities to temnospondyls and anthracosaurs.

A postbranchial lamina on the clavicle, not coded here, has been identified in the dvinosaur **Thabanchuia (*Witzmann, 2013*) and in a number of *stereospondyls: *Schoch & Witzmann (2011)* described it in **Plagiosuchus and **Trematolestes; D. M. has seen it on a clavicle of **Metoposaurus that, as of 2015, was to be catalogued by the University of Opole (Poland), and in another that is on exhibit in the museum of Krasiejów (Poland); *Yates & Warren (2000*: fig. 7) depicted it in **Benthosuchus, **Paracyclotosaurus, **Lyrocephaliscus and **Koskinonodon—they only called it "anterior flange (prescapular process)", but F. Witzmann (pers. comm. 2015) confirms that this structure is most likely identical to the postbranchial lamina. The homology of the postbranchial laminae on the cleithrum of non-temnospondyls and the clavicle of temnospondyls remains unclear at the moment.

Interestingly, the parasymphysial bone in the lower jaw (PSYM 1(0), ch. 147) has the same distribution as the postbranchial lamina on the cleithrum, missing data and aïstopods excepted (Fig. 24).

Noting the presence of anthracosaurs and *Tulerpeton*-like animals soon after the beginning of the Carboniferous, *Anderson et al. (2015)* drew attention to the old idea that all or almost all limbed vertebrates are part of the smallest clade that contains Anthracosauria and Temnospondyli. "If *Tulerpeton* represents the earliest occurrence of embolomeres (as appears to be suggested by the close similarity between humeral and femoral morphology)" (*Anderson et al., 2015*: 24), the last common ancestor of

anthracosaurs and temnospondyls would have lived in the Devonian. However, *Anderson et al. (2015)* provided no support for this complex of ideas, which contradicts the current consensus (including RC07), beyond citing *Lebedev & Coates (1995)*; they did not mention the reanalysis and refutation of the latter by *Laurin (1998b)*. Unsurprisingly, then, these historical hypotheses are not supported by our analyses; as discussed by *Pawley (2006)*, the similarities between *Tulerpeton* and Anthracosauria are synapomorphic at that level, but were further modified by Temnospondyli and various other branches—they represent intermediate states. A fortiori, even if some or all extant amphibians are temnospondyls, there is currently no strong reason to think that the origin of the tetrapod crown-group happened in the Devonian as *Anderson et al. (2015)* speculated. The highest support values that keep *Tulerpeton* away from Anthracosauria are BPO = **67**, BPE = 22 and PP = **86** (Figs. 18–20); the lack of support in the parsimony analyses of the extended taxon sample is likely due to the unstable positions of *Eucritta*, Temnospondyli, *Caerorhachis* and the *Parrsboro jaw which can all intervene between *Tulerpeton* and Anthracosauria.

A topology somewhat similar to the above scenario was found by *Klembara et al. (2014)*, where a weakly supported (*Crassigyrinus* (*Whatcheeria*, Anthracosauria)) clade and a weakly supported clade composed of Baphetidae and Temnospondyli were found. However, the second clade came out—likewise with weak support—as closer to Amniota than the first clade, which appeared just one node crownward of *Ichthyostega*. Containing 36 ingroup taxa (a quarter of them seymouriamorphs; *Tulerpeton* was not sampled), an all-zero outgroup (*Klembara & Ruta, 2004b*: 86—see *Marjanović & Laurin, 2008*, for discussion of this practice) and 156 characters (many of course focused on seymouriamorphs), their matrix seems at least a priori less well suited to resolving this problem than ours.

At present, the oldest known anthracosaurs date (as mentioned) from the middle Tournaisian (*Anderson et al., 2015*), while the oldest known temnospondyls appear only after Romer's Gap (*Balanerpeton*, possibly *Caerorhachis*, perhaps *Casineria*: *Milner & Sequeira, 1994*; *Paton, Smithson & Clack, 1999*; *Ruta, Milner & Coates, 2002*; no temnospondyls have been identified in the middle Tournaisian material reported by *Anderson et al., 2015*, or *Clack et al., 2016*). This is congruent with the hypothesis that Temnospondyli and the tetrapod crown-group, which is likewise currently unknown before the end of Romer's Gap, are more closely related to each other than to Anthracosauria. However, current understanding of the diversity of limbed vertebrates during Romer's Gap is still very poor; the current absence of evidence may turn out not to be evidence of absence.

### The "Parrsboro jaw"

An incomplete natural mold of a partly crushed lower jaw, the *Parrsboro jaw in our expanded taxon sample, has only two positions in our parsimony analyses (Fig. 24C): it is either the sister-group of *Caerorhachis* (BPE = 31; contradicting a node with a PP of 48) or that of all traditional temnospondyls (against a PP of 45). Both positions lie within the large range found by *Sookias, Böhmer & Clack (2014)* in

their much smaller analysis. Analysis EB recovers what amounts to an intermediate position, rootward of Temnospondyli and Gephyrostegidae but crownward of *Silvanerpeton*.

Immediately, a character comes to mind that is absent from the present matrix but would influence the position of the *Parrsboro jaw: the presence/absence of denticles on the prearticular. Presence, a plesiomorphy shared with, for example, colosteids (*Bolt & Lombard, 2001*), is found in the *Parrsboro jaw and in *Caerorhachis* but not, to our knowledge, in (other) temnospondyls or for that matter any anthracosaur-grade animals. A denticle field that covers most of the lingual face of the prearticular (extending onto the splenial at least in *Caerorhachis*), instead of being restricted to the dorsal side, even seems to be entirely limited to *Caerorhachis* and the *Parrsboro jaw among all sarcopterygians.

### Anthracosaurian phylogeny

In the analyses of the original taxon sample (R1–R3 and B1), Anthracosauria is resolved as in RC07 when it lies rootward of Temnospondyli (the only option in R2), but in two other ways, with *Pholiderpeton scutigerum* as the sister-group of either (*Proterogyrinus* + *Archeria*) or (*Pholiderpeton attheyi* + *Anthracosaurus*), when it lies crownward of Temnospondyli. Bootstrap support greater than 50% or a posterior probability greater than 75% exists for three nodes: Anthracosauria (not including *Silvanerpeton*; BPO = **54**; BPE = 20; PP = **97**), Embolomeri (BPO = **52**; BPE = 21; PP = **99**), Embolomeri except *Proterogyrinus* (PP = **94**; against a BPO of 33 and a BPE of 31) and *Pholiderpeton attheyi* + *Anthracosaurus* excluding *Pholiderpeton scutigerum* (BPO = **57**; PP = **68**; against a BPE of 18).

Adding taxa narrows these topologies down to that of RC07, even though Anthracosauria is consistently crownward of Temnospondyli in Analyses R4–R6 (against a BPE of 6 and a PP of 48). The three added OTUs form a clade (resolved as (*NSM 994 GF 1.1 (*Palaeoherpeton*, *Neopteroplax*)); against a BPE of 28 and a PP of 62), which nests with *Anthracosaurus* (BPE = 28; PP = 58).

Like RC07, we consistently find that *Pholiderpeton scutigerum* and *Pholiderpeton attheyi* are not sister-groups (BPO = **57**; BPE = 25; PP = **68**). A logical consequence (if para- or polyphyletic genera are to be avoided) would be to reinstate the genus name *Eogyrinus* Watson, 1926, for *Pholiderpeton attheyi*. However, we refrain from performing a taxonomic act because *Pawley (2006*: chapter 6*)* did find these two species as sister-groups using a different and larger character sample (but the same sample of possible anthracosaurs as RC07, minus *Silvanerpeton*), not to mention the weak support for our result.

We are more confident that the traditional taxon Eogyrinidae, comprising all embolomeres except *Proterogyrinus*, *Archeria* and *Anthracosaurus*, is paraphyletic (BPO = **57**; BPE = 28; PP = **68**). These taxa were only found to form a clade in the much smaller analysis of *Buchwitz et al. (2012)*, which was focused on chroniosuchians.

### Temnospondyl large-scale phylogeny

I shall refrain here from discussion of the temnospondyls, although work being done at present, by Baird and Carroll, for example, suggests progress toward sorting out

true phyletic lines among the Rhachitomi in preference to the somewhat artificial grouping which I used in my 1947 classification.

– *Romer (1964*: 154*)*

More than 50 years after this hopeful statement, temnospondyl phylogeny remains poorly understood (Fig. 25). RC07, fittingly, found the base of Temnospondyli to be a polytomy of Edopoidea (= *Edops* + Cochleosauridae), Dvinosauria, *Balanerpeton*, Dendrerpetidae and an *Eryops*-Dissorophoidea clade (Fig. 25A); *stereospondylomorphs were not included in the matrix. Most of the mathematically possible resolutions of this polytomy have been found in more focused but still large analyses: *Pawley (2006*: chapter 5; Fig. 25B*)* found *Edops* alone as the sister-group to all other temnospondyls (except *Caerorhachis*, see above and below), among which a (*Capetus* (Dendrerpetidae, *Balanerpeton*)) clade, Cochleosauridae, Dvinosauria, paraphyletic *stereospondylomorphs and *Eryops* were successively closer to Dissorophoidea. *Ruta (2009*; Fig. 25C*)* found Edopoidea as a whole to be the sister-group to all other temnospondyls, among which *Capetus*, (*Eryops* + *Stereospondylomorpha), *Dendrerpeton confusum* and "*D. acadianum*" were successively closer to a trichotomous clade formed by *Balanerpeton*, Dvinosauria and Dissorophoidea; the only two *stereospondylomorphs in the matrix were *Sclerocephalus* and *Glanochthon*. (The OTU called "*Dendrerpeton acadianum*" most likely included *Dendrysekos*.) *McHugh (2012*; Fig. 25D*)* found *Neldasaurus*, otherwise considered a dvinosaur, to be the sister-group of all other temnospondyls; those were divided into, on the one hand, a clade that contained the *stereospondyls and the neotenic dissorophoids inside a clade formed by the remaining dvinosaurs, and on the other hand a clade that contained (Dendrerpetidae + non-aquatic dissorophoids), (*Balanerpeton* + *Capetus*), Edopoidea and *Eryops* as successively closer relatives of the remaining *stereospondylomorphs. After deleting the colosteid *Greererpeton* from his taxon sample, *Schoch (2013)* recovered Edopoidea and (*Balanerpeton* + Dendrerpetidae) successively closer to a trichotomy of Dvinosauria, Dissorophoidea and Eryopiformes (= *Eryops* + Stereospondylomorpha)—when he used TNT; PAUP 3.1 failed to find any of the shortest trees (*Schoch, 2013*: 682). *Dilkes (2015a)* analyzed an expanded and corrected version of *Schoch's (2013)* dataset. Given the full sample of 73 OTUs (*Dilkes, 2015a*: fig. 10), he found a polytomy of *Capetus*, *Iberospondylus*, (Dendrerpetidae + *Balanerpeton*), Edopoidea and a clade which contained all other temnospondyls; that clade was itself a polytomy of *Zatracheidae, Eryopidae, Dissorophoidea, a clade containing the *non-stereospondyl stereospondylomorphs and three separate clades of *stereospondyls. Deleting 25 OTUs, but not *Greererpeton*, resulted (*Dilkes, 2015a*: fig. 11A) in a single eryopiform clade that contained *Eryops* (the remaining eryopid) and the *non-stereospondyl stereospondylomorphs as a grade within which a single *stereospondyl clade was nested, and also revealed a (Dvinosauria (*Zatracheidae, Dissorophoidea)) clade, but continued to resolve the relationships of these clades in two very different ways (Figs. 25E and 25F). Only the additional deletion of *Capetus* and *Iberospondylus (*Dilkes, 2015a*: fig. 11B) made Eryopiformes consistently the sister-group of the (Dvinosauria (*Zatracheidae, Dissorophoidea)) clade, followed on the outside by

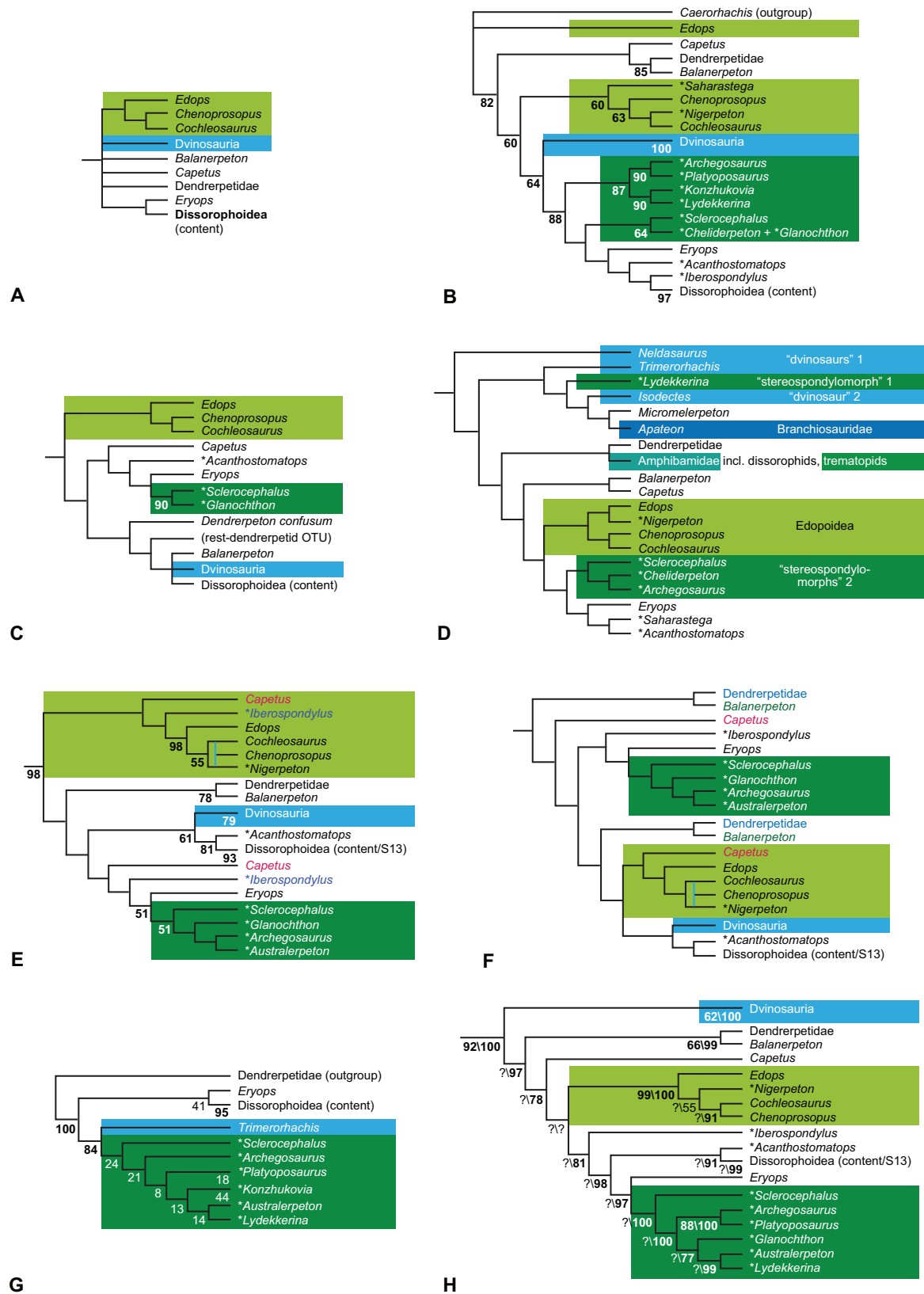

**Figure 25 Hypotheses on temnospondyl phylogeny since 2006.** Taxa absent from our expanded taxon sample are omitted, those we added to the sample of RC07 are marked with an asterisk. Dvinosauria and Dissorophoidea are collapsed where possible. The colored rectangles (labeled in D)

and the terms "Dissorophoidea (content)" and "Dissorophoidea (S13)" are consistent with Figs. 1 and 10–20. The matrices of (A–C) are ultimately based on that of *Ruta, Coates & Quicke (2003)*, so they—and ours (Fig. 26)—should not be considered fully independent of each other. Numbers below internodes are published bootstrap percentages; percentages below 50 were, except in (G), not reported. (A) Strict consensus of RC07; non-temnospondyls omitted. Dissorophoidea is marked in boldface for being extant because it contains Lissamphibia; this is the only analysis shown here that included any lissamphibians. RC07 (app. 4) performed a bootstrap analysis but did not publish the results for the part of the tree shown here. (B) *Pawley (2006*: chapter 5*)*. "\**Cheliderpeton* + \**Glanochthon*" was a single OTU called "*Cheliderpeton* spp."; the name *Glanochthon* had not yet been coined. \**Acanthostomatops* was included as part of a Zatracheidae OTU. (C) *Ruta (2009)*, strict consensus of analysis without reweighting; non-temnospondyls omitted. \**Glanochthon latirostris* was called by its previous name, "*Cheliderpeton latirostre*". The "rest-dendrerpetid OTU" was called "*Dendrerpeton acadianum*", but may have contained information from other dendrerpetids as well; see Materials and Methods: Treatment of OTUs: Dendrerpetidae. (D) *McHugh (2012)*; non-temnospondyl omitted. Dissorophidae (as two clades, one of which contains *Broiliellus brevis*) and Trematopidae (also as two clades) are nested in three different places within Amphibamidae. All bootstrap percentages of the nodes shown here are below 50 and were not published. (E, F) All equally parsimonious topologies (8 MPTs) from the analysis of 48 taxa in *Dilkes (2015a)*; non-temnospondyls omitted. Note (in F) that Dendrerpetidae and *Balanerpeton* are sister-groups in all MPTs. To see the individual MPTs, we repeated the analysis (calculation time: 00:04:36.5), finding MPTs of the same number, length and indices as published by *Dilkes (2015a)*. The bootstrap percentages are from *Dilkes (2015a*: fig. 11B*)*, which shows an analysis where *Capetus* and \**Iberospondylus* were omitted; the topology is otherwise identical to (E). (G) *Pereira Pacheco et al. (2017)*, focused on stereospondylomorphs. Two species of \**Platyoposaurus* and three of \**Konzhukovia* were included as separate OTUs. The tree by *Eltink et al. (2016)* is fully compatible. (H) *Pardo, Small & Huttenlocker (2017)*; non-temnospondyls and lissamphibians omitted (the MPTs differ only in the positions of the lissamphibians and \*\**Chinlestegophis*). Numbers are bootstrap percentages followed by Bayesian posterior probabilities in % (boldface if ≥ 75, not published < 50).

(Dendrerpetidae + *Balanerpeton*) and finally Edopoidea as in *Schoch (2013)* and in Fig. 25E. Using the dendrerpetid *Dendrysekos* as the outgroup, *Eltink & Langer (2014)* found the ingroup to be divided into (*Eryops* + Dissorophoidea) and (Dvinosauria + \*Stereospondylomorpha); theirs was a \*stereospondylomorph-focused analysis where *Trimerorhachis* was the only included dvinosaur. The same holds for *Pereira Pacheco et al. (2017*; Fig. 25G) and for *Eltink et al. (2016)*, who omitted Dissorophoidea altogether. *Pardo, Small & Huttenlocker (2017*: supplementary information part D; Fig. 25H) accepted most of the corrections by *Dilkes (2015a*—not mentioned by *Pardo, Small & Huttenlocker, 2017)* but added taxa, omitted others, and omitted the characters Dilkes had added. Dvinosauria emerged as the sister-group to the rest of the ingroup under both parsimony and Bayesian inference (*Pardo, Small & Huttenlocker, 2017*: fig. S7B and 2 = S7A, respectively); (Dendrerpetidae + *Balanerpeton*), *Capetus*, Edopoidea, \**Iberospondylus*, \*\**Peltobatrachus*, Eryopiformes, \*Zatracheidae and \*\**Lapillopsis* (previously considered a stereospondyl) were found successively closer to Dissorophoidea in all MPTs, with the positions of *Capetus* and \*\**Peltobatrachus* slightly destabilized in the Bayesian analysis.

This diversity of topologies appears to result in large part from the use of different outgroups. *Pawley (2006)* used *Caerorhachis* as the outgroup in chapter 5 after finding it to be in a trichotomy with \**Casineria* and all other temnospondyls in chapter 6 (her main analysis of the phylogeny of the limbed vertebrates). *Ruta (2009)* included a variety of early limbed vertebrates, but entirely omitted the taxa that form the amniote total group in the trees of RC07 (see Fig. 1), including Gephyrostegidae, Anthracosauria and *Caerorhachis*—the closest relative of Temnospondyli in his trees is Baphetoidea. *McHugh (2012)* used *Greererpeton* as the outgroup. *Schoch (2013)* chose the anthracosaur *Proterogyrinus*; in analyses where he included *Greererpeton* in the ingroup, *Greererpeton* either fell out as the sister-group to Temnospondyli or, "in some variant analyses of the large dataset (66 taxa, 'no swapping' option in PAUP),

*Greererpeton* nests with the dvinosaurs, and in another variant *Greererpeton* and *Neldasaurus* form a 'clade' at the very base of the tree (66 taxa, 'no swapping' option, TNT). Whereas this reflects similarities in the postorbital skull shared by the two taxa, it also shows that aquatic taxa with particular features are sometimes attracted despite the obvious homoplastic status of these characters (indicated by multiple conflicting evidence)." This attraction between *Greererpeton* and *Neldasaurus* replicates the result of *McHugh (2012)*, as does, to a lesser extent, the attraction of *Greererpeton* to the dvinosaurs as a whole. Unfortunately, however, *Schoch (2013*: 693*)* did not report the lengths of these trees; the lack of branch-swapping means that the local optima found in each tree-building replicate were not explored any further, which makes it likely that the reported trees were suboptimal. Notably, however, *Greererpeton* did not have such effects in the analyses of a version of that matrix improved by *Dilkes (2015a)*. In the next version of the same matrix, *Pardo, Small & Huttenlocker (2017*: supplementary information part D*)* kept *Proterogyrinus* as the outgroup, and *Greererpeton* emerged as the sister-group to the rest of the ingroup, followed as mentioned by the monophyletic dvinosaurs.

Our analyses of the original taxon sample (R1–R3, B1; Figs. 26A–26C) consistently place the colosteids far away from Temnospondyli. Yet, of those MPTs in which Temnospondyli is crownward of Anthracosauria (which is all in R2), some (all in R2; Fig. 26A) find Dvinosauria (which has a BPO of 35) as the sister-group to all other temnospondyls (except *Caerorhachis* in some from R1; BPO = 40). This clade of all other temnospondyls (BPO = 18) consists (against a BPO of 30) of a dichotomy of Dissorophoidea on the one side and an unusual clade on the other side in which (*Balanerpeton* + Dendrerpetidae), *Capetus* and *Eryops* (BPO = 19) are successively closer to a surprisingly highly nested Edopoidea (monophyletic: BPO = 26). Other MPTs from R1 and R3 (Fig. 26B) find Temnospondyli (always excluding *Caerorhachis*) as containing a backbone of (Dissorophoidea (*Eryops* (Cochleosauridae (*Capetus* (*Trimerorhachis* (*Balanerpeton*, Dendrerpetidae)))))) (against a BPO of 30); (*Isodectes* + *Neldasaurus*) is the sister-group of *Trimerorhachis* or of all other temnospondyls, among which *Edops* is one node closer to or one node farther from Dendrerpetidae than *Capetus* is. Finally, those MPTs (from R1 and R3; Fig. 26C) where Temnospondyli is rootward of Anthracosauria consistently resolve it as a Hennigian comb of (*Edops* (Cochleosauridae (*Eryops* (*Capetus* (Dvinosauria (Dendrerpetidae, Dissorophoidea)))))), with *Balanerpeton* as the sister-group of either of the last two. This position of *Edops* is reminiscent of *Pawley (2006*: chapter 5; Fig. 25B*)*. Analysis B1, where Temnospondyli is again rootward of Anthracosauria, finds a sort of compromise topology, with Dvinosauria on the outside (BPO = 18) except for *Caerorhachis*, *Eryops* next to Edopoidea (BPO = 19), Dissorophoidea next to (*Balanerpeton* + Dendrerpetidae) (BPO = 30), and very weak support overall (Fig. 18).

Analyses R4–R6 (Figs. 26C and 26D) find Temnospondyli rootward of Anthracosauria—so far rootward that Temnospondyli contains *Eucritta* and (Colosteidae + Baphetidae) successively closer to the clade of traditional temnospondyls (a surprising return to 20th-century classifications, contradicting a BPE of 20 and a PP of **86**)

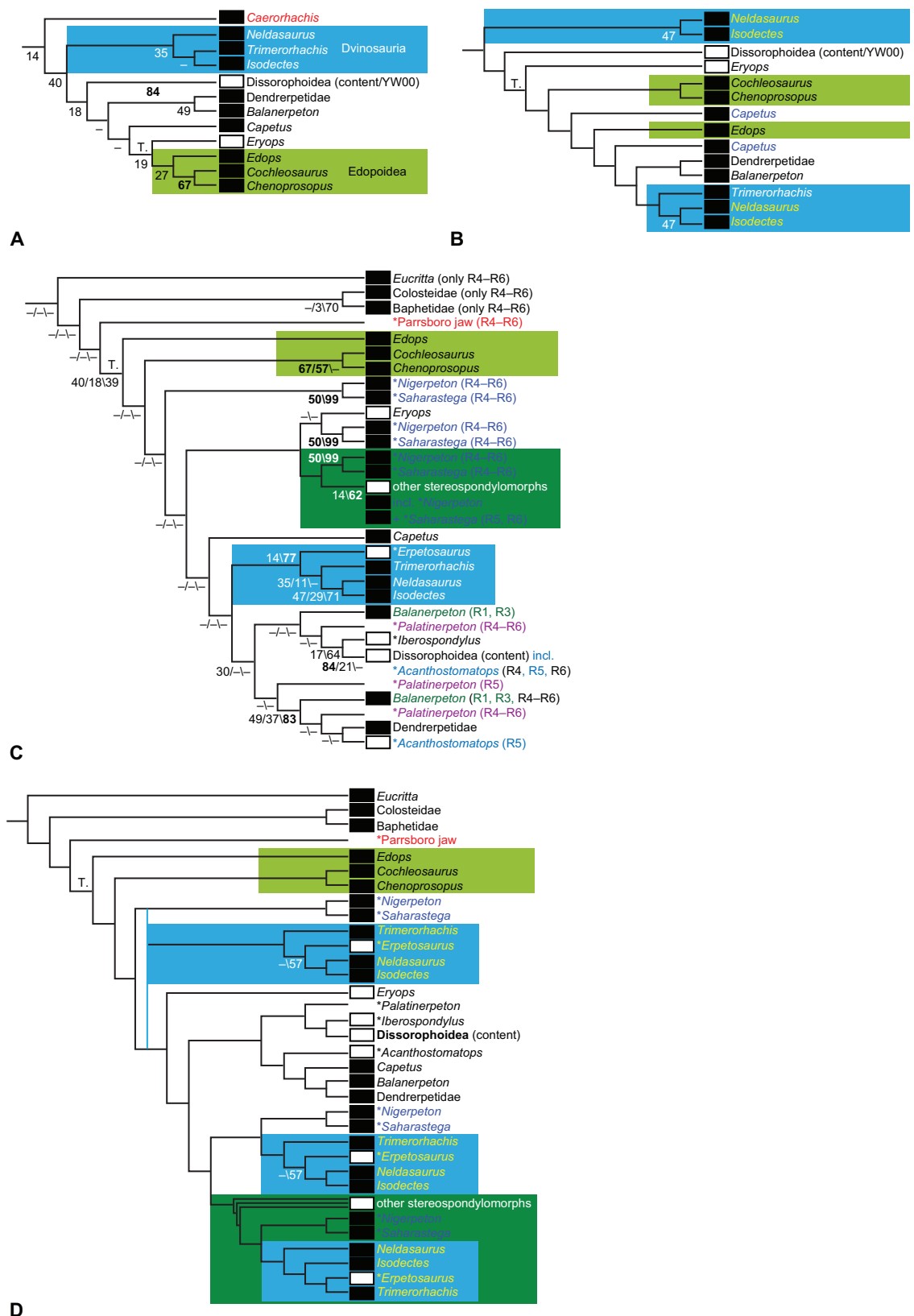

**Figure 26 Temnospondyl phylogeny in our analyses, distribution of the intertemporal bone, and the application of the name Temnospondyli.** Dissorophoidea and Stereospondylomorpha are collapsed; modern amphibians and various "lepospondyls" (see Fig. 12) are omitted. Filled

rectangles indicate presence of the intertemporal (state 0 of ch. 32—PAR 2/POSFRO 3/INTEMP 1/SUTEMP 1), empty rectangles its absence (state 1, or state 2 in the case of *Erpetosaurus*; see App. S1); the absence of a rectangle for *Palatinerpeton* and the *Parrsboro jaw represents missing data. T., Temnospondyli as defined by Schoch (2013); the entire depicted clade is Temnospondyli as defined by Yates & Warren (2000). (A) Topology found in Analyses R1–R3 (original taxon sample) in MPTs where Temnospondyli is closer to Amniota than Anthracosauria. Only one position of *Caerorhachis* is shown; the other is far outside Temnospondyli. Numbers are BPO. (B) Additional topologies found in R1 and R3 (original taxon sample) in MPTs where Temnospondyli is closer to Amniota than Anthracosauria. Numbers are BPO of nodes absent from (A). (C) Topologies found in R1 and R3 (original taxon sample) and R4–R6 (expanded taxon sample) in MPTs where Anthracosauria is closer to Amniota than Temnospondyli. Only one position of the *Parrsboro jaw is shown, the other being well outside Temnospondyli. Numbers are BPO/BPE\PP, or BPE\PP where BPO is inapplicable. (D) Additional topologies found in R5 (Fig. 16). Numbers are BPE\PP.             

and optionally the *Parrsboro jaw (see above). In keeping with its rootward position, the clade of traditional temnospondyls is resolved in R4 and R6 as in corresponding trees from R1 and R3, namely as (*Edops* (Cochleosauridae (Eryopiformes (*Capetus* (Dvinosauria ((*Balanerpeton*, Dendrerpetidae) (*Iberospondylus*, Dissorophoidea))))))); in R5, Dvinosauria sometimes (Fig. 26D) enters Stereospondylomorpha, and Eryopiformes sometimes becomes a grade (with Stereospondylomorpha closer to Dissorophoidea than to *Eryops*), in which latter case Dvinosauria can fall out rootward of it or as the sister-group of Stereospondylomorpha, that is, as part of Limnarchia Yates & Warren, 2000. The corresponding bootstrap and Bayesian trees (Analyses B2, EB: Figs. 19 and 20) find Temnospondyli crownward of *Eucritta*, Colosteidae, Baphetidae, *Caerorhachis* and Anthracosauria and consequently find a quite different temnospondyl topology, with an almost basal dichotomy of Dissorophoidea against most of the rest, but the highest BPE throughout the base of the temnospondyl tree is only 14 (for Dvinosauria); similarly, only five very small temnospondyl clades have PPs above **75**—Dvinosauria has **77**, but (*Isodectes* + *Neldasaurus*) has 71, (Dissorophoidea (*Iberospondylus*, *Mordex*)) has 64, Stereospondylomorpha has 62, Dvinosauria except *Trimerorhachis* has 57, a novel (*Edops* + *Cochleosaurus*) clade has 47, Temnospondyli has 39, Dissorophoidea has 38, Temnospondyli excluding *Palatinerpeton* has 35, (*Chenoprosopus* (*Nigerpeton*, *Saharastega*)) has 33.

Dissorophoidea is never found closer to *Eryops* than Stereospondylomorpha is; in other words, Euskelia Yates & Warren, 2000, is always limited to *Eryops* (BPE of Eryopiformes = 2, PP = 14) or to (*Eryops* (*Nigerpeton*, *Saharastega*)) (against a BPE of 15 and a PP of 33).

We conclude that progress in resolving temnospondyl phylogeny may come from analyses that have not only dense sampling of temnospondyls and a long list of maximally independent characters, but also a large, dense sample of other taxa, because the closest relatives of Temnospondyli have not been identified and the position of Temnospondyli rootward or crownward of the anthracosaurs, most evidently, has an enormous impact on temnospondyl phylogeny. In other words, the problem of temnospondyl phylogeny cannot be solved in isolation.

Schoch (2013: 689) defined the name Temnospondyli as applying to "[t]he least inclusive clade containing *Edops craigi* and *Mastodonsaurus giganteus*". This definition may be applied to our trees—as done in Fig. 26—by assuming that **Mastodonsaurus* would group with the other *stereospondylomorphs or, in their absence, with

*Eryops*. In Analyses R4–R6 and some MPTs from R1 and R3 (Figs. 26C and 26D), in other words all those where the anthracosaurs are closer to Amniota than the temnospondyls are, Temnospondyli would then have its usual contents, because *Edops* is found as the sister-group to all other traditional temnospondyls. Conversely, in Analyses R2, B1, B2, EB and the remaining MPTs from R1 and R3 (Figs. 26A and 26B), Temnospondyli would always exclude Dissorophoidea; in R2, B1, B2, EB and some MPTs from R1 and R3, Temnospondyli would even contain nothing but *Eryops*, Stereospondylomorpha and Edopoidea (Fig. 26A). Furthermore, *Capetus*, *Balanerpeton*, Dendrerpetidae and Dvinosauria fall outside of Temnospondyli under this definition in the MPTs found by *Pardo, Small & Huttenlocker (2017*: fig. S7B; Fig. 25H*)*. Therefore, we strongly recommend against using this definition, or any other minimum-clade definition with so few internal anchors (specifiers), for this name. We have used the definition by *Yates & Warren (2000)*, which applies the name Temnospondyli to the clade encompassing all organisms that are more closely related to *Eryops* than to the "microsaur" *Pantylus*.

### Stereospondylomorpha

Throughout the analyses with added taxa (Figs. 14–17, 26C and 26D), the stereospondylomorph temnospondyls—*Sclerocephalus*, *Cheliderpeton*, *Glanochthon*, *Archegosaurus*, *Platyoposaurus*, *Konzhukovia*, and the stereospondyls *Lydekkerina* and *Australerpeton*—form a clade, either alone (BPE = 14; PP = 62) or with *Nigerpeton* + *Saharastega* (some MPTs from R5 and R6) and Dvinosauria (some MPTs from R5), even though no characters intended to resolve stereospondylomorph phylogeny were included in the matrix.

*Sclerocephalus* or, in R4 and some trees from Analyses R5 and R6, a clade formed by *Sclerocephalus* and *Cheliderpeton* (BPE = 27; against PP = 50) is consistently found as the sister-group to the rest. Within this remainder (BPE = 11; PP = 57), those trees where *Nigerpeton* and *Saharastega* do not join, namely all from Analyses EB and R4 as well as some from R5 and R6, group *Konzhukovia* with *Platyoposaurus* with astonishing support (PP = **98**), and both of them with *Australerpeton* (PP = **81**) and *Archegosaurus* (PP = 60). Analysis EB also groups *Glanochthon* with *Lydekkerina* with negligible support (PP = 31); otherwise, *Lydekkerina* is the sister-group of all the above. Despite keeping *Nigerpeton* and *Saharastega* out (BPE = 14), Analysis B2 groups *Australerpeton* and *Archegosaurus* (BPE = 30) with the likewise extremely long-snouted *Platyoposaurus* (BPE = 11). In the remaining MPTs from R5 and R6, a clade of *Archegosaurus* and *Glanochthon* (or, perhaps surprisingly, (*Archegosaurus* (*Cheliderpeton*, *Glanochthon*))) as well as a clade of *Konzhukovia* and *Lydekkerina* (BPE = 28) occur.

*Lydekkerina* and *Australerpeton* are never found as sister-groups. Most likely we have exhausted the scope of the present character sample, and the stereospondylomorphs are sorted by noise more than by signal. Still, the position of *Sclerocephalus* with respect to the others agrees—ignoring the dvinosaurs in Analysis R5—with *Eltink & Langer (2014)*, *Dilkes (2015a)*, *Pardo, Small & Huttenlocker (2017)* and arguably *McHugh (2012)*.

*Cisneros et al. (2015)* found Dvinosauria nested within their small sample of stereospondylomorphs. We do not interpret this result as corroborating our finding of the same in the mentioned trees from R5, because it could easily be due to their small sample of non-dvinosaurs in general—or to the fact that *Eryops* was included in the outgroup so that Eryopiformes was forced to include Dvinosauria a priori. Furthermore, *Cisneros et al. (2015)* did not order any characters.

### Dissorophoidea

Something of a consensus on dissorophoid phylogeny has been forming. Recent analyses (*Ruta, 2009*; *Maddin et al., 2013b*; *Holmes, Berman & Anderson, 2013*; *Dilkes, 2015a*; *Pardo, Small & Huttenlocker, 2017*) have generally found Dissorophidae (represented here by *Broiliellus brevis* alone) and Trematopidae (*Acheloma, Phonerpeton, Ecolsonia,* *Mordex*—the latter here included in a phylogenetic analysis for the first time) as sister-groups (forming Olsoniformes *Anderson et al., 2008b*), and many have found the traditional "branchiosaur" *Micromelerpeton* to have no connection to the branchiosaurids (*Apateon, Leptorophus, Schoenfelderpeton,* *Tungussogyrinus,* *Branchiosaurus*); sometimes, *Micromelerpeton* is placed as the sister-group to all other dissorophoids together (*Maddin et al., 2013b*; *Dilkes, 2015a*). RC07, however, upheld the more traditional topology where the trematopids lie outside a clade formed by the other dissorophoids while *Micromelerpeton* forms the sister-group of Branchiosauridae (most recently found by both analyses presented in *Pardo, Small & Huttenlocker, 2017*: fig. S6, and by the Bayesian but not the parsimony analysis in their fig. S7). Further, in the dispute over whether *Ecolsonia* is a trematopid or a dissorophid (*Berman, Reisz & Eberth, 1985*), RC07 found a sort of compromise position in that *Ecolsonia* came out one node closer to *Broiliellus brevis* and the remaining dissorophoids than to (*Acheloma + Phonerpeton*).

This position remains a possibility for *Ecolsonia* in some trees from Analyses R1–R3 and R5. The other trees, as well as all trees from Analyses R4, R6 and EB, place *Ecolsonia* as a trematopid (BPO = **56**; BPE = 38; PP = **94**) like *Ruta (2009)*, *Maddin et al. (2013b)*, *Holmes, Berman & Anderson (2013)*, *Dilkes (2015a)* and *Pardo, Small & Huttenlocker (2017)*, but continue to find a clade of all non-trematopid dissorophoids (BPO = **55**; BPE = 14; PP = 48 excluding *Branchiosaurus* and *Acanthostomatops*, see below). Within that clade, *Broiliellus* can be the sister-group to most or all of the rest (which has the values BPO = 39, PP = 60) or to Amphibamidae alone (BPE = 9); the latter option is the only one in Analyses R4 and R6. Despite our best efforts to eliminate the effects of ontogeny on their scores, *Micromelerpeton* and Branchiosauridae remain sister-groups throughout the analyses with the original taxon sample (BPO = **61**). This does not change when taxa are added (BPE = 27, PP = 55), except that *Branchiosaurus*, as it turns out, never joins the clade formed by the "branchiosaurids" of the original taxon sample (see below).

As expected, *Mordex* settles at the base of the trematopid clade (R4, R6, some MPTs from R5: (*Mordex* (*Ecolsonia* (*Acheloma, Phonerpeton*))); BPE = 28) or grade (some MPTs from R5). Analysis EB finds it slightly more rootward, next to *Iberospondylus* (PP = 46).

Interestingly, none of our trees find all supposed branchiosaurids to form a clade; such a clade is contradicted by a BPE of 8 and a PP of 60. *Branchiosaurus* is found closer to *Broiliellus* + Amphibamidae (R4, R6, some MPTs from R5) or closer to the root of Dissorophoidea (the other MPTs from R5; BPE = 8; PP = 48), yet *Micromelerpeton* always stays with the bulk of "branchiosaurids" as described above (BPE = 27; PP = 55). *Tungussogyrinus*, its somewhat froglike ilium (state ILI 9(1)) and Early Triassic age notwithstanding, nests with *Branchiosaurus* in R4, R6 and some MPTs from R5 (BPE = 16), next to the three branchiosaurids from the original sample in other trees from R5, next to them plus *Micromelerpeton* in yet others from R5 and in EB (PP = 54), and next to *Schoenfelderpeton* in the remaining MPTs from R5. We think that the present matrix cannot test whether the "branchiosaurids" are a clade (*Fröbisch & Schoch, 2009a*; *Maddin et al., 2013b*) or a polyphyletic assemblage of larval and neotenic dissorophoids.

RC07 could not resolve if the conventional amphibamids in their sample (*Amphibamus*, *Platyrhinops*, *Eoscopus*, *Doleserpeton*) formed a clade with respect to *Micromelerpeton* and the branchiosaurids. The analyses with the original taxon sample (R1–R3) find them as a clade (BPO = **60**; also containing the salientians in R3 as the sister-group of *Doleserpeton*). This remains the case when taxa are added (BPE = 25 and PP = 43 for a clade that also contains *Gerobatrachus*). *Micropholis*, however, joins this clade only in some MPTs from R5 (as the sister-group of *Eoscopus*, or one node rootward of *Gerobatrachus*); elsewhere, it is the sister-group of the "branchiosaur" clade described above. In Analyses R4, *Doleserpeton* and *Gerobatrachus* are sister-groups (BPE = 40; against a PP of 51); in R5, *Doleserpeton* and *Gerobatrachus* are equally parsimoniously found as the sister-group of Lissamphibia, with the other one of the two being the next more distant relative; in R6, *Doleserpeton* is closer to Batrachia. All this remains to be tested with much larger samples of potential amphibamids and amphibamid-related characters, without reducing the sample of other temnospondyls.

While we are quite skeptical about a close relationship of *Micromelerpeton* to some or all branchiosaurids (compare with *Wiens, Bonett & Chippindale, 2005*), the main alternative in the literature—Micromelerpetidae as the sister-group to all other dissorophoids (*Schoch, 2013*; *Maddin et al., 2013b*; *Dilkes, 2015a*)—does not seem to be well supported either; it never occurs in our MPTs and contradicts the bootstrap and Bayesian trees as well (BPO = **84**; BPE = 27; PP = 60). The name Dissorophoidea should certainly not be defined with *Micromelerpeton* as one of only two specifiers, as *Schoch (2013)* has done. That said, the older definition of Dissorophoidea by *Yates & Warren (2000*: 86*)* as all organisms closer to **Dissorophus* than to *Eryops* (**Dissorophus* being very closely related to *Broiliellus*) applies to a clade that contains, in the trees from some of our analyses, not only the traditional dissorophoids but also various other temnospondyls, like *Balanerpeton*, Dendrerpetidae, *Capetus*, Dvinosauria and, in some MPTs from R5 (Fig. 16), even Stereospondylomorpha.

*Schoch (2013)* listed dvinosaur-like character states of micromelerpetids and used them to argue for a sister-group relationship of Micromelerpetidae to the rest of Dissorophoidea (accepted without comment by *Fröbisch et al., 2015*) together with a sister-group relationship (or nearly so) of Dissorophoidea and Dvinosauria. A close relationship of

Dvinosauria and Dissorophoidea is never found in our MPTs (contradicted only, however, by BPO = 30, BPE = 6 and PP = 64). We wonder if the similarities listed by *Schoch (2013)* are instead homoplastic and indicate that the micromelerpetids are a "dissorophoid version of a dvinosaur"—a clade that generally had a dvinosaur-like lifestyle as aquatic predators with more or less anguilliform locomotion but, unlike dvinosaurs, was capable of dispersing over land in the adult stage (at least in the one species, *M. credneri*, that is known to have had non-neotenic, skeletally mature adults in one population: *Lillich & Schoch, 2007*; *Schoch, 2009*). This might even explain why dvinosaurs and micromelerpetids have not been found in the same sites (see *Schoch & Milner, 2014*; *Cisneros et al., 2015*): maybe micromelerpetids were competitively excluded from bodies of water that dvinosaurs could reach, but were able to diversify in inland water bodies out of the reach of dvinosaurs.

In the matrix of *Dilkes (2015a)*, a revision and extension of that by *Schoch (2013)*, Dvinosauria and Dissorophoidea are held together exclusively or almost exclusively by paedomorphic adaptations to an aquatic lifestyle ("Node D" in *Dilkes, 2015a*: 81): hypobranchials ossified in adults; lack of ossification of the glenoid facet of the scapula and of the carpals, pubis and tarsals; and weak torsion of the humerus. The remaining character state, jaw joints and occipital condyle(s) being at the same transverse level, may also belong here; it is in fact absent in the more mature known *Micromelerpeton* specimens (*Boy, 1995*; *Schoch, 2009*: fig. 2a, b; see also our ch. 146 in App. S1). While the sister-group of Dissorophoidea in the tree of *Dilkes (2015a*: fig. 13*)* is the terrestrial Zatracheidae (see below), followed by Dvinosauria on the outside, Dissorophoidea would be reconstructed as ancestrally aquatic because Micromelerpetidae is found as the sister-group to the other dissorophoids, among which Branchiosauridae is sister to an amphibious/terrestrial clade composed of Amphibamidae, Dissorophidae and Trematopidae; we consider this position of Branchiosauridae unlikely because of the evidence for a lack of a clear distinction between amphibamids and branchiosaurids (*Vallin & Laurin, 2004*; *Fröbisch & Schoch, 2009a*; *Werneburg, 2012a*; *Maddin et al., 2013b*; and references therein; these results are more or less congruent with the MPTs from our analyses with added taxa, where such a clear distinction is likewise absent).

### Other added temnospondyls

*Nigerpeton* was described as a cochleosaurid (*Steyer et al., 2006*; *Sidor, 2013*). This was confirmed by *Pawley (2006)*, *Schoch (2013)*, *Dilkes (2015a)* and *Pardo, Small & Huttenlocker (2017)*; *McHugh (2012)* found it and *Edops* together to form the sister-group of Cochleosauridae (Fig. 25D). In our analyses (Figs. 26C and 26D), it is always found next to *Saharastega* (BPE = **50**; PP = **99**).

*Saharastega* was described as a temnospondyl of rather uncertain position (*Damiani et al., 2006*). In dorsal view, its skull is unusual for a temnospondyl in some ways, which led *Yates (2007)* to suggest that it was a seymouriamorph. Its palate, however, is unremarkable for a temnospondyl and quite unlike that of any seymouriamorph; we also note that the skull roof is very reminiscent of the temnospondyl **Macrerpeton* (*Schoch & Milner, 2014*: fig. 38B, E). *Pawley (2006*; Fig. 25B*)* found it to be the sister-group

of all other cochleosaurids; *McHugh (2012*; Fig. 25D*)* recovered it as the sister-group of Zatracheidae, in this matrix represented by *Acanthostomatops (see below); *Schoch (2013)*, followed by *Dilkes (2015a)*, explicitly excluded it as too incomplete.

The *Saharastega*-*Nigerpeton* clade is not found in or close to Cochleosauridae in any MPTs (Figs. 26C and 26D). R4 places it next to *Eryops*, next to Stereospondylomorpha or next to all traditional temnospondyls except *Edops* and Cochleosauridae; R6 adds a position inside Stereospondylomorpha, R5 a further one, where it can be joined by Dvinosauria. Analysis B2, finally, finds it as the sister-group of Edopoidea (BPE = 4), and EB places it next to the cochleosaurid *Chenoprosopus* (PP = 33), together forming the sister-group (PP = 25) of a novel *Edops*–*Cochleosaurus* clade (PP = 47). We hope for larger matrices and a restudy of **Macrerpeton* more detailed than that by *Montanari (2012*: chapter II*)*. *Saharastega* and *Nigerpeton* are both unusually large for Paleozoic temnospondyls, have rather long snouts, and are known almost only from skulls with unusual and not very good preservation; and while *Nigerpeton* shares rare features with the cochleosaurids, a fully stereospondylous intercentrum (state TRU VER 13-14(0); see ch. 265 in App. S1) is catalogued as part of the holotype (D. Marjanović, pers. obs. 2016).

*Iberospondylus* is better preserved than the above two, but no less enigmatic. In both descriptions (*Laurin & Soler-Gijón, 2001*, *2006*) it was recovered as the sister-group to a clade that contained *Eryops*, **Parioxys* (on which see Materials and Methods: Treatment of OTUs: Taxa that were not added), **Zatrachys* (*Zatracheidae; see below), the included dissorophoids and the stereospondylomorph *Sclerocephalus*—a position identical (but for the taxon sample) to that found by *Pardo, Small & Huttenlocker (2017*; Fig. 25H*)*. *Pawley (2006)* found *Iberospondylus* as the sister-group of Dissorophoidea (Fig. 25B, together with **Parioxys*); *McHugh (2012)* did not include or mention it; *Dilkes (2015a*: 69*)* recovered it as an edopoid or an eryopid with a reduced taxon sample (Figs. 25E and 25F). Our analyses all side with Pawley's in finding *Iberospondylus* (Figs. 26C and 26D), or in the case of EB an *Iberospondylus*-*Mordex* clade (with PP = 46), as the sister-group of Dissorophoidea (BPE = 17; PP = 64). *Iberospondylus* and Dissorophoidea share the dorsal process on the caudoventral tip of the quadrate (state QUA 1(1)).

*Acanthostomatops* represents Zatracheidae, a clade with terrestrial adults long thought to lie close to *Eryops* and/or Dissorophoidea. *Pawley (2006*; Fig. 25B*)* found it next to (*Iberospondylus* + Dissorophoidea), *McHugh (2012*; Fig. 25D*)* found it nested within Eryopidae together with *Saharastega*, *Dilkes (2015a*: fig. 11; Figs. 25E and 25F*)* recovered (Dvinosauria (*Zatracheidae, Dissorophoidea)) in his analyses of reduced taxon samples, *Pardo, Small & Huttenlocker (2017*: fig. 2, S7; Fig. 25H*)* found (Eryopiformes (*Zatracheidae (**Lapillopsis*, Dissorophoidea))) based on a closely related matrix. In our Analyses R4 and R6, and some MPTs from R5, unexpectedly, *Acanthostomatops* is highly nested within Dissorophoidea, forming part of the sister-group of *Broiliellus* + Amphibamidae together with a *Branchiosaurus*-*Tungussogyrinus* clade. In other MPTs from R5, *Acanthostomatops*, *Branchiosaurus* and optionally *Tungussogyrinus* lie one node rootward (instead of crownward) of the "branchiosaur" clade. In yet others,

*Acanthostomatops* is the sister-group of (*Capetus* (*Balanerpeton*, Dendrerpetidae)), together forming the sister-group to (*Palatinerpeton* (*Iberospondylus*, Dissorophoidea)); yet others have a (*Balanerpeton* (*Palatinerpeton* (Dendrerpetidae, *Acanthostomatops*))) clade next to *Iberospondylus* + Dissorophoidea. Analysis B2 finds *Acanthostomatops* alone as the sister-group of *Iberospondylus* + Dissorophoidea (BPE = 11), while EB places it next to all dissorophoids except Trematopidae (PP = 34). While we never find *Acanthostomatops* next to *Eryops*, we occasionally did so in the previous version of our matrix (*Marjanović & Laurin, 2016*), even though practically all that has changed in the temnospondyl part of the matrix since then is a series of corrections to *Nigerpeton* and *Saharastega*. Evidently, the position of *Acanthostomatops* is not strongly constrained by our character sample (Figs. 26C and 26D) despite concerning the least modified of all zatracheids.

*Palatinerpeton* is poorly known and has an unexpected combination of features. Unsurprisingly, then, it has never before been included in a large phylogenetic analysis. *Boy (1996)* presented a detailed phylogenetic hypothesis that put it close to Stereospondylomorpha and to an eryopid-*zatracheid-**Parioxys* clade, and less close to *Capetus*; of all other temnospondyls, however, only Edopoidea was included in the investigation which polarized its characters with reference to non-temnospondyl outgroups. *Schoch & Witzmann (2009a)* found it as the sister-group of Eryopiformes, not close to *Acanthostomatops*, which instead emerged as the sister-group of Dissorophoidea. Despite the dearth of data, Analyses R4 and R6 as well as some MPTs from R5 place *Palatinerpeton* as the sister-group to either Dendrerpetidae (against a BPE of 37 and a PP of **83**) or *Iberospondylus* + Dissorophoidea (against a BPE of 11 and a PP of 64; Figs. 26C and 26D); the bootstrap analysis conducted under the same conditions (B2) similarly places *Palatinerpeton* as the sister-group to (*Capetus* (*Balanerpeton*, Dendrerpetidae)) with a BPE of 5, all together forming the sister-group of (*Acanthostomatops* (*Iberospondylus*, Dissorophoidea)) with a BPE of 6, an arrangement similar to some MPTs from R5 (Fig. 26C). The remaining MPTs from R5 (also Fig. 26C) find *Palatinerpeton* next to Dendrerpetidae + *Acanthostomatops*. Analysis EB, in contrast, places it next to all other temnospondyls together (PP = 35; Fig. 20). *Palatinerpeton* could become important for the question of temnospondyl phylogeny; a μCT scan could perhaps reveal the unexposed surfaces and fill in much of its missing data.

*Erpetosaurus* was originally thought to be a colosteid, but has long been recognized as an unusual dvinosaur with a few colosteid-like features and was redescribed as such by *Milner & Sequeira (2011)*. We confirm this as discussed above under Colosteidae (BPE = 14; PP = **77**) and shown in Figs. 26C and 26D.

### Chroniosuchia

A clade composed of *Chroniosaurus* and *Bystrowiella* (BPE = 42; PP = 67) has a fully resolved position (Fig. 27G) one node crownward of *Gephyrostegus* or *Casineria* (depending on the latter's position) (BPE = 7; PP = 55) and one node rootward of *Solenodonsaurus* (or, in Analysis EB, of Seymouriamorpha including *Solenodonsaurus*;

BPE = 6; PP = 44). This forms a kind of compromise between the positions found in earlier phylogenetic analyses, while disagreeing with all of them: the closely related analyses of *Clack & Klembara (2009)*, *Clack et al. (2012b)* and *Witzmann & Schoch (2017)* found the included chroniosuchians in a polytomy with *Silvanerpeton*, Anthracosauria and a "gephyrostegid"-seymouriamorph-amniote-"microsaur" clade (Fig. 27D); *Klembara, Clack & Čerňanský (2010)* resolved them as anthracosaurs, agreeing with earlier suggestions (*Laurin, 2000*, and references therein) that the chroniosuchians were the geologically youngest anthracosaurs; *Schoch, Voigt & Buchwitz (2010)* and *Buchwitz et al. (2012)* found them to be the sister-group of the only included lepospondyl—which *Buchwitz et al. (2012)* and table 1 of *Schoch, Voigt & Buchwitz (2010)* stated to be the poorly known *Asaphestera*, while the text of *Schoch, Voigt & Buchwitz (2010)* mentioned *Pantylus* instead; *Klembara et al. (2014*; Fig. 27E) recovered them as the sister-group of a gephyrostegid-seymouriamorph clade, though their bootstrap tree has them as the sister-group of Gephyrostegidae alone. All these analyses relied on quite small data matrices; however, unlike us, *Schoch, Voigt & Buchwitz (2010)* and *Buchwitz et al. (2012)* included more than two chroniosuchians in their analyses (the latter's matrix building on the former's).

Our analyses thus place Chroniosuchia as the sister-group to a clade with terrestrial adults, with which they share various adaptations to walking and bearing weight that all more rootward taxa, including the "gephyrostegids" and *Casineria, lack (with exceptions nested within the temnospondyls).

### Solenodonsaurus

*Solenodonsaurus* was long thought to lie particularly close to Diadectomorpha + Amniota (*Laurin & Reisz, 1999*; *Vallin & Laurin, 2004*; Fig. 27A). RC07 found it just outside the smallest clade of diadectomorphs, amniotes and "lepospondyls" (Fig. 27C); we consistently find it just outside the smallest clade formed by the above and all seymouriamorphs (which has BPO = 22; BPE = 3; Figs. 27F and 27G), except in Analysis EB, where *Solenodonsaurus* is the sister-group to all other seymouriamorphs (PP = 51).

*Danto, Witzmann & Müller (2012)*, who redescribed *Solenodonsaurus*, unexpectedly found it to be a lepospondyl instead; however, their changes to the scores of *Solenodonsaurus* in the matrix of *Ruta, Coates & Quicke (2003)* clearly contradict their own text and/or illustrations in several cases (App. S1).

### Seymouriamorpha

Uniquely, RC07 found Seymouriamorpha to be a grade, with *Seymouria* and *Kotlassia* successively closer to a (*Solenodonsaurus* (Lepospondyli (Amniota, Diadectomorpha))) clade than a clade formed by the remaining seymouriamorphs (Figs. 1 and 27C). In all of our trees, *Seymouria* is part of a clade with some or all other traditional seymouriamorphs; in Analyses R1 and R3 (Figs. 27F and 27G), *Kotlassia* can be the sister-group of that clade or one node rootward of Seymouriamorpha (this is the only option in R2), and the (*Leptoropha + Microphon*) clade can be nested within it (always in R2; Fig. 27F) or lie one node crownward of it. The corresponding bootstrap tree (B1) shows (*Kotlassia + Seymouria*),

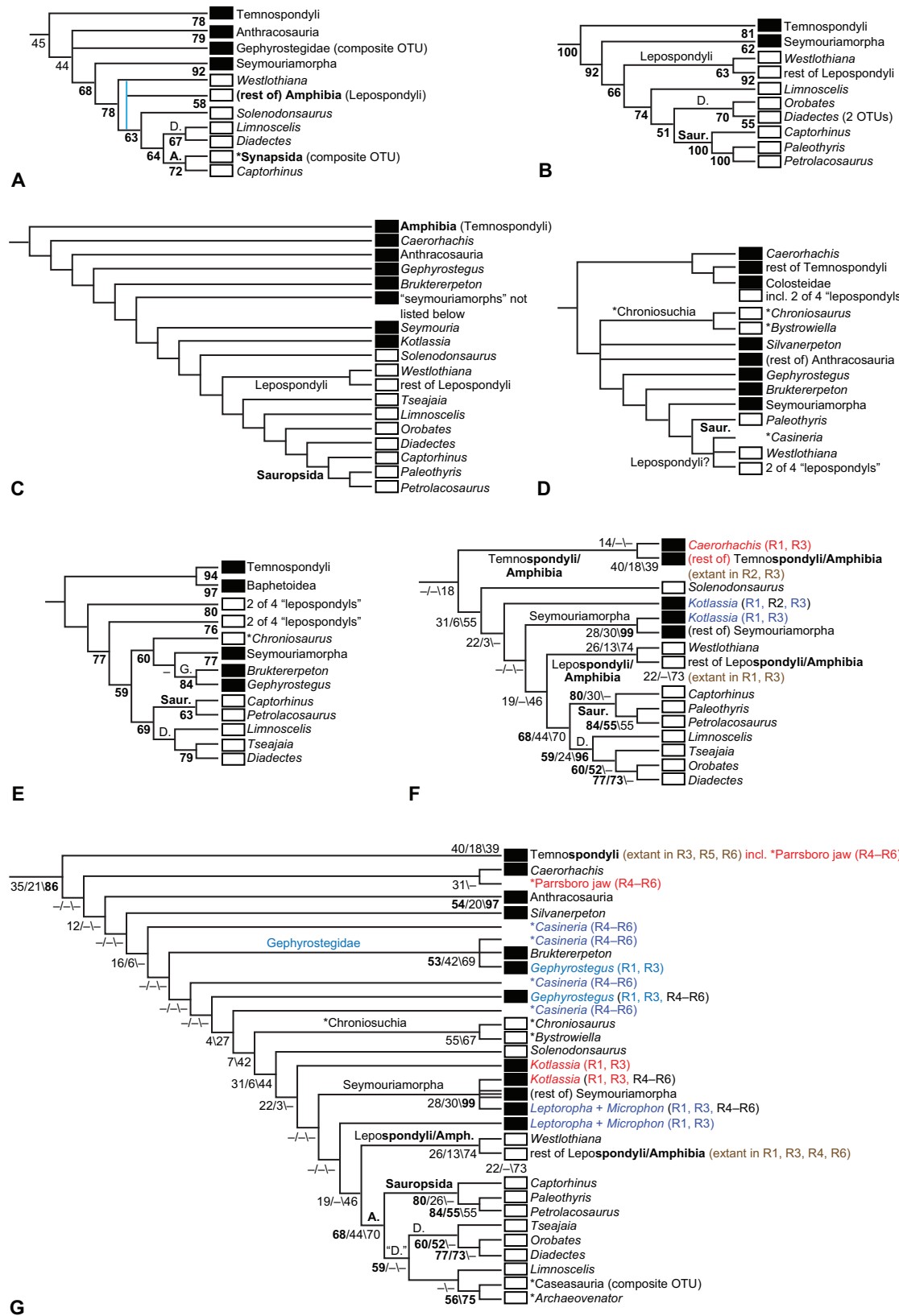

**Figure 27 Amniote relationships and distribution of the intertemporal bone.** Taxa not included in our analyses are omitted; those included in the expanded but not the original taxon sample are marked with an asterisk. Several named clades are collapsed. The matrices of (B), (C), (F) and (G) are

ultimately based on that of *Ruta, Coates & Quicke (2003)*, so they should not be considered fully independent of each other. Numbers are bootstrap percentages. Filled rectangles indicate presence of the intertemporal (state 0 of ch. 32—PAR 2/POSFRO 3/INTEMP 1/SUTEMP 1; see App. S1), empty rectangles its absence (any other state); the absence of a rectangle indicates missing data. A., Amniota; D., Diadectomorpha; Saur., Sauropsida. (A) *Vallin & Laurin (2004)*. (B) *Pawley (2006*: chapter 6*)*. (C) RC07; "rest of Lepospondyli" excludes the adelospondyls (see Figs. 1 and 24A), bootstrap values were not published. (D) *Clack et al. (2012b*; lacking *Bystrowiella)* and *Witzmann & Schoch (2017)*; all bootstrap percentages in this part of the tree are below 50 in both versions and were not published. (E) *Klembara et al. (2014)*. Bootstrap percentages below 50 were not published, but note that a sister-group relationship of *Chroniosaurus* and Gephyrostegidae (G.) is supported by a bootstrap value of 53%. (F) Topologies from our Analysis R2 and some MPTs from R1 and R3, where Temnospondyli is closer to Amniota than Anthracosauria. Numbers are BPO/BPE\PP. (G) Topologies found in Analyses R4–R6 and the remaining MPTs from R1 and R3. Numbers below nodes are BPO/BPE\PP; the space for BPO is omitted where inapplicable. "D." indicates Diadectomorpha for the original but not the extended taxon sample.

supported by a BPO of 26, as the sister-group to all other seymouriamorphs (BPO of Seymouriamorpha = 28). The addition of *Karpinskiosaurus, however, rearranges seymouriamorph phylogeny and prevents *Kotlassia*, *Leptoropha* and *Microphon* from leaving this clade (Fig. 27G): throughout Analyses R4–R6, the latter two are the sister-group of all other seymouriamorphs, among which a clade of (*Seymouria* (*Kotlassia*, *Karpinskiosaurus*)) is the sister-group to the remainder (Discosauriscidae; BPE = 27).

The only seymouriamorph clade with good support is (*Leptoropha* + *Microphon*) at a BPO of **88**, a BPE of **85** and a PP of **100**. All other bootstrap percentages, including those for Seymouriamorpha as a whole, are below 40 in both analyses; Analysis EB finds a PP of **91** for Discosauriscidae and one of **99** for, as mentioned, Seymouriamorpha excluding *Solenodonsaurus*, while the other branches stay below a PP of 60.

Neither the classification of *Bulanov (2003)*, which was not based on a phylogenetic analysis, nor—with the exception of the *Seymouria*-*Karpinskiosaurus* clade found in Analysis EB (PP = 27)—the results of the analysis by *Klembara et al. (2014*: fig. 8*)* are congruent with any of our results. From this and the very low support values (though see above for PP), we conclude that at least the relationships of *Utegenia*, *Ariekanerpeton* and (*Leptoropha* + *Microphon*) to each other and to *Discosauriscus* remain unclear; while the matrix by *Klembara et al. (2014)* has a denser sampling of seymouriamorphs and of characters that are relevant to their phylogeny, our matrix has a much larger sample of non-seymouriamorphs to root the seymouriamorph tree.

Curiously, Analysis B2 finds Seymouriamorpha inside the amniote-"lepospondyl" clade, as the sister-group to the diadectomorph-amniote clade (Fig. 19); in other words, Seymouriamorpha and the "lepospondyls" switch places compared to all other MPTs and B1. This signal is not, or not only, due to our revision, but was already present in the original matrix, as shown by its occurrence in some MPTs from O3 (Data S7). If corroborated by future studies, this topology would amount to a reversal to the textbook hypothesis of the mid-late 20th century (which was still assumed, for unstated reasons, by *Fröbisch & Schoch, 2009a*: fig. 1, and by *Berman, 2013*). However, the sample of characters where seymouriamorphs and diadectomorphs share the same apomorphic state may already be exhaustive or nearly so in the present matrix, and the bootstrap support for this topology is very low (BPE = 9; contradicting a BPO of 19 and a PP of 46).

### Amniota and Diadectomorpha

Uniquely, RC07 found the diadectomorphs to form a Hennigian comb with respect to the sauropsids (Figs. 1 and 27D): (*Tseajaia* (*Limnoscelis* (*Orobates* (*Diadectes* (*Captorhinus* (*Paleothyris*/*Protorothyris*, *Petrolacosaurus*))))))). This was due to numerous misscores, at least some of which were evidently caused by reliance on ancient literature about badly preserved specimens. In Analyses R1–R3 we find (Figs. 27F and 27G) a monophyletic Diadectomorpha (BPO = **59**; BPE = 24; PP = **96**), resolved as (*Limnoscelis* (*Tseajaia* (*Orobates*, *Diadectes*))) like in the latest published analyses (*Reisz, 2007*; *Kissel, 2010*; *Berman, 2013*; further supported by *Berman, Reisz & Scott, 2010*), as the sister-group of Sauropsida; the clade of all diadectomorphs except *Limnoscelis* is found in all of our parsimony analyses and supported in both bootstrap analyses (BPO = **60**; BPE = **52**; against a PP of **76**), and the (*Diadectes* + *Orobates*) clade enjoys some of the highest values in the entire trees (BPO = **77**; BPE = **73**; against a PP of 59).

As mentioned (Materials and Methods: Treatment of OTUs: Taxa added as new OTUs for a separate set of analyses), the three amniotes included by RC07 are all sauropsids, so RC07 were unable to test whether the diadectomorphs are amniotes. We therefore added the synapsid OTUs \*Caseasauria (composite) and \*Archaeovenator, which robustly come out as sister-groups (BPE = **56**; PP = **75**). Surprisingly, however, diadectomorph monophyly breaks down; more precisely, throughout the parsimony analyses with added taxa, *Limnoscelis* never emerges as a diadectomorph (Fig. 27G). Instead, Analyses R4–R6 find both clades as amniotes, specifically recovering Amniota as (Sauropsida (Diadectomorpha (*Limnoscelis* (\*Caseasauria, \*Archaeovenator)))); in terms of the topology found by Analyses R1–R3, \*Caseasauria and \*Archaeovenator are not placed next to Sauropsida (as they are in Analysis B2: BPE = 30), but next to *Limnoscelis*.

Under the conditions of Analysis R4, only one additional step is required in Mesquite to restore Amniota and Diadectomorpha as sister-groups—at the cost of rendering *Captorhinus* a synapsid; moving it back into Sauropsida needs one step more. Analysis EB weakly supports a very different arrangement where *Limnoscelis* is a diadectid so that *Tseajaia* is the sister-group to all other diadectomorphs (PP = **76**), Diadectomorpha is the sister-group of *Captorhinus* (PP = 49), and both together lie on the synapsid side of Amniota (PP = 44). All this is in stark contrast to the much stronger support for the smallest clade that contains all of the taxa mentioned in this section: BPO = **68**, BPE = 44, PP = 70. The support for *Captorhinus* as a sauropsid depends very strongly on the taxon sample (BPO = **80**; BPE = 26; contradicted by a PP of 49), as does that for (*Paleothyris* + *Petrolacosaurus*) to a lesser extent: BPO = **84**; BPE = **55**; PP = 55.

Many of the characters presented by *Berman (2013)* as supporting the membership of Diadectomorpha in Amniota are not parsimony-informative given his taxon sample; the others are already included in our matrix. Any satisfactory resolution of amniote origins will require a larger sample of amniotes and possibly hitherto unrecognized characters. Further, several of those features of the description of *Tseajaia* by *Moss (1972)* that were not revisited by *Berman, Sumida & Lombard (1992)* seem unusual in the light of more recent work on diadectomorphs and may warrant redescription.

## Westlothiana

Despite numerous changes to its scores, *Westlothiana* stays (Fig. 27) in the position found by RC07 and *Pawley (2006)*: it is the sister-group to all other "lepospondyls". Although bootstrap support is low (BPO = 26) or only exists for a similar topology (BPE = 10 for *Westlothiana* + *Utaherpeton*; BPE = 7 for both + all other amphibians except the adelospondyls; BPE = 13 for Amphibia), there is moderate Bayesian support (PP = 74).

### *The lepospondyl problem*

> [ . . . ] the fact that we find throughout the Carboniferous and early [sic] Permian varied series of small, non-labyrinthodont amphibians, which I have classed as lepospondyls in a broad use of that term, presents an evolutionary problem for which we have at present no solution.
>
> – *Romer (1964*: 158*)*

> Recent research is beginning to shed new light on the anatomy and relationships of rare and problematic forms, such as lepospondyls [ . . . ] [four references]. Several issues related to lepospondyl interrelationships are likely to undergo extensive revision in the near future. Published analyses of lepospondyls reveal a disconcerting lack of agreement, to the point that almost any pattern of relationships has been proposed [ . . . ] [11 references].
>
> – *Ruta, Coates & Quicke (2003*: 290*)*

> And what is a lepo . . . ?
> – Gary Johnson, candidate for president of the USA, in a TV interview in 2016

For much of the 20th century, all limbed vertebrates (or nearly so) except Lissamphibia and Amniota were divided into Labyrinthodontia and Lepospondyli. When phylogenetic analysis began to be introduced in the mid-1990s, these two taxa had largely come to be seen as wastebaskets, Labyrinthodontia basically containing the large-bodied taxa and Lepospondyli the small-bodied ones (except small temnospondyls). It was expected that Labyrinthodontia would turn out to be paraphyletic and Lepospondyli to be polyphyletic. The first prediction has held up, if perhaps less radically than expected; the latter has fared much less well, in that analyses like those of *Ruta, Coates & Quicke (2003)*, *Vallin & Laurin (2004*; Fig. 28A) or *Maddin, Jenkins & Anderson (2012*; Fig. 28F)* turned up a lepospondyl clade lying not close to Temnospondyli or Colosteidae as some had surmised, but right next to Diadectomorpha + Amniota (Fig. 27), reminiscent of the old idea of a close relationship between amniotes and "microsaurs" that Romer had so thoroughly excluded in the mid-20th century. (See, however, *Pardo & Anderson (2016)* and *Pardo et al. (2017, 2018)*.)

RC07 found a novel variant supported by previously neglected characters: they upheld the picture described above, except for finding that the adelospondyls (*Acherontiscus* and Adelogyrinidae) were not lepospondyls or anywhere near, forming instead a clade with the faraway colosteids (Figs. 1 and 28D). However, several plesiomorphic scores for the adelospondyls, which kept them away from the remaining "lepospondyls" and were
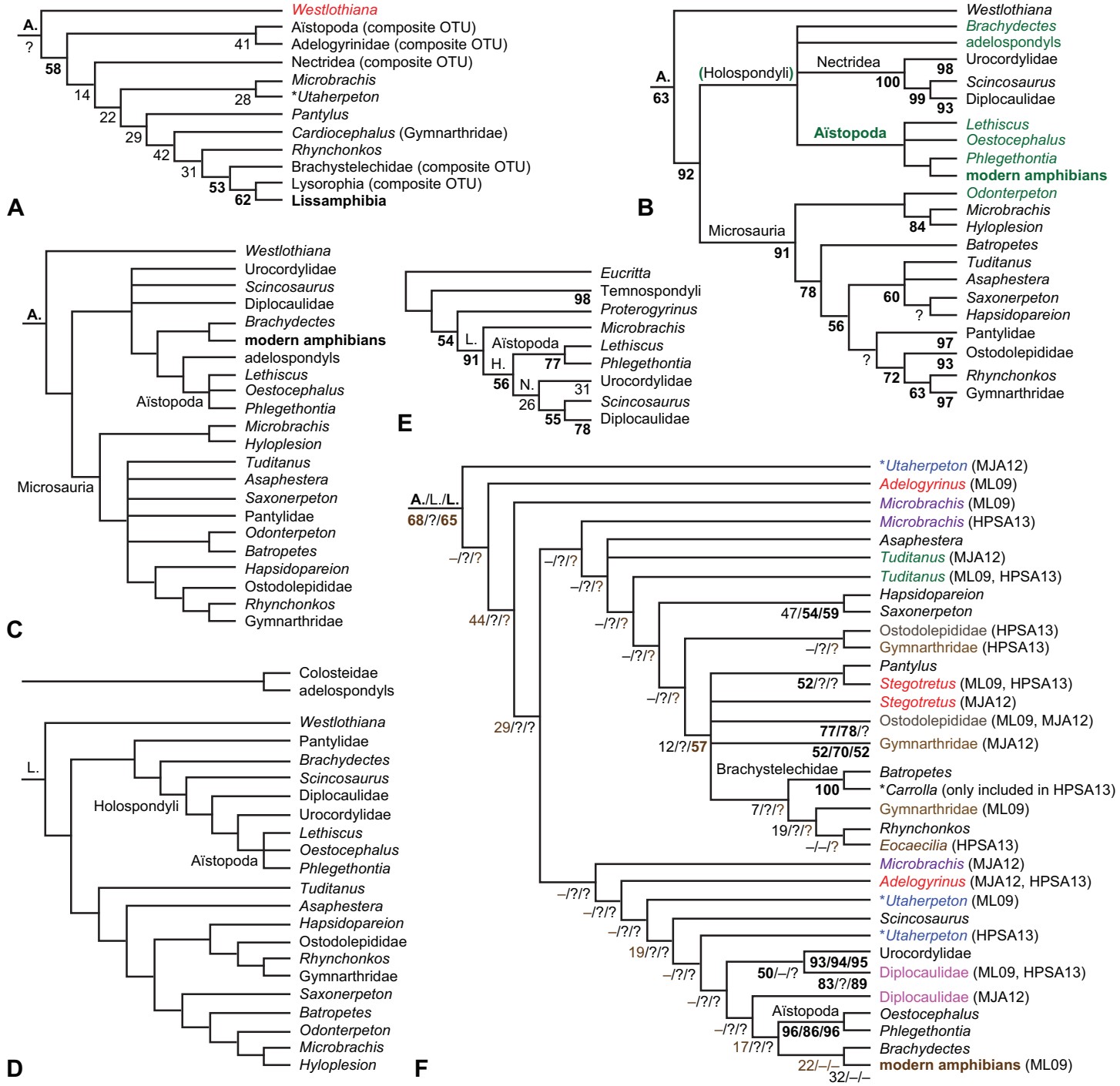

**Figure 28 Hypotheses on "lepospondyl" relationships since 2004.** Various named clades are collapsed. Taxa not included in our analyses are omitted; those included in the expanded but not the original taxon sample are marked with an asterisk. A., Amphibia; HPSA13, *Huttenlocker et al. (2013)*; L., Lepospondyli; MJA12, *Maddin, Jenkins & Anderson (2012)*; ML09, *Marjanović & Laurin (2009)*. (A) *Vallin & Laurin (2004)*. The other position of *Westlothiana* lies outside Amphibia and is not shown (see Fig. 27A). Numbers are published bootstrap percentages. (B) *Pawley (2006*: fig. 88*)*. Numbers are bootstrap percentages from fig. 63 (those below 50 were not reported), which shows an analysis that omitted the taxa written in dark green but found a congruent topology. (C) *Pawley (2006*: fig. 91*)*: cranial characters and data as in (B), but the postcranial ones unchanged from *Ruta, Coates & Quicke (2003)*. (D) RC07. (E) *Milner & Ruta (2009)*; entire tree except the aïstopod **Ophiderpeton*. Adelospondyls and lysorophians were not included. H., Holospondyli; N., Nectridea. Numbers are published bootstrap percentages. (F) Matrices derived from that of *Anderson et al. (2008a)*, namely: (1) ML09; (2) MJA12 as reanalyzed by *Marjanović & Laurin (2018)*, lacking the corrections proposed by ML09 and those by

*Sigurdsen & Green (2011)* but containing those by *Maddin & Anderson (2012)* and new ones by MJA12; (3) HPSA13, lacking all of the proposed corrections mentioned here but containing new ones. Note that ML09 found the LH, MJA12 the TH and HPSA13 the PH, and that ML09 and HPSA13 did not find *Eocaecilia* in the same position. Numbers are bootstrap percentages from ML09, MJA12 and HPSA13 in this order (the latter two did not report percentages below 50). *Pardo, Small & Huttenlocker (2017*: fig. S6*)* analyzed an unpublished matrix based on MJA12 with some updates based on HPSA13; the MPTs and the Bayesian tree are almost fully congruent with (F), especially with MJA12, but neither bootstrap values nor posterior probabilities were published.

shared with the colosteids, appear indefensible. For example, RC07 scored the adelospondyls as having partially covered lateral-line canals on the skull (our ch. 100: state SC 1(1) for *Dolichopareias*, SC 1(1 or 2) for *Acherontiscus* and *Adelospondylus*; state 1 is "mostly enclosed with short sections in grooves", state 2 is "mostly in grooves with short sections enclosed"), but the grooves (*Carroll, 1969a*; *Andrews & Carroll, 1991*) appear wide and shallow and were most likely genuinely interrupted, with the lateral-line organ perhaps continuing in the skin away from the bone—rather than lying under the bone surface—or perhaps being interrupted itself. This condition, which we have scored as part of the existing state SC 1(3)—lateral-line organ entirely in grooves, not covered by bone—is best seen in *Archegosaurus* (*Witzmann, 2006*) and was evidently also present in *Proterogyrinus* (*Holmes, 1984*), which RC07 had also misscored as having state 1 or 2.

Analysis R2 (Fig. 29B) groups the adelospondyls with the aïstopods and urocordylids among the "lepospondyls" that are dragged into Temnospondyli by the constraint on Lissamphibia. All other phylogenetic analyses place the adelospondyls with the urocordylids and aïstopods in highly nested positions within a "lepospondyl"/amphibian clade (PP = **76**; Figs. 29A and 29C). Moving the adelospondyls next to Colosteidae on a tree from Analysis R1 requires only two extra steps; moving them into Colosteidae on a tree from R4 (as the sister-group to *Aytonerpeton*, together forming the sister-group of *Colosteus* + *Greererpeton*) requires no less than six extra steps. The bootstrap analyses do not strongly support any position; Analysis B1 places the adelospondyls just rootward of Holospondyli (the highest BPO that keeps them away from Holospondyli is 25, the highest that keeps them away from Colosteidae is 35), Analysis B2 finds the adelospondyls next to a clade of all other amphibians (BPE = 13 rootward, 7 crownward).

In spite of our results, including a node with PP = **86** (*Caerorhachis* and everything more crownward: Fig. 20), we consider it possible that the adelospondyls do not belong together with most or all of the other "lepospondyls", but are rather colosteid-grade animals (as found again in *Witzmann & Schoch, 2017*). One reason is the fact that their ceratobranchial bones, not considered in the present matrix, retain longitudinal grooves for gill arteries (*Witzmann, 2013*). Such grooves indicate internal gills (*Schoch & Witzmann, 2011*) and are not found in the only other possibly aquatic "lepospondyl" from which hyobranchial bones are known, the lysorophian *Brachydectes* (*Wellstead, 1991*; *Witzmann, 2013*); there is further no trace of internal gills in extant amphibians—the gills of tadpoles, although soon covered by a lid, are homologous to external gills and thus to the septa between internal gills (*Schoch & Witzmann, 2011*, and references therein). In this light, we would like to draw attention to the embolomerous vertebrae of *Acherontiscus*, where the intercentra are much larger (state TRU VER

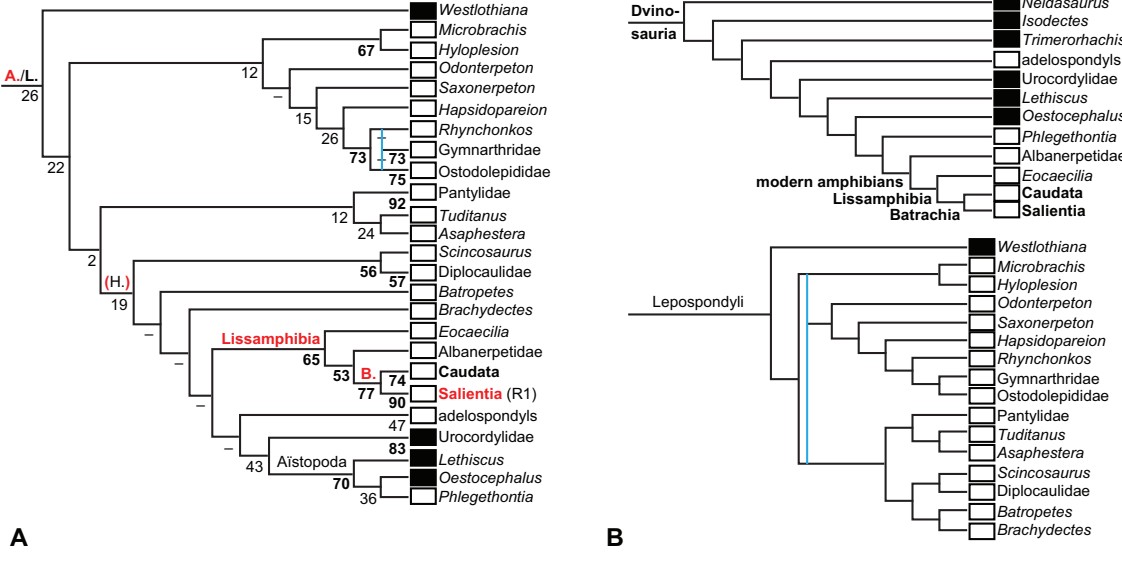

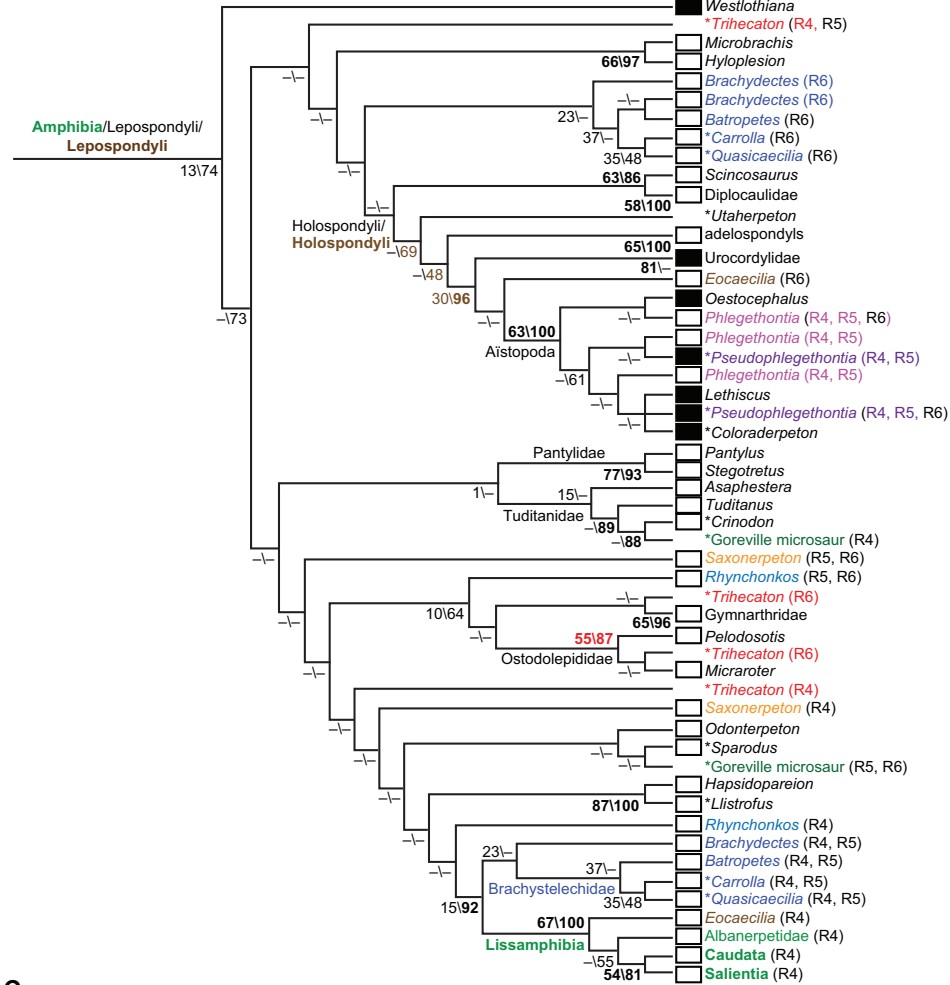

**Figure 29 The relationships of "lepospondyls" in our analyses, and the distribution of the supratemporal bone.** Various named clades are collapsed. Filled rectangles indicate presence of the supratemporal (state 2 or lower of ch. 32—PAR 2/POSFRO 3/INTEMP 1/SUTEMP 1; see App.

S1), empty rectangles its absence (state 3 or 4); the absence of a rectangle indicates missing data. A., Amphibia; B., Batrachia; H., Holospondyli; L., Lepospondyli. (A) Topology found in Analyses R1 and R3. Numbers below internodes are BPO; compare unsupported nodes to Fig. 18. (B) Topology found in R2. (C) Topology found in R4–R6. Numbers are BPE\PP; compare unsupported nodes to Figs. 19 and 20. Numbers in red or brown depend on *Trihecaton* or *Eocaecilia*, respectively, not being members of the clades in question; the opposite holds for clade names in color.

13-14(0), our ch. 265) than in any other "lepospondyls" that are known to have intercentra. We furthermore wonder if the single-piece centra of adelogyrinids are pleuro- or intercentra—in the adelogyrinid **Palaeomolgophis*, many centra articulate with two successive neural arches (*Andrews & Carroll, 1991*), so perhaps the adelogyrinids were stereospondylous. (We have scored them as unknown for this question, as we have done with all other OTUs that have single-piece centra throughout the entire column; for details and discussion, see ch. 259—TRU VER 7—in App. S1.)

In any case, the adelospondyls themselves are relatively well supported as a clade (BPO = 47; BPE = 40; PP = **100**), as is Adelogyrinidae to the exclusion of *Acherontiscus* (BPO = **76**; BPE = **75**; PP = **95**). These results agree with those of RC07 (Fig. 1).

Other than *Westlothiana* and (exclusively in B2) the adelospondyls and *Utaherpeton*, all "lepospondyls" form a clade—with or without some or all modern amphibians, depending on our constraints—in all of our analyses except R2 (BPO = 22; BPE = 1; PP = 73; Fig. 29B). The relationships within this clade remain unclear. A basic pattern that emerges from our results, however (Fig. 29), is a basal (or, in some MPTs from R2, nearly basal) dichotomy between a clade that contains most "microsaurs" and another that contains Holospondyli. This is shared with RC07 (Fig. 28D), *Anderson et al. (2008a)* and the matrices derived from the latter's (*Marjanović & Laurin, 2009*; *Maddin, Jenkins & Anderson, 2012*; *Huttenlocker et al., 2013*; *Pardo, Small & Huttenlocker, 2017*: fig. S6; Fig. 28F); only Analysis EB has Holospondyli deeply nested in a long "microsaur" grade (PP = 40). Analyses R1, R3 and sometimes R2 (BPO = 23; BPE = 15) find *Microbrachis* and *Hyloplesion* on the "microsaur" side like RC07 (Figs. 28D, 29A and 29B); the remaining MPTs from R2 find them outside the dichotomy (Fig. 29B); R4–R6 place them on the holospondyl side (Fig. 29C). RC07 found the tuditanids on the "microsaur" side but the pantylids on the holospondyl side; we find both on the holospondyl side (or diplocaulid-scincosaurid side) in R1–R3 (BPO = 2), but, like the Anderson-lab matrices, both on the "microsaur" side in R4–R6 (BPE = 1). RC07 and the Anderson-lab matrices recovered *Brachydectes* on the holospondyl side but *Batropetes* on the "microsaur" side; we find both on the holospondyl side (or diplocaulid-scincosaurid side) in R1–R3 and R6 (BPO = 25; BPE = 6, PP = **76**), but both on the "microsaur" side in R4 and R5. This latter topology is reminiscent of the one found by *Vallin & Laurin (2004)*, where Aïstopoda, Adelogyrinidae and Nectridea—each a single composite OTU—clustered at the "lepospondyl" base, from which a long "microsaurian" stem then separated a Lysorophia–Lissamphibia clade (Fig. 28A).

The alternative to the basal dichotomy, the long "microsaur" grade of Analysis EB, may be considered suspicious: the "microsaurs" are phenetically similar to the

amniotes, and remain close to them (except for the brachystelechids), while the other "lepospondyl" groups cluster as many internodes away from Amniota as possible (BPO = 19; against a BPE of 7; PP = **76**).

Even so, the limbless aïstopods and the adelospondyls, for which extremities are unknown and among which *Adelogyrinus* is well enough preserved that we have scored it as limbless (state HUM 18/DIG 1(0); for Discussion see ch. 219 in App. S1 as well as *Andrews & Carroll, 1991*: 252), never form an exclusive clade in our analyses (Fig. 29)—even though they did in *Germain (2008a*: fig. 5.15; simplified in *Marjanović & Laurin, 2013*: fig. 4C).

As in RC07, "nectridean" monophyly is never found; instead (Fig. 29) there is a strong attraction between the urocordylid "nectrideans" and the aïstopods, which form an exclusive clade in all analyses (BPO = 43; BPE = 30; PP = **96**). Except in Analysis R2 (Fig. 29B), however, all "nectrideans", aïstopods, adelospondyls (except in B2) and the "microsaur" *Utaherpeton* (except in B2) consistently form either an exclusive clade (Analyses R4 and R5) or a grade toward *Brachydectes*, brachystelechid "microsaurs" and/or modern amphibians (Analyses R1, R3, R6 and EB; BPO = 25; BPE = 15; PP = 57; Figs. 29A and 29C). Arguably, R2 is an example of the second pattern, because there, too, Diplocaulidae and *Scincosaurus* are closer to *Brachydectes* and the brachystelechid *Batropetes* than to Urocordylidae (Fig. 29B). In support of the second pattern, five extra steps are needed in Mesquite under the conditions of R1 to create the first pattern, that is, to make (*Scincosaurus* + Diplocaulidae) the sister-group of (adelospondyls (Urocordylidae, Aïstopoda)); one more is required to make the latter clade the sister-group of Diplocaulidae alone as in RC07.

Consistently, and unlike in RC07, Diplocaulidae and *Scincosaurus* are sister-groups (BPO = **56**; BPE = **63**; PP = **86**). This agrees with other recent analyses (*Ruta, Coates & Quicke, 2003*; *Pawley, 2006*; *Milner & Ruta, 2009*; *Pardo, Small & Huttenlocker, 2017*: fig. S6A but not B—Figs. 28B and 28E). Unlike in all earlier versions of our revised matrix (*Marjanović & Laurin, 2015, 2016*), urocordylid phylogeny is resolved, with the original taxon sample making *Ptyonius* the sister-group of the other two as in RC07 and *Germain (2008a*: fig. 5.15), while adding taxa puts *Sauropleura* in that position; urocordylid monophyly, unsurprisingly, is well supported (BPO = **56**; BPE = **63**; but contradicting a PP of 65). Diplocaulidae (BPO = **57**; BPE = **58**; PP = **100**) contains (*Diplocaulus* + *Diploceraspis*) (BPO = BPE = PP = **100**) and (*Keraterpeton* + *Diceratosaurus*) (BPO = BPE = 43; PP = 68), the latter clade contradicting RC07 and *Germain (2010)*; *Batrachiderpeton* is the sister-group of the former clade (BPO = 46; BPE = 39; PP = **94**). Sampling almost no *stereospondyls, our matrix cannot test the hypothesis by *Pardo (2011, 2014)* and *Pardo et al. (2018)* that *Diplocaulus*, *Diploceraspis* and **Ductilodon*—but not the "keraterpetids"—are **brachyopoid temnospondyls; we await full publication of that work (see also *Marjanović & Laurin, 2013*), which will contain a redescription of *Diploceraspis* and **Ductilodon*.

Even though we updated the scores of *Lethiscus* based on *Pardo et al. (2017)*, who for the first time found the aïstopods as whatcheeriid-grade animals, and even though we added *Coloraderpeton* mostly from the same source, we recover Aïstopoda nested as
deeply among the "lepospondyls" as in all previous analyses. Attempts to move the aïstopods, the urocordylids, *Utaherpeton* and the adelospondyls next to the colosteids in Mesquite appear to require 10 additional steps on a tree from Analysis R1 and 22 on one from R4, contradicting a BPO of 35, a BPE of 20 and a PP of **86**; placing Aïstopoda + Urocordylidae next to *Ossinodus* (crownward of Whatcheeriidae and optionally of *Tulerpeton*) while leaving the adelospondyls with the colosteids takes a total of 25 extra steps in R1 as far as we can determine in Mesquite, against a BPO of **67** and a PP of **86**; moving all these "lepospondyls" next to *Ossinodus* (one node crownward of *Densignathus*, two of Whatcheeriidae) takes an impressive total of 32 extra steps in R4, despite contradicting a BPE of only 22. Trees illustrating these topologies are contained in Data S3.

This could, on the one hand, be due to the fact that our matrix almost completely lacks braincase characters, which *Pardo et al. (2017)* sampled more densely than ever before. Most conspicuously, we were unable to score the remarkable persistence of the buccohypophyseal canal in (at least) *Lethiscus* and *Coloraderpeton*, a plesiomorphy not otherwise reported from anywhere crownward of **Tiktaalik*. We have, however, scored such features as the exposure of the ventral cranial fissure/suture caudal to the parasphenoid (at least in the midline) in *Lethiscus*, *Coloraderpeton* and apparently *Pseudophlegethontia* (*Anderson, 2003b*: fig. 2A), the lack of a lingual lamina on the angular in all three, the spiracular notches and palatal dentition of the first two, or the mostly or fully enclosed mandibular lateral-line canal and the preopercular bone of *Coloraderpeton* (see ch. 142, 161, 69, 103, 104, 107, 108, 110, 118, 101, 81—PASPHE 9, ANG 2-3, SQU 3, VOM 3, VOM 4, VOM 8, VOM 9, PAL 1, ECT 5, SC 2, PREOPE 1—in App. S1). Although we suspect that the supposed parasymphysial of *Lethiscus* and *Coloraderpeton*—a very large, massive, toothless bone—is actually a mentomandibular ossification of Meckel's cartilage, we are currently unable to test this idea and have scored the bone as the parasymphysial, which has not otherwise been reported from anywhere crownward of the anthracosaurs (teal in Fig. 24).

Conversely, however, postcranial characters are underrepresented in the matrix by *Pardo et al. (2017)* even compared to ours. Being deliberately restricted to taxa with good braincase data (*Pardo et al., 2017*: supplementary information part A), their taxon sample furthermore lacks any "lepospondyls" other than *Lethiscus*, *Coloraderpeton* and a few "recumbirostrans" (on which see below).

On the other hand, the quality of our data for *Oestocephalus* could have misled our analyses. Although *Carroll (1998a)* expressed strong doubts about "lepospondyl" affinities for aïstopods and a preference for much more rootward placements not unlike the ones eventually found by *Pardo et al. (2017)*, he described—and illustrated in his specimen drawings—several apomorphies of *Oestocephalus* that are unexpected for such a rootward position, for example a large lingual lamina on the angular. To test these anatomical interpretations, it will be necessary to restudy the specimens. *Pardo et al. (2017)* did not include *Oestocephalus* in their matrix; *Anderson, Pardo & Holmes (2018)* reported a large, well preserved skull which will add valuable information once fully described.
It will be interesting to see in future analyses what the braincase characters used by *Pardo et al. (2017)* will do to this area of the tree once they are added to our matrix. In particular, will the urocordylids follow the aïstopods rootward (*Pardo et al., 2018*), in keeping with such plesiomorphies (not coded in our matrix) as the prearticular denticle row or the huge distal Meckelian fenestra of *Sauropleura* (*Bossy, 1976*; *Bossy & Milner, 1998*), or will their vertebral similarities to the other "nectrideans" keep them in a rest-"lepospondyl" clade, or will all "nectrideans" come with them in spite of the similarities between the brachystelechid "microsaurs", the lysorophians, the modern amphibians and *Scincosaurus* in particular?

Plesiomorphies unexpected for "lepospondyls" are not limited to the head skeleton in aïstopods. *Wellstead (1982*: 204, fig. 8A*)* reported a cololite in *Lethiscus*, commenting only that "[i]ts segmented appearance is likely due to a spiral valve in the intestine". The spiral valve is plesiomorphic for gnathostomes, found today in chondrichthyans and lungfish; in extant tetrapods, however, it is absent, and cololites suggest its absence in temnospondyls as well (*Godfrey, 2003*: fig. 2). Independently of *Pardo et al. (2017)*, this finding—if correct—implies a phylogenetic position for Aïstopoda rootward of Temnospondyli.

Aïstopoda is perhaps of particular interest because some of its members were terrestrial beyond reasonable doubt, even though *Coloraderpeton must have been obligatorily aquatic on account of its mandibular lateral-line canal (*Pardo et al., 2017*): *Rößler et al. (2012)* reported two indeterminate aïstopods from Chemnitz (eastern Germany), where a forest was covered by volcanic ash around the Artinskian–Sakmarian boundary in the Cisuralian (see also *Spindler et al., 2018*: 319). The forest floor is preserved in situ; no water was involved in the deposition. This confirms the taphonomic and morphological arguments reviewed by *Anderson (2002)* and *Germain (2008a*: chapters II, III; *2008b)*, to which we add the absence of lateral and ventral keels on the centra, which are found (state TRU VER 15(1)—ch. 266 in App. S1) in all other potential anguilliform swimmers in this matrix. Considering the age of *Lethiscus*, which lies at the very end of Romer's Gap, it is conceivable that the "first step on land" was done "without limbs" (*Germain, 2008a*: chapter III)—and that Aïstopoda contains a separate origin of a terrestrial lifestyle.

Following our corrections to *Lethiscus* after *Pardo et al. (2017)*, aïstopod monophyly is well supported (BPO = **70**; BPE = **63**; PP = **100**), yet not found in any trees from Analysis R2, where the aïstopods form a grade toward the modern amphibians. The phylogeny of Aïstopoda is unchanged from RC07 given the original taxon sample, but the resolution largely breaks down when taxa are added (Fig. 29).

In addition to finding the aïstopods to be whatcheeriid-grade animals, *Pardo et al. (2017)* found all other "lepospondyls" they sampled—*Brachydectes* and a selection of "microsaurs"—to form a clade within Sauropsida. Although this was supported (*Pardo et al., 2017*: extended data fig. 7b) by bootstrap percentages of 71 for Sauropsida including the animals in question ("Recumbirostra"; see below), 95 for Amniota and 96 for Amniota + *Limnoscelis* (the only sampled diadectomorph), we wonder if this result is an artefact of the taxon and character samples (or of misscores). For instance, the hyobranchial apparatus of *Brachydectes* or *Pantylus* would imply several reversals from the ancestral amniote condition or massive homoplasy among all other amniotes; similarly,

many of the sampled "microsaurs" retain well-developed dorsal scales (e.g. CG78; Figs. 3 and 4), but not one traditional amniote does. A much larger matrix will be required to test this question. It is puzzling, too, that *Pardo et al. (2017)* derived a large part of their matrix from those of *Maddin, Jenkins & Anderson (2012)* and *Huttenlocker et al. (2013)*, which are in turn derived from that of *Anderson et al. (2008a)*, but did not implement or mention any of the changes to the latter that were proposed (or repeated from *Marjanović & Laurin, 2009*) by *Sigurdsen & Green (2011)*, even though they cited *Sigurdsen & Green (2011)* as the source of a new character.

### The interrelationships of the "microsaurs"

> The Tuditanomorpha appears to be a natural assemblage [. . . ]. Among the Microbrachomorpha, the four genera are so distinct from one another that each has been placed in a monotypic family. It is possible that this is not a natural assemblage, but these forms share more features in common with each other than any do with other groups of Paleozoic tetrapods.
>
> – CG78: 11–12

> Although there is considerable question concerning the nature and degree of interrelationship within the Tuditanomorpha, the included families share a great many features. [. . . ] The remaining microsaurs, in contrast, are a more varied group, and may or may not have a significant common heritage. [. . . ] They are here classified as a natural group, but this may not prove to be the case.
>
> – CG78: 113

It is not an overstatement to say that "microsaur" phylogeny is a mess. No two analyses, usually even if based on successive versions of the same matrix, have recovered the same topology or nearly so (Fig. 28). Ours are no exception (Fig. 29).

Microsaur monophyly does not occur in any of our trees. A clade with a membership similar to that of Tuditanomorpha *Carroll & Gaskill, 1978*, is not found in any of our analyses either, although Analysis R6 comes close by finding a clade composed of all "tuditanomorphs" and the "microbrachomorph" *Odonterpeton*. Analyses B1, B2 and EB feature an (*Odonterpeton* (*Microbrachis*, *Hyloplesion*)) clade reminiscent of Microbrachomorpha *Carroll & Gaskill, 1978* (BPO = 23, BPE = 15 and PP = 35 for the whole), and under the conditions of R1 three extra steps suffice in Mesquite to create such a clade; but even so, this clade (which was also found by RC07) is not close to the sampled brachystelechids (*Batropetes*, \**Carrolla* and \**Quasicaecilia*, the first two observed by D. M.), of which *Batropetes* was cautiously included in Microbrachomorpha by CG78 (the others were not yet known).

That said, the discrepancies between the reconstructions by CG78, the specimen drawings by CG78, and D. Marjanović's personal observations of the holotype of *Odonterpeton* are such that *Odonterpeton* will have to be redescribed. For example, the supposed suture between the left parietal and the supposed postparietal is a series of unconnected cracks; *Odonterpeton* consequently has no identifiable postparietal at all (state POSPAR 1-2(2); see ch. 39 in App. S1 for details), and symmetry is restored by an additional

curve to the left at the caudal end of the suture between the parietals that is omitted from the drawings by CG78 (presumably because it is not visible in strict dorsal view).

Following the redescription of *Microbrachis* and *Hyloplesion* by Olori (2015), these two OTUs are found as sister-groups throughout all of our analyses, as they were by RC07; the support for this clade is unusually high (BPO = **67**; BPE = **66**; PP = **97**).

Anderson (2007b: 205–206) coined the name Recumbirostra and carefully defined it as applying to "the clade descended from the most recent common ancestor of *Pantylus*, *Cardiocephalus sternbergi*, *Rhynchonkos*, and *Micraroter*, but not including *Tuditanus* or *Microbrachis*". This definition, and therefore this name, cannot be applied to the trees from any of our analyses: while *Rhynchonkos*, Gymnarthridae (incl. *Cardiocephalus*) and Ostodolepididae (incl. *Micraroter*) form an exclusive clade of elongate "microsaurs" in all analyses except R4 (variously joined by *Trihecaton* in R6 and EB; BPO for clade = 22, BPE = 26, PP = 64; Fig. 29), the pantylids (*Pantylus* and *Stegotretus*; see below for *Sparodus*) are never closer to it than *Tuditanus* and/or *Microbrachis* are (BPO = 12; BPE = 1; PP = 40)—indeed, as described below, Pantylidae and Tuditanidae always fall out as sister-groups (Fig. 29) except in EB.

The original taxon sample contained a single brachystelechid, *Batropetes*. In Analyses R1 and R3, it is the sister-group of a clade consisting of *Brachydectes* (the only included lysorophian), the adelospondyls, Urocordylidae, Aïstopoda and all available modern amphibians (Fig. 29A). Analysis B1, however, recovers *Batropetes* and *Brachydectes* as sister-groups (BPO = 37). These two arrangements cannot be distinguished in R2, where the entire clade that forms the sister-group of *Brachydectes* in R1 lies within the temnospondyls. When taxa are added, the three brachystelechids (*Batropetes*, *Carrolla*, *Quasicaecilia*) form an exclusive clade (BPE = 37) in R4, R5 and some MPTs from R6 (Fig. 29C). In the remaining MPTs from R6, they form a grade toward *Brachydectes*; in EB, they form a grade toward the modern amphibians instead, with *Brachydectes* on the outside (PP = **88**). Only in R2, R4, R5 and perhaps R6 can the brachystelechids even be considered part of the "microsaur" grade.

Pawley (2006: 239) reported to have found the "microsaurs" *Tuditanus* and *Asaphestera* (both classified in Tuditanidae by CG78) to score identically except for missing data. They differ in our matrix. Indeed, the added tuditanid *Crinodon* (see below) has more in common with *Tuditanus* than *Asaphestera* does (Fig. 29C), despite being easy to distinguish from both. Although *Asaphestera* is found as a tuditanid in all parsimony analyses (see below), Analysis EB finds it one node crownward of Tuditanidae (PP = 36).

As in RC07, though unlike in the Anderson-lab matrices (Fig. 28F), *Hapsidopareion* (together with *Llistrofus*, see below) is never the sister-group of the supposed (CG78) hapsidopareiid *Saxonerpeton* (kept apart by BPO = 26), except in Analysis B2 (BPE = 10) and EB (PP = 35), although they are only one internode apart in R1 and R2. In all parsimony analyses, Tuditanidae (*Tuditanus*, *Asaphestera*; in R4–R6 also *Crinodon*; in R4 and B2 further the *Goreville microsaur) and Pantylidae (*Pantylus*, *Stegotretus*; in B2 also *Sparodus*) consistently form a clade, though it is less well supported (BPO = 12; BPE = 1; against a PP of 40) than either Tuditanidae (BPO = 24; BPE = 15; PP = **89**

excluding *Asaphestera*) or Pantylidae (BPO = 92; BPE = 36, BPE = **77** without *Sparodus*; PP = **88**, PP = **93** without *Sparodus*) on their own. In EB, Tuditanidae forms the sister-group to all other "lepospondyls" (which, uniquely, include *Asaphestera*: PP = 36) except *Westlothiana* (PP = 73).

### Added "microsaurs"

*Carrolla*, *Quasicaecilia* and *Crinodon* have been treated above (see also the next section for the first two).

*Utaherpeton* was described as one of the oldest "microsaurs"; the original description (*Carroll, Bybee & Tidwell, 1991*) already noted similarities to the "nectrideans". Our results (Fig. 29C), much like those of *Marjanović & Laurin (2009*; Fig. 28F) but unlike those of *Vallin & Laurin (2004*: fig. 28F and fig. 35A) or *Pardo, Small & Huttenlocker (2017*: fig. S6), place *Utaherpeton* inside the holospondyl clade, next to the clade composed of the adelospondyls, urocordylids, aïstopods and, in R6, *Eocaecilia* (PP = 69; against a BPE of 10). We deduce that, rather than being just another "microbrachomorph" "microsaur", *Utaherpeton* could occupy a crucial position close to the origin of Holospondyli and deserves future attention. The corresponding bootstrap tree, however, interestingly makes it the sister-group of *Westlothiana*, a position that agrees vaguely with the original description but has negligible support (BPE = 10; against a PP of **76**).

The *"Goreville microsaur" (*Lombard & Bolt, 1999*) either falls out as a tuditanid next to *Crinodon* (R4; PP = **88**; Fig. 29C) or nests with *Sparodus* (R5, R6). The bootstrap trees place it in Tuditanidae, with very low support (BPE = 15), although Bayesian support is considerable (PP = **89**).

*Sparodus* was considered a gymnarthrid by CG78 and *Carroll (1988)*. In our analyses such a relationship is never found. In R4–R6, *Sparodus* forms a clade with *Odonterpeton* and (in R5 and R6) the *"Goreville microsaur"; the same cautionary notes about *Odonterpeton* as above apply. *Pantylus* and *Stegotretus* nest next to *Sparodus* in Analyses B2 and EB (BPE = 36; PP = **88**). A pantylid position would not be surprising considering the enormous palatine and especially coronoid tusks (*Carroll, 1988*; D. Marjanović, pers. obs. of the same specimen—Figs. 3 and 4).

*Trihecaton* has hitherto been neglected since CG78 because its skull is almost entirely unknown. The rest of the skeleton including the lower jaw, however, is mostly preserved and articulated. Because the postcranium is so close to complete and lower-jaw characters are well represented in this matrix, D. M. has scored *Trihecaton* directly from the specimens (which probably belong to the same individual). Analyses R4 and R5 resolve its position next to (Holospondyli (*Microbrachis*, *Hyloplesion*)) or, in R4, one node rootward of *Saxonerpeton*; R6 places it next to *Micraroter* within Ostodolepididae or next to Gymnarthridae in different MPTs (Fig. 29C). Analyses B2 and EB place it next to the Gymnarthridae-*Rhynchonkos*-Ostodolepididae clade (BPE = 10; PP = 64). Evidently, further taxa and further postcranial characters should be added to future analyses to resolve the potentially interesting position of this animal.

*Llistrofus* is always the sister-group of *Hapsidopareion* (BPE = **87**; PP = **100**), as expected (*Bolt & Rieppel, 2009*).

### Lissamphibian origins

As summarized in Tables 1, 3 and 4, our revision of the matrix of RC07 supports the lepospondyl hypothesis over the temnospondyl hypothesis, which in turn is more parsimonious than the polyphyly hypothesis. Adding OTUs, including *Gerobatrachus*, to the matrix increases the difference between the LH and the TH by one step and the difference between the TH and the PH by two steps, still favoring the LH over its alternatives, contrary to the results of RC07 (Fig. 1), *Sigurdsen & Green (2011)*, *Maddin, Jenkins & Anderson (2012)*, *Pardo, Small & Huttenlocker (2017*: fig. S6*)* and *Pardo et al. (2017)*; the LH further has a posterior probability of **92** (Fig. 20).

As discussed above (Materials and Methods: Robustness analyses), a reliable statistical test for whether the differences between these trees are distinguishable from random is not available. The null hypothesis (that they are not) cannot currently be rejected, except perhaps for the difference between the LH and the PH, and that only under the original taxon sample (Table 4).

Monophyly of the modern amphibians is found in all analyses (except of course those constrained against it, see Table 1: R3, R6) and is well supported: BPO = **65**; BPE = **67**; PP = **100**. Note that adding taxa does not decrease the support.

In Analysis R1 (Figs. 10, 11, 29A and 30L), the "lepospondyls" closest to Lissamphibia are a clade of adelospondyls, urocordylids and aïstopods, a novel and unexpected result. Next closest is *Brachydectes*, the only sampled lysorophian. A sister-group relationship between lissamphibians and lysorophians, not found here, has long been considered an integral part of the lepospondyl hypothesis (though see *Marjanović & Laurin, 2013*, for discussion) and would have been consistent with the results of *Vallin & Laurin (2004)*, *Pawley (2006*: fig. 91*)* and *Marjanović & Laurin (2008, 2009*: supplementary figure*)*, as shown in Figs. 28, 30A, 30C and 30E. On the outside follows *Batropetes*, the only brachystelechid "microsaur" in the original taxon sample; this is congruent with the results of *Vallin & Laurin (2004)*. Three extra steps are needed in Mesquite to render *Batropetes* and *Brachydectes* each other's sister-group, one more is required to move *Batropetes* closer to Lissamphibia than *Brachydectes*.

However, when taxa are added to the parsimony analysis (R4; Figs. 14, 29C and 30M), *Brachydectes* is the sister-group of Brachystelechidae, which includes *Batropetes* (BPO = 37; BPE = 23), both together forming the sister-group of Lissamphibia (BPO = 20; BPE = 14). Analysis EB uniquely finds the modern amphibians nested inside the brachystelechids, closest to *Quasicaecilia* + *Carrolla* (PP = **88**), followed by *Batropetes* (PP = 45) and then by *Brachydectes* (PP = **92**).

The sister-group of the smallest clade formed by all modern amphibians, *Brachydectes* and the brachystelechids is *Scincosaurus* + Diplocaulidae in Analyses R1 and EB (PP = 57). R4 instead places *Rhynchonkos* in this position (contradicting a BPE of 26 and a PP of **76**); it has historically played a large role in the PH (*Carroll & Holmes, 1980*; *Carroll, 2007*) and was recovered in the same position by *Vallin & Laurin (2004*;

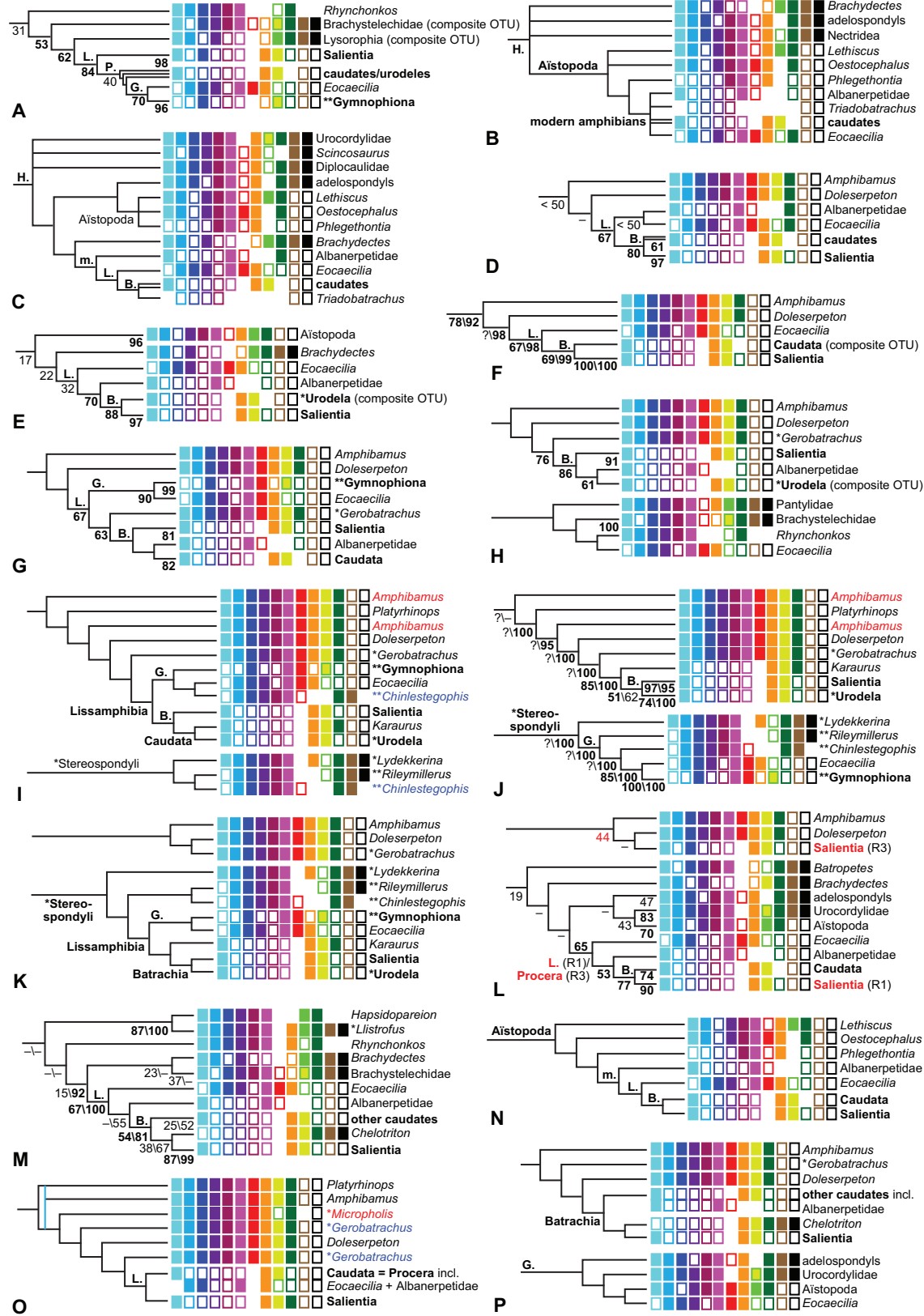

**Figure 30 The phylogeny of modern amphibians and their closest relatives in recently published (A–K) and our (L–P) analyses, and the distribution of several character states.** Albanerpetidae is a composite OTU throughout. Numbers in (A), (D), (E), (G), (H) are bootstrap

percentages. B., Batrachia; G., Gymnophionomorpha (not shown when *Eocaecilia* is the only included member); L., Lissamphibia; m., modern amphibians (when not synonymous with L.); P., Procera. (A) *Vallin & Laurin (2004)*; the two unnumbered "caudate" nodes have bootstrap values below 30%. (B) *Pawley (2006*: fig. 88; see Fig. 28B*)*. (C) *Pawley (2006*: fig. 91; see Fig. 28C*)*. (D) RC07. We write "< 50" where RC07 (app. 4) wrote "no bootstrap support in a 50% majority-rule consensus" and "–" where they wrote "no boots[t]rap support compatible with a 50% majority-rule consensus". Caudate monophyly was found in some MPTs and, as shown here, 61% of the bootstrap replicates. (E) *Marjanović & Laurin (2009)*. (F) *Sigurdsen & Green (2011*: fig. 2B, 4*)*. Numbers are bootstrap percentages from unweighted parsimony (left; not published if below 50) and Bayesian posterior probabilities (right; in %). (G) *Maddin, Jenkins & Anderson (2012)*. Bootstrap percentages below 50 were not published. (H) *Huttenlocker et al. (2013)*. Bootstrap percentages below 50 were not published. (I–K) MPTs from *Pardo, Small & Huttenlocker (2017)*; see *Marjanović & Laurin (2018)*. Numbers in (J) are bootstrap percentages (left; not published if below 50) and Bayesian posterior probabilities (right; in %). The ribs of **Rileymillerus* are scored under the assumption that the tentatively referred postcrania really belong to a juvenile of the same taxon as the holotype, which is a skull (*Bolt & Chatterjee, 2000*); this is at least not contradicted by the postcrania of **Chinlestegophis* (*Pardo, Small & Huttenlocker, 2017*). (L) R1 (unconstrained), R3 (constrained for the PH; both original taxon sample). Numbers are BPO given that Salientia has the position it has in R1. (M) R4 (unconstrained, expanded taxon sample). Numbers are BPE\PP. (N) R2 (constrained against the LH, original taxon sample). (O) R5 (constrained against the LH, expanded taxon sample); *Micropholis* also has two positions outside the clade shown here. (P) R6 (constrained for the PH, expanded taxon sample). Except where mentioned otherwise below, the rectangles show state 0 (filled) or 1 (empty) of characters 18 (LAC 1—cyan), 39 (POSPAR 1–2—teal), 45 (POSFRO 1—blue), 53 (TAB 1/SQU 4—violet), 60 (POSORB 1—reddish purple), 71 (JUG 1—magenta), 72 (JUG 2–6—red), 77 (QUAJUG 1—orange), 102 (VOM 1–13—bright green and greenish yellow, see below), 160 (ANG 1—dark green), 246 (RIB 3—brown) and 250 (RIB 7—black). Absence of a rectangle means missing data, inapplicability or ambiguous optimization at the root of a collapsed clade, except that *Lydekkerina* is polymorphic for character 72. State 1 of the unordered character 39 does not occur in this figure, so empty rectangles show state 2. State 2 of the ordered character 72 is limited to Pantylidae in this figure and therefore shown as state 1. The ordered character 102 has three states: vomer about as wide as long or wider (0, filled greenish yellow rectangles); intermediate (1, empty bright green rectangles); at least 2½ times as long as wide (2, filled bright green rectangles). Rectangles with a bright green rim filled in with greenish yellow indicate ambiguous optimization of state 1 or 2. Character 18: presence (0) or absence (1) of lacrimal; 39: presence (0) or absence (2) of postparietals; 45: presence (0) or absence (1) of postfrontal; 53: presence (0) or absence (1) of tabular; 60: presence (0) or absence (1) of postorbital; 71: presence (0) or absence (1) of jugal; 72: contact between maxilla and quadratojugal at ventrolateral edge of skull (0), jugal separates maxilla and quadratojugal and forms part of ventrolateral edge of skull (1 and 2); 77: presence (0) or absence (1) of quadratojugal; 102: see above; 160: presence (0) or absence (1) of angular; 246: ribs mostly straight (0, empty rectangles) or ventrally curved in at least part of the trunk (1, filled rectangles); 250: trunk ribs longer (0) or shorter (1) than three successive articulated vertebrae in adults. Note that we do not distinguish absence due to loss of an ossification center from absence as a separate bone due to ontogenetic fusion; in more or less all cases the ontogeny is insufficiently well known to decide. For further derivations within Caudata and Salientia, see the text. *Adelospondylus* has state 60(1), *Notobatrachus* is polymorphic for 77. The optimizations for character 72 differ for Aïstopoda between (P) and the rest due to the larger taxon sample in (P); likewise, the optimizations for character 160 vary for Caudata/Urodela depending on taxon samples.

Fig. 30A*)*. B1 and B2 place Holospondyli as a whole next to (Lissamphibia (*Batropetes*/ ***Brachystelechidae***, *Brachydectes*)) with very weak support (BPO = 25; BPE = 6).

When the TH is enforced on the original taxon sample through a constraint (R2; Figs. 12, 29B and 30N), the lissamphibians do not nest with *Doleserpeton* as expected, or even within Dissorophoidea at all. Instead, they nest within the dvinosaurs—and take the aïstopods (which become paraphyletic with respect to Lissamphibia), the urocordylids and the adelospondyls with them. While it seems remarkable that such a surprising novel topology is more parsimonious than any conventional version of the TH, the modest magnitude of this difference—to move all lissamphibians next to *Doleserpeton* (and restore lepospondyl monophyly) requires three extra steps in Mesquite, as many as to enforce the PH (Analysis R3; Fig. 30L)—and the lack of statistical significance for the difference to Analysis R1 suggest that we should ascribe this result to insufficiencies in the matrix. Indeed, adding taxa (R5; Figs. 15 and 30O) "rectifies" the constrained topology by placing Lissamphibia among the amphibamid dissorophoids, far from the dvinosaurs and without turning any "lepo-" into temnospondyls; however, both *Doleserpeton* and *Gerobatrachus* are found as sister-groups of Lissamphibia

with equal parsimony, failing to resolve the disagreement between *Sigurdsen & Bolt (2010)*, who favored *Doleserpeton* in this role, and *Anderson et al. (2008a)*, *Maddin & Anderson (2012)*, *Maddin, Jenkins & Anderson (2012*—Fig. 30G), *Pardo, Small & Huttenlocker (2017*—Figs. 30I–K*)* and *Pardo et al. (2017)*, who found \*Gerobatrachus to lie closer to (or even within) Lissamphibia.

*Pardo, Small & Huttenlocker (2017*: fig. S6*)* added \*\*Chinlestegophis and \*\*Rileymillerus to a matrix slightly modified from that of *Maddin, Jenkins & Anderson (2012)*, which contains almost only dissorophoids and "lepospondyls", and found them as temnospondyls—though not as lissamphibians, despite finding the TH. Satisfied that \*\*Chinlestegophis, \*\*Rileymillerus and the lissamphibians were temnospondyls, *Pardo, Small & Huttenlocker (2017)* then added them to a matrix that contained temnospondyls and modern amphibians but no "lepospondyls"; that matrix is based on *Schoch (2013)*, accepting (J. Pardo, pers. comm. 2017; D. Marjanović, pers. obs. 2017), though not citing, most of the score changes proposed by *Dilkes (2015a)* but omitting the characters Dilkes had added and greatly changing the taxon sample. Bayesian analysis and some of the MPTs resulting from this second matrix found the caecilians, \*\*Chinlestegophis and \*\*Rileymillerus as stereospondyls while the batrachians remained dissorophoids (*Pardo, Small & Huttenlocker, 2017*: fig. 2C, S7). However, lissamphibian monophyly—within Stereospondyli or Dissorophoidea—is equally parsimonious (*Marjanović & Laurin, 2018*; Figs. 30I–30K). While \*\*Chinlestegophis and \*\*Rileymillerus are clearly stereospondyls, we therefore—and for a number of anatomical reasons—remain uncertain for the time being whether the caecilians are closely related to them. However, it is clear that a very large matrix—sampling stereospondyls, "lepospondyls" and many characters—will be necessary to adequately test this question.

Such a matrix will also need to be carefully scrutinized for accuracy before analysis because accidental misscores are, as we show here, common and because small changes routinely have disproportionate and unpredictable impacts on the results. In this context, we note that even though the matrices of *Maddin & Anderson (2012)*, *Maddin, Jenkins & Anderson (2012)*, *Huttenlocker et al. (2013)* and *Pardo, Small & Huttenlocker (2017*: fig. S6*)* are largely derived from that of *Anderson et al. (2008a)*, and that of *Pardo et al. (2017)* is by half, none of these publications addressed or mentioned the changes to that matrix proposed by *Marjanović & Laurin (2009)* and *Sigurdsen & Green (2011)*, despite their impact on the results in the latter publications. Indeed, the matrix and character list of *Pardo, Small & Huttenlocker (2017*: fig. S6*)* remain altogether unpublished.

Other than new braincase characters and one character from *Maddin & Anderson (2012)*, the matrix of *Pardo et al. (2017)* was composed from those of *Clack et al. (2012a)* and *Huttenlocker et al. (2013)*. It is thus not much better equipped to test lissamphibian origins than the latter.

The albanerpetid \*\*Shirerpeton was published too recently for us to score (*Matsumoto & Evans, 2018*). It reinforces our impression that Albanerpetidae should not be omitted from phylogenetic analyses of lissamphibian origins (as done by *Pardo, Small & Huttenlocker, 2017*: fig. 2, S7), but instead has potentially crucial new information to offer. For example, a suproccipital bone had never been documented in a modern

amphibian, its presence was scored as unknown for Albanerpetidae in RC07, and we have kept this score (despite the suggestive median caudodorsal thickening in the fused braincase of *Albanerpeton*—*Maddin et al., 2013a*); **Shirerpeton* unambiguously has a suproccipital, which, as *Matsumoto & Evans (2018)* noted, strongly resembles that of *Brachydectes* in its shape and spatial relationships. The suproccipital, or at least its exposure on the skull surface, is absent in all temnospondyls that are well enough known, including *Doleserpeton*; among "microsaurs", it is absent in Gymnarthridae and not exposed in *Stegotretus*, but present and exposed elsewhere, including in *Batropetes*.

### Lissamphibian phylogeny

All salientians form a clade (BPO = **90**; BPE = **87**; PP = **99**) in all analyses. The salientian *Liaobatrachus* is consistently found as the sister-group of *Notobatrachus* + *Vieraella* (BPE = **82**; PP = **100**). A clade formed by the latter two (BPO = **96**) to the exclusion of more crownward salientians (like *Liaobatrachus*: BPE = **68**; PP = 58) has not been found elsewhere (*Dong et al., 2013*); this may be due to the character sample, though perhaps the taxon sample with its many but distant outgroups also plays a role.

Contrary to RC07, or to *Pardo, Small & Huttenlocker (2017*: fig. 2C, S6A, S7; Figs. 30J and 30K)*, the two caudates *Karaurus* and *Valdotriton* form a clade in all MPTs derived from the original taxon sample (BPO = **74**). Seemingly bizarre things happen when taxa are added, including the further caudates *Beiyanerpeton*, *Pangerpeton* and *Chelotriton*, but this is not surprising: because only two caudates were originally included, the matrix does not contain any characters that are intended to resolve caudate phylogeny.

The extremely peramorphic salamandrid *Chelotriton* (*Marjanović & Witzmann, 2015*) is the most salientian- and temnospondyl-like caudate, falling out as the sister-group of Salientia in Analyses R4, R6 and EB (BPE = 38; PP = 67; Figs. 30M and 30P), and as the sister-group of a clade that contains all other non-salientian modern amphibians in Analysis R5. Evidently, its peramorphosis pulls it out of Caudata given the present character sample, and indeed toward the base of Lissamphibia or Batrachia in constrained trees where the closest relatives of Lissamphibia or Batrachia are temnospondyls; but this is not enough to pull the modern amphibians into the temnospondyls, or to support the Procera hypothesis (on which see below) unless the TH is enforced (Fig. 30O). We conclude that ontogeny does not as severely "discombobulate" phylogeny (*Wiens, Bonett & Chippindale, 2005*: title) in this case as one could have expected.

The more moderately peramorphic *Pangerpeton* is always found within a caudate clade or grade. In R4–R6 it lies next to a *Valdotriton*-*Karaurus*-*Beiyanerpeton* clade (which has a BPE of 21 and, in R5, is either the sister-group of Albanerpetidae + *Eocaecilia* or contains it as the sister-group of *Valdotriton*); Analysis EB places it next to *Beiyanerpeton* + *Karaurus* (PP = 57).

*Beiyanerpeton* combines paedomorphosis with unexpected plesiomorphies. Possibly due to this balance, it is neither attracted to nor repelled from the caudate root: in R4–R6, it is always found as the sister-group of *Karaurus* (BPE = 46; PP = 46), which is normally

considered a stem-caudate, but found in various nested positions in the caudate clade or grade in our analyses (a result we ascribe to the character sample).

In trees that support the LH (all unconstrained searches: Analyses R1, R4, B1, B2, EB; Figs. 30L and 30M), the Batrachia hypothesis is consistently recovered: Salientia and all caudates form a clade to the exclusion of Albanerpetidae and *Eocaecilia* (BPO = **77**; BPE = **54**; PP = **81**). This agrees with all recent molecular analyses (*Pyron, 2014*: supplementary file amph_shl.tre; *Irisarri et al., 2017*), with *Pardo, Small & Huttenlocker (2017*: fig. 2, S7, but not S6A*)* and, apart from Albanerpetidae, with *Maddin, Jenkins & Anderson (2012)*. Given the original taxon sample, Salientia nests as the sister-group of Caudata (R1); otherwise, it forms the sister-group of the unusually peramorphic and therefore frog-like caudate *Chelotriton (see above) followed by a clade containing all other caudates. The sister-group of Batrachia is Albanerpetidae (R1, R4 and EB; BPO = **53**; PP = 55) or *Eocaecilia* (BPE = 44).

In Analysis R5 (Figs. 15 and 30O), which is constrained against the LH and supports the TH, Salientia forms the sister-group to all other modern amphibians (Procera); in other words, it lies as close as possible to the amphibamids, while Albanerpetidae and *Eocaecilia* (both not constrained) nest within the caudate grade. Procera was likewise found by *Vallin & Laurin (2004)*, *Pardo, Small & Huttenlocker (2017*: fig. S6A, but not S7*)* and *Pardo et al. (2017*: extended data fig. 7*)*. In R2 (Fig. 30N), in which various holospondyls are the closest relatives of lissamphibians and all of the above are nested among temnospondyls, *Eocaecilia* always lies outside Batrachia, while Albanerpetidae is the sister-group to Lissamphibia as a whole; moving Albanerpetidae back into Lepospondyli (as the sister-group of *Brachydectes*) cannot be done in Mesquite without adding at least eight steps.

The constraint for the PH specified only the backbone ((Salientia, *Doleserpeton*) (*Brachydectes*, *Eocaecilia*)); the positions of the caudates and Albanerpetidae were left open. Both groups fall out as lepospondyls (next to *Eocaecilia*) given the original taxon sample (R3; Fig. 30L). This version of the PH, where Salientia alone belongs to the temnospondyls while Caudata and Gymnophionomorpha are lepospondyls, has not been proposed since *Carroll & Holmes (1980)*. To move Caudata into Temnospondyli (as the sister-group of Salientia) requires two extra steps in Mesquite; to move Caudata + Albanerpetidae into Temnospondyli costs a total of three extra steps and comes at the cost of nesting this clade within Salientia (next to *Notobatrachus + Vieraella*). However, the caudates including albanerpetids become temnospondyls when taxa—*Chelotriton?—are added (R6; Fig. 30P).

In summary, Salientia is attracted to Amphibamidae and to Caudata (especially *Chelotriton), and possibly pushed away by "lepospondyls"; Albanerpetidae is drawn toward brachystelechids + lysorophians and *Eocaecilia* as well as toward Caudata, and repelled by Amphibamidae.

Several of the attractions of Albanerpetidae will likely be reinforced by **Shirerpeton*, which was published too recently for us to score (*Matsumoto & Evans, 2018*). For example, **Shirerpeton* is the only modern amphibian known to possess a suproccipital, arguing for a position outside Lissamphibia; the postfrontal is absent as in batrachians

but not caecilians; the supratemporal or tabular, retained in *Eocaecilia* but no other known lissamphibians so far, appears to be present; at the same time, the squamosal and the pterygoid of **\*\*Shirerpeton* form a tube filled by the rod-like quadrate, a feature shared specifically with Caudata.

## Characters

### Continued problems with the character sample

> Taxon and character sets are now large enough to be mined for large-scale evolutionary trends [ . . . ].
>
> – *Coates, Ruta & Friedman, 2008*: 572

> As the number of analyzed characters increases, the accuracy and resolution of all methods also increase [ . . . ]. These results suggest that no matter which method is applied to a dataset, it should be a goal for morphological datasets to include as many characters as possible if the most accurate estimates of topology are to be obtained.
>
> – *O'Reilly et al., 2018*: 116

Our mergers of redundant characters are the largest factor in the decrease of the number of parsimony-informative characters from 333 to 277 (re-split to 280 for Analysis EB). This is less than three times the number of taxa (which is 102 in the original sample, 158 in the expanded sample). Many of the characters now have multiple states, so the total number of apomorphic character states has decreased much less than the number of characters suggests; still, the low number of characters (compared with the number of taxa) explains why the present matrix does not provide a fully resolved, robust large-scale phylogeny of the limbed vertebrates. That the number is indeed low for our taxon samples is confirmed by the low bootstrap values and by the short internal branches found in Analysis EB (Figs. 18, 19 and 21). It has become common for phylogenetic analyses of a similarly large or smaller sample of extinct amniotes to have far larger numbers of characters (*Godefroit et al., 2013*: 101 OTUs, 992 characters [all binary]; *Lee et al., 2014*: 120 OTUs, 1,251 characters [all binary]; *Tschopp, Mateus & Benson, 2015*: 81 OTUs, 477 characters; *Ezcurra, 2016*: 96 OTUs, 600 characters; *Simões et al., 2016*: 193 OTUs, 610 characters; *Shelley, Williamson & Brusatte, 2016*: 169 OTUs, 693 characters; *Baron, Norman & Barrett, 2017*: 74 OTUs, 404 parsimony-informative characters (counted by *Mortimer, 2017*); *Buscalioni, 2017*: 97 OTUs, 425 characters; *Cau et al., 2017*, "first dataset": 199 OTUs, 1,314 parsimony-informative characters [all binary; D. Marjanović, pers. obs.]; *Cau, 2018a*: 132 OTUs, 1,431 parsimony-informative characters [all binary]!). Indeed, the matrices of *McGowan (2002)*, *Vallin & Laurin (2004)*, RC07, and *Anderson et al. (2008a)* each contain characters that the three others lack, and a cursory look at those of *Pardo, Small & Huttenlocker (2017*—76 OTUs, 322 parsimony-informative characters [D. Marjanović, pers. obs.]*)* and *Pardo et al. (2017*—58 OTUs, 340 parsimony-informative characters*)* reveals that even they lack characters present in ours; the supermatrix by *Sigurdsen & Green (2011)*, compiled from those of *Vallin & Laurin (2004)*, RC07 and *Anderson et al. (2008a)*, has 335 characters that are

parsimony-informative even though the taxon sample is restricted to 25 OTUs. Clearly, then, the present character sample could easily be greatly expanded. Therefore, we strongly doubt the first statement quoted above.

In addition to being small, the character sample is not random. We think that postcranial characters in particular are underused in the present matrix. This is implied by its craniocentricity—190 of the 277 characters, a bit more than two thirds, describe the skull, lower jaw or teeth. Further, the endochondral braincase (*Maddin, Jenkins & Anderson, 2012*; *Szostakiwskyj, Pardo & Anderson, 2015*; *Pardo, Szostakiwskyj & Anderson, 2015*; *Ascarrunz et al., 2016*; *Pardo & Anderson, 2016*; *Pardo et al., 2017*, *2018*; *Anderson, Pardo & Holmes, 2018*) is represented by only three characters (which all concern the occiput), and the hyobranchial skeleton (*Witzmann, 2013*; *Anderson, Pardo & Holmes, 2018*)—even the much-discussed stapes (*Lombard & Bolt, 1979*, *1988*; *Bolt & Lombard, 1985*, *1992*; *Laurin, 1998a*; *Schoch, 2002*; *Clack, 2003*; *Robinson, Ahlberg & Koentges, 2005*; *Witzmann, 2006*; *Sigurdsen, 2008*)—has not been considered at all. The analyses of *Pawley (2006)*, who added a net total of 53 postcranial characters to the 95 of the matrix of *Ruta, Coates & Quicke (2003*—the preceding version of RC07*)* and found different results, support our suspicion that the present matrix and the resulting trees are affected by accidental sampling bias. For an overlapping taxon sample, *Ruta (2011)* found 157 characters from the appendicular skeleton alone, where the present matrix has 53.

Being the largest one available of its kind and having been scrutinized for problems of coding and scoring, our matrix is currently the best—or least bad—starting point for comparisons and discussions as carried out below. Future analyses of the phylogeny of early limbed vertebrates will, however, certainly need much larger character samples to reach reliable conclusions.

### Surprising reversals

It is not surprising that homoplasy is omnipresent (as shown by the tree indices) in a matrix with 102 or 158 OTUs that span 380 million years. Some reversals, though, are unexpected even within this context. In this section we present bones that are unambiguously optimized (by parsimony in Mesquite on the MPTs from Analyses R1–R6; Mesquite presents uncertainty as such rather than restricting itself to ACCTRAN or DELTRAN) as having been lost and then reappeared. Some of these have been discussed before (*Sigurdsen & Green, 2011*, and references therein), mostly in the context of lissamphibian origins. We do not, at present, take any position on whether such reversals should be considered particularly implausible (see *Wiens, 2011*; *Botelho et al., 2014*; *Diaz & Trainor, 2015*; *Ossa-Fuentes, Mpodozis & Vargas, 2015*).

RC07 did not specify any state changes as irreversible (or weighted them in other ways), and their trees (Fig. 1) already require some of the reappearances of lost bones that we find in our parsimony analyses. We have preferred not to make changes to the matrix or the analysis procedures that are not clearly necessary; because the development genetics of most (if not all) of the characters in this matrix are insufficiently well understood, and because supernumerary skull bones—some of which may or may not be

atavistic reappearances—are known in several taxa with sufficiently large sample sizes (e.g., *Greererpeton*: *Smithson, 1982*; *Micromelerpeton* and other "branchiosaurs": *Boy, 1972*; *Discosauriscus*: *Klembara, 1993*; *Klembara, Tomášik & Kathe, 2002*), we have not coded any state changes as impossible.

One of the apomorphies which support the sister-group relationship of Urocordylidae and Aïstopoda, and thus "nectridean" paraphyly, in Analyses R1 and R3–R6 is the reappearance of the supratemporal bone in the skull roof (ch. 32: transition from state PAR 2/POSFRO 3/INTEMP 1/SUTEMP 1(3) to state (2), plotted in Fig. 29; for details on all characters see App. S1); this bone is lost in all other "lepospondyls" except the most rootward one, *Westlothiana* (and lost again in the aïstopod *Phlegethontia*, along with several other skull roof bones). Note that this reversal is inherited from RC07 (Figs. 1 and 28D). Yet more surprising is the fact that the supratemporal is long and unusually narrow in urocordylids, aïstopods (except *Lethiscus*), and probably *Westlothiana* (ch. 51: state SUTEMP 3(1), otherwise found only in *Orobates* and *Archaeovenator*); this condition adds support to the hypothesis that the supratemporal of urocordylids and aïstopods is indeed a supratemporal rather than a neomorph.

In some trees from R5, the intertemporal, too, reappears up to twice (transitions to state PAR 2/POSFRO 3/INTEMP 1/SUTEMP 1(0); Fig. 26D). This happens when *Capetus*, *Balanerpeton* and Dendrerpetidae are nested close to Dissorophoidea; in some of those trees, *Nigerpeton* and *Saharastega* (and optionally Dvinosauria) are highly nested among the stereospondylomorphs, necessitating a second reappearance.

The lost parasymphysial bone in the lower jaw, if correctly identified, reappears once in the aïstopods *Lethiscus* and *Coloraderpeton* (ch. 147: state PSYM 1(0); compare Fig. 24).

The postsplenial bone of the lower jaw reappears (ch. 157: transitions to state POSPL 1(0)) up to three times in the original taxon sample, and up to twice in the expanded sample. The loss in Diadectomorpha + Amniota is reportedly reversed in *Petrolacosaurus* (*Reisz, 1981*; see discussion in App. S1), concerning all MPTs from R1–R6. In Analyses R1–R3 and R6, where the diplocaulids are close enough to *Batropetes* and *Brachydectes* which lack postsplenials, the presence of the postsplenial in *Diplocaulus* (*Douthitt, 1917*)—contrasting with absence in the diplocaulids *Diploceraspis* and *Batrachiderpeton* (as correctly scored by RC07: *Beerbower, 1963*; *Bossy & Milner, 1998*)—is unambiguously optimized as a reversal as well. The third reappearance occurs in the adelogyrinids in R1 and R3.

The anterior tectal is optimized as disappearing early and then reappearing in the colosteid *Aytonerpeton* (see ch. 6—TEC 1). This happens not only in the MPTs from Analyses R4–R6, where Colosteidae and Baphetidae are sister-groups within Temnospondyli, but also when we move Colosteidae in these trees (in Mesquite) to the more traditional place it has in Analyses R1–R3, because the anterior tectal is scored as absent in *Pederpes* and *Crassigyrinus*. The homology of this bone is not quite clear, however, and most snout tips in this part of the tree are suboptimally preserved; see ch. 6 and 7 in App. S1 for discussion.

The preopercular bone reappears (ch. 81: state PREOPE 1(0)) in the aïstopod *Coloraderpeton* after a very long absence. We suspect that this is an artefact of our character sample and that Aïstopoda belongs far rootward of where we find it (*Pardo et al., 2017*); further research on other aïstopods, notably **Ophiderpeton*, may be helpful. The same holds for the return of the mandibular lateral-line canal and its fully or mostly enclosed condition in *Coloraderpeton*; with all other aïstopods currently scored as lacking this canal altogether, the transition from state 4 to state 1 or 0 of the ordered ch. 101 (SC 2) counts as (at least) three autapomorphies.

The suture between the basisphenoid and the basioccipital ("ventral cranial suture") is usually covered ventrally by the parasphenoid in limbed vertebrates crownward of *Crassigyrinus*. The parasphenoid is, at least in the midline, too short to cover it (ch. 142: state PASPHE 9(1)) in the aïstopods *Coloraderpeton* and probably *Pseudophlegethontia*, and *Lethiscus* even has an exposed open fissure there if the scan presented by *Pardo et al. (2017)* has successfully distinguished spongy bone from rock (which may not be the case; J. Pardo, pers. comm. 2017). (We have kept state 2, the covered condition, for *Oestocephalus*; this has more recently turned out to be wrong (*Anderson, Pardo & Holmes, 2018*). In *Phlegethontia*, the braincase is fused up to such an extent that the caudal margin of the parasphenoid cannot be recognized, so we have kept its score unknown.) While this is an unexpected reversal in our trees, it also occurs in the diadectomorph *Tseajaia* and some individuals of *Diadectes*, including apparently all of *D. absitus* (*Moss, 1972*; *Berman, Sumida & Martens, 1998*); and in the dissorophoid temnospondyl *Acheloma* the lateral parts of the suture are covered very late in ontogeny (*Maddin, Reisz & Anderson, 2010*).

Unlike in the earlier version of this matrix discussed by *Marjanović & Laurin (2013)*, losses of the posterior (or rather distal) coronoid (ch. 175: POST COR 1(1)) are no longer necessarily reversed on any of our MPTs except those of R6, where the bone reappears in *Coloraderpeton*. The precise homology of the bone that bears the lingual toothrow in the caecilian lower jaw and fuses to the dentary in early ontogeny remains, as far as we know, to be investigated; in the absence of evidence for separate bones, we have scored *Eocaecilia* as lacking any coronoids despite the presence of the lingual toothrow on the so-called pseudodentary (a bone of compound ontogenetic origin in extant caecilians). We have scored *Diplocaulus* as unknown; the dentary is unusually broad in dorsal view and bears a short lingual toothrow around the symphysis (*Douthitt, 1917*: fig. 2.5), and the area is damaged in all specimens that *Douthitt (1917*: 17*)* had at his disposal, so we cannot exclude that part of the supposed dentary belongs to a coronoid (or indeed several) and the suture is simply too eroded to be visible.

*Sigurdsen & Green (2011)* noted that their reevaluation of the matrix by *Vallin & Laurin (2004)* continued to support the LH, but required the return of the lost postparietal (ch. 39 in our matrix: POSPAR 1-2(0)), postfrontal (ch. 45: POSFRO 1(0)) and tabular (ch. 53: TAB 1/SQU 4(0); see also below) in *Eocaecilia*. Our unconstrained Analyses R1 and R4 do not require any of these reversals, and neither do R3 and R6, which are constrained for the PH (Figs. 30L, 30M and 30P). Conversely, the same three reversals are required on all MPTs from R2 and R5 (both constrained against the LH, supporting

the TH; Figs. 30N and 30O). In other words, the losses of these three bones actually favor the LH and the PH over the TH.

It remains to be seen, however, if the possible tabular of *Eocaecilia*—here scored as such (as explained by *Marjanović & Laurin, 2008*)—really is a tabular or instead a supratemporal. According to *Jenkins, Walsh & Carroll (2007)*, it is not known if that bone reaches the caudal margin of the skull table (as expected of a tabular), although it seems to D. Marjanović (pers. obs. of the holotype, MNA V8066) that it does; see ch. 53 (TAB 1/SQU 4) in App. S1 for details. *Pardo, Small & Huttenlocker (2017)* misinterpreted the reconstruction in *Jenkins, Walsh & Carroll (2007*: fig. 1) as indicating that the shape and size of this bone were known, and argued for considering it a supratemporal based on this erroneous premise. Whether the similarity of the unusually large squamosal of *Eocaecilia* to the squamosal and the unusually large tabular of **\*\*Chinlestegophis* combined (*Pardo, Small & Huttenlocker, 2017*: fig. 3; not mentioned in the text) is indicative of homology remains to be tested.

Analysis R5 always nests *Eocaecilia* and Albanerpetidae within Caudata (Fig. 15). Albanerpetidae was not included in the matrix of *Vallin & Laurin (2004)*. The jugal is present (ch. 71: JUG 1(0)) in *Eocaecilia* and Albanerpetidae, but lost in Batrachia; thus, Analysis R5 requires that an exclusive common ancestor of *Eocaecilia* and the albanerpetids has regained the lost jugal (which yields a total of three steps for this character over the entire tree). This reversal does not occur in any trees from R1 or R4 (Figs. 30L and 30M); *Brachydectes* has lost the jugal, but is never found as the sister-group of the modern amphibians (unlike in *Vallin & Laurin, 2004*—Fig. 30A), so its loss is always optimized as a separate event from the loss at the root of Batrachia (for a total of two steps). Thus, the presence of the jugal in *Eocaecilia* and Albanerpetidae indirectly but unambiguously supports the LH over the TH, quite contrary to the conclusions of *Sigurdsen & Green (2011)*. (Similarly, R6 nests Albanerpetidae within Caudata, requiring the same reversal for the PH.)

Of the other characters mentioned by *Sigurdsen & Green (2011)* in this context, the losses of the lacrimal (ch. 18: LAC 1(1); *Eocaecilia*, \**Chelotriton* + some or all salientians—unknown in *Triadobatrachus*, *Diploceraspis*, *Phlegethontia*), the prefrontal (ch. 12: PREFRO 1(1); limited to salientians crownward of *Triadobatrachus*, which retains the prefrontal: *Ascarrunz et al., 2016*) and the quadratojugal (ch. 77: QUAJUG 1(1); *Valdotriton*, \**Beiyanerpeton*, part of *Notobatrachus*, *Batropetes*, *Brachydectes*—unknown in \**Carrolla* and \**Quasicaecilia*) do not unambiguously reverse in any of our trees regardless of constraints (Figs. 30L–30P).

The palatine is scored as present (ch. 114: PAL 8(0)) in only two batrachians in this matrix: the salientian *Triadobatrachus* and the caudate \**Beiyanerpeton*. No reversal is required in any MPTs from Analyses R1–R3. In R4 and R6, these occurrences of state 0 are optimized as two separate reversals; in R5, only that of \**Beiyanerpeton* remains an unambiguous reversal, to which some MPTs from R5 (supporting the TH) add the occurrence in *Eocaecilia* as a further one. Judging from comparisons to the literature, all these apparent reversals seem to be clearly due to our small sample of salientians, caudates and caudate-related characters.

An angular (ch. 160: ANG 1(0)) has been reported in the caudates *Karaurus* and *Chelotriton* (see App. S1), but in no other lissamphibians in our sample (see, however, *Jia & Gao, 2016a*, *2016b*, for further occurrences in caudates). The former's condition is consistently optimized as a reversal; so is the latter's in Analysis R4 (Fig. 30M). The same cautionary notes about our taxon and character sample apply, particularly because *Chelotriton* is a crown-group salamandrid (*Marjanović & Witzmann, 2015*, and references therein); moreover, the ventral view of *Karaurus* has not been adequately described or illustrated and is not visible on the cast we had access to.

Only three "microsaurs" are scored for the presence (ch. 167: ANT COR 1(0)) or absence (state 1) of the anterior (mesial) coronoid: presence has been described in *Microbrachis*, absence in *Pantylus* and *Batropetes*. An unambiguous reversal does not occur among "microsaurs" according to the analyses of the original taxon sample (but does happen in *Lethiscus*); the condition of *Microbrachis* is optimized as a reversal in R4–R6, where *Microbrachis* and *Hyloplesion* are placed next to Holospondyli rather than among other "microsaurs". As far as the crushed material allows, the lower jaw of *Microbrachis* should be reinvestigated.

The "nectrideans" *Urocordylus* (*Bossy & Milner, 1998*, and references therein) and *Diceratosaurus* (*Jaekel, 1903*; *Bossy, 1976*; A. Milner, pers. comm. 2009; D. Marjanović, pers. obs. of MB.Am.776, CM 25468, CM 34617, CM 81504 and CM 81508; contra *Bossy & Milner, 1998*: 99) have five fingers per hand (state 2 of ch. 276 = DIG 1-2-3-4), even though many other "lepospondyls", including other "nectrideans" like *Sauropleura* and *Keraterpeton*, are known to have four or three (states 3 and 4). For *Diceratosaurus* at least, this is not a case of malformation during limb regeneration as sometimes found in the temnospondyl *Micromelerpeton* (*Fröbisch, Bickelmann & Witzmann, 2014*): all five fingers, and indeed metacarpals, have distinct lengths and widths, with the longest and thickest in the middle (*Jaekel, 1903*; D. Marjanović, pers. obs. of MB.Am.776, the same specimen). Furthermore, *Urocordylus* and *Diceratosaurus* are not close relatives; *Urocordylus* is a urocordylid like *Sauropleura* (BPO = **83**; BPE = **81**; PP for Urocordylidae incl. Aïstopoda = **96**), while *Diceratosaurus* is a diplocaulid like *Keraterpeton* (BPO = **57**; BPE = **63**; PP = **100**).

Thus, according to our unconstrained results, the supratemporal reappeared once after having been lost, and pentadactyly even did so twice; the parasymphysial and the anterior coronoid may have done so once or not at all, the intertemporal (in the expanded taxon sample) twice, once or never. Perhaps unexpectedly, the shortest trees that are constrained against the LH (and consequently support the TH: Analysis R2) additionally require the postparietal, the postfrontal and the tabular of *Eocaecilia* to be interpreted as reversals; expanding the taxon sample (Analysis R5) further extends this list to the jugal found in *Eocaecilia* and Albanerpetidae. Conversely, given the expanded taxon sample, the lost palatine and the lost angular each returned twice according to our unconstrained results (R4), and once or twice according to constrained analyses (R5), but issues with our taxon and character samples are evident.

***Other recently discussed characters that are included in this matrix***

The character that describes whether the maxilla and the quadratojugal touch at the ventrolateral margin of the skull (ch. 72: state JUG 2-6(0)) or the jugal intervenes between them (states 1 and 2) is only applicable when all three of these bones are present, a fact neglected by *Sigurdsen & Green (2011)* who highlighted this character several times in the text and in their table 4, arguing that it groups "lepospondyls and amniotes, but not [...] modern amphibians" (p. 467). This character does indeed reverse from state 1 to state 0 in *Eocaecilia* in the MPTs from Analysis R4 (and R6), while it does not reverse in those from Analysis R5, which support the TH; however, it also reverses seven additional times elsewhere in the MPTs from Analyses R4 and R6, compared to six in the MPTs from Analysis R5, out of a total of 24 steps in each of the three analyses. We see no reason to ascribe great subjective significance to this character, which is also known to have changed states within Amniota several times; indeed, the occurrence of state 0 in *Caseasauria is unambiguously optimized as a reversal in all of our trees. The holotype of *Lydekkerina even has state 0 on the left and state 1 on the right side (*Hewison, 2007*).

*Sigurdsen & Green (2011)* divided the length/width ratio of the vomer into two states. In our matrix this character, 102 (VOM 1-13), has three states (because RC07 rendered it as two characters). The widest vomers (state 0) occur in *Ymeria, Ventastega, Colosteus, *Spathicephalus, the "core amphibamids" (*Eoscopus, Platyrhinops, Amphibamus, Doleserpeton, *Gerobatrachus; synapomorphic among them), the seymouriamorph *Ariekanerpeton*, and all lissamphibians (unknown in Albanerpetidae and *Triadobatrachus*) except *Eocaecilia* and *Pangerpeton. Under the LH (Analysis R4; Fig. 30M), the occurrence of the intermediate state (1) in *Eocaecilia* is plesiomorphic—indeed, the absence of state 0 is symplesiomorphic with *Eusthenopteron*—and that in *Pangerpeton is equally parsimoniously optimized as either plesiomorphic or a reversal. Under the TH (Analysis R5; Fig. 30O), state 0 is synapomorphic among amphibamids and lissamphibians, and state 1 in *Eocaecilia* and *Pangerpeton requires either two reversals or (equally parsimoniously) a shared reversal followed by a return to state 1. Under the PH (Analysis R6; Fig. 30P), state 1 in *Pangerpeton is unambiguously a reversal from state 0, while state 1 in *Eocaecilia* is not. Altogether, this ordered three-state character requires 21 steps in Analysis R4 and 20 in R5 and R6. This character is, in other words, rather inconclusive.

The length and curvature of the ribs was given special attention by *Sigurdsen & Green (2011*; and references therein*), in that they connected short, straight ribs with buccal pumping as the mode of lung ventilation seen in extant amphibians, and long, curved ones with aspiration by active expansion of the ribcage as seen, plesiomorphically, in amniotes. We find much to disagree with here. First, long straight ribs exist (e.g. *Diplocaulus*), as do short curved ribs (e.g. *Acherontiscus*); consequently, RC07 treated these features as two characters, which remain in our matrix as ch. 246 (RIB 3)—"Ribs mostly straight (0) or ventrally curved in at least part of the trunk (1)"—and 250 (RIB 7)—"Trunk ribs longer (0) or shorter (1) than three successive articulated vertebrae in adults". Second, short ribs (straight or not) may require buccal pumping, but long curved ribs do not require aspiration breathing and do not even automatically make it possible, as the trivial

cases of the long but entirely immobile ribs of **turtles and such **pterosaurs as **Pteranodon demonstrate; as long as the necessary research on the mobility of the ribs and their muscle attachment sites has not been done, we see no reason to think that such animals as anthracosaurs or lysorophians breathed air in a way much different from **extant caecilians, which use buccal pumping and inhale several times for each time they exhale, thus compensating for the fact that their heads are much smaller than their lungs (Carrier & Wake, 1995). Third, such research may be difficult in taxa without ossified rib heads; that includes even such terrestrial animals as the chroniosuchian **Bystrowiella (Witzmann & Schoch, 2017). The hypothesis that buccal pumping is not a lissamphibian or temnospondyl autapomorphy, but rather an ancient plesiomorphy (in agreement with Lebedev & Coates, 1995, and Witzmann, 2016), is further supported by the fact that it forms the last stage of air inhalation in the extant actinopterygian **Polypterus (Graham et al., 2014). In any case, states RIB 3(0) and RIB 7(1), both found in all lissamphibians in this matrix except *Chelotriton, support the TH and the PH over the LH by one step each in the trees with the expanded taxon sample (Figs. 30M, 30O and 30P); the PH ranks with the TH here because these states are optimized as synapomorphies of Eocaecilia and Aïstopoda under the PH.

It may prove interesting that the ribs of **Chinlestegophis, briefly mentioned but not illustrated by Pardo, Small & Huttenlocker (2017), are curved (state RIB 3(1); Figs. 30I–30K) and considerably longer than those of any lissamphibians except certain salamandrids like *Chelotriton, though it is not clear whether state RIB 7(0) was reached (J. Pardo, pers. comm. 2017).

### Preaxial polarity in limb development

Although not included in our matrix, this character has featured prominently in several recent publications, so we take the opportunity to review the latest developments here.

In frogs and amniotes today, the digits form in a roughly postaxial-to-preaxial (caudocranial) sequence: IV first, then III and V, then II, then I. In salamanders with free-living larvae, the opposite is observed: first I and II, then III, then IV, then (in the foot) V. Fröbisch, Carroll & Schoch (2007) documented the latter (as an ossification sequence) in the dissorophoid temnospondyl Apateon and proposed that this could be a synapomorphy of (at least some) temnospondyls and salamanders. Marjanović & Laurin (2013) discussed this, finding the optimization ambiguous due to the lack of an outgroup, and pointed out that distal carpal/tarsal I ossified first in the few temnospondyls, but also anthracosaurs and colosteids, of which incompletely ossified carpi/tarsi are known; distal carpal/tarsal 4 is known to ossify first only in amniotes and diadectomorphs. Recently, Glienke (2015: fig. 6O–S) has documented that distal tarsal 1 and/or 2 ossified first in the "lepospondyl" Batropetes. This bolsters the inference that the resemblance of Apateon and salamanders is symplesiomorphic, and that the amniote/frog pattern of development (see Marjanović & Laurin, 2013, for further discussion) is an autapomorphy of Amniota + Diadectomorpha as well as of Salientia or part thereof.

Olori (2015) found a long delay in the ossification of the fifth toe in the "lepospondyls" Microbrachis and Hyloplesion; because this is shared with Sauropsida, Olori (2015) suggested that these "microsaurs" may have shared the frog/amniote pattern (repeated

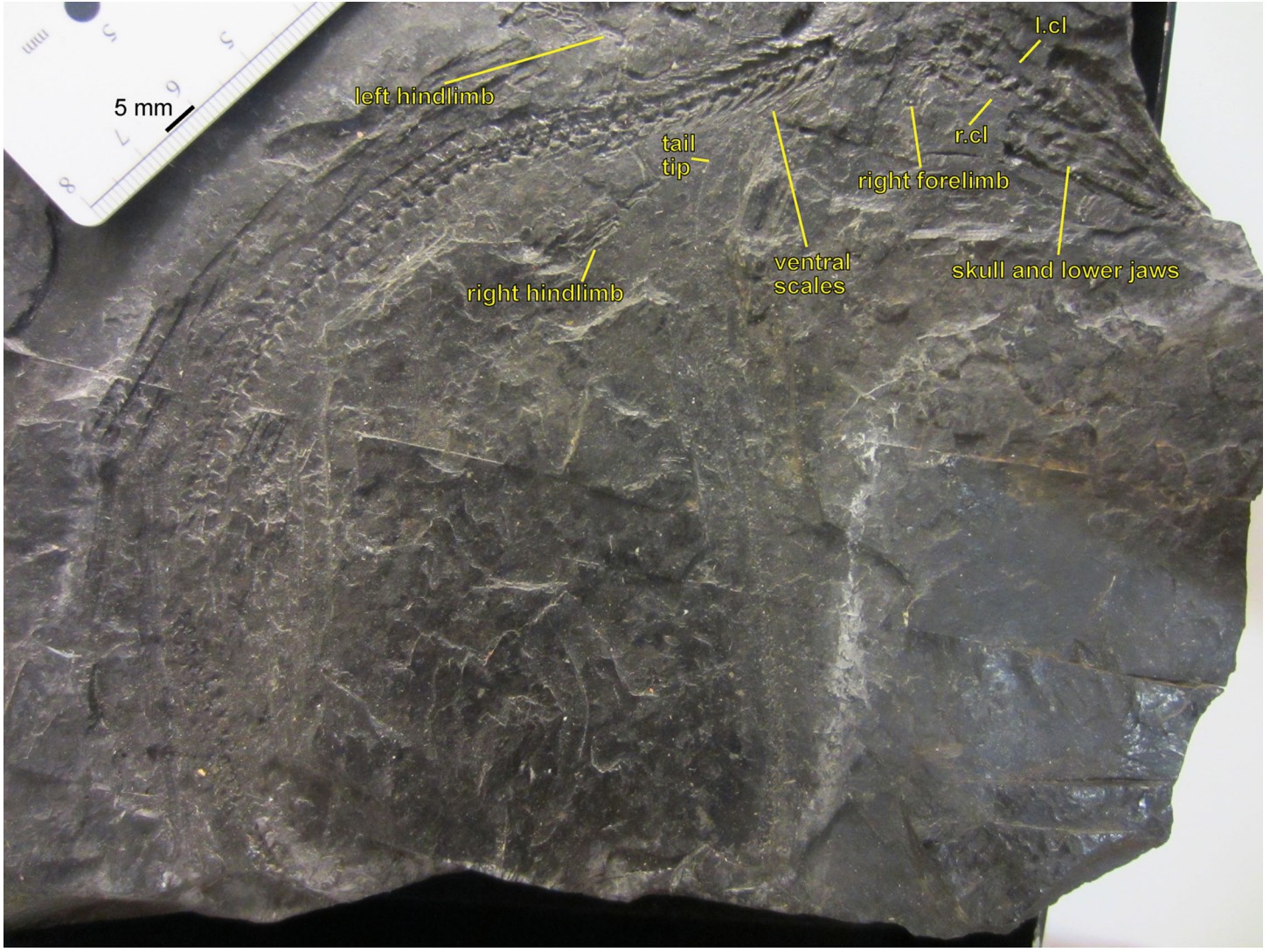

**Figure 31 NHMW 1983/32/11, a natural mold of an articulated skeleton referred to *Sauropleura scalaris*.** Abbreviations: l.cl, left clavicle; r.cl, right clavicle. Photo taken by D. M.

without further discussion by *Fröbisch et al., 2015*). We do not find this convincing, especially in the light of *Batropetes*.

At this opportunity we would like to report that distal tarsal 2 appears to be the only ossified tarsal in NHMW 1983/32/11, a large articulated specimen of *Sauropleura scalaris* (Figs. 31 and 32). At least for the time being, our analyses continue to find *Sauropleura* as a "lepospondyl". Further, we wonder if a poorly mineralized round object proximal to metacarpal I in *Casineria* is distal carpal 1; other carpals are not mineralized (Fig. 6; *Paton, Smithson & Clack, 1999*: fig. 2, 3).

*Jia & Gao (2016b)* pointed out that the ossified parts of radius and tibia are considerably longer and wider than those of ulna and fibula, respectively, in the smallest known larva of the Early Cretaceous crown-group salamander *Nuominerpeton*. They considered this fact to be evidence of a preaxial-to-postaxial sequence in development. If this inference

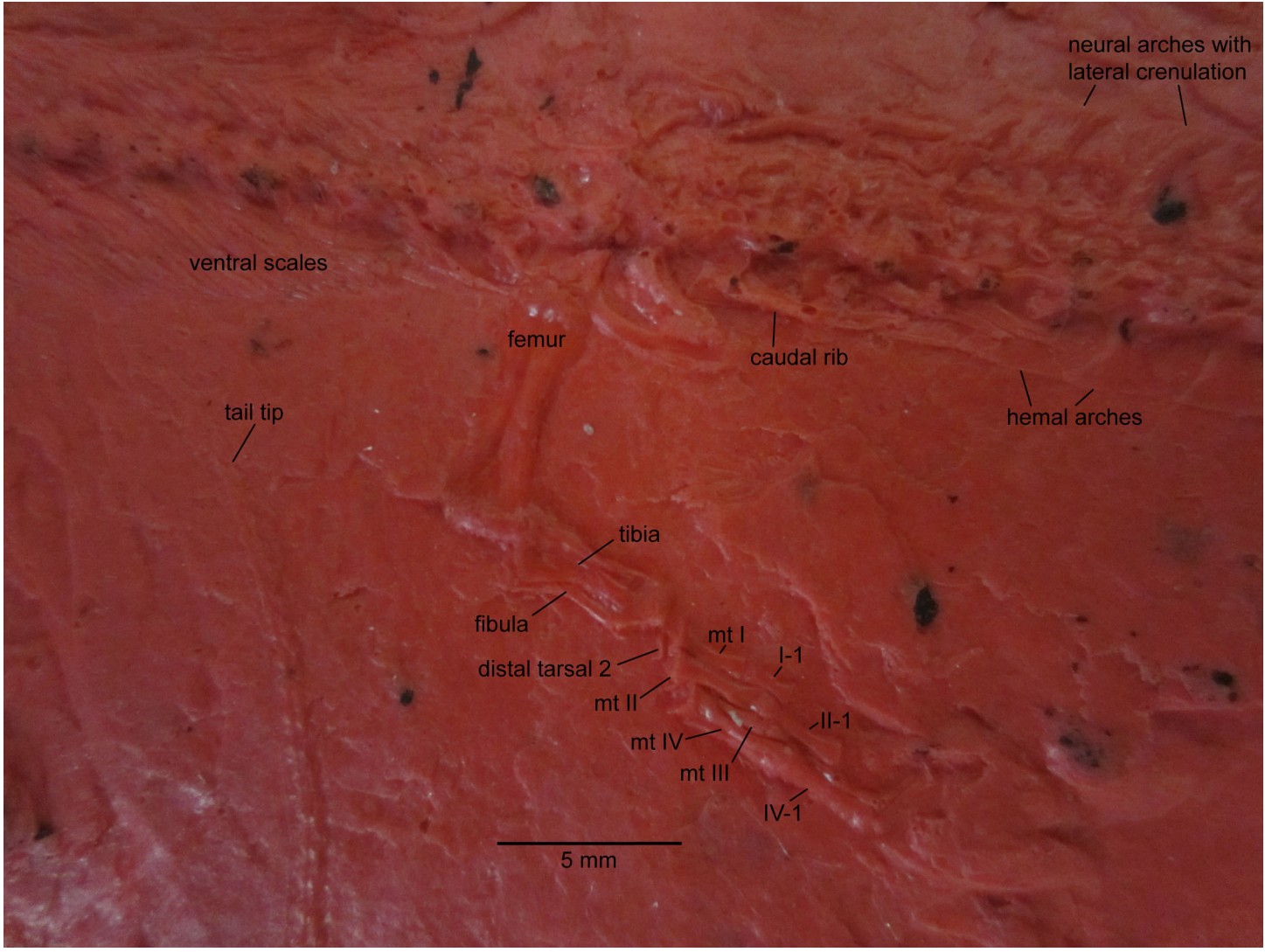

**Figure 32 Area around the right hindlimb of a latex cast of the *Sauropleura* specimen shown in Fig. 31.** Abbreviation: mt, metatarsal. Photo taken by D. M.

holds, preaxial polarity is very clearly plesiomorphic: state RAD 2(0), radius longer than ulna (see ch. 222 in App. S1), is found in our matrix in *Eusthenopteron*, *Panderichthys*, *Acanthostega*, *Crassigyrinus* (whose endochondral skeleton is generally poorly ossified) and the skeletally immature temnospondyl *\*Tungussogyrinus*.

### Other characters that are potentially important for the origin of lissamphibians but not considered in this matrix

Several matrices for the analysis of the phylogeny of the limbed vertebrates have coded phalangeal formulae as one or more characters. It appears that the first four toes plesiomorphically have 2-3-4-5- in the pentadactyl clade, as retained by most amniotes and most "lepospondyls," while reductions in the number of phalanges (2-2-3-4- or fewer) occur in temnospondyls and modern amphibians. However, intermediate states are

found in Albanerpetidae, where *Celtedens*—the only representative for which feet are known—has lost only one phalanx (2-3-4-4-; *McGowan, 2002*), and in *Batropetes*, which has lost two (2-3-4-3-; *Glienke, 2015*: fig. 6O–S). We have not looked into this further, but so far it seems that phalangeal formulae do not support the TH or the PH over the LH.

*Sigurdsen & Bolt (2009)* have pointed out that lissamphibians share a particularly large radial condyle on the humerus with dissorophid and amphibamid temnospondyls, and *Sigurdsen & Green (2011)* used this as a character in their supermatrix. *Marjanović & Laurin (2013)* have discussed complications. D. M. has since seen two humeri of *Diadectes* (AMNH 4380 and AMNH 4709); their radial condyles are proportionally at least as large as those of *Doleserpeton* or \*\**Dissorophus* (*Sigurdsen & Bolt, 2009*: fig. 3).

*Sigurdsen (2008)* drew renewed attention to the fact that the perilymphatic duct in the inner ear runs on different sides of the lagena in lissamphibians and amniotes, and presented evidence that it was positioned in *Doleserpeton* as in frogs in particular. The conclusion that this resemblance must be synapomorphic rather than symplesiomorphic rests on the assumption that the amphibian and the amniote conditions must have evolved independently from the lack of a perilymphatic duct as seen in \*\**Latimeria*. We are not convinced by this latter inference; it is not clear to us why the duct could not have twisted around the lagena at some point, or conversely if the condition is homologous among all amniote clades.

*Marjanović & Laurin (2014)* discussed the branchial denticles and their occasional confusion with gill rakers (which most recently occurs in *Jia & Gao, 2016a*; for an animal that has branchial denticles and mineralized gill rakers, see \*\**Amia*—*Grande & Bemis, 1998*), but did not comment on an interesting fact: Plesiomorphically, there is one row of denticle-bearing dermal bone plates on each of the four ceratobranchials (per side). In branchiosaurids, the branchial denticles were apparently used for filter-feeding (the way gill rakers often are), so that each of the three gill slits (per side) was framed by two rows of denticle-bearing plates—ceratobranchials 1 and 4 bearing one each, ceratobranchials 2 and 3 bearing two each, for a total of six. Of the four lissamphibians (all of them currently considered crown-group salamanders) known to retain these plates, \**Beiyanerpeton* has four rows (*Gao & Shubin, 2012*), and apparently so does \*\**Qinglongtriton* (not quite clear from the illustrations or the text of *Jia & Gao, 2016a*)—but \*\**Chunerpeton* (*Gao & Shubin, 2003*) and \*\**Seminobatrachus* (*Skutschas & Gubin, 2012*) have six. Homoplasy must be assumed under both the TH and the LH (and even under the PH).

Alone among modern amphibians, most anurans have a tympanic ear, and apparently so did at least *Notobatrachus* among stem-salientians (*Báez & Nicoli, 2004*). Osteological correlates of this character complex have often been thought to occur in various temnospondyls, including but not limited to *Doleserpeton* and \*\**Chinlestegophis*. We mostly disagree with these identifications, which we will discuss at length in a future work; an incomplete draft is included in the first preprint version of this paper (*Marjanović & Laurin, 2015*).

## CONCLUSIONS

### Matrix accuracy

Morphological data matrices for phylogenetic analyses routinely contain problematic scores and questionably constructed characters (*Maddison, 1993*; *Strong & Lipscomb, 1999*; *O'Keefe & Wagner, 2001*; *Wiens, 2001*; *Vallin & Laurin, 2004*; *Wiens, Bonett & Chippindale, 2005*; *Pawley, 2006*; *Marjanović & Laurin, 2008*, *2009*; *Brazeau, 2011*; *Sigurdsen & Green, 2011*; *Bardin, Rouget & Cecca, 2014*; *Dilkes, 2015a*; *Watanabe, 2016*; *Simões et al., 2017*; *Mortimer, 2017*; *Langer et al., 2017*; *Spindler et al., 2018*: online resource 3). The matrix by RC07 is no exception. In 9 years of work, we have found it necessary (App. S1) to merge many redundant characters (the main cause for decreasing the total number of parsimony-informative characters from 333 to 277 without loss of information), redefine non-additive binary characters (of which one state was described while the other state was scored for all other conditions in RC07), order several potentially continuous characters (*Slowinski, 1993*; *Wiens, 2001*; *Grand et al., 2013*; *Rineau et al., 2015*; *Baron, Norman & Barrett, 2017*: supplementary information: 4–9) and make 4,200 changes to individual scores (documented and justified in App. S1, shown in App. S2 except for parsimony-uninformative characters, counted in Data S4), which includes splitting each of two composite characters in two and adding states to a few other characters.

Neither the original matrix nor our changes show evidence of bias (e.g. for or against a hypothesis on lissamphibian origins: Tables 6–12; $p > 0.1$). Rather, apart from the changes necessitated by works published since 2007, most of the 2,471 scores of RC07 that disagree with the literature or our observations of specimens without our having redefined a state are best explained as typographic or similar accidental, unsystematic errors.

### Phylogeny

The dataset by RC07 is the largest one that has so far been used in an analysis of the phylogeny of early limbed vertebrates, a field that contains the controversial question of the origin(s) of the extant amphibian clades. From analyses of our revised version of this dataset (App. S2, Data S3), we conclude:

- After our attempts to improve the quality of coding and scoring, the matrix supports different results from those found by RC07, most notably the "lepospondyl hypothesis" on the origin of Lissamphibia (LH) rather than the "temnospondyl hypothesis" (TH). This does not change when the taxon sample is expanded or when Bayesian inference is used instead of parsimony; the expanded taxon sample includes the lissamphibian-like dissorophoids *Gerobatrachus*, *Micropholis* and *Tungussogyrinus* as well as the extremely peramorphic and therefore accidentally temnospondyl-like salamander *Chelotriton* (*Marjanović & Witzmann, 2015*).

- Many parts of the tree are too sparsely sampled in characters to be strongly supported. The bootstrap values along the "trunk" of the tree are low; while the LH and the "polyphyly hypothesis" may entail statistically distinguishable topologies unless taxa are added, the TH is indistinguishable from both (Table 4); and while some Bayesian

posterior probabilities are high (notably 1.00 for lissamphibian monophyly and 0.92 for the LH), moderately high values are also found for more debatable groupings (in particular, Aïstopoda, Adelogyrinidae and *Acherontiscus* are supported as rather close to Lissamphibia by a PP of 0.76). Future analyses with a taxon sample of similar or larger size will need to have much larger character samples.

It follows that the "temnospondyl hypothesis" should not be taken for granted as the default by studies in evolutionary biology.

Similar conclusions hold for less popular questions which nonetheless have sometimes important implications for evolutionary biology:

The chroniosuchians (represented by *Chroniosaurus* and *Bystrowiella* in the expanded taxon sample) are neither anthracosaurs (*Laurin, 2000*; *Klembara, Clack & Čerňanský, 2010*) nor lepospondyls (*Buchwitz et al., 2012*) nor seymouriamorphs (*Klembara et al., 2014*) in our trees; instead, they lie between the more rootward gephyrostegids, which may be followed rootward by the temnospondyls, and the more crownward *Solenodonsaurus*, which is followed crownward (not rootward as in RC07) by Seymouriamorpha (or may even be a seymouriamorph, as weakly supported by our Bayesian analysis). Character conflict causes either the aquatic anthracosaurs or the ecologically diverse temnospondyls to lie more rootward depending on the taxon sample, the analysis method, and even constraints on the positions of extant amphibians. *Caerorhachis* lies in the same region of the tree; it may or may not be a temnospondyl as found by *Pawley (2006)*. Rather than *Casineria* being close to amniote origins, we cannot so far distinguish it from *Caerorhachis* (as previously noted by *Pawley, 2006*); we find it in the gephyrostegid grade, rootward of the chroniosuchians. This is corroborated by three (possibly four) previously overlooked features of *Casineria* that would be unexpected for a close relative of Amniota. *Westlothiana* retains its position from RC07: it is the sister-group to all other amphibians. *Iberospondylus* emerges particularly close to Dissorophoidea; *Nigerpeton* and *Saharastega* are well supported as sister-groups, and recovered around *Eryops* or around or even in Stereospondylomorpha in the MPTs, but with weak bootstrap support for a position next to Edopoidea and weak Bayesian support for a position in it; temnospondyl phylogeny otherwise remains largely mysterious and very strongly depends on the relative positions of temnospondyls and anthracosaurs, though a sister-group relationship of *Eryops* (the only sampled eryopid) and Stereospondylomorpha is likely. "Nectridean" monophyly is never found. "Microsaurian" monophyly is sometimes approximated, but the brachystelechids sometimes cannot even be considered part of a "microsaur" grade; neither can *Utaherpeton*. Despite the variety of results, Tuditanomorpha, Microbrachomorpha and Recumbirostra are not found; *Microbrachis* and *Hyloplesion* are sister-groups, however. *Perittodus* emerges rootward of *Ichthyostega*; *Densignathus* lies just crownward or just rootward of Whatcheeriidae, followed crownward by *Tulerpeton*, all of which are rootward of *Crassigyrinus* and Colosteidae; *Sigournea*, *Doragnathus* and *Diploradus* are found as baphetoids close to *Spathicephalus*, apparently baphetids; the "St. Louis tetrapod" and *Aytonerpeton* are unambiguous colosteids; the "Parrsboro jaw" may be close to temnospondyl origins and/or to *Caerorhachis*.

Within the LH, the closest relatives of lissamphibians including albanerpetids are not the lysorophians alone (represented by *Brachydectes*), but most parsimoniously a clade composed of the latter and the brachystelechid "microsaurs" (*Batropetes*, *Carrolla* and *Quasicaecilia*); the Bayesian analysis even nests Lissamphibia within Brachystelechidae with rather strong support (PP = 0.88). Within the TH, our constrained analyses find *Doleserpeton* and *Gerobatrachus* to be equally good (or bad) candidates for the lissamphibian sister-group.

## Phylogenetics

In spite of generally low support, several poorly known taxa are placed unambiguously in our trees. Isolated lower jaws and the like, not to mention headless skeletons like *Casineria* or the "microsaur" *Trihecaton*, should not be excluded from phylogenetic analyses a priori; they are by no means guaranteed to add more noise than signal, let alone noise that conflicts with the signal. Wildcard behavior is hard to predict and evidently does not, in our dataset, correlate with amounts or proportions of missing data.

Not all settings for phylogenetic analyses make it likely that all, or any, of the MPTs will be found. *Maddin, Jenkins & Anderson (2012)* recovered neither any of the MPTs that are consistent with their matrix, nor the vast majority of the trees that have the length of the shortest trees they reported (*Marjanović & Laurin, 2018*); similarly, at least one of the analyses performed by *Skutschas & Gubin (2012)* and *Schoch (2013)* failed to find any trees of the shortest possible length (reported but not commented on by *Skutschas & Gubin, 2012*, and *Schoch, 2013*, respectively), and *Baron, Norman & Barrett (2017)* found only 93 of 16,632 MPTs (*Watanabe, 2017a*, *2017b*; *Langer et al., 2017*). Care should be taken to run enough addition-sequence replicates to find all optimal islands, and to allow sufficient time for the branch-swapping algorithms to explore these islands thoroughly. However, the "parsimony ratchet" procedure of RC07 found all MPTs that were compatible with their matrix—even though it did not find any MPTs, or any trees less than 35 steps longer than the MPTs, when applied to the much smaller matrix of *Skutschas & Gubin (2012*, as reported there*)*.

We encourage testing of the behavior of Bayesian inference with large paleontological matrices (missing data clustered by body part; non-coeval tips), which is currently not well understood. Homoplasy-free characters are uncommon enough in our matrix that a prerequisite for parsimony with implied weights is most likely not fulfilled.

Further progress on the questions of phylogeny discussed above—and on many others—will require the description of new fossils and the redescription of known ones, but also the mergers of existing matrices to increase the character sample (and to a lesser degree the taxon sample) together with a special effort to keep data matrices free from typographic errors and similar accidental misscores, from the effects of heterochrony and from redundant characters.

In the meantime, we hope to have laid a solid base for future additions of data by creating a "reasonably clean" dataset that does not contradict current knowledge. We have documented all of our changes to the matrix of RC07 in App. S1 and S2 and invite further scrutiny. A matrix of morphological data is a matrix of hypotheses and must

not be taken for granted as a set of measured facts; and it must not be assumed that any cell in a matrix could not influence the resulting trees.

## ABBREVIATIONS

| | |
|---|---|
| **AMNH** | Properly AMNH DVP or AMNH FARB—Department of Vertebrate Paleontology or Collection of Fossil Amphibians, Reptiles and Birds at the American Museum of Natural History, New York. |
| **App./app.** | Appendix (our S1 or S2)/appendix (of cited works); all App.-Tables are part of App. S1. |
| **BEG** | Vertebrate Paleontology Laboratory, The University of Texas at Austin (formerly Bureau of Economic Geology). |
| **BPE** | Bootstrap percentage given the expanded taxon sample (bootstrap analysis B2). |
| **BPO** | Bootstrap percentage given the original taxon sample (bootstrap analysis B1). |
| **CG78** | *Carroll & Gaskill (1978)*. |
| **ch.** | Character. |
| **ci** | Consistency index (in lowercase to make the numbers stand out). |
| **CM** | Carnegie Museum of Natural History, Pittsburgh. |
| **LH** | "Lepospondyl hypothesis" on the origin of lissamphibians. |
| **hi** | Homoplasy index. |
| **ITRI** | Inter-tree retention index (*Grand et al., 2013*). This is an asymmetric measure of how many of the three-item statements that constitute a reference tree are shared by the tree in question, taking into account the number of degrees of freedom. |
| **MB.Am.** | "Amphibian" in the vertebrate paleontology collection of the Museum für Naturkunde, Berlin. |
| **MCZ** | Properly MCZ VPRA—"reptile" or "amphibian" in the vertebrate paleontology collection of the Museum of Comparative Zoology, Harvard University, Cambridge (Massachusetts). |
| **MGUH** | Geologisk Museum, Statens Naturhistoriske Museum, Københavns Universitet, Copenhagen ("Museum Geologicum Universitatis Hafniensis"). |
| **MNHN** | Muséum national d'Histoire naturelle, Paris. |
| **MNN MOR** | Moradi collection of the Musée national du Niger, currently kept at the Burke Museum, University of Washington, Seattle. |
| **MPT** | Most parsimonious tree. |
| **NHMW** | Naturhistorisches Museum Wien, Vienna. |
| **NMS** | National Museum of Scotland, Edinburgh. |
| **NSM** | Nova Scotia Museum, Halifax. |
| **OTU** | Operational taxonomic unit (a line in a data matrix). |
| **pers. comm.** | personal communication. |
| **pers. obs.** | personal observation. |

| PH | "Polyphyly hypothesis" on the origin of "lissamphibians". |
|---|---|
| PIN | Paleontological Institute of the Academy of Sciences of the Russian Federation, Moscow. |
| PP | Posterior probability in percent given the expanded taxon sample (exploratory Bayesian analysis EB). |
| PU-ANF | "Amphibian" specimens from Puertollano (Spain) in the vertebrate collection of the Department of Paleontology, Geology division, Universidad Complutense, Madrid; formerly kept at the Museum für Naturkunde, Berlin. |
| rc | Rescaled consistency index. |
| RC07 | *Ruta & Coates (2007)*. |
| ri | Retention index. |
| TH | "Temnospondyl hypothesis" on the origin of lissamphibians. |
| TMM | Vertebrate Paleontology Laboratory, The University of Texas at Austin (formerly Texas Memorial Museum). |
| USNM | National Museum of Natural History, Smithsonian Institution, Washington (DC). |
| YPM | Yale Peabody Museum, New Haven (Connecticut). |

## ACKNOWLEDGEMENTS

This work began as chapter V of the doctoral thesis of Damien Germain supervised by M. L. (*Germain, 2008a*). D. Germain did not have time to complete this large task, so additional work came to be done as chapter 5 of the doctoral thesis of D. M., again supervised by M. L. (*Marjanović, 2010*; parts were presented in *Marjanović & Laurin, 2013*). This, too, remained incomplete for lack of time. Further work was later done by D. M.; this is presented here for the first time (apart from the preprints of this work: *Marjanović & Laurin, 2015*, *2016*, *2018*). We thank Damien Germain for sharing his data and expertise with us; he did not feel that he had contributed enough to this study to coauthor it.

We warmly thank José Grau (formerly Museum für Naturkunde, Berlin) for coming up with, and implementing, ways to run the bootstrap and Bayesian analyses in parallel on a computer he had access to. Previously, they took a week each, even though we had restricted the bootstrap analyses to very few heuristic replicates per bootstrap replicate and extremely few rearrangements per heuristic replicate, restrictions that were no longer necessary this time.

Valentin Rineau (CR2P, Paris) kindly calculated the ITRI (Table 5); hardware and software issues prevented us from doing that ourselves.

We are indebted to Ronan Allain and Bernard Battail for prolonged access to casts of *Triadobatrachus* and *Karaurus* at the MNHN (as well as to the original of *Triadobatrachus* for shorter periods). D. M. further thanks Jasons Anderson and Pardo, Andrew and Angela Milner, Marcello Ruta, Rainer Schoch, Trond Sigurdsen, Florian Witzmann, David Černý, Arjan Mann, Ralf Werneburg, John Nyakatura and Sabine Glienke for discussions, Andrew Milner in particular for discussion of the postcranium of

*Isodectes* and David Černý for discussion of the current state of Bayesian inference in phylogenetics (in comments to *Mortimer, 2017*); Florian Witzmann, Oliver Hampe and the librarian Annegret Henkel (now retired) for access to dead-tree literature; Ursula Göhlich, Mathias Harzhauser and Alexander Lukeneder for access to specimens of *Microbrachis*, *Hyloplesion*, *Sauropleura* (Figs. 31 and 32), *Sparodus* (Figs. 3 and 4) and others at the NHMW, Ursula Göhlich for permission to publish photos (Figs. 3, 4, 31 and 32), Amy Henrici and Matt Lamanna for access to many specimens of *Chenoprosopus*, *Isodectes*, *Diceratosaurus*, *Ptyonius*, *Tseajaia* and others at the CM (twice), Michael Brett-Surman and Hans-Dieter Sues for access to specimens of *Chenoprosopus*, *Isodectes*, *Eryops*, *Tuditanus*, *Odonterpeton* and others at the USNM, J. Chris Sagebiel and Timothy Rowe for access to specimens of *Acanthostega*, *Ichthyostega*, *Greererpeton*, *Gephyrostegus*, *Eryops*, *Seymouria*, *Diadectes*, *Paleothyris*, *Carrolla* and others at the BEG/TMM, Carl Mehling for access to specimens of *Ichthyostega*, *Colosteus*, *Trimerorhachis*, *Eryops*, *Phonerpeton*, *Doleserpeton*, *Platyrhinops*, *Diadectes*, *Tuditanus*, *Diceratosaurus* and others at the AMNH, Mark Norell for access to a microscope in his office at the AMNH (that's the office with the amazing view), Jessica Cundiff for access to specimens of *Ichthyostega*, *Edops* (including the large, heavy skull on exhibit and its separately kept right stapes), *Eryops*, *Acheloma*, *Phonerpeton*, *Eocaecilia*, *Archeria*, *Paleothyris*, *Ptyonius*, *Lethiscus*, *Eothyris*, *Oedaleops* and others at the MCZ, Stig Walsh for access to *Casineria* and a microscope with a digital camera at the NMS, for explaining that the almost Я-shaped glyph in the specimen number on the part and counterpart of *Casineria* is a G and for permission to publish photos (Figs. 5–7), Florian Witzmann, Oliver Hampe, Thomas Schossleitner and Rodrigo Soler-Gijón for access to specimens of *Gephyrostegus*, *Batropetes*, *Scincosaurus*, *Diceratosaurus*, *Iberospondylus*, *Sclerocephalus* and *Chelotriton* at the Museum für Naturkunde (Berlin), Chris Sidor, Meredith Rivin and Ron Eng for access to the collection of the Burke Museum at the University of Washington (Seattle) where all known material of *Nigerpeton* and *Saharastega* is currently kept, Jason Anderson for access and transportation to a specimen of *Asaphestera* (currently on loan in his lab at the University of Calgary), not to mention others that will feature in future publications, Frederik Spindler, Ralf Werneburg, Ronny Rößler and Jörg Schneider for access to and discussion of the only known skull of *Madygenerpeton* during a conference-associated excursion in 2014, during which the abovementioned two aïstopod specimens from Chemnitz (*Rößler et al., 2012*) were also shown, and Antonia Bookbinder and family, Keating Holland, Maryann Haggerty and Erin Machniak for hospitality during collection visits. Adam Bodzioch and Jakub Kowalski kindly enabled D. M. to participate in the excavations in Krasiejów (Poland) of 2015 and to partially prepare a *Metoposaurus* clavicle discovered there by Weronika Kulikowska. Finally, D. M. would like to thank the security staff at the AMNH for successfully wrestling with the new badge system in 2013.

Andrew Milner, Laura Nicoli, Jasons Anderson and Pardo, Florian Witzmann, Jorge Bar, Guilhem Carbasse, José R. Guzmán Gutiérrez, Graeme Lloyd, Krzysztof Rogoż, Thiago Carlisbino, Eduardo Ascarrunz and Ralf Werneburg sent electronic reprints, some of which would have been impossible to acquire otherwise; Eduardo Ascarrunz and John Nyakatura shared their digital models of *Triadobatrachus* (*Ascarrunz et al., 2016*)

and *Orobates* (*Nyakatura et al., 2015*); Trond Sigurdsen sent a version of the NEXUS file of RC07 to clear up our confusion about different-looking versions. The authors of *Ascarrunz et al. (2016)* and *Pardo & Anderson (2016)* kindly allowed us to use their manuscripts before publication.

We apologize for any omissions from these lists; their order is meant to be chronological.

D. M. thanks David Peters for making a claim about *Eocaecilia* and *Microbrachis* that turned out to be incorrect but led D. M. to consulting the literature and correcting two scores for *Acherontiscus* and *Adelospondylus*.

The reviewers Alexander Pyron, Michael Buchwitz and Jason Pardo took the time to read the huge manuscript three times (J. Pardo five times), for which we are very grateful; their comments helped clarify our explanations. Thomas Arbez kindly helped us proofread it. We also greatly appreciate Erik Seiffert's handling of the manuscript as editor during its most difficult period, and thank Peter Binfield for allowing that to happen.

### Funding
At the beginning of the study period (up to 2010), David Marjanović was a beneficiary of the University Student Subsidy Office (Studienbeihilfenbehörde) of the Republic of Austria; from February 2012 to February 2014, he received an unnumbered postdoctoral grant from the Alexander von Humboldt Foundation. Since April 2018 he has been supported by his parents. The funders had no role in study design, data collection and analysis, decision to publish, or preparation of the manuscript.

### Grant Disclosures
The following grant information was disclosed by the authors:
The Alexander von Humboldt Foundation.

### Competing Interests
The authors declare that they have no competing interests.

### Author Contributions
- David Marjanović conceived and designed the experiments, performed the experiments, analyzed the data, contributed reagents/materials/analysis tools, prepared figures and/or tables, authored or reviewed drafts of the paper, approved the final draft.
- Michel Laurin conceived and designed the experiments, performed the experiments, contributed reagents/materials/analysis tools, authored or reviewed drafts of the paper, approved the final draft.

### Data Availability
All data are included in the Supplementary Information.

This is not the complete list of our sources for score changes; additional references are cited in App. S1.

## Supplemental Information

Supplemental information for this article can be found online at http://dx.doi.org/10.7717/peerj.5565#supplemental-information.

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
