# Peer review of "Phylogeny of Paleozoic limbed vertebrates reassessed through revision and expansion of the largest published relevant data matrix"

_PeerJ, doi:10.7717/peerj.5565_

## Round 0.1 · original submission · Major Revisions

Dear authors,

I have now three reports about your manuscript. All consider that it is an interesting contribution and recommend its publication in PeerJ. However, all agreed that the draft needs some more work to fix a series of weaknesses. The main concerns are focused on the lack of a clear delimitation of the aims of the manuscript as well as on its extensive and unnecessary length. Please, pay attention to these issues regarding the advice of the reviewers below and follow their recommendations step by step in order to improve your work. I particularly agree that the evolution of the middle ear may be the subject for a separate paper, where you would describe with more detail the information that you provided in this manuscript, taking into account the suggestions made by Reviewer 1 (Jason Pardo). Even though I expect that you modify the manuscript accordingly to one and all the suggestions made by the reviewers, I would like that you consider deeply carefully the following issues:

1- Figures should be improved. You need to include images with a better resolution or in focus and provide clear and complete labelling and scale bars for all of them.
2- Codification of ontogenetic (paedomorphic and juvenile) characters should be clarified and if possible, properly identified.
3- Consider the option to leave long discussions of some characters for forthcoming papers or move them to the supplementary material section.
4- Identify the characters that were added to the original matrix of Ruta & Coates, 2007 in the revised matrix; perhaps the use of different colours suggested by Reviewer 2 (Michael Buchwitz) may be appropriate.
5- Provide a stratigraphic control for the studied taxa and characters, as an independent criterion to test alternative hypotheses.
6- Consider the possibility of include additional figures to facilitate the access to more simplified and understandable trees.

I very look forward to see the revised version of this manuscript soon.
With my best regards,
Graciela Piñeiro

·

Basic reporting

The authors have done an awe-inspiring job of renovating the Ruta & Coates matrix, the largest matrix for tetrapods. Their attention to detail and justification for changes is phenomenal.

Experimental design

I don't have first-hand experience with most of these taxa, but I trust the author's interpretations. I suspect that some paelontologists might disagree with some of their findings or changes, but since they have been so diligent in recording their thought processes, everything is available for inspection and further discussion and interpretation.

Validity of the findings

The authors do a great job of discussing the empirical realities and philosophical difficulties of paleontological phylogenetics. Support and resolution decrease in many cases, which is the sad reality, but this places us in a vastly improved position to move forward scientifically.

Additional comments

I have three primary comments:

1) It's a shame more extant taxa aren't coded, to facilitate a combined morphological+molecular analysis in the future.

2) It's a shame that likelihood-based analyses weren't explored, to determine if this conflicted with parsimony, and changed outcomes.

3) It's a shame that morphological clock/stratigraphic analyses weren't performed, as the temporal ordering of character states can exert a strong influence on topological resolution and support.

However, these are very minor concerns compared to the wealth of information presented, and can now be explored in future studies, rather than necessarily in this one.

·

Basic reporting

The manuscript by David Marjanović & Michel Laurin on the phylogeny of Palaeozoic limbed vertebrates is well to very well written and organized. Considering the number of pages, style of writing and subdivision it has the character of monograph and fulfills the terms of an original research study.

Apart from a short section entitled 'Nomenclature' (which would also work as a sub-chapter of the 'Introduction') the manuscript follows the PeerJ standard style of subdivision.

The figures are relevant, of high quality and well described. Some of the detail photographs could do with an additional labeling of anatomical directions to make orientation easier (2-6, 24-35, i.e. through arrows). The phylogenetic tree schemes are well adjusted to fit on one page. Coloured branch underlays illustrating alternative positions of certain groups occur for the first time in Fig. 8 but a general legend of these underlays is provided for the first time in Fig. 18 - perhaps this legend should be included in the manuscript as a figure of its own (before Fig. 8).

All raw data for phylogenetic analyses are supplied in 3 separate NEXUS files and as a "human-readable" table in Appendix 2.

No sub-headings are used in abstract which covers most of the important aspects and data of the present study - with the exception of some notable details: (1) number of characters used in the original and in the revised analyses; (2) total number of steps found for the most parsimonious trees and those of the constrained analyses; (3) names of the significance/robustness tests; (4) the analyses carried out are described without the labels R1 to R12 used below - I recommend to include them in the abstract as well (to make the correspondence clear).

The introduction is comprehensive and provides a solid background about (a) early tetrapod evolution and (b) phylogenetic approaches chiefly based on the morphology of extinct animals, thereby the studies relevant for these topics and the present approach are cited.
* * *
Experimental design

The authors' main concern is the update and correction of an often used and cited cladistic dataset presented in a study by Ruta & Coates (2007) in order to carry out phylogenetic analyses which shed light on crucial problems of early tetrapod phylogeny. Based on (a) the authors' own study of relevant fossil material (primary original research sensu PeerJ author + review guidelines) and (b) excerption of many recent studies about Palaeozoic 'basal' tetrapods, this work is done with much scrutiny and the resulting data form a large part of manuscript file (appendices: 196 pages), not much shorter than the main body of the manuscript (text: 123 p.; figures + tables + captions: 75 p.).

All modifications to the character matrix of Ruta & Coates (2007) are listed in Appendix 1 whereas the "human-readable" version of the revised matrix is provided in Appendix 2. Scanning through this large data table it would be helpful to have these many changes undertaken by Marjanović & Laurin readily visualized - e.g. black font = unchanged; blue font = characters with changed definitions or merged characters; red font = changed individual cells and so on. I notice that this kind of colour-coding constitutes a quite large amount of work considering the c. 28,000 cells of the matrix (without additional taxa) but would make the whole table somewhat easier to countercheck.

Introducing the aims of their study the authors explain that "[their] work cannot pretend to solve the question of lissamphibian origins [...]. It merely tries to test whether the trees found by Ruta & Coates (2007) still follow from their matrix [...] after a thorough effort [of improvement]" (L. 298FF) - which is an understatement and in a certain way contradicted by the authors themselves in the following paragraphs where they announce the inclusion of further taxa ('operational taxonomic units') and employment of several analyses and tests to compare alternative hypothesis for the origin of lissamphibians. Perhaps the first statement should be reformulated and adapted to the somewhat more ambitious rationale this study got by the time it was submitted. Given the long list of issues treated in the various subchapters of the 'Discussion' section, the authors should mention in the 'Aims' section in somewhat greater detail which further problems of early tetrapod(omorph) phylogeny (apart from lissamphibian origins) they want tackle in their present approach. Despite these points, the research question can be regarded as well defined within the 'Introduction' section as a whole.

In the 'Methods & materials' subchapters entitled "Treatment of characters", "Modifications to individual cell" and "Treatment of OTUs" the authors provide the boundary conditions they have followed in their revision of the dataset by Ruta & Coates (2007). From time to time the authors' understandable urge to make their own comprehensive work intelligible to others leads them to the discussion of more general aspects of morphology-based cladistics. In their place I would rather refrain from lecturing the reader about the art and theory of character coding (and how others have failed in that business) and merely focus on the set of rules they now have rigorously imposed on their own revised dataset. The authors' choice of additional taxa is well justified in the subchapters 'Taxa added as new OTUs [...]' and 'Taxa that were not added' - I wonder, however, how much of an influence the inclusion and omission of certain lissamphibians (i.e. Chelotriton, Czatkobatrachus) might have had on the resulting lissamphibian relations.

In contrast with the chapters about dataset preparation the chapters 'Phylogenetic analyses" and "Robustness analyses" are relatively short. Nevertheless, the authors provide all necessary details for the replication of their analyses as well as the necessary NEXUS files to run them in PAUP (the software they used for all parsimony and robustness analyses discussed in the manuscript).

In order to compare the most parsimonious trees of their analyses (which support the lepospondyl hypothesis of lissamphibian ancestry = LH) to trees in agreement with two alternative hypotheses (temnospondyl hypothesis = TH and polyphyly hypothesis = PH), the authors repeat their analyses with and without the application of one or two constraints (see L. 1779FF). Although these constraints had the desired effects on tree topology, the authors do not explain how they there chosen and I wonder whether some alternative constraints (e.g. Doleserpeton closer to Eocecilia than Brachydectes to enforce TH) could result in more parsimonious (TH) trees than those found by Marjanović & Laurin.

In the last paragraph of the 'Methods' section the authors describe the use of three significance tests - Kishino-Hasegawa test, Templeton test and winning sites test - implemented in PAUP - through which the results of the unconstrained and constrained parsimony analyses are compared. They state that the null hypothesis (H0) is that "two tested trees are statistically indistinguishable" (= are equally well supported by the underlying data matrix (?)). At least in case of the Kishino-Hasegawa test it is also important to look at the alternative hypothesis (HA) because in cases where a known MPT is compared to a less parsimonious tree it might be more reasonable to employ the one-sided version of this test and not the usual two-sided version; accordingly the significance (alpha) levels or p values have to be adapted in order to avoid a false non-rejection of the null hypothesis (see e.g. Schmidt (2009) in Lemey, Salemi and Vandamme (eds.): The Phylogenetic Handbook: a Practical Approach to Phylogenetic Analysis and Hypothesis testing. Cambridge Univ. Press). I would strongly recommend that the authors check the respective null and alternative hypotheses and provide further references about the application of these tests and reasonable significance levels used in comparable phylogenetic and robustness analyses (alpha = 0.05 is not always a good choice).
* * *
Validity of the findings

The results of the phylogenetic and robustness analyses are presented in great detail - as data tables, tree figures and long written descriptions. Thereby the authors also include negative/inconclusive results (as required according to the PeerJ guidelines) - especially the non-significant test results.

A critical point in the author's interpretation of the significance test results (Table 4) lies in the application of several pair-wise tests based on the same cladistic dataset (e.g. R1 vs R2; R1 vs R3; R2 vs R3) due to the higher risk that with a higher number of test cases the null hypothesis of one individual is test wrongly rejected due to chance ("p hacking"). Therefore usually a correction, such as the Bonferroni correction, has to be applied, e.g. in case of 3 pair-wise tests only p <0.05/3 instead of p <0.05 leads to the rejection of the null hypothesis. Accordingly none of the discussed test results would be significant. (Perhaps I'm mistaken here - the authors should check how this matter is dealt with in similar approaches.)

In the following 'Discussion' chapter Marjanović & Laurin present a rather detailed and arguably balanced interpretation of the presented results. Thereby not only the problem of lissamphibian ancestry but various phylogenetic relationships among and within groups of limbed vertebrates as well as aspects character evolution are discussed in depth (which I found somewhat disproportionate because most of these points were not addressed in the 'Aims' chapter at the beginning).

The conclusion is well stated, linked to original research question and limited to aspects discussed in the previous manuscript section. That the authors emphasize the lepospondyl hypothesis of lissamphibian ancestry as an alternative to the often favoured temnospondyl hypothesis is consistent with the dataset and results presented. A detailed critical evaluation of the individual changes and omissions the authors made to the matrix of Ruta & Coates (2007) and whether biases towards or against a certain theory played a role in the presented revision is not in the scope of this review.
* * *
Additional comments

While the manuscript has many perks compared to others (clarity, consistency, thoroughness), conciseness is arguably not among them. I have some (non-binding) suggestions for increasing the digestibility of the manuscript by means of omission - especially in the longish 'Materials and methods' section: In "Which matrix are we critizing" the authors compare the official version of the Ruta & Coates (2007) matrix to an inofficial version. This chapter and "Changes by Ruta & Coates (2007) to the matrix of Ruta, Coates & Quicke (2003)" might be obsolete if the authors take the published study of Ruta & Coates (2007) and nothing else as a starting point of their approach - If necessary some of these points could be addressed in Appendix 1 (list of all corrections). For the same reason the re-analysis of Maddin, Jenkins & Anderson (2012) could also be spared. In the discussion part much space is given to the debate about internal relationships of major groups and the groupings of individual taxa. Given the larger perspective of the study set in introduction part it is perhaps not necessary to comment on every scrap of phylogenetic outcome, especially if the taxa sample of the discussed clades is small/not representative and if the vagueness of the groups found is rather high (low bootstrap values).

One issue is notably absent in the 'Discussion' section: The concordance between first occurrences in the fossil record and phylogeny (e.g. order of branching) is not discussed in a separate subchapter and generally rarely considered by the authors (an exception is chapter 'The interrelationships of Anthracosauria, [...] and Temnospondyli'). Since the stratigraphic fit provides an independent (posterior) criterion to compare alternative phylogenies I would have expected more about it.
The number of figures (35) is certainly not too large for a manuscript of 400 pages and the authors could even add some more to increase the accessibility of their paper: Especially the 'Discussion' part would benefit from (a) simplified tree subsets that illustrate alternative hypotheses/outcomes (see e.g. the types of graphics Sterling Nesbitt used in his 2011 monograph on archosaur phylogeny); (b) illustration of character evolution by means of simplified trees on which critical changes in important character complexes are mapped.

For some reason - to add a personal note (?) - the authors use quite often direct quotations of older and more recent works at the beginning of a new chapter or subchapter, usually without a further immediate comment whether they agree or disagree or why this particular statement is relevant for the respective part of the manuscript. In L. 1214, Conrad & Norell (2015) are quoted, but their statement "We recommend the inclusion of as many fossils as possible in any phylogenetic analysis." was made in a study about a diverse extant group (anguimorphs), i.e. in a context far away from the deep-time problems Marjanović & Laurin try to tackle. In L. 2464 a statement in obvious disagreement with authors' point of view is quoted uncommented (as if to provoke the reader's derision) whereas in L. 2680 the same authors are quoted with another statement which appears to be in agreement with the opinion of Marjanović & Laurin. Even though I appreciate attempts to make a dry scientific text more readable, my recommendation is to reduce the number of quotations and/or integrate them better in the main text.

Another point: The authors' style of commenting (the work of others) is often far from complimenting and some might find it unnecessarily offensive. In. L. 286 they write that a former cooperator "was unable to complete this large task" while it would have been politer and no less honest to state that this task exceeded the scope of his thesis. Is it really necessary to speculate about others' intentions as in L. 582FF or to imply/spread the suspicion that certain fossil specimens have not been studied first-hand as in L. 826FF or to lament about the additional work the authors themselves had due to the lacking exactness of others (L. 844FF) or to criticize that a once planned update study has not been realized (L. 1586FF)? A scientific manuscript is usually not the adequate place to usher exasperation about others. Usually all contradicting observations or principal differences in the manner how a problem is approached can be stated in factual language. (In case of doubt it is always safe to apply Thumper's rule: "If you can't say something nice, don't say nothing at all." - Thumper in Felix Salten's Bambi, 1942.)

In the first chapter of their introduction the authors make the following statement: "[A] matrix of morphological data is a matrix of hypotheses. The characters, their states, and the relationships between the states are hypotheses [...]; the taxa are hypotheses that rely on hypotheses [...]; and even given all these, each cell in a data matrix is still a hypothesis that relies on hypotheses of homology, ontogeny and preservation – some are close enough to observed facts, others less so" (L. 203FF). Notwithstanding the rigorousness necessary for any kind of morphology-based phylogenetics I would like to remind the authors of the problem that characters (as merely human-perceived subsets of the overall morphology), character states and relationships between states usually lack the falsifiability of other scientific hypotheses. Despite the many rules and conventions so duly cited by the authors there is no generalizable, let alone unambiguous, way to translate a complex morphology into a cladistic dataset. A good example are continuously distributed proportions or ratios which are coded as (ordered) discrete character states: You can measure them and try to define reasonable boundaries for these character states but despite everyone's best afford someone else will define different character states and in the end you might be convinced of your own scheme but cannot falsify those chosen by others based on the same measurements and equally reasonable assumptions. (Acceptance of the limitations of morphology-based cladistics can be liberating!)

Further points:

L. 1709: That Chroniosaurus is the only chroniosuchian from Russia for which a thorough description exists might not be entirely correct - There are two Russian monographs which also cover Chroniosuchus in greater detail: IVAKHNENKO, M. F. and TVERDOKHLEBOVA, G. I. 1980. Systematics, morphology, and stratigraphic significance of the Upper Permian chroniosuchians from the east of the European part of the USSR. Izdatelstvo Saratovskogo Universiteta, Saratov, 67 pp. and GOLUBEV, V. K. 2000a. Permian and Triassic chroniosuchians and biostratigraphy of the Upper Tatarian series in Eastern Europe. Trudy Paleontologicheskogo Instituta, Moscow, 172 pp.

L. 1928: 'Figure 8' and some of the following tree figures are directly mentioned in the text, I wonder whether that was an intentional break of the otherwise followed convention to put all figure references in brackets.

L. 2411: The reference for Ruta (2011) is missing.

L. 4295 - Seems unfair to attribute a lepospondyl relationship of chroniosuchians to Buchwitz et al. (2012) as they were not really interested in the above-anthracosaur-grade group which gave rise to the Chroniosuchia but merely added some further taxa and osteoderm features to the dataset of Schoch et al. (2010) whose analysis yielded such a lepospondyl affinity.
* * *
In summary, I have found only few points that require substantial reconsideration and thus would regard "minor revisions" without re-review as sufficient.

Michael Buchwitz, 19th February 2016

·

Basic reporting

This manuscript reports on a massive amount of completed work. I am concerned, however, that this manuscript contains several separate papers with separate aims. This has contributed both to the extensive length of the manuscript and to the complex and sometimes meandering nature of the text itself. A more focused approach is necessary, and it may be preferable to split this study into several papers.

There are a few issues in terms of standard practices:

- Some specimen figures lack appropriate scale bars. This must be amended if these descriptions are to remain in the text.

- The authors extensively cite recent grey literature (conference abstracts and unpublished theses) even in cases where work is definitely forthcoming. While the effort to acknowledge ongoing research by other groups is greatly appreciated (full disclosure, one of these theses is mine), I would rather see citations limited to cases where they are referring to final peer-reviewed versions of this research or, if these are not yet available, to ignore them until a peer-reviewed version exists.

- The authors began work with a matrix that is apparently not the published matrix from the original paper. This seems inappropriate given that the original matrix is available as an appendix in the original paper.

Experimental design

Although the analyses reported have been carried out proficiently, I am concerned somewhat about how the analyses line up with (or, rather, do not line up with) the rationale of the authors.

If the goal of the paper is to dig into the dataset of Ruta & Coates 2007, then the paper needs to focus on this specifically. A clear rationale needs to be developed, because the impact of R&C 2007 on wider phylogenetic treatments of early tetrapod phylogeny is not immediately apparent. Other workers have not discarded their own pet analyses in favor of R&C 2007.

If the goal is to attempt to create a grand-scale tetrapod tree in order to examine larger scale trends in tetrapod evolution, why use R&C 2007? If the R&C 2007 matrix is as flawed as the authors claim, why use it for this purpose? There are many alternate datasets out there, and one might think that the correct approach is to merge character and taxon sampling across ALL these analyses.

The authors explicitly use chimaeric OTUs, a practice I am not happy about. Melding taxa into a single OTU is an assumption of monophyly and, more importantly, an assumption of morphological congruity that is simply not justified in this case. Which codings come from which taxon? How are coding decisions resolved when they are in conflict? What are the effects of not coding these as chimaeric taxa? While this sort of treatment does occur in some other matrices (including the matrix they are criticizing), it is generally bad practice, and it is unclear why the authors criticize this practice when others do it, and then engage in the same practice.

The authors make a passing mention about paedomorphic and juvenile characters that are not coded for certain taxa. Which characters these are, and which taxa they were not coded for is not reported, nor is this practice justified. How was the juvenile status of these animals determined? How was the ontogenetic variability assessed? How were boundary cases resolved? I am aware the authors have done this in past analyses, but this has been strongly criticized on the grounds that many “paedomorphic taxa” and juvenile characters are not necessarily identified via objective criteria nor with reference to known ontogenetic sequences. I believe that this would be best accomplished as a second parallel analysis specifically examining the effects of ontogeny on early tetrapod phylogeny, possibly as a separate publication. However, I want to see an original version of the analysis without the ontogenetic recodings.

Validity of the findings

The authors present results of a number of phylogenetic analyses, including a number of constraint trees. These include some novel results. The problem is that the analyses reported here were designed to test one specific question (how different treatments of the R&C 2007 matrix affect inferred support for hypotheses of lissamphibian origins) but are interpreted in context of a number of additional questions (anthracosaur relationships, lepospondyl relationships, polarity of character evolution deep in the tetrapod stem, etc). I think this may be overinterpretation on the parts of the authors, given their admission that they have not gone back and resampled character matrices which adequately characterize morphology relevant to evolution in these parts of the tree. I would prefer to see the authors focus on the methodological strengths of their analyses (by demonstrating that there is only equivocal support for the temnospondyl hypothesis) rather than focusing on results which may be trivial inasmuch as a tree-building algorithm will eventually produce a tree and that tree will ultimately have topology.

The authors also dedicate a substantial amount of space to discussing evolution of characters on their tree, noting how much homoplasy they see in character evolution. They particularly focus on the evolution of the middle ear. This section confuses the middle ear with tympanic middle ears. This discussion would be improved by a model-based approach to ancestral state reconstruction as well as differentiation between the stem-tetrapod ear condition and secondarily reduced middle ears as seen in various living and extinct lineages. It would also be improved by a more detailed discussion of both middle and inner ear morphology across the fin-to-limb transition and within early tetrapod diversity (e.g. Clack 2003, Brazeau 2006, etc.). Adding these analyses and descriptions would further encumber this already large manuscript. In my opinion, this would be best approached as a second, distinct paper where the authors could implement a diversity of ancestral state reconstruction methods as well as delve into the comparative morphology of the middle ear in early tetrapods in more detail.

Additional comments

I am very glad to see the analyses that the authors present here, and their primary conclusion (that it’s too early to disregard the lepospondyl hypothesis of lissamphibian origins) is well taken. The core analyses and interpretations that deal with this specific issue are strong and are a welcome addition to the early tetrapod literature. The problems with this manuscript appear when the authors stray from these core analyses to try to either address larger questions of tetrapod phylogeny, when the authors attempt to produce ancestral state reconstructions for characters with widespread convergence, and when the authors try to incorporate new observational data for taxa which are either unfigured or poorly figured in the literature.

My preference would be for the authors to restrict the analyses and interpretation to the core study here (i.e. does R&C 2007 contain equivocal support the lepo hypothesis?) and leave a lot of the taxon addition, redescription, and discussion about larger trends to subsequent papers. I think this would both allow the authors to make their phylogenetic points more clearly and would allow other workers to access new morphological information and character evolution modeling and discussion more easily. The manuscript could be further streamlined by moving large discussions about character lists, taxon lists, and so on to supplementary material or appendices.

I would very much like to see a version of this manuscript after revision. I am willing to provide more detailed comments on the text at the authors’ or editor’s request.

---

## Round 0.2 · Minor Revisions

I was asked to step in as the Editor for this article and the submission was returned to the 3 prior reviewers.

You have attended to many of the comments from the first round, but there are still a few areas of the text that could be improved. Due to the length of the paper, it is important that the message follows though the whole text. Review 2 has suggested some minor areas that need further clarification and Reviewer 3 has given some excellent - and time intensive - comments that should be considered. It is recommend that the authors thoroughly read the reviewer comments, update the text where appropriate, or provide a detailed rebuttal as to why recommended changes were not made. Once the revised manuscript is returned, it will then be determined if further review is necessary.

·

Basic reporting

In terms of the basic standards of scientific reporting, the MS is outstanding. It is written in clear, plain English, and all of the methods, materials, and subjective choices made are well-described and justified.

Experimental design

Again, design is excellent. The authors have done a great job of justifying their platform and revisions.

Validity of the findings

The findings seem to be exceptionally robust, and should provide a platform for a great deal of additional future work.

Additional comments

I have reviewed the revised MS and the authors' rebuttal letter. They seem to have done an excellent job in revision. There will always be minor points of disagreement, particularly for a dataset of this size, but I think we have reached a good point for publication. In particular, the in-depth examination of each character, reduction of redundant information, and correction of suboptimal codings, is particularly important. Along with other recent work such as Simoes et al. (2016; Cladistics) on Squamata, hopefully we will see a new generation of morphological matrices conducted to such a high standard of quantitation.

·

Basic reporting

(This re-review is mainly based on the document file with tracked changes which I studied to see whether my points/suggestions from the earlier review have been taken into account.)

Figures: The coloured branch underlays used in various tree schemes are now labeled/explained in each figure caption which is an adequate solution.

Abstract: Has been changed to include some noteworthy details and abbreviations which provide a better correspondence to the following manuscript parts.

Raw data: The authors added font colours to their "human-readable" version of the character matrix in Appendix 2 in order to visualize changes made to the original version of Ruta & Coates (2007).
* * *
Experimental design

Aims of the study: Has been changed in certain details to be in better correspondence with the discussion section and looks okay now.

Methods and materials: Authors did a good job adding chapters explaing why they left out continuous characters and used parsimony analysis instead of other methods (i.e. maximum likelihood or Bayesian approaches). The use of constraints to enforce alternatives (temnospondyl hypothesis = TH; polyphyly hypothesis = PH) to the most parsimonious trees (which represent the lepospondyl hypothesis of lissamphibian ancestry = LH) is well explained in the revised version of the manuscript.

Robustness analysis: An explanatory paragraph regarding the applicability of the employed significance tests has been added but the interpretation of the results might still pose a problem (see section below).
* * *
Validity of the findings

Significance tests: The authors write about the meaning of the results of their robustness analyses (L.56: "Statistically, the TH (R2, R5, R8, R12) is not distinguishable from the LH or the PH, but the LH (R1, R4, R7, R10) and the PH (R3, R6, R9, R11) may be distinguishable from each other under both taxon samples and both reversibility settings. A reliable test is not available." In the Conclusions section they write (L.4237): "The bootstrap values along the “trunk” of the tree are low, and while the LH and the “polyphyly hypothesis” may entail statistically distinguishable topologies, the TH is indistinguishable from both (Table 4)."

I think the authors should explain to the reader in somewhat greater detail why only few tests yielded significant results (in accordance with the way these tests have been designed): Considering the dataset as a whole, especially the overall degree of consistency/homoplasy, and the low difference in tree length, the lepospondyl hypothesis is not a significantly better explanation than the temnospondyl hypothesis (it appears to be significantly a better model than the polyphyly hypothesis, though). If the difference in tree length had been somewhat larger (e.g. 30 instead of 10+ steps) or the consistency index somewhat higher, one of the hypotheses (LH vs TH) might have been be better supported and the null hypothesis (no significant difference between the two) would have been rejected. Arguably the non-rejection is not a sign that the tests are not working or appropriate (implicitly suggested by the authors in L. 2688FF) but that the dataset yields no solution which hypothesis is better (one could say: LH as more parsimonious alternative is not parsimonious enough...).

(Table 4 heading: "Random variation" might be misleading, perhaps better formulation: "The null hypothesis that the difference in tree length between the two alternative topologies is due to chance is not rejected (for significance levels of 0.05, 0.01, ...)."
* * *
Additional comments

Manuscript length/conciseness: The authors omitted some of the parts from the beginning of the Methods sections and simplified their explanation which dataset is revised for what reason.

Figures: The authors added some more schemes to increase the accessibility of their paper, namely simplified trees and tree subsets that illustrate published alternative hypotheses, constraint settings and outcomes as well as trees on which critical changes in important character complexes are mapped - looks good.

Chroniosuchians: The suggested Russian literature has been considered/cited.

Other points from the first review: have either been considered or they represented mere suggestions in the first place (not required changes).
* * *
In accordance with my earlier review and the comments above the study may be published as is.

Michael Buchwitz, 30th January 2017

·

Basic reporting

This manuscript, “Reevaluation of the largest published morphological data matrix for phylogenetic analysis of Paleozoic limbed vertebrates”, examines the strength of phylogenetic support in a published analysis of lissamphibian origins (Ruta & Coates 2007, here referred to as RC2007) for temnospondyl (temnospondyl hypothesis, TH) and lepospondyl (lepospondyl hypothesis, LH) origins of modern amphibians. The authors accomplish this by “correcting” the original matrix by reinterpreting character state delineations in some cases, taking steps to reduce nonindependence issues within the dataset, adding taxa, and revisiting issues of purported heterochrony in early tetrapods and lissamphibian origins. In this regard, the paper resembles several other recent papers by the authors, but provides insight into the construction of this specific dataset and, more broadly, the risks involved in taking a phylogenetic character matrix at face value. This study is strong where it compares the performance of their modified matrices with the original, and is weak when the authors use these results as the basis for discussion of broader trends in tetrapod evolution. Again, I strongly believe that a more focused approach that emphasizes the role of this study as a test of the effects of methodological differences between working groups would benefit this manuscript.

Many of my suggestions from the first round of reviews remain; the manuscript feels largely unchanged, it still lacks experimental focus, and the takeaway message is difficult to determine without substantial prior knowledge of what the authors are doing. I again strongly recommend that the authors reduce the length of the manuscript and narrow the focus of the studies contained within. As a rule of thumb, I suggest that the authors give careful consideration to the distinction between claims supported by the core experiments reported on in this manuscript and those which are not. The core experiments reported here show that the matrices used to support the TH contain latent support for the LH, and these differences hinge upon differences in how real anatomical data is transformed into artificially-categorical character data for phylogenetic analysis. This work is well-executed and, while I may personally disagree with some specifics, is scientifically defensible and publishable. However, these core experiments are augmented by a range of other comments, asides, assertions, and notes which very strongly assert specific interpretations without the associated data and rigor that would be expected in a standalone work. These deficiencies are due to the fact this manuscript attempts to do everything rather than a reflection of the quality of work these authors do (or the quality of the core experiments, which is high), but this is not the right format to present many of these ideas.

As an example, the authors make the attempt in the Materials and Methods to combine Caerorhachis and Casineria into a single operational taxonomic unit (OTU) for the sake of phylogenetic analysis. These are animals from different localities and different time intervals. They may fit vaguely into the same grade of tetrapod or they may not; prior studies have suggested that these are very different types of animals, but Casineria has not been described outside of a brief report. The only suggestion that these might be the same type of animal was raised in an unpublished thesis (Pawley 2006, which the authors cite extensively). The authors take this for granted and establish a single composite OTU encompassing data from these two animals. They make a vague effort to defend this by presenting a few photographs of Casineria, without the sort of anatomical description necessary to sufficiently interpret these figures and assess the authors’ assertions. If the authors were to have presented these interpretations in a standalone work, it would be rejected out of hand because the data presented here is far less than what is standard within the field. I am very familiar with both authors’ work and I know that they hold themselves and their colleagues to higher standards than this in standalone work. This highlights other issues as well seen throughout the manuscript: grey literature is repeatedly cited as an authority when it should not be, the authors usurp ideas from recent grey literature without waiting for those ideas to pass peer review, and strong assertions are made based on interpretations (or reinterpretations) of specimens or illustrations on very ambiguous bases. Again, this is only one example (there are many more throughout the paper).

On the subject of grey lit, the authors in their response also argue that abstracts are peer-reviewed and thus citeable, but this is not strictly the case; the observations and results reported in an abstract are not peer-reviewed and there is no guarantee that the content and conclusions are accurate or will remain the same before manuscript submission and/or acceptance for publication. The authors explain in their response that they use grey literature “only when they have to” but I don’t think this is sufficient justification. Until published, this work should be treated as if it doesn’t exist, because a version of record does not in fact exist. In most of these cases, I don’t think the grey literature cited is actually critical to the paper, and the authors’ discussion of other people’s results is well outside the realm of normal practice. This will not be taken well by some people, and I don’t think it does anyone (the authors or the journal) any good to violate these norms in this way. The manuscript is not weaker with these discussions removed.

In my initial review, I stated that I would prefer that the manuscript be split into several shorter manuscripts. I still believe this is the correct prescription here. I believe the core analyses could easily be introduced, reported on, and discussed in less than 35 manuscript pages, excluding figures and references, with a focus on the research problem (i.e. the Paleozoic origins of modern amphibians) addressed by the original paper (RC2007) and how differences in character handling affect those phylogenetic results. I think that the analytical core of this study is worthwhile and should be published, but the authors MUST first separate this from the speculations and incomplete studies throughout this manuscript which do not meet the necessary standards of rigor.

Experimental design

The experimental design of the core experiments is acceptable, but additional work presented throughout the manuscript does not meet standards of the field, including attempts to produce composite OTUs from taxa that have never been synonymized and have never been shown to exist in the same part of the tree, and reporting on ‘lepospondyl’ ossification sequences based on single specimens. None of these are essential to the core analyses of the paper, so their continued inclusion is unnecessary.

Some of the experiments (e.g. exclusion of ‘paedomorphic’ characters) are not fully justified and probably require experimental validation of both assumptions as well as methods. These experiments are not essential to the central project reported on here so this can be amended by simply excising them. The amount of work necessary to validate the approaches taken here would encumber this manuscript even more, with no direct benefit to the central research question.

Validity of the findings

As before, the central conclusion of this manuscript is useful and valid: there is latent support for the Lepospondyl Hypothesis in datasets that have been used to support the Temnospondyl Hypothesis. I have not scrutinized each individual coding and each individual character revision in detail, so I can’t comment on whether the authors have in fact “corrected” the dataset or have simply provided an alternate interpretation of the morphological variation present among early tetrapods. This distinction doesn’t affect the validity of the project (it’s a perfectly valid exercise either way) but the latter is less likely to be treated as superceding the original work. This has no bearing on the decision to accept or reject, but should guide the way the authors portray and weigh their results.

The authors also draw a lot of conclusions on what this expanded analysis means for broader trends in vertebrate evolution, with specific reference to taxa added by the authors here and/or outside the primary focus of the original study and dataset. Phylogenetic analyses will always produce results, and sometimes those results are novel, but not all results (including novel results) are meaningful. As I said in my first review, I would recommend that the authors make a greater effort to identify the research goals of the original study and the research goals of their study, and restrict their conclusions and interpretation to results directly relevant to these research goals.

Some of the discussion items and character ancestral state reconstructions are probably not meaningful within the context of the current study. I am glad to see that the authors removed the middle ear discussion, which was out of place, but there are other similar discussions that really should be reconsidered as well. For example, I would like to see the Caerorhachis/Casineria discussion completely dropped, as this can only be addressed with a more comprehensive description of these taxa that meets basic standards for anatomical description and systematic synonymy. There are a number of other discussions that the authors should consider extracting from this manuscript and using as a basis for future research.

Additional comments

General comments for the author
I still believe that the core analyses here represent a valid exercise and that they should be published. There is essentially nothing the authors need to add to this manuscript, but there is still a substantial amount of material which I believe needs to be removed or moved to separate manuscripts. In a study like this, it is tempting to draw extensive interpretations of the results and/or comment upon where such similar studies might go, but the authors should recognize the difference between topologies that are meaningful in context of the original research question (i.e. does the RC2007 matrix strongly support the temnospondyl hypothesis or not) versus those which differ from previously published analyses but are not directly interpretable without substantial extra work. The authors need to carefully weigh completeness against focus; here focus is probably more important than completeness, as this analysis is not meant to be a final complete treatment of early tetrapod relationships.
I have attached some section-specific comments to guide revision, but these are rough guidelines only and the authors should seek brevity over strict adherence with these suggestions. These are still broad in scale as opposed to specific action items.
As before, I look forward to seeing a revised version of this manuscript.


Specific Comments for Authors

Introduction –

P5: I would prefer to see the epigraphs removed. They are distracting.

P5-6: The review of why phylogenies are important is perhaps unnecessary. Worry only about the rationale for this particular study. You don’t need to justify the existence of the field to the readers.

P6: As one of the other reviewers pointed out in the first round of reviews, accusing other workers of error is needlessly confrontational. If the problem is that the RC2007 matrix has a lot of nonindependence, then this should be the focus of the study.

Background—
P 7-8: You throw a lot of taxonomic groups at the reader in quick succession. Either describe each of these groups briefly at first appearance, or treat the phylogenetic issues in terms of their implications for tetrapod diversification i.e. inclusive or exclusive tetrapod crown, nature of stem-amphibians, nature of stem-amniotes, etc. The latter would be my preference.

P8: Again, confrontational language concerning efforts of prior workers.

P8: Discussion of D. Germain’s contributions can be relegated to a statement in “Acknowledgments”

Aims—
The aims and rationale are not clearly developed here. Additionally, speculation on what readers and/or reviewers might believe is probably unwarranted; this reviewer, for example, is a little bewildered by the addition of 47 taxa, including taxa for which there may not be reliable descriptive data available.

Nomenclature—
I would prefer to see this entire section streamlined and relegated to the supplement. I will freely admit that I am not particularly interested in nomenclatural debates, and I suspect most readers feel the same way. If the authors are concerned about readers wanting them to justify their nomenclatural decisions, then directing them to a supplement is appropriate. In some cases, this might be better served by a standalone note in a journal that specifically handles this sort of thing.

On a specific note, the authors specify several features of anatomical nomenclature that are perhaps not consistent with general usage or with the underlying biology. “Orbit” for example indicates the opening in the skull that accommodates the eye. Other fenestrae can be continuous with the orbit, but that does not mean that these fenestrae are one and the same. Ultimately the lack of bone in these areas is likely induced early in development, and induced differently in the orbital region and the temporal region. Where possible, I suggest the authors distinguish between the orbit (which houses the eye) and the temporal fenestra (which ostensibly accommodates the origins of some/all of the jaw adductors). This functions perfectly well in mammals where the postorbital bar is incomplete, and I can’t see why that can’t work for the small number of lepospondyls (i.e. just Brachydectes) with an analogous condition. I recognize that the authors are most concerned here with employing simple descriptive terminology rather than using homology-based terminology that could end up being somewhat circular, but in cases where each component is clearly delineated (as is the case in both batrachians and in lysorophians) I don’t see the need for such anatomical agnosticism.

Treatment of characters—
I am not really happy with this section. It misuses some of the literature and simplifies a rather complex problem in ways that are often simply incorrect. To begin, the Brazeau quote cited here is taken out of context and actually means something different than what the authors seem to intend. Brazeau is specifically talking about conditional coding and multistate characters, specifically in terms of how different approaches to coding will change how treelength is calculated. He is not precisely addressing the issue of character nonindependence except in the specific case where absence of a single structure may reappear as a state in multiple characters. O’Keefe and Wagner (2001) describe post-hoc methods of searching for groups of characters which contain strongly cohesive intra-group signal but strongly conflictive inter-group signal. They make mention of different ways that characters might not be independent of each other, but they do not delineate specific ways in which characters may violate assumptions of independence. Elsewhere, grey literature is cited in a substantive manner (full disclosure: this is a portion of my masters thesis, which is part of an in-prep manuscript), and developmental genetics studies are mentioned but are actually employed in a slightly less than accurate manner. Specifically, both the Kangas & al. and Harjunmaa & al. papers are discussing impact of NF-κB signaling on several “distinct” features of tooth morphology (cusp number, cusp morphology, and shape), i.e. when you disrupt NF-κB signaling, you disrupt a few features of both primary and secondary enamel knot formation and persistence. This is a single signaling system that plays out in four dimensions (space plus time) to create a finite range of morphologies of the complete tooth. It would be incorrect to describe these as correlations. It would be more correct to describe these as conditions where the character is incorrectly conceptualized or atomized. This also ought to be distinguished from pleiotropy, where a gene product or pathway is involved in patterning, morphogenesis, or differentiation in multiple distinct systems and thus disruption of the pathway can have multiple distinct effects on the phenotype. There are both biological and operational reasons to consider these violations of character independence separately, but this is also not discussed by the authors here. Similarly, this is very different still from correlations associated with, say, correlated evolution of functional systems (e.g. simultaneous exaggeration of origin and insertion structures for a single muscle). And so on and so forth. Ideally the authors would also examine how violations of character independence do in fact appear in molecular data and how these are handled in molecular analyses. Molecular data is not immune to these problems (e.g. secondary structures in ribozymes as well as secondary and tertiary structure of proteins), and the solutions that have been employed by molecular workers do not necessarily align with the solutions employed here.
I am not bringing any of this up to convince the authors to write more here. I do not think that is a worthwhile exercise at this time because this issue requires serious consideration in a paper of its own. I am trying to convince the authors to take the complexity of this problem more seriously and to consider leaving out a discussion that misrepresents the state of the literature, misrepresents the problems involved, and may mislead readers. I do not imagine that this current manuscript will be cited as an authority on how to handle violations of character independence in morphological datasets, so the authors’ use here of space might be better spent on other things.

The authors also rehash an earlier argument of theirs that the data collection and handling must accommodate concerns of heterochrony (Ontogeny discombobulates phylogeny section) citing a paper by Wiens and colleagues. It is important to note that the character deletion method of Wiens and colleagues did not actually solve the phylogenetic incongruity issue at all, and they still recovered “paedomorphic” and “peramorphic” clades, and Wiens et al. (2005) don’t really suggest this approach as a long-term solution to heterochrony issues. In addition, this specific situation (salamanders) is probably more complex than simple heterochrony and is not really a good exemplar for how to handle ontogenetically variable morphology, particularly in early tetrapods, which seem to have less stereotyped ontogenetic trends. Moreover, we do not have particularly good postnatal ontogenetic sequences for most early tetrapods, so it is unclear how the authors identify these sorts of things. The lack of a clear methodology supported by data and analysis makes me worry that this may just be speculation. Again, I feel that this deserves an independent treatment where the relevant data can be reviewed, compared, and employed towards meaningful ends.

Tail fin skeleton (p19)—
The authors combine codings from an early tetrapod tail of uncertain affinities with an existing OTU that does not exhibit this morphology. This decision makes no sense; Archeria lacks the condition described by Clack 2011, and there is no reason to believe that the animal reported by Clack 2011 had any of the morphology we see in Archeria. The authors should reverse this decision. It will likely not substantially change their results.

Modifications to individual cells (p20-25)—
Move this section to the supplement and remove editorializing, particularly criticisms of R&C2007’s methodology.

The albanerpetid neck (p 25)—
What is the point of the comparison between ‘microsaurs’ and albanerpetids? Doesn’t this essentially assume relationships a priori? Inference of homology by comparison is not unreasonable, but the authors explicitly reject that approach elsewhere.

Ignored information (p. 25-26)—
The authors discuss grey literature on aistopods and Crassigyrinus but do not employ this data in the analysis. My preference would be to remove this section in its entirety.

Treatment of OTUs (p 26-28)—
The authors use a composite OTU for Protorothyris and Paleothyris, even though it would be easy to simply recode this OTU from scratch. It’s not entirely clear why the authors review the history of Dendrerpeton, but it would again appear that it would be easy enough to limit this OTU to the articulated Joggins animal. Additional OTUs are established later as well, and I’m not really entirely certain why.

I am not convinced that the authors understand the importance of Rhynchonkos in the analysis. Rhynchonkos has historically been considered to represent a transitional state between general ‘microsaurs’ and early caecilians (represented by Eocaecilia). Thus, inclusion of this taxon is specifically intended to test the Polyphyly Hypothesis. Szostakiwskyj et al. (2015) redescribed the collection of skulls attributed to Rhynchonkos and found that three species were present. Caecilian-like characteristics were found to be roughly distributed among all three taxa, but no one taxon has the full suite of “Rhynchonkos” characters sensu Carroll & Currie (1975). Limiting the OTU to the type skull is only half the job; the other two taxa (Dvellecanus and Aletrimyti) do need to be addressed as well. I am not a huge fan of increasing the taxon sample here, but this is the one case where I think it could be justified, given that these three taxa are explicitly central to one of the hypotheses being tested.

I do not understand the authors’ comments on Brachydectes. I am aware that the authors have attempted to reinterpret the skull based on drawings in the literature, and that this section exists in context of those prior works. However, Pardo & Anderson (2016) is the most extensive and up to date description of this animal. I am glad that the authors agree with our interpretations of the skull of this animal (we put quite a lot of time into that study) but that the authors feel the need to state that they have confirmed our conclusions by consulting a few casts is strange. Just refer to the recent description which you are following for cranial morphology and don’t worry about whether or not you have sufficiently explained the history of your past interpretations or the steps you personally took to decide that you concur with the published description.

There is no debate as far as I am aware about the status of the Speiser Shale microsaur. It is Proxilodon bonneri. It does not belong to Euryodus. It has not formed a monophyletic grouping with Euryodus for several decades, and Proxilodon is the valid genus name.

The authors argue in an unprecedented way that Casineria and Caerorhachis are the same animal or otherwise are closely related. Published phylogenetic analyses do not place these animals as close relatives, nor are they known from the same localities, nor are they the same age. Compatibility of phylogenetic codings for a small number of characters is not sufficient to justify wrapping these two taxa into one OTU, and any sort of taxonomic revision would require a separate in depth treatment of the anatomy of both taxa. Moreover, I’m not sure inclusion of these animals actually matters with respect to the central research question of this study, so their exclusion is essentially trivial.

Similarly, it is unclear why the authors choose to unite Hapsidopareion and Llistrofus into one OTU rather than treat them separately or work with only one. The description of Llistrofus is more complete and extensive, so I’d recommend sticking with that one, but really the authors should just stick with whatever was in the original publication.

Added taxa (p 35-45)—
This section should be moved to supplement and pared down. Taxa do not need to be introduced in detail.

Taxa not added (p 45-46)—
There is no reason to include this section.

Section on maximum parsimony (p 49-50)—
This section can be removed in entirety. Parsimony is industry standard so this choice doesn’t need to be justified in this manner. In addition, some of the argumentation used here is probably incorrect or otherwise does not justify the decision to use nonparameteric methods exclusively.

Reanalysis of Maddin, Jenkins, & Anderson 2012 (p 51)—
Why is this section included? We are 50 pages in and now we’re being introduced to an analysis of a completely different dataset. Why? Please reconsider this inclusion.

Analysis results (p56-87)—
The authors then present a full 30 pages of manuscript discussing fine details of each tree recovered. I would prefer to see the authors focus on the forest rather than the trees. Is the purpose of the study to examine support for different hypotheses of amphibian origins? If so, then focus on where amphibians are recovered and remark only upon major reorganization of the tree. Fine changes within groups (e.g. the interrelationships of diverse groups like recumbirostran microsaurs or amphibamid temnospondyls) are probably not meaningful, especially considering that this dataset is limited in terms of scope of both taxa sampled and characters sampled. If the purpose of the study is to examine the stability of deeper parts of the tree, then this needs to be highlighted, but fine interrelationships of OTUs is not of particular interest to this reader in this context.


Discussion—
(P87-90)
The authors begin the discussion by criticizing their resulting matrix and by stating a number of caveats. I’m unsure of what this is meant to accomplish and whether this is an appropriate place to criticize the makeup of the matrix. Specifically, the authors complain about the anatomical makeup of the matrix. This does not seem to be the right place for this, and this section would be improved by editing the text to be somewhat less despondent. RC2007 didn’t employ many neurocranial characters, but then the neurocranium of most of their taxon sample had not been well-studied at the time, particularly for temnospondyls and ‘lepospondyls.’ Otherwise, the focus on the skull is due to the richness of this dataset and ease of coding information from sutural patterns. There is also no evidence that postcranial characters are better at resolving phylogeny than cranial characters are (the neurocranium is a different story). The neologism “craniocentrism” can probably be dropped entirely; this usage seems to imply bias and malpractice, which is not the case.

Phylogenetic relationships (p90-127)—
This section is honestly too long to give the point by point discussion that it requires. Personally, I would like to see this section dramatically reduced (10 pages maximum, 3 pages recommended, 3 paragraphs ideal) and address only general trends where they relate specifically to the aims of this study. There is a lot of speculation here, some of which takes the form of hinting at privileged information presented by others at conferences. This is particularly apparent to me in the authors’ section on ‘lepospondyls.’ The authors hint very strongly at lepospondyl polyphyly, including some very specific possible relationships of specific groups of lepospondyls, with reference to grey literature by third parties (conference abstracts and an unpublished thesis) reporting on extensive primary anatomical work on these groups. However, the authors themselves present no such data here and their results do not strongly hint at lepospondyl polyphyly. I recognize that this is a case of excitement by the authors about work in progress rather than an attempt to scoop ideas generated from substantial anatomical work, but the authors should be more careful here both because this could be misinterpreted and because this is likely to confuse readers who are not “in the know.” In other cases, the authors dedicate substantial text to poorly-known taxa which are probably miscoded due to discrepancies between the published descriptions and the actual specimens. Again, this is likely to confuse readers and does not improve the manuscript.

Reversals in character evolution (p127-132)—
The authors here review evidence for re-appearance of several bones in the early tetrapod skull in various lineages. This section contains a lot of speculation associated with specific instances but the importance of this pattern is not discussed. One possibility is that the conceptual framework within which we understand the cranial exoskeleton is fundamentally incorrect, so our understanding of gains and losses misrepresents the actual situation. Another possibility is that the backbone of the phylogeny presented here is incorrect, so these characters never reverse in the first place. A third possibility is that these characters violate the assumption of time-reversibility employed in ancestral state reconstruction (see e.g. Goldberg & Igic, 2008), and the tree is correct but the ancestral state reconstruction is not. I would not be particularly surprised if all three are true in different parts of the tree. However, this really deserves its own treatment as a separate study (or research program). It would be more appropriate for the authors to summarize these phenomena briefly (~1-2 paragraphs) and acknowledge that this may be methodological rather than biological.

Preaxial polarity in limb development (p 132-134)—
This section appears to specifically be a retort to recent work by Frobisch et al. (2015) and stands apart from the remainder of the manuscript. The authors introduce new observations (50 pages into the discussion) and speculatively reinterpret other work. I am not convinced by this section at all. I am convinced that a study of the ossification sequence in the limbs of Sauropleura (and Ptyonius, not mentioned here) would be a substantial contribution to the field, but a brief reference to a single specimen is not the same as scoring dozens to hundreds of specimens (which are available for study) for ossification presence/absence. Additionally, the issue of limb ossification polarity is not limited to paleontology alone; there is a developmental genetics literature on this subject that must be addressed as well. In all, I would very much like to see the authors complete a study on this issue, but this section is simply not appropriate here. My suggestion is to remove it entirely.

Errata (p137-139)—
The authors here suggest corrections in papers of theirs published in other journals. This is not an appropriate venue for this. Errata should be sent to each journal where it can remain connected with the original text. This section should be removed.

---

## Round 0.3 · Major Revisions

Dear Authors,

Due to its length - 432 pages! - this paper has been difficult to compare and contrast between iterations/reviews to identify where updates have been made. As such, it is very important to obtain insightful peer review. The most recent iteration was sent out for further review by Reviewer 3, as I believe they have spent a lot of time and effort reading the paper and presenting detailed comments/questions/concerns. Based on the most recent review, there appears to still be some dissatisfaction with the amendments made to the text/analysis.

Because the paper has been through a number of iterations and because Review 3 still has concerns over the text/analysis, I sent the submission to another PeerJ Editor to get a second opinion on how to proceed. We both agree that Reviewer 3 is being fairly even-handed in their detailed evaluation of the work and think the authors should incorporate more of the constructive feedback provided. To ensure such changes are made, we agreed to assess the paper as 'Major Revisions'.

In particular, we would like to authors to focus on three major topics:

1) Data coding and analysis: we agree that the authors need to reconsider using composite taxa coding and that there should be some type of sensitivity analysis in terms of modeling assumptions. This would include the algorithm used (ACCTRAN/DELTRAN) and also ancestral state reconstruction (parsimony/likelihood).
2) Toning down of wording: in particular, making broad sweeping generalized statements that are not unambiguously supported by the data. Please be more conservative in the interpretation of the results.
3) Unpublished work: we also ask the authors to reconsider using unpublished work in their study e.g. theses, or papers in prep/press. Although this is not unethical, it is not very collegial.

Further, it would be of great benefit to further reduce the length of the paper - where possible - to make it more digestible.

We hope that making such changes to the paper will help to strengthen the study, making it more valuable to the scientific record.

With many thanks.

·

Basic reporting

Unchanged.

Experimental design

Unchanged.

Validity of the findings

Unchanged.

Additional comments

I have reread the manuscript by Marjanovic & Laurin, now retitled “Reproducibility in phylogenetics: reevaluation of the largest published morphological data matrix for phylogenetic analysis of Paleozoic limbed vertebrates.” Some of the changes (e.g. the new aims sections) improve clarity of the manuscript. A number of other aspects of the manuscript remain either unchanged or unaddressed, despite in some cases representing critical flaws in the analysis or presentation of results. I remain unconvinced by some aspects of the authors’ justification for leaving these flaws unfixed, but this manuscript has now been in review for over a year and a half and it seems ridiculous to further delay a decision. I have provided a brief set of general concerns as a record of my thoughts on the matter, but I don’t expect these will be addressed and thus do not anticipate another round of reviews will be necessary.

I still strongly believe there are some problems with the manuscript as it is currently written. Some of these are aesthetic and should not be considered an impediment to publication, some are disagreements about the norms of publication and collegiality and should be given more serious consideration even if the authors ultimately do not agree. A few are more serious concerns about methodology and directionality of inference in paleontology, but while I consider these choices to be methodologically inappropriate, changes are unlikely to affect the results as they have been presented.

I strongly disagree with a few aspects of the analyses completed here, and feel strongly that these should be amended to the standards of the field before publication. I am not particularly concerned about the addition or not of Aletrimyti and Dvellecanus (my comments there were perhaps misinterpreted), but I am concerned about the choice to create composite taxa in a number of parts of the tree. Coding a composite taxon means that you are inferring the presence of a species or individual where all the coded character states were present in the same organism. In several of the cases here there is both serious reason to doubt this as well as serious reason to think that the character handling may have an effect on the results. I am particularly concerned about the incorporation of morphology from a generalized stem-tetrapod tail into codings of derived embolomeres, and of composite coding of two partial early (likely stem) tetrapods which require further descriptive attention. Composite coding of several plesiomorphic ‘microsaurs’ may also have small local effects on the phylogenetic relationships of that group but I’m unsure of whether this will have broader effects on the tree and interpretations thereof. I’ve been consistent in my stance that the use of composite taxa is not really appropriate from my first review in February 2016, so I’m a little disappointed that we have reached this point without this issue being addressed.

I am still unsure of many of the interpretations of tetrapod phylogeny and evolution that are presented by the authors in the discussion. Individual character reconstruction on a large sample of trees such as this is unlikely to be meaningful. In the case of digit ossification sequences, I still maintain that it is preferable for the authors to generate high quality sequence data from a large sample of animals rather than assume that this information can be supported or falsified by a single individual. In the case of character reversals, ancestral state reconstruction is nontrivial and the assumptions of your methods strongly affects your results. Factors as simple as employing ACCTRAN versus DELTRAN in a simple parsimony ancestral states reconstruction (which itself is not preferable) have the potential to affect whether you will find “surprising reversals” or expected convergences on a given tree. Again, this feels to me like the authors are attempting to do too much and as a result are giving only cursory attention to problems that deserve more complete attention. There is a difference between saying “here are some observations that deserve greater attention” and saying “these are our interpretations of our results, and we therefore dispute the current state of the literature” and in some cases I believe the authors veer too much towards the latter.

In terms of comparing trees and analyses, there are obviously many levels at which the authors could address this problem. I’ve suggested that the authors stick to the approach that requires the least additional work (i.e. “does the dataset, when treated differently, contain latent support for the Lepospondyl Hypothesis”). This has the narrowest implications but also requires the least work and is most strongly supported by the results already presented by the authors. I strongly believe that further/broader conclusions will require more extensive work, which is why I have suggested taking a more conservative approach. Generalization in my opinion would require testing of randomness of the purported corrections, testing the statistical significance of differences between trees, and calculating metrics of overall tree similarity between the RC2007 tree and the ones reported here. If the authors are arguing that the methodology of RC2007 hid ambiguous evidence in favor of other hypotheses, the data presented are sufficient to support that, which is why I have suggested restricting the conclusions to this more defensible ground. I cannot suggest offhand a list of specific analyses which must be completed, but I can say that the analyses presented do not support all the inferences made.

I still feel that references to recent theses and meeting abstracts (some of which are currently in review or in press) are still inappropriate. Yes, these datasets are out there and these ideas are out there, but aggressively mining works in progress and responding to those ideas before a version of record exists is generally not considered good collegial practice. Having an answer to everything is not necessary, and having an answer in the literature before the version of record is even published is even less necessary. There is literally no impact on the paper itself, so I am unsure of why the authors are so adamant about keeping these references in the paper and discussing the preliminary results and ideas contained therein.

That said, I do recognize that we are entering into the realm of diminishing returns. The effects of changes to the analyses are unlikely to be substantial without substantial additional effort, and the authors have been adamant enough about resisting suggested changes that it seems unlikely the issues I’ve brought up will be addressed. The argument that the necessary experiments would take two months is not particularly compelling to me (my work intersects with a field where reviewer-requested and editor-required experiments may take upwards of a year or even two) but I am also willing to be realistic about the impact of these experiments on the overall shape of the manuscript.

---

## Round 0.4 · Major Revisions

The authors have Appealed the last decision and offered to provide a revised manuscript that addresses the concerns that were raised in the last round of review (which is included below). I have been asked to consider this appeal and so I am issuing this "Major Revisions" decision to give the authors the opportunity to upload a revised manuscript, a tracked changes document, and a detailed accounting of their rebuttal/revisions. I will then consider this new revision further.

· Appeal

Appeal


· · Academic Editor

Reject

The previous editor is now unavailable so I have been asked to step in. I have examined this paper's history of reviews and revisions and rebuttals. My judgement is that there has not been sufficient responsiveness to reasonable requests by the third reviewer. Two reviewers' shorter reviews were satisfied but this last reviewer put more effort into their reviews and drilled to the core of some important issues. The revisions indicated some reluctance to make revisions, especially re-analyses, and in the latest rebuttal the excuse is made that the paper needs to be accepted w/o further delay (i.e. analyses requested) because the first author needs publications for their career and grant proposal(s). While I am very sympathetic to this general problem that early career researchers, in particular, face, this does not excuse the paper from needing to meet the standards of peer review.

Thus the final decision on the paper needs to be based upon judgement of what those standards are. I have considered this carefully and decided that, after 4 rounds of review/revision, we cannot justify further review for reasons noted above. We cannot keep reviewing until fatigue wears out the non-authors and the authors get what they want without due diplomacy. This, in my view, would circumvent a fair peer review process. I agree with the third reviewer's main points, in particular, regarding inappropriate discussion of *recent* (not 2006) unpublished work by active researchers, and methodological choices or assumptions that are not acceptable and may be biasing the analysis. At some point a journal must stop burdening its editorial and review system if a paper is not meeting the standards of the journal, which in this case include reasonably sound science and professionalism. Hence at this point I unfortunately must render the decision to reject the paper.

I expect that the authors, whom I respect, will be unhappy with this decision for reasons noted in their own recent rebuttal, and given how long the peer review/revision process has been with this paper. The preprint of the paper, including running commentaries, will remain and can be cited in grant proposals/CVs so that evidence of research effort (certainly large effort in this case; but arguably a risky "putting all eggs in one basket" approach) is discoverable. I am sympathetic to concerns such as those but have been forced to make a difficult decision and weighed the options heavily. It is one of the harder decisions I have had to make as an editorial board member at 3 journals over >10 years. I hope that the reasoning, if not accepted, is respected as I have respected the effort put into this study and all other factors demanding attention from an editor and colleague.

John Hutchinson
Academic Editor, PeerJ

·

Basic reporting

see below

Experimental design

see below

Validity of the findings

see below

Additional comments

I have looked through the tracked changes in the modified manuscript. I am happy to see that the authors have taken a first step towards addressing the issues I have been pointing out for a year and a half now, but I still feel that their efforts here are incomplete. I am a bit perplexed as to why the authors have elected to add additional sections “testing” the phylogenetic analyses of unrelated papers and discussing reported results, after I already stated that I felt the discussion of the Maddin et al. 2012 paper and results was out of place. The underlying problem, again, is that the authors are attempting to do everything at once (additional paragraphs and sections have now been added!) instead of restricting themselves to a more modest scope. I apologize in advance for the length of this review, but I think all these points need to be made.

I am glad to see that the authors have accepted the need to treat Casineria and Caerorhachis separately, and the results of the subset of analyses where they did this confirm that additional work is still required before these can be treated as the same animal in phylogenetic analyses. As such, it is clearly necessary to treat these animals separately in the remaining analyses, as I have been saying since my first review in February 2016. The authors say in their response that “the burden of evidence lies on those who wish to show that Casineria is not identical to Caerorhachis” which is completely counter the standards of the field. Both are currently recognized as valid taxa, so by the fundamental standards of the field of taxonomy, the burden of evidence is on those who wish to synonymize them. Such work requires anatomical description and taxonomic revision, which the authors have not done. This is itself a completely separate problem from the results of phylogenetic analysis and of the overall relationships of Casineria and it is unclear to me why the authors confound those two issues.

I am still not convinced by the authors’ arguments and results with respect to the decision to incorporate the Five Points tail into the character codings of Archeria. There are tails of Archeria and supraneurals are not preserved. The Five Points tail is not phylogenetically discernable on its own; yes, it has embolomerous vertebrae but there is no inherent reason to believe that this alone is a synapomorphy of a clade. Unlike other examples that the authors list, this is a case where the monophyly of (Archeria+Five-Points) is assumed a priori rather than the result of phylogenetic analysis and non-phylogenetic taxonomic revisions.

While I am personally relieved that my own embargoed results are no longer in danger of being reported on before publication, the authors have done nothing to amend similar cavalier approaches to work in progress, prep, or press by Drs. Porro and McHugh. I understand that Dr. Pawley has been out of the field for some time now and is not actively working towards publishing her thesis research necessitating citation of her thesis for certain points, but Drs. Porro and McHugh are both very active in the field and the authors are citing very recently presented research (from 2015!). If the authors insist on referring to those works in progress, then my gut feeling is that it would be appropriate to ask written permission to refer to that work. It seems easier to simply bypass this by citing only versions of record. It would be irresponsible and self-serving of me to object to this only when it was my own work that was under discussion.

In terms of Sauropleura and sequence polymorphism, “ontogenetic trajectories” are themselves just trends. Regulation of timing of developmental events is not as tight as the regulation of these events themselves, and there is generally some variability in timing in living populations, which is why prior studies of fossil limb ossification sequences have required assessment of large numbers of specimens. Furthermore, in cases where a digit is unossified, it is not always clear whether you’re looking at a situation where that element had not ossified yet at time of death, or whether you are looking at an animal that simply did not have that digit either due to a developmental anomaly or due to injury. This is why surveying large numbers of specimens is absolutely necessary. One specimen is not a trend. Given that premature chondrogenesis in digits I and II in salamanders is driven by a novel salamander-specific gene (Prod1) that regulates expression of the chondrogenic transcription factor Sox9 and signaling ligand Bmp2 under control of retinoic acid signaling in the anterior forelimb, the authors are arguing de phylo that this entirely novel form of cross-talk between signaling pathways either arose multiple times in parallel or was lost to pseudogenization in three independent lineages (amniotes, frogs, caecilians) while remaining entirely absent among sarcopterygian outgroups. That is an extraordinary claim which requires extraordinary evidence. One specimen that is missing one digit is not extraordinary evidence. A large-scale study of limb morphology and growth trajectories in Sauropleura AND a comprehensive morphological and phylogenetic treatment of Sauropleura AND genetic dissection of purported “preaxial dominance” (if present) in Neoceratodus is the minimal effort necessary to make the argument that preaxial dominance is the primitive condition for tetrapods. I think these studies should be done and I would be happy to see these authors do it, but these conclusions must be made after collecting the data, not before.

I am not particularly convinced by the authors’ justification of their approach to ancestral character state reconstruction. They clarify that they are simply looking at Mesquite output of basic parsimony econstruction here. There is enough work in recent years that has shown that simple ancestral state reconstruction is often misleading if not incorrect, and that more explicit model-based investigations are necessary. Good examples of this are the models of Goldberg & Igic (2008) and Maddison et al. 2007 among others. Claiming high rates of highly-specific convergence and high rates of unexpected character reversals requires more rigorous testing than is presented here.

In terms of the scope of this work, I have repeatedly stated that this work’s strength is that it provides an assessment of how much of the matrix of R&C2007 is either mistaken or ambiguous, and the effects of that ambiguity on the resulting phylogenetic conclusions of R&C2007. I’m sure some of these are true mistakes, but many are ambiguities or judgment calls. The reality is that morphological phylogenetic analyses (all of them, including mine and including the authors’) are full of such judgment calls and ambiguities, either in the coding or even in the framing of character states. In fact, this even applies to molecular analyses, where base pair calls as well as contig alignment calls contain a degree of uncertainty. Unfortunately, the format of this study is such that it is unclear which differences in coding between R&C2007 and the authors of this manuscript are true errors versus which are judgement calls versus which might be errors by Drs. Marjanovic and Laurin. Furthermore, there is no effort to produce generalizable results that might be of use to morphological phylogeneticists. For example, are certain types of character more prone to miscoding? Are certain types of character codings essentially author-specific judgment calls? Does this mean that the tendencies of certain authors to exclusively work in a specific taxonomic group AND to have certain coding biases will potentially create unexpected phylogenetic confounds? Ideally this sort of work would involve additional experiments to constrain those interpretations and allow generalizations, but the current analyses are sufficient for a more limited discussion.

The work is less strong in evaluating the different hypotheses of lissamphibian origins. Lissamphibian origins is major focus of research among paleontologists working on early tetrapod evolution, but the state of the field at the time of R&C2007 was that the support for different hypotheses of lissamphibian origins was extremely weak due largely to the substantial morphological distance between Paleozoic tetrapod groups and the earliest representatives of each lissamphibian order. As a result, the past decade of research has focused intently on collection of new data. This includes studies of modern lissamphibian histology and development (e.g. Maddin & Anderson 2012), ontogeny (Maddin et al. 2010), and new sources of data (particularly the braincase) for lepospondyls, temnospondyls, and modern lissamphibians. More importantly, new evo-devo studies from a number of labs are beginning to make some real sense of lissamphibian origins more generally, bringing this research problem more in line with more mature fields of research such as dinosaur origins or mammal origins. Preliminary phylogenetic treatments that incorporate this new understanding of lissamphibian and early tetrapod morphology have shown that inclusion of these data has the potential to dramatically change our interpretations of both the phylogenetic relationships of lissamphibians as well as the broader relationships of these early tetrapod groups in question. As such, if the goal is to evaluate lissamphibian relationships in a definitive way (rather than to ask methodological questions about reproducibility in state coding) then non-inclusion of these new characters is a glaring oversight that many workers in the field will recognize immediately, particularly when the central conceit of this manuscript (that it has corrected all the problems in other groups’ papers) is a confrontational one.

Where this work is not strong is in presenting a cohesive phylogenetic analysis of all early tetrapods. As I stated above, the field has advanced substantially from where it was when Ruta and Coates were building their dataset, and this study does not incorporate the new morphological data (and new understanding of older morphological data) that have been reported since. The authors state in their response that the rapidly changing state of the field might make the current work obsolete unless sections describing work in progress by other groups are included. I would instead argue that it is the non-treatment of rich character sets which have been built and improved upon in the past 10 years (Maddin et al. 2012 is a key example) that threatens the longevity of this study. More to the point, if the goal is to produce a phylogenetic analysis of all early tetrapods, then basing the entire analysis on a single dataset which has not been widely adopted by the field does not seem to be the ideal way to get this done.

This is why I keep emphasizing that the authors need to make a serious effort to figure out what this paper is about. The analyses presented here are sufficient for a paper of more modest scope. They are not sufficient if the goal is to present an authoritative summary of early tetrapod phylogeny writ large or to offer a final resolution of the lissamphibian origins problem. I think many of the problems with this paper stem from the fact that it attempts to do the latter, and I think the authors will be somewhat disappointed if they hope this will be treated by the rest of the field as a landmark study. Ultimately it is not my place as a reviewer to dictate the appropriate scope (that is for the authors to decide) but I worry that adequately supporting a more expansive scope will require large time-consuming additions and modifications to the underlying dataset as well as a change in analytical approach. This manuscript is already far too long and adding more is not the way to fix the problems that are present here.
I fully agree that this review process has taken far longer than is ideal. My criticisms have not changed from the beginning of this process and it is unfortunate that only now are the authors beginning to address the methodological and collegial shortcomings of the paper. Given the duration between draft submissions (~9 months between revision 1 and revision 2), it seems unreasonable to blame the editorial staff or reviewers. Again, from the beginning I have suggested that this manuscript be split into 3-4 separate publications, each of which would be a more manageable unit of publication for the authors, editors, reviewers, and readers alike.

I have consistently argued for revisions that will require the absolute least amount of additional work. I think it is obvious at this point that the Casineria/Caerorhachis, Archeria/Five-Points, and Hapsidopareion/Llistrofus OTUs need to be broken up and all analyses rerun. Discussion of Casineria and Caerorhachis and justification for combining these taxa needs to be removed because the final version of this manuscript cannot use these composite taxa. I think it is also necessary to remove the remaining discussions of unpublished work in progress/preparation/press by other workers or to gain explicit permission to discuss that work. I would like to see better concordance between the scope of the paper and the analyses reported; currently the paper has an extremely broad scope but the analyses presented do not sufficiently support all the claims made. My preference would be for the authors to restrict the scope to what can be easily supported by the analyses employed (i.e. a comparison of different treatments of a single base dataset showing the impacts of how ambiguity is resolved by practitioners, possibly pointing out in broad strokes the less robust parts of the current consensus phylogeny) because I do not believe that the data and analyses presented here are sufficient to support the broader goals of the authors. If the authors insist on keeping the scope at its current scale (or increasing it, as they seem to be doing with each additional revision) then I personally believe that their claims need to be backed up by additional analyses to meet the standards of the field. Unfortunately, the manuscript is so large that it is difficult to give clear direction except on specific cases.

Ultimately, I don’t want to stand in the way of publication of this paper. The edits I have requested are mostly relatively simple but are necessary to maintain a semblance of adherence to taxonomic guidelines and collegiality in a manuscript which may not be well-received by many in the field due to disagreements about methodology, direction of inference, and tone. No one in the field will accept the a priori synonymization of Casineria and Caerorhachis, nor will they accept the fusion of the Five Points tail with Archeria. Substantial additional work must be completed before these choices can be justified a posteriori. Many in the field will take the premature discussion of unpublished results as personal attacks. The inclusion of re-assessments (read: polemic criticisms) of other unrelated published matrices into this manuscript will be taken by some as a personal attack. The casual dismissal of years of meticulous data collection in favor of ambiguous interpretations of single specimens will be taken by some as arrogant dismissiveness. As one of the authors whose work is criticized here, I realize that this is not personal or mean-spirited. I think some of these criticisms are unreasonable, incorrect, or irrelevant, but I am not interested in standing in the way of their publication. However, other workers may well take the contents of this manuscript as hostile and insulting. It has been my goal here to suggest ways to mitigate some of these effects by amending inappropriate taxonomic practices, moderating language, reducing discussion of other researcher’s unpublished results which may affect the impact of the resulting publications, and limiting the scope of the paper to more defensible ground. Unless a good faith effort is made to address these issues, my review is unlikely to change, and I’m happy to leave further decisions to the editor’s discretion.

---

## Round 0.5 · Minor Revisions

Dear Dr. Marjanović,

As you know, I was asked to serve as an adjudicating editor following the appeal you submitted in response to John Hutchinson’s “reject” decision. Your manuscript has now been through five rounds of review, and has been handled by a total of four different editors. Obviously this is a highly unusual and complex situation, particularly as the manuscript received a “minor revisions” decision in Round 2, only to be followed by a subsequent “major revisions” decision in Round 3, and a “reject” decision in Round 4.

I have received two reviews of the revised manuscript that you submitted after your appeal was considered and approved. Unfortunately, they are strongly at odds. Reviewer 1 (Pyron) recommends that the paper be accepted without any additional revisions, while Reviewer 3 (Pardo) recommends that the paper be rejected. I have been unable to secure additional reviews, but I note that in a previous round Reviewer 2 (Buchwitz) recommended that the paper be accepted. Given the long and complicated history of this submission, I have also consulted with the PeerJ Paleontology Section Editors Andrew Farke and Laura Wilson. Our consensus decision is that your manuscript should be Accepted with some remaining minor revisions (details below).

I appreciate that an enormous amount of effort has been put into this study and that you have accommodated many of the reviewers’ requests (adding a Bayesian analysis, splitting up composite OTUs, etc.). All of the reviewers agreed that the core reanalysis of Ruta and Coates (2007) (or RC07) is solid and clearly deserving of publication, but two of the three reviewers were concerned that the text had far outgrown this specific goal. After all, your study isn’t simply a reevaluation of RC07, it is also a very significant augmentation of RC07, and that point isn’t sufficiently highlighted. In my opinion the problem starts immediately, with the title of the paper -- only a small part of the study, as it currently stands, is strictly about “reproducibility” (that would be a far, far shorter paper). I think it would be worth your while to come up with a title that better encompasses the scope of what you are trying to accomplish here, so that readers have a better sense of what to expect.

One concern that both Buchwitz and Pardo raised is that some passages come across as unnecessarily combative or insulting. I appreciate that researchers from different academic and cultural backgrounds can have quite different conceptions of how criticism should be framed in academic discourse, and perhaps it is the case that you legitimately do not understand why Buchwitz and Pardo have expressed concern. Regardless, I would urge you to think carefully about whether any passages might be pointlessly creating ill will among colleagues. Our debates are far more likely to remain objective and productive in the long term if criticisms are communicated dispassionately and free of ridicule.

Through five long rounds of review, Jason Pardo has made a number of thoughtful recommendations. I appreciate that you have made several changes in response to his suggestions (the Caerorhachis/Casineria and Hapsidopareion/Llistrofus OTUs were split up, CM 34638 was removed from the analysis, discussion of the “middle ear” was removed, etc.). These changes have improved the paper. The overarching concern that was repeatedly raised by Pardo is the scope of the paper; he argued that the manuscript could be broken up into multiple papers, and I agree -- the current manuscript covers so much territory, and there are so many digressions interspersed throughout the text, that it is frankly very hard to follow at times. It is clear after five rounds of review that you have no intention of breaking this manuscript up into multiple papers, but any additional effort you could make to reduce the size of the text would likely improve readability. The reanalyses of the Maddin et al. (2012) and Pardo et al. (2017) matrices stood out to me as examples of particularly unnecessary detours that seemed like total distractions from the primary goals of your study; please remove these from the main text.

Pardo was also concerned about citation of “grey literature” (abstracts, masters theses, etc.). Fortunately some of the studies concerned are now citable as peer-reviewed papers. Regardless, I am not of the opinion that abstracts and theses are off-limits for citation; I have certainly cited abstracts and theses in my own papers, primarily as a way (in my eyes at least) of politely recognizing the priority of somebody’s work (particularly in cases where peer-reviewed publication of such work has been significantly delayed).

Despite the concerns mentioned above, this study is extremely impressive. It is a significant contribution to the study of early tetrapod phylogeny and deserves to be published. As you revise your manuscript I hope that you will make sincere efforts to further reduce its size (including cutting out the reanalyses of the Maddin et al. and Pardo et al. matrices), and that you will pay close attention to how your criticisms are expressed, removing any unnecessarily confrontational or inflammatory statements.

·

Basic reporting

All of the PeerJ criteria for basic reporting are satisfied.

Experimental design

All of the PeerJ criteria for experimental design are satisfied.

Validity of the findings

All of the PeerJ criteria for validity are satisfied.

Additional comments

I have reviewed this MS for a third time, apparently having been omitted for two previous rounds of review. As I've noted previously, the amount of effort expended and data presented are simply staggering. This defies an attempt to evaluate every aspect of the MS.

Instead, the main goals of review should be to ensure: i) reproducibility, ii) documentation of all methodological and analytical steps, and iii) absence of any major errors of fact or logic.

In this regard, I think the current MS satisfies all criteria. Regardless of disagreements about character states within particular taxa, the authors provide their rationale for all changes, allowing future researchers to cross-check their analyses. This will be invaluable as tetrapod systematics progresses.

In particular, the authors added a Bayesian analysis as I requested, and their methodological settings and comparison to the parsimony trees is illuminating. Hopefully, future researchers will begin to apply more complex models of morphological evolution (e.g., Pyron 2017 Syst. Biol.), but this is a great first start, to assess the potential for long-branch attraction to skew the estimated trees.

I think it's unsurprising that additional examination and clarification of taxa and characters actually decreases confidence in estimated results; as the authors note, hopefully this will spur the additional labor needed to commence additional collection of data. If we're lucky, the matrix presented here will form the foundation of great future strides in understanding.

·

Basic reporting

Paper is long and meandering.

Digressions and casual discussions need to be eliminated.

Paper still lacks a clear rationale that supports the experiments outlined.

Citations and responses to works in progress by other research groups need to be removed.

General paper structure appears to promise a methodological investigation of how reproducible matrix construction is. Instead, paper claims to deliver a phylogeny of tetrapod relationships and a resolution to the lissamphibian origins problem. In actuality, it is very uncertain what the paper actually delivers at all.

Experimental design

Experimental design is not clear. Many experiments are reported but how these support the rationale is unclear. Basic summary statistics on character recoding are not reported, making it difficult to assess whether any patterns in supposed errors exist. Sensitivity analyses are not conducted or reported, making it impossible to tell why the results differ. Statistics of tree similarity are not reported, making it difficult to tell whether differences between trees are actually meaningful.

This is not the first paper the authors have written with this methodology, and it seems that no matter the source dataset, these authors get the same general set of results. It is unclear whether everyone else's dataset is bad, or whether the authors are introducing biases in morphological interpretation or character treatment. One way to solve this problem is to bring in outside collaborators from other research groups and statistically compare the resulting matrices. This is obviously not done here either.

Validity of the findings

The actual findings of the authors are difficult to interpret because their phylogenetic experiments are not appropriately controlled and do not employ appropriate sensitivity analyses.

The findings that the authors appear to want the reader to accept (overall phylogenetic patterns and resolution of the lissamphibian origins problem) are not valid for the reason that they are not actually tested here. The experiments here test whether it is possible to break the underlying dataset of Ruta & Coates 2007, but does not test how much change is necessary to do so (this would require sensitivity analysis) nor does it test how much change is actually reasonable to impose here (that would require replicate efforts by several independent researchers/groups).

The specific conclusions of the authors are difficult to determine, but much of the discussion is only tentatively linked to the experiments reported and is not supported even by the data reported here.

---

## Round 0.6 · Minor Revisions

Thank you for the efforts that you have made to shorten your manuscript and further address the comments of the reviewers. The current version is much improved, and I am of the opinion that this submission will be acceptable for publication with two additional minor changes: 1) Please remove the Table of Contents to be consistent with PeerJ’s formatting requirements, and 2) remove the “tests find no bias (p > 0.1)” passage in the abstract, as this statement is meaningless without additional details about which tests were employed. In the same vein, the following passage in the abstract cannot be interpreted without additional information about which test was employed, and so I recommend deletion: “Statistically, the TH (R2, R5) is not distinguishable from the LH or the PH; LH (R1) and PH (R3) may be distinguishable from each other given the original taxon sample at p > 0.035. A test for the upper bound of p is not available.” The abstract should be comprehensible without reference to the main text.

---

## Round 0.7 · accepted · Accept

Thank you for making the requested changes. I now consider the manuscript to be acceptable for publication in PeerJ. Congratulations, and I look forward to seeing your paper in print!

#